# Firefly genomes illuminate parallel origins of bioluminescence in beetles

**Timothy R Fallon[1,2†], Sarah E Lower[3,4†], Ching-Ho Chang[5], Manabu Bessho-Uehara[6,7,8], Gavin J Martin[9], Adam J Bewick[10], Megan Behringer[11], Humberto J Debat[12], Isaac Wong[5], John C Day[13], Anton Suvorov[9], Christian J Silva[5,14], Kathrin F Stanger-Hall[15], David W Hall[10], Robert J Schmitz[10], David R Nelson[16], Sara M Lewis[17], Shuji Shigenobu[18], Seth M Bybee[9], Amanda M Larracuente[5], Yuichi Oba[6], Jing-Ke Weng[1,2]\***

[1]Whitehead Institute for Biomedical Research, Cambridge, United States; [2]Department of Biology, Massachusetts Institute of Technology, Cambridge, United States; [3]Department of Molecular Biology and Genetics, Cornell University, Ithaca, United States; [4]Department of Biology, Bucknell University, Lewisburg, United States; [5]Department of Biology, University of Rochester, Rochester, United States; [6]Department of Environmental Biology, Chubu University, Kasugai, Japan; [7]Graduate School of Bioagricultural Sciences, Nagoya University, Nagoya, Japan; [8]Monterey Bay Aquarium Research Institute, Moss Landing, United States; [9]Department of Biology, Brigham Young University, Provo, United States; [10]Department of Genetics, University of Georgia, Athens, United States; [11]Biodesign Center for Mechanisms of Evolution, Arizona State University, Tempe, United States; [12]Center of Agronomic Research, National Institute of Agricultural Technology, Córdoba, Argentina; [13]Centre for Ecology and Hydrology (CEH), Wallingford, United Kingdom; [14]Department of Plant Sciences, University of California Davis, Davis, United States; [15]Department of Plant Biology, University of Georgia, Athens, United States; [16]Department of Microbiology Immunology and Biochemistry, University of Tennessee HSC, Memphis, United States; [17]Department of Biology, Tufts University, Medford, United States; [18]NIBB Core Research Facilities, National Institute for Basic Biology, Okazaki, Japan

**\*For correspondence:**
wengj@wi.mit.edu

[†]These authors contributed equally to this work

**Competing interests:** The authors declare that no competing interests exist.

**Abstract** Fireflies and their luminous courtships have inspired centuries of scientific study. Today firefly luciferase is widely used in biotechnology, but the evolutionary origin of bioluminescence within beetles remains unclear. To shed light on this long-standing question, we sequenced the genomes of two firefly species that diverged over 100 million-years-ago: the North American *Photinus pyralis* and Japanese *Aquatica lateralis*. To compare bioluminescent origins, we also sequenced the genome of a related click beetle, the Caribbean *Ignelater luminosus*, with bioluminescent biochemistry near-identical to fireflies, but anatomically unique light organs, suggesting the intriguing hypothesis of parallel gains of bioluminescence. Our analyses support independent gains of bioluminescence in fireflies and click beetles, and provide new insights into the genes, chemical defenses, and symbionts that evolved alongside their luminous lifestyle.
DOI: https://doi.org/10.7554/eLife.36495.001

**eLife digest** Glowing fireflies dancing in the dark are one of the most enchanting sights of a warm summer night. Their light signals are 'love messages' that help the insects find a mate – yet, they also warn a potential predator that these beetles have powerful chemical defenses. The light comes from a specialized organ of the firefly where a small molecule, luciferin, is broken down by the enzyme luciferase.

Fireflies are an ancient group, with the common ancestor of the two main lineages originating over 100 million years ago. But fireflies are not the only insects that produce light: certain click beetles are also bioluminescent.

Fireflies and click beetles are closely related, and they both use identical luciferin and similar luciferases to create light. This would suggest that bioluminescence was already present in the common ancestor of the two families. However, the specialized organs in which the chemical reactions take place are entirely different, which would indicate that the ability to produce light arose independently in each group.

Here, Fallon, Lower et al. try to resolve this discrepancy and to find out how many times bioluminescence evolved in beetles. This required using cutting-edge DNA sequencing to carefully piece together the genomes of two species of fireflies (*Photinus pyralis* and *Aquatica lateralis*) and one species of click beetle (*Ignelater luminosus*). The genetic analysis revealed that, in all species, the genes for luciferases were very similar to the genetic sequences around them, which code for proteins that break down fat. This indicates that the ancestral luciferase arose from one of these metabolic genes getting duplicated, and then one of the copies evolving a new role.

However, the genes for luciferase were very different between the fireflies and the click beetles. Further analyses suggested that bioluminescence evolved at least twice: once in an ancestor of fireflies, and once in the ancestor of the bioluminescent click beetles.

More results came from the reconstituted genomes. For example, Fallon, Lower et al. identified the genes 'turned on' in the bioluminescent organ of the fireflies. This made it possible to list genes that may be involved in creating luciferin, and enable flies to grow brightly for long periods. In addition, the genetic information yielded sequences from bacteria that likely live inside firefly cells, and which may participate in the light-making process or the production of potent chemical defenses.

Better genetic knowledge of beetle bioluminescence could bring new advances for both insects and humans. It may help researchers find and design better light-emitting molecules useful to track and quantify proteins of interest in a cell. Ultimately, it would allow a detailed understanding of firefly populations around the world, which could contribute to firefly ecotourism and help to protect these glowing insects from increasing environmental threats.

DOI: https://doi.org/10.7554/eLife.36495.002

## Introduction

Fireflies (Coleoptera: Lampyridae) represent the best-studied case of bioluminescence. The coded language of their luminous courtship displays (*Figure 1A*; *Video 1*) has been long studied for its role in mate recognition (*Lloyd, 1966*; *Lewis and Cratsley, 2008*; *Stanger-Hall and Lloyd, 2015*), while non-adult bioluminescence is likely a warning signal of their unpalatable chemical defenses (*De Cock and Matthysen, 1999*), such as the cardiotoxic lucibufagins of *Photinus* fireflies (*Meinwald et al., 1979*). The biochemical understanding of firefly luminescence: an ATP, $Mg^{2+}$, and $O_2$-dependent luciferase-mediated oxidation of the substrate luciferin (*Shimomura, 2012*), along with the cloning of the luciferase gene (*de Wet et al., 1985*; *Ow et al., 1986*), led to the widespread use of luciferase as a reporter with unique applications in biomedical research and industry (*Fraga, 2008*). With >2000 species globally, fireflies are undoubtedly the most culturally appreciated bioluminescent group, yet there are at least three other beetle families with bioluminescent species: click beetles (Elateridae), American railroad worms (Phengodidae) and Asian starworms (Rhagophthalmidae) (*Martin et al., 2017*). These four closely related families (superfamily Elateroidea) have homologous luciferases and structurally identical luciferins (*Shimomura, 2012*), implying a single origin of beetle bioluminescence. However, as Darwin recognized in his 'Difficulties on Theory' (*Darwin, 1872*), the

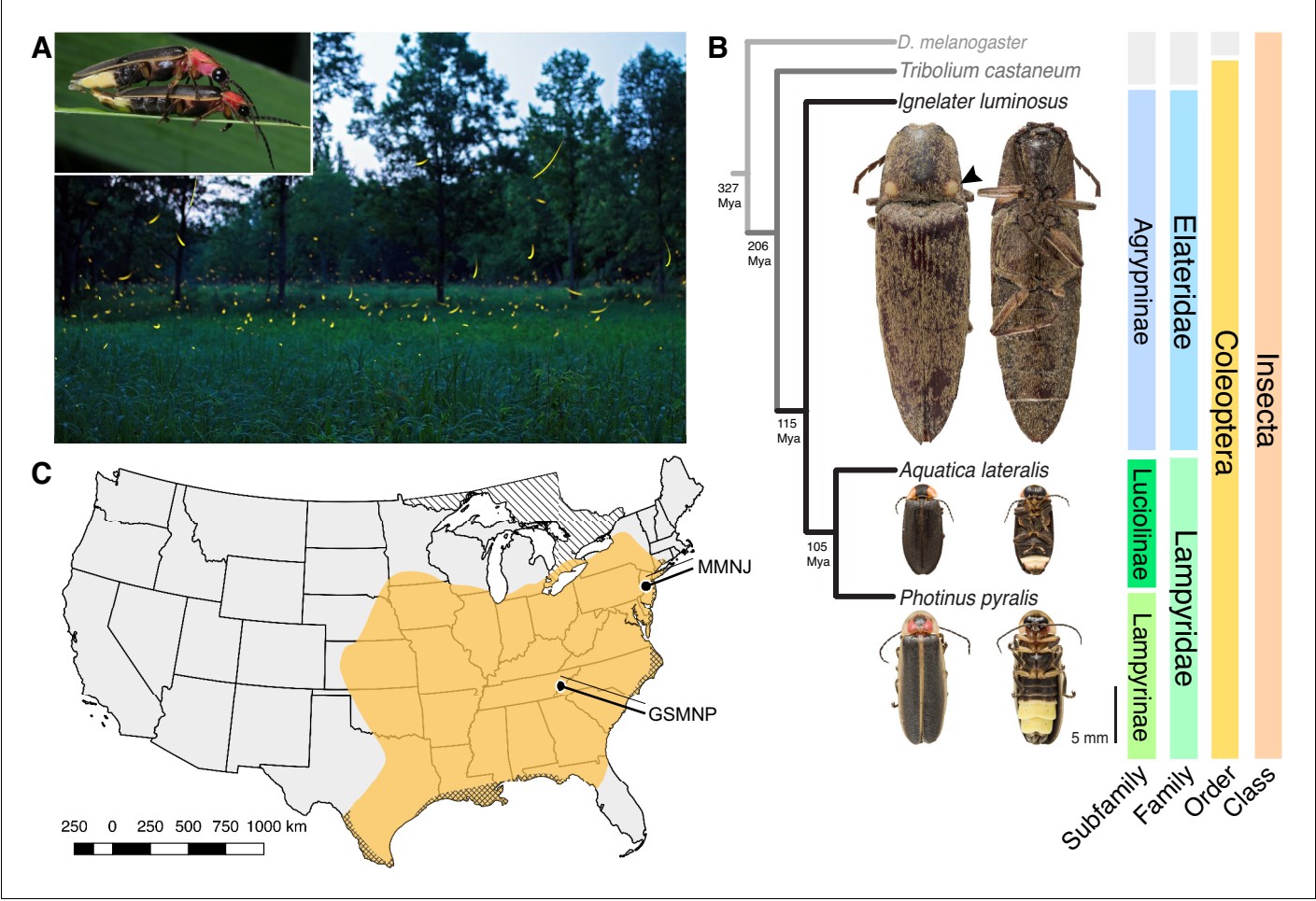

**Figure 1.** Geographic and phylogenetic context of the Big Dipper firefly, *Photinus pyralis*. (**A**) *P. pyralis* males emitting their characteristic swooping 'J' patrol flashes over a field in Homer Lake, Illinois. Females cue in on these species-specific flash patterns and respond with their own species-specific flash (*Lloyd, 1966*). Photo credit: Alex Wild. Inset: male and female *P. pyralis* in early stages of mating. Photo credit: Terry Priest. (**B**) Cladogram depicting the hypothetical phylogenetic relationship between *P. pyralis* and related bioluminescent and non-bioluminescent taxa with *Tribolium castaneum* and *Drosophila melanogaster* as outgroups. Numbers at nodes give approximate dates of divergence in millions of years ago (mya) (*Misof et al., 2014*; *Mckenna et al., 2015*). Right: Dorsal and ventral photos of adult male specimens. Note the well-developed ventral light organs on the true abdominal segments 6 and 7 of *P. pyralis* and *A. lateralis*. In contrast, the luminescent click beetle, *I. luminosus*, has paired dorsal light organs at the base of its prothorax (arrowhead) and a lantern on the anterior surface of the ventral abdomen (not visible). (**C**) Empirical range of *P. pyralis* in North America, extrapolated from 541 reported sightings (Appendix 1.2). Collection sites of individuals used for genome assembly are denoted with circles and location codes. Cross hatches represent areas which likely have *P. pyralis*, but were not sampled. Diagonal hashes represent Ontario, Canada.

DOI: https://doi.org/10.7554/eLife.36495.003

light organs amongst the luminous beetle families are clearly distinct (*Figure 1B*), implying independent origins. Thus, whether beetle bioluminescence is derived from a single or multiple origin(s) remains unresolved.

To address this long-standing question, we sequenced and analyzed the genomes of three bioluminescent beetle species. To represent the fireflies, we sequenced the widespread North American 'Big Dipper Firefly', *P. pyralis* (*Figure 1A,C*) and the Japanese 'Heike-botaru' firefly *Aquatica lateralis* (*Figure 1B*). *P. pyralis* was used in classic studies of firefly bioluminescent biochemistry (*Bitler and McElroy, 1957*) and the cloning of luciferase (*de Wet et al., 1985*), while *A. lateralis*, a species with specialized aquatic larvae, is one of the few fireflies that can be reliably cultured in the laboratory (*Oba et al., 2013a*). These two fireflies represent the two major firefly subfamilies, Lampyrinae and Luciolinae, which diverged from a common ancestor over 100 Mya (*Figure 1B*) (*Misof et al., 2014*;

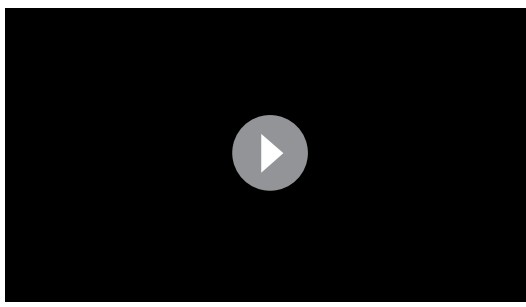

**Video 1.** A *Photinus pyralis* courtship dialogue.
DOI: https://doi.org/10.7554/eLife.36495.004

*Mckenna et al., 2015*). To facilitate evolutionary comparisons, we also sequenced the 'Cucubano', *Ignelater luminosus* (*Figure 1B*), a Caribbean bioluminescent click beetle, and member of the '*Pyrophorus*' used by Raphaël Dubois (1849-1929) to first establish the enzymatic basis of bioluminescence in the late 1800s (*Dubois, 1885*; *Dubois, 1886*). Comparative analyses of the genomes of these three species allowed us to reconstruct the origin(s) and evolution of beetle bioluminescence.

# Results

## Sequencing and assembly of firefly and click-beetle genomes

*Photinus pyralis* adult males were collected from the Great Smoky Mountains National Park, USA (GSMNP) and Mercer Meadows New Jersey, USA (MMNJ) (*Figure 1C*), and sequenced using short-insert, mate-pair, Hi-C, and long-read Pacific Biosciences (PacBio) approaches (*Appendix 4—table 1*). These datasets were combined in a MaSuRCA (*Zimin et al., 2013*) hybrid genome assembly (Appendix 1.5). The *Aquatica lateralis* genome was derived from an ALL-PATHs (*Butler et al., 2008*) assembly of short insert and mate-pair reads from a single adult female from a laboratory-reared population, whose lineage, dubbed 'Ikeya-Y90', was first collected 25 years ago from a now extinct population in Yokohama, Japan (Appendix 2.5). A single *Ignelater luminosus* adult male, collected in Mayagüez Puerto Rico, USA, was used to produce a high-coverage Supernova (*Weisenfeld et al., 2017*) linked-read draft genome (Appendix 3.5), which was further manually scaffolded using low-coverage long-read Oxford Nanopore MinION sequencing (Appendix 3.5.4).

The gene completeness and contiguity statistics of our *P. pyralis* (Ppyr1.3) and *A. lateralis* (Alat1.3) genome assemblies are comparable to the genome of the model beetle *Tribolium castaneum* (*Figure 2F*; Appendix 4.1). The *I. luminosus* genome assembly (Ilumi1.2) is less complete, but is comparable to other published insect genomes (*Figure 2F*; Appendix 4.1). Protein-coding gene-sets for our study species were produced via an EvidenceModeler-mediated combination of homology alignments, *ab initio* predictions, and *de novo* and reference-guided RNA-seq assemblies followed by manual gene curation for gene families of interest (Appendix 1.10; 2.8; 3.8). These coding gene annotation sets for *P. pyralis, A. lateralis,* and *I. luminosus* are comprised of 15,773, 14,285, and 27,557 genes containing 94.2%, 90.0%, and 91.8% of the Endopterygota Benchmarking Universal Single-Copy Orthologs (BUSCOs) (*Simão et al., 2015*), respectively. Protein clustering via predicted orthology indicated 77% of genes were found in orthogroups with at least one other species (*Figure 2E*; *Appendix 4—figure 1*). We found the greatest orthogroup overlap between the *P. pyralis* and *A. lateralis* genesets, as expected given the more recent phylogenetic divergence of these species. Remaining redundancy in the *P. pyralis* assembly and annotation, as indicated by duplicates of the BUSCOs and the assembly size (*Figure 2F*; *Appendix 4—table 2*) is likely due to the heterozygosity of the outbred input libraries (Appendix 1). The higher BUSCO completeness of the assemblies as compared to the genesets (*Appendix 4—table 3*), suggests that future manual curation efforts will lead to improved annotation completeness.

To enable the characterization of long-range genetic structure, we super-scaffolded the *P. pyralis* genome assembly into 11 pseudo-chromosomal linkage groups using a Hi-C proximity-ligation linkage approach (*Figure 2A*; Appendix 1.5.3). These linkage groups contain 95% of the assembly (448.8 Mbp). Linkage group LG3a corresponds to the X-chromosome based on expected adult XO male read coverage and gene content (Appendix 1.6.4.1) and its size (22.2 Mbp) is comparable to the expected X-chromosome size based on sex-specific genome size estimates using flow cytometry (~26 Mbp) (*Lower et al., 2017*). Homologs to *T. castaneum* X-chromosome genes were enriched on LG3a over every other linkage group, suggesting that the X-chromosomes of these distantly related beetles are homologous, and that their content has been reasonably conserved for >200 MY (Appendix 1.6.4.1) (*Mckenna et al., 2015*). We hypothesized that the *P. pyralis* orthologs of known

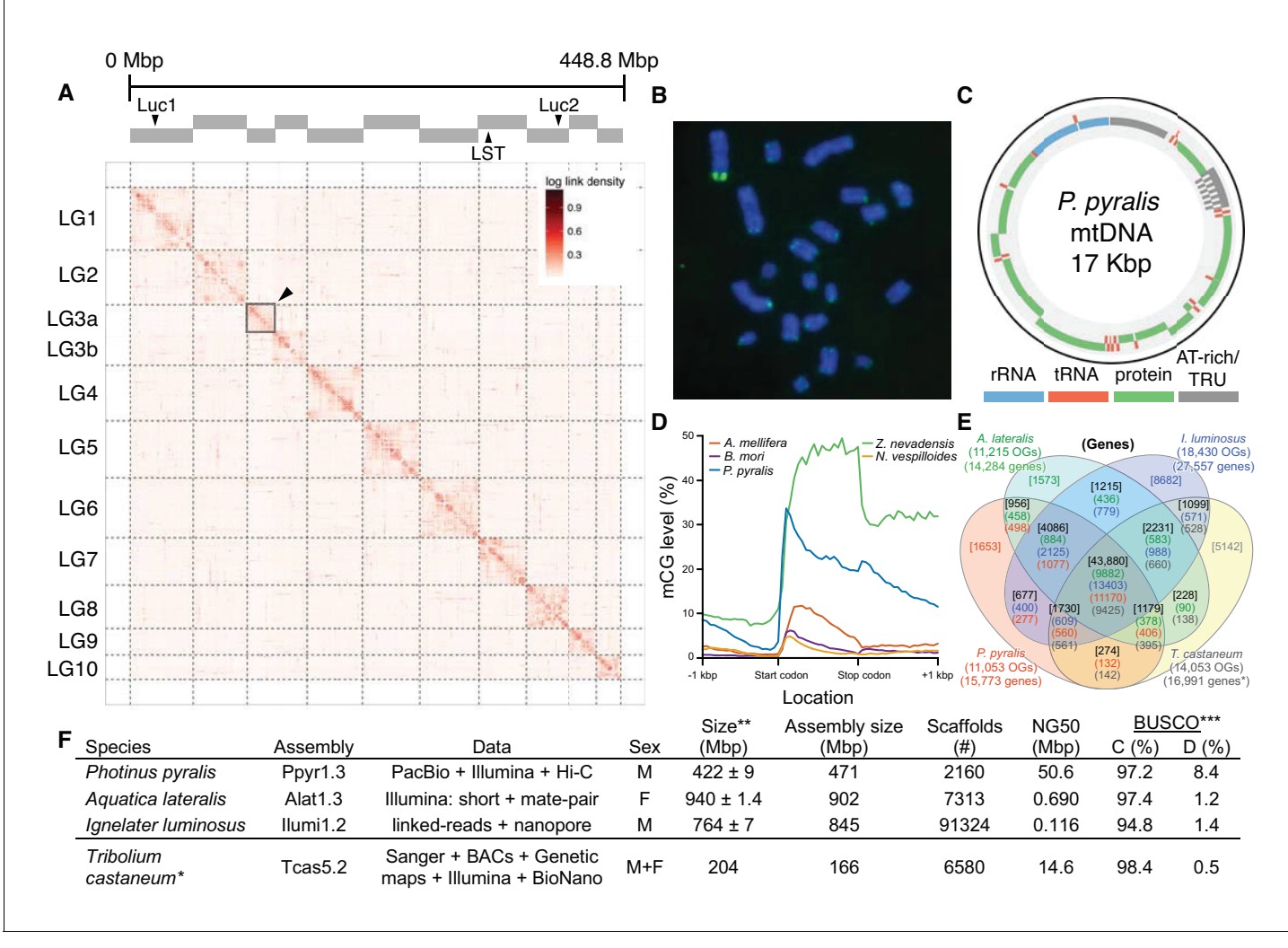

Figure 2. *Photinus pyralis* genome assembly and analysis. (A) Assembled Ppyr1.3 linkage groups with annotation of the location of known luminescence-related genes, combined with Hi-C linkage density maps. Linkage group 3a (box with black arrow) corresponds to the X chromosome (Appendix 1.6.4.1). (B) Fluorescence in situ hybridization (FISH) on mitotic chromosomes of a *P. pyralis* larvae. The telomeric repeats TTAGG (green) localize to the ends of chromosomes stained with DAPI (blue). 20 paired chromosomes indicates that this individual was an XX female (Appendix 1.13). (C) Genome schematic of *P. pyralis* mitochondrial genome (mtDNA). Like other firefly mtDNAs, it has a tandem repetitive unit (TRU) (Appendix 1.8). (D) mCG is enriched across gene bodies of *P. pyralis* and shows methylation levels that are at least two times higher than other holometabolous insects (Appendix 1.12). (E) Orthogroup (OGs) clustering analysis of genes with Orthofinder (**Emms and Kelly, 2015**) shows a high degree of overlap of the *P. pyralis*, *A. lateralis*, and *I. luminosus* genesets with the geneset of *Tribolium castaneum*. Numbers within curved brackets (colored by species) represent gene count from specific species within the shared orthogroups. Numbers with square brackets (black color) represent total gene count amongst shared orthogroups. OGs = orthogroups, *=Not fully filtered to single isoform per gene. See Appendix 4.2.1 for more detail. Intermediate scripts and species-specific overlaps are available as **Figure 2—source data 1**. (F) Assembly statistics for presented genomes. *=*Tribolium castaneum* model beetle genome assembly (**Tribolium Genome Sequencing Consortium et al., 2008**) **=Genome size estimated by FC: flow cytometry. *P. pyralis* n = 5 females (SEM) *I. luminosus* n = 5 males (SEM), *A. lateralis* n = 3 technical-replicates of one female (SD). ***=Complete (C), and Duplicated (D), percentages for the Endopterygota BUSCO (**Simão et al., 2015**) profile (Appendix 1.4, 2.4, 3.4, 4.1).

DOI: https://doi.org/10.7554/eLife.36495.005

The following source data is available for figure 2:

Source data 1. *Figure 2E*. Orthogroup clustering analysis.
DOI: https://doi.org/10.7554/eLife.36495.006
Source data 2. Excel file of *Figure 2F* table.
DOI: https://doi.org/10.7554/eLife.36495.007

bioluminescence genes, including the canonical luciferase *Luc1* (**de Wet et al., 1985**) and the

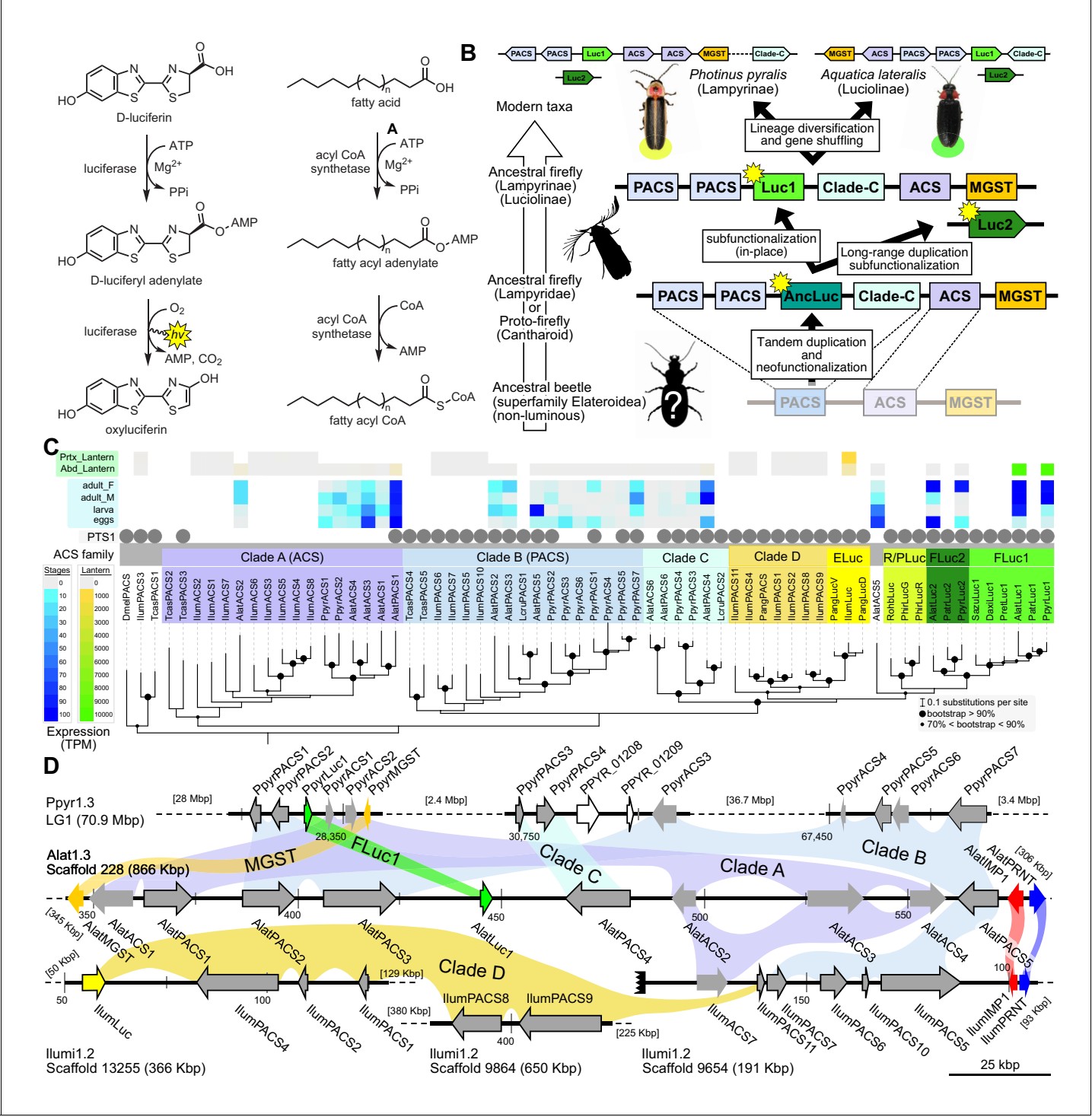

**Figure 3.** A genomic view of luciferase evolution. (A) The reaction scheme of firefly luciferase is related to that of fatty acyl-CoA synthetases. (B) Model for genomic evolution of firefly luciferases. Ranging from genome structures of luciferase loci in extant fireflies (top), to inferred genomic structures in ancestral species (bottom). Arrow (left) represents ascending time. Not all adjacent genes within the same clade are shown. (C) Maximum likelihood tree of luciferase homologs. Grey circles above gene names indicate the presence of peroxisomal targeting signal 1 (PTS1). Color gradients indicate the transcript per million (TPM) values of whole body in each sex/stage (grey to blue) and in the prothorax or abdominal lantern (grey to orange to green). Tree and annotation visualized using iTOL (*Letunic and Bork, 2016*). Prothorax and abdominal lantern expression values for *I. luminosus* are from whole prothorax plus head, and metathorax plus the two most anterior abdominal segments. Fluc = firefly luciferases, Eluc = elaterid luciferases, R/ PLuc = rhagophthalmid/phengodid luciferases. (Appendix 4.3.2) Gene tree, gene accession numbers, annotation, and expression values are available as *Figure 3—source data 1*. (D) Synteny analysis of beetle luciferase homologs. Nine of the 14 *A. lateralis* PACS/ACS genes closely flank AlatLuc1 on

*Figure 3 continued*

scaffold 228, while 4 of the 13 *P. pyralis* PACS/ACS genes are close neighbors of PpyrLuc1 on LG1, with a further seven genes 2.4 Mbp and 39.1 Mbp away on the same linkage-group. Although the *Luc1* loci in *P. pyralis* and *A. lateralis* are evidently derived from a common ancestor, the relative positions of the most closely related flanking PACS/ACS genes have diverged between the two species. *IlumLuc* was captured on a separate scaffold (Ilumi1.2_Scaffold13255) from its most most closely related PACSs (*IlumPACS8, IlumPACS9*) on Ilumi1.2_Scaffold9864, although three more distantly related PACS genes (*IlumiPACS1, IlumiPACS2, IlumiPACS4*) are co-localized with *IlumLuc*. In contrast, a different scaffold (Ilumi1.2_Scaffold9654) shows orthology to the firefly *Luc1* locus. The full Ilumi1.2_Scaffold13255 was produced by a manual evidence-supported merge of two scaffolds (Appendix 3.5.4). Genes with a PTS1 are indicated by a dark outline, except for the genes with white interiors, which instead represent non-PACS/ACS genes without an identified homolog in the other scaffolds. Co-orthologous genes are labeled in the same color in the phylogenetic tree and are connected with corresponding color bands in synteny diagram. Genes and genomic regions are to scale (Scale bar = 25 Kbp). Gaps excluded from the figure are shown with dotted lines and are annotated with their length in square brackets. Scaffold ends are shown with rough black bars. MGST = Microsomal glutathione S-transferase, IMP = Inositol monophosphatase, PRNT = Polyribonucleotide nucleotidyltransferase. Figure produced with GenomeTools 'sketch' (v1.5.9) (*Gremme et al., 2013*). Figure production scripts available as *Figure 3—source data 2*.

DOI: https://doi.org/10.7554/eLife.36495.008

The following source data is available for figure 3:

**Source data 1.** Gene tree, gene accession numbers, annotation, and expression values for *Figure 3C*.

DOI: https://doi.org/10.7554/eLife.36495.009

**Source data 2.** Bash scripts for *Figure 3D* figure production.

DOI: https://doi.org/10.7554/eLife.36495.010

specialized luciferin sulfotransferase *LST* (*Fallon et al., 2016*), would be located on the same linkage group to facilitate chromosomal looping and enhancer assisted co-expression within the light organ. We, however, found these genes on separate linkage groups (*Figure 2A*).

In addition to nuclear genome assembly and coding gene annotation, we also assembled the complete mitochondrial genomes (mtDNA) of *P. pyralis* (*Figure 2C*; Appendix 1.8) and *I. luminosus* (Appendix 3.10), while the mtDNA sequence of *A. lateralis* was recently published (*Maeda et al., 2017*). These mtDNA assemblies show high conservation of gene content and synteny, with the exception of the variable ~1 Kbp tandem repeat unit (TRU) found in the firefly mtDNAs.

As repetitive elements are common participants and drivers of genome evolution (*Feschotte and Pritham, 2007*), we next sought to characterize the repeat content of our genome assemblies. Overall, 42.6%, 19.8%, and 34.1% of the *P. pyralis*, *A. lateralis*, and *I. luminosus* assemblies were found to be repetitive, respectively (Appendix 1.11; 2.9; 3.9). Of these repeats 66.7%, 39.4%, and 55% could not be classified as any known repetitive sequence, respectively. Helitrons, DNA transposons that transpose through rolling circle replication (*Kapitonov and Jurka, 2001*), are among the most abundant individual repeat elements in the *P. pyralis* assembly. Via in situ hybridization, we identified that *P. pyralis* chromosomes have canonical telomeres with telomeric repeats (TTAGG) (*Figure 2B*; Appendix 1.13).

DNA methylation is common in eukaryotes, but varies in degree across insects, especially within Coleoptera (*Bewick et al., 2017*). Furthermore, the functions of DNA methylation across insects remain obscure (*Bewick et al., 2017*; *Glastad et al., 2017*). To examine firefly cytosine methylation, we characterized the methylation status of *P. pyralis* DNA with whole genome bisulfite sequencing (WGBS). Methylation at CpGs (mCG) was unambiguously detected at ~20% within the genic regions of *P. pyralis* and its methylation levels were at least twice those reported from other holometabolous insects (*Figure 2D*; Appendix 1.12). Molecular evolution analyses of the DNA methyltransferases (DNMTs) show that direct orthologs of both DNMT1 and DNMT3 were conserved in *P. pyralis*, *A. lateralis,* and *I. luminosus* (*Appendix 4—figure 2*; Appendix 4.2.3), implying that our three study species, and inferentially likely most firefly lineages, possess mCG. Corroborating this claim, $CpG_{[O/E]}$ analysis of methylation indicated our three study species had DNA methylation (*Appendix 4—figure 3*).

## The genomic context of firefly luciferase evolution

Two luciferase paralogs have been previously described in fireflies (*Oba et al., 2013a*; *Bessho-Uehara et al., 2017*). *P. pyralis Luc1* was the first firefly luciferase cloned (*de Wet et al., 1985*), and its direct orthologs have been widely identified from other fireflies (*Oba, 2014*). The luciferase paralog *Luc2* was previously known only from a handful of Asian taxa, including *A. lateralis*

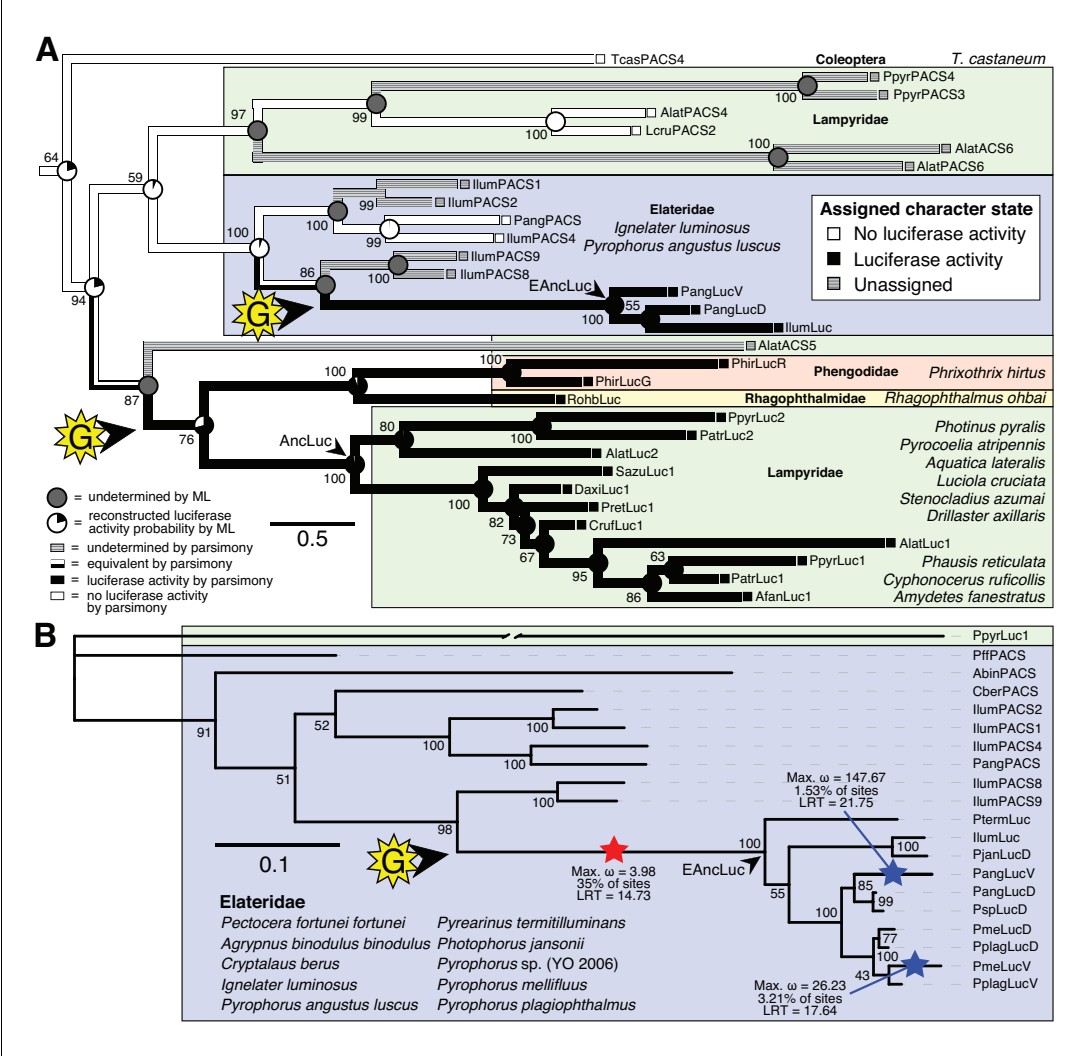

**Figure 4.** Parallel evolution of elaterid and firefly luciferase. (**A**) Ancestral state reconstruction recovers at least two gains of luciferase activity in bioluminescent beetles. Luciferase activity (top right figure key; black: luciferase activity, white: no luciferase activity, shaded: undetermined) was annotated on extant firefly luciferase homologs via literature review or inference via direct orthology. The ancestral states of luciferase activity within the putative ancestral nodes were then reconstructed with an unordered parsimony framework and a maximum likelihood (ML) framework (bottom left figure key; Appendix 4.3.3). Two gains ('G') of luciferase activity, annotated with black arrows and yellow stars, are hypothesized. These hypothesized gains occurred once in a gene within the common ancestor of fireflies, rhagophthalmid, and phengodid beetles, and once in a gene within the common ancestor of bioluminescent elaterid beetles. Scale bar is substitutions per site. Numbers adjacent to nodes represents node support. NEXUS and newick files available as **Figure 4—source data 1** (**B**) Molecular adaptation analysis supports independent neofunctionalization of click beetle luciferase. We tested the molecular adaptation of elaterid luciferase using the adaptive branch-site REL test for episodic diversification (aBSREL) method (**Smith et al., 2015**) (Appendix 4.3.4). The branch leading to the common ancestor of elaterid luciferases (red star) was one of three branches (red and blue stars) recovered with significant (p<0.01) evidence of positive selection, with 35% of sites showing strong directional selection ($\omega$ or max $d_N/d_S = 3.98$), which we interpret as signal of the initial neofunctionalization of elaterid ancestral luciferase (EAncLuc) from an ancestor without luciferase activity. As the selected branches with blue stars are red-shifted elaterid luciferases (**Oba et al., 2010a**; **Stolz et al., 2003**), they may represent the post-neofunctionalization selection of a few key sites via sexual selection of emission colors. Specific sites identified as under selection using Mixed Effect Model of Evolution (MEME) and Phylogenetic Analysis by Maximum Likelihood (PAML) methods are described in Appendix 4.3.4. The tree and results from the full adaptive model are shown. Branch length, with the exception of the PpyrLuc1 branch which was shortened, reflects the number of substitutions per site. Numbers adjacent to nodes represents node support. Figure was produced with iTOL (**Letunic and Bork, 2016**). Gene tree, metadata, and coding nucleotide multiple sequence alignment available as **Figure 4—source data 2**.

DOI: https://doi.org/10.7554/eLife.36495.011

The following source data is available for figure 4:

**Source data 1.** NEXUS and Newick files for luciferase ancestral state reconstruction in **Figure 4A**.

DOI: https://doi.org/10.7554/eLife.36495.012

*Figure 4 continued on next page*

*Figure 4 continued*

**Source data 2.** Gene tree, metadata, and coding nucleotide multiple sequence alignment for Elaterid luciferase homolog branch selection test.
DOI: https://doi.org/10.7554/eLife.36495.013

(*Oba et al., 2013a*; *Bessho-Uehara et al., 2017*). Previous investigations of these Asian taxa have shown that *Luc1* is responsible for light production from the lanterns of adults, larvae, prepupae and pupae, whereas *Luc2* is responsible for the dim glow of eggs, ovaries, prepupae and the whole pupal body (*Bessho-Uehara et al., 2017*). From our curated genesets (Appendix 1.10; 2.8), we unequivocally identified two firefly luciferases, *Luc1* and *Luc2*, in both the *P. pyralis* and *A. lateralis* genomes. Our RNA-Seq data further show that in both *P. pyralis* and *A. lateralis*, *Luc1* and *Luc2* display expression patterns consistent with previous reports. While *Luc1* is the sole luciferase expressed in the lanterns of both larvae and adults, regardless of sex, *Luc2* is expressed in other tissues and stages, such as eggs (*Figure 3C*). Notably, *Luc2* expression is detected in RNA libraries derived from adult female bodies (without head or lantern), suggesting detection of ovary expression as described in previous studies (*Bessho-Uehara et al., 2017*). Together, these results support that since their divergence via gene duplication prior to the divergence of Lampyrinae and Luciolinae, *Luc1* and *Luc2* have established different, but conserved roles in bioluminescence throughout the firefly life cycle.

Firefly luciferase is hypothesized to be derived from an ancestral peroxisomal fatty acyl-CoA synthetase (PACS) (*Figure 3A*) (*Oba et al., 2003*; *Oba et al., 2006a*). We found that, in both firefly species, *Luc1* is genomically clustered with its closely related homologs, including PACSs and non-peroxisomal acyl-CoA synthetases (ACSs), enzymes which can be distinguished by the presence/absence of a C-terminal peroxisomal-targeting-signal-1 (PTS1). We also found nearby microsomal glutathione S-transferase (MGST) family genes (*Figure 3D*) that are directly orthologous between

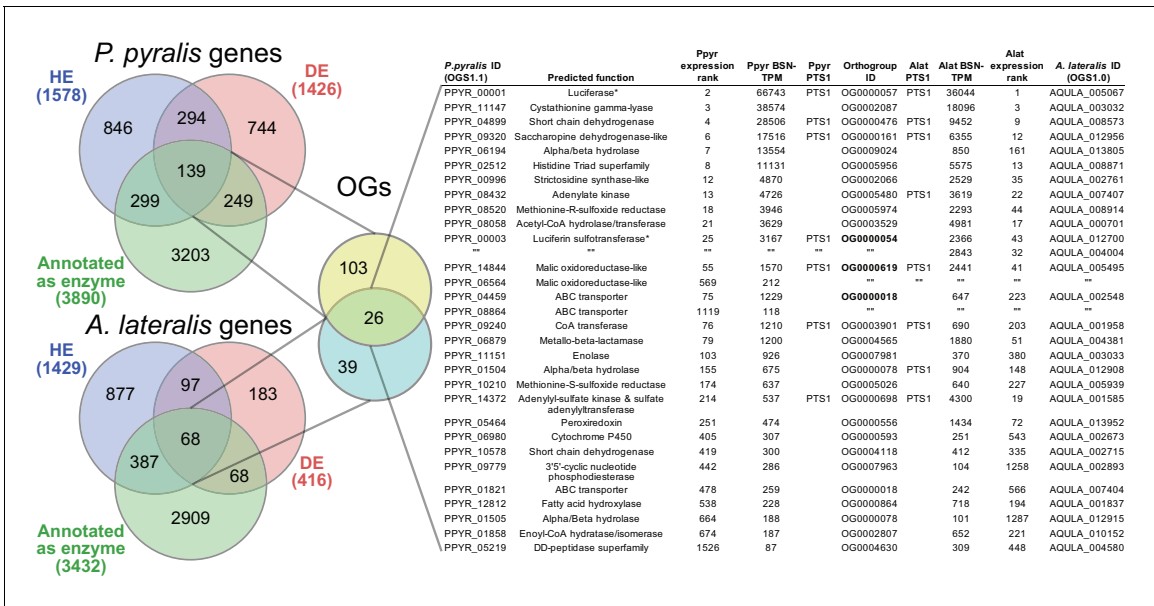

| P.pyralis ID (OGS1.1) | Predicted function | Ppyr expression rank | Ppyr BSN-TPM | Ppyr PTS1 | Orthogroup ID | Alat PTS1 | Alat BSN-TPM | Alat expression rank | A. lateralis ID (OGS1.0) |
|---|---|---|---|---|---|---|---|---|---|
| PPYR_00001 | Luciferase* | 2 | 66743 | PTS1 | OG0000057 | PTS1 | 36044 | 1 | AQULA_005067 |
| PPYR_11147 | Cystathionine gamma-lyase | 3 | 38574 | | OG0002087 | | 18096 | 3 | AQULA_003032 |
| PPYR_04899 | Short chain dehydrogenase | 4 | 28506 | PTS1 | OG0000476 | PTS1 | 9452 | 9 | AQULA_008573 |
| PPYR_09320 | Saccharopine dehydrogenase-like | 6 | 17516 | PTS1 | OG0000161 | PTS1 | 6355 | 12 | AQULA_012956 |
| PPYR_06194 | Alpha/beta hydrolase | 7 | 13554 | | OG0009024 | | 850 | 161 | AQULA_013805 |
| PPYR_02512 | Histidine Triad superfamily | 8 | 11131 | | OG0005956 | | 5575 | 13 | AQULA_008871 |
| PPYR_00996 | Strictosidine synthase-like | 12 | 4870 | | OG0002066 | | 2529 | 35 | AQULA_002761 |
| PPYR_08432 | Adenylate kinase | 13 | 4726 | | OG0005480 | PTS1 | 3619 | 22 | AQULA_007407 |
| PPYR_08520 | Methionine-R-sulfoxide reductase | 18 | 3946 | | OG0005974 | | 2293 | 44 | AQULA_008914 |
| PPYR_08058 | Acetyl-CoA hydrolase/transferase | 21 | 3629 | | OG0003529 | | 4981 | 17 | AQULA_000701 |
| PPYR_00003 | Luciferin sulfotransferase* | 25 | 3167 | PTS1 | **OG0000054** | | 2366 | 43 | AQULA_012700 |
| "" | | | | | "" | | 2843 | 32 | AQULA_004004 |
| PPYR_14844 | Malic oxidoreductase-like | 55 | 1570 | PTS1 | **OG0000619** | PTS1 | 2441 | 41 | AQULA_005495 |
| PPYR_06564 | Malic oxidoreductase-like | 569 | 212 | | "" | "" | | | |
| PPYR_04459 | ABC transporter | 75 | 1229 | | **OG0000018** | | 647 | 223 | AQULA_002548 |
| PPYR_08864 | ABC transporter | 1119 | 118 | | "" | "" | "" | | |
| PPYR_09240 | CoA transferase | 76 | 1210 | PTS1 | OG0003901 | PTS1 | 690 | 203 | AQULA_001958 |
| PPYR_06879 | Metallo-beta-lactamase | 79 | 1200 | | OG0004565 | | 1880 | 51 | AQULA_004381 |
| PPYR_11151 | Enolase | 103 | 926 | | OG0007981 | | 370 | 380 | AQULA_003033 |
| PPYR_01504 | Alpha/beta hydrolase | 155 | 675 | | OG0000078 | PTS1 | 904 | 148 | AQULA_012908 |
| PPYR_10210 | Methionine-S-sulfoxide reductase | 174 | 637 | | OG0005026 | | 640 | 227 | AQULA_005939 |
| PPYR_14372 | Adenylyl-sulfate kinase & sulfate adenylyltransferase | 214 | 537 | PTS1 | OG0000698 | PTS1 | 4300 | 19 | AQULA_001585 |
| PPYR_05464 | Peroxiredoxin | 251 | 474 | | OG0000556 | | 1434 | 72 | AQULA_013952 |
| PPYR_06980 | Cytochrome P450 | 405 | 307 | | OG0000593 | | 251 | 543 | AQULA_002673 |
| PPYR_10578 | Short chain dehydrogenase | 419 | 300 | | OG0004118 | | 412 | 335 | AQULA_002715 |
| PPYR_09779 | 3'5'-cyclic nucleotide phosphodiesterase | 442 | 286 | | OG0007963 | | 104 | 1258 | AQULA_002893 |
| PPYR_01821 | ABC transporter | 478 | 259 | | OG0000018 | | 242 | 566 | AQULA_007404 |
| PPYR_12812 | Fatty acid hydroxylase | 538 | 228 | | OG0000864 | | 718 | 194 | AQULA_001837 |
| PPYR_01505 | Alpha/Beta hydrolase | 664 | 188 | | OG0000078 | | 101 | 1287 | AQULA_012915 |
| PPYR_01858 | Enoyl-CoA hydratase/isomerase | 674 | 187 | | OG0002807 | | 652 | 221 | AQULA_010152 |
| PPYR_05219 | DD-peptidase superfamily | 1526 | 87 | | OG0004630 | | 309 | 448 | AQULA_004580 |

**Figure 5.** Comparative analyses of firefly lantern expression highlight likely metabolic adaptations to bioluminescence. Enzymes which are highly expressed (HE), differentially expressed (DE), and annotated as enzymes via InterProScan are shown in the Venn diagrams for their respective species. Those genes in the intersection of the two sets which are within the same orthogroup (OGs) as determined by OrthoFinder are shown in the table. Many-to-one orthology relationships are represented by bold orthogroups and blank cells. See Appendix 4.2.2 for more detail. *=genes of previously described function. Underlying expression quantification and Venn analysis available on FigShare: (DOI: 10.6084/m9.figshare.5715151)
DOI: https://doi.org/10.7554/eLife.36495.014

The following source data is available for figure 5:

**Source data 1.** Table of *Figure 5* highly expressed, differentially expressed, orthogroup overlapped genes.
DOI: https://doi.org/10.7554/eLife.36495.015

both species, Genome-wide phylogenetic analysis of the luciferases, PACSs and ACSs genes indicates that *Luc1* and *Luc2* form two orthologous groups, and that the neighboring PACS and ACS genes near *Luc1* form three major clades (*Figure 3C*): Clade A, whose common ancestor and most extant members are ACSs, and Clades B and C whose common ancestors and most extant members are PACSs. *Luc1* and *Luc2* are highly conserved at the level of gene structure—both are composed of seven exons with completely conserved exon/intron boundaries (*Appendix 4—figure 4*; *Appendix 4—figure 5*), and most members of Clades A, B, and C also have seven exons. The exact syntenic and orthology relationships of the ACS and PACS genes adjacent to the *Luc1* locus remains unclear, likely due to subsequent gene divergence and shuffling (*Figure 3C,D*).

Luc2 is located on a different linkage-group from *Luc1* in *P. pyralis* and on a different scaffold from *Luc1* in *A. lateralis,* consistent with the interpretation that *Luc1* and *Luc2* lie on different chromosomes in both firefly species. No PACS or ACS genes were found in the vicinity of *Luc2* in either species. These data support that tandem gene duplication in a firefly ancestor gave rise to several ancestral PACS paralogs, one of which neofunctionalized in place to become the ancestral luciferase (*AncLuc*) (*Figure 3B*). Prior to the divergence of the firefly subfamilies Lampyrinae and Luciolinae around 100 Mya (Appendix 4.3), this *AncLuc* duplicated, possibly via a long-range gene duplication event (e.g. transposon mobilization), and then subfunctionalized in its transcript expression pattern to give rise to *Luc2*, while the original *AncLuc* subfunctionalized in place to give rise to Luc1 (*Figure 3B*). From the shared *Luc* gene clustering in both fireflies, we infer the structure of the pre Luc1/Luc2 duplication *AncLuc* locus contained one or more ACS genes (Clade A), one or more PACS genes (Clade B/C), and one or more MGST family genes (*Figure 3B*).

## Independent origins of firefly and click beetle luciferase

To resolve the number of origins of luciferase activity, and therefore bioluminescence, between fireflies and click beetles, we first identified the luciferase of *I. luminosus* luciferase (*IlumLuc*), and compared its genomic context to the luciferases of *P. pyralis* and *A. lateralis* (*Figure 3D*). Unlike some other described bioluminescent Elateridae, which have separate luciferases expressed in the dorsal prothorax and ventral abdominal lanterns (*Oba et al., 2010a*), we identified only a single luciferase in the *I. luminosus* genome which was highly expressed in both of the lanterns (*Figure 3C*; Appendix 3.8). The exon number and exon-intron splice junctions of *IlumLuc* are identical to those of firefly luciferases, but unlike the firefly luciferases which have short introns less than <100 bp long, *IlumLuc* has two long introns (*Appendix 4—figure 4*). We found several PACS genes in the *I. luminosus* genome which were related to *IlumLuc* and formed a clade (Clade D) specific to the Elateridae (*Figure 3C,D*). *IlumLuc* lies on a 366 Kbp scaffold containing 18 other genes, including three related Clade D PACS genes (Scaffold 13255; *Figure 3D*; *Figure 4*); however, the Clade D genes that are most closely related to *IlumLuc* are found on a separate 650 Kbp scaffold (Scaffold 9864; *Figure 3D*). We infer that the *IlumLuc* locus is not orthologous to the extant firefly *Luc1* locus, as *IlumLuc* is not physically clustered with Clade A, B or C ACS or PACS genes (*Figure 3C,D*). We instead identified a different scaffold in *I. luminosus* that is likely orthologous to the firefly *Luc1* locus (Scaffold 9654; *Figure 3D*). This assessment is based on the presence of adjacent Clade A and B ACS and PACS genes, as well as orthologous exoribonuclease family (PRNT) and inositol monophosphatase family (IMP) genes, both of which were found adjacent to the *A. lateralis Luc1* locus, but not the *P. pyralis Luc1* locus (*Figure 3D*). Interestingly, *IlumPACS11*, the most early-diverging member of Clade D, was also found on Scaffold 9654 (*Figure 3D*). This finding is consistent with an expansion of Clade D following duplication of the *IlumPACS11* syntenic ancestor to a distant site. Overall, these genomic structures are consistent with independent origins of firefly and click beetle luciferases.

We then carried out targeted molecular evolution analyses including the known beetle luciferases and their closely related homologs. Ancestral state reconstruction of luminescent activity on the gene tree using Mesquite (*Maddison and Maddison, 2017*) recovered two independent gains of luminescence as the most parsimonious and likely scenario: once in click beetles, and once in the common ancestor of firefly, phengodid, and rhagophthalmid beetles (*Figure 4A*; Appendix 4.3.3). In an independent molecular adaptation analysis utilizing the coding nucleotide sequence of the elaterid luciferases and their close homologs within Elateridae, 35% of the sites of the branch leading to the ancestral click beetle luciferase showed a statistically significant signal of episodic positive selection with $d_N/d_S > 1$ ($\omega$ or max $d_N/d_S = 3.98$) as compared to the evolution of its paralogs using the aBSREL branch-site selection test (*Smith et al., 2015*) (*Figure 4B*; Appendix 4.3.4). This implies

that the common ancestor of the click beetle luciferases (*EAncLuc*) underwent a period of accelerated directional evolution. As the branch under selection in the molecular adaptation analysis (*Figure 4B*) is the same branch of luciferase activity gain via ancestral reconstruction (*Figure 4A*), we conclude that the identified selection signal represents the relatively recent neofunctionalization of click beetle luciferase from a non-luminous ancestral Clade D PACS gene, distinct from the more ancient neofunctionalization of firefly luciferase. Based on the constraints from our tree, we determine that this neofunctionalization of *EAncLuc* occured after the divergence of the elaterid subfamily Agrypninae. In contrast, we cannot determine if the original neofunctionalization of *AncLuc* occurred in the ancestral firefly, or at some point during the evolution of 'cantharoid' beetles, an unofficial group of beetles including the luminous Rhagophthalmidae, Phengodidae and Lampyridae among other non-luminous groups, but not the Elateridae (*Branham and Wenzel, 2003*). There is evidence for a subsequent luciferase duplication event in phengodids, but not in rhagophthalmids, that is independent of the duplication event that gave rise to *Luc1* and *Luc2* in fireflies (*Figures 3C* and *4*). Altogether, our results strongly support the independent neofunctionalization of luciferase activity in click beetles and fireflies, and therefore at least two independent gains of luciferin-utilizing luminescence in beetles.

## Metabolic adaptation of the firefly lantern

Beyond luciferase, we sought to characterize other metabolic traits which might have co-evolved in fireflies to support bioluminescence. Of particular importance, the enzymes of the *de novo* biosynthetic pathway for firefly luciferin remain unknown (*Oba et al., 2013b*). We hypothesized that bioluminescent accessory enzymes, either specialized enzymes with unique functions in luciferin metabolism or enzymes with primary metabolic functions relevant to bioluminescence, would be highly expressed (HE: 90th percentile; Appendix 4.2.2) in the adult lantern, and would be differentially expressed (DE; Appendix 4.2.2) between luminescent and non-luminescent tissues. To determine this, we performed RNA-Seq and expression analysis of the dissected *P. pyralis* and *A. lateralis* adult male lantern tissue compared with a non-luminescent tissue (Appendix 4.2.2). We identified a set of predicted orthologous enzyme-encoding genes conserved in both *P. pyralis* and *A. lateralis* that met our HE and DE criteria (*Figure 5*). Both luciferase and luciferin sulfotransferase (LST), a specialized enzyme recently implicated in luciferin storage in *P. pyralis* (*Fallon et al., 2016*), were recovered as candidate genes using four criteria (HE, DE, enzymes, direct orthology across species), confirming the validity of our approach. While a direct ortholog of LST is present in *A. lateralis*, it is absent from *I. luminosus*, suggesting that LST, and the presumed luciferin storage it mediates, is an exclusive ancestral firefly or cantharoid trait. This finding is consistent with previous hypotheses of the absence of LST in Elateridae (*Fallon et al., 2016*), and with the overall hypothesis of independent evolution of bioluminescence between the Lampyridae and Elateridae.

Moreover, we identified several additional enzyme-encoding HE and DE lantern genes that are likely important in firefly lantern physiology (*Figure 5*). For instance, adenylate kinase likely plays a critical role in efficient recycling of AMP post-luminescence, and cystathionine gamma-lyase supports a key role of cysteine in luciferin biosynthesis (*Oba et al., 2013b*) and recycling (*Okada et al., 1974*). We also detected a combined adenylyl-sulfate kinase and sulfate adenylyltransferase enzyme (*ASKSA*) among the lantern-enriched gene list (*Appendix 4—figure 8*), implicating active biosynthesis of 3'-phosphoadenosine-5'-phosphosulfate (PAPS), the cofactor of LST, in the lantern. This finding highlights the importance of LST-catalyzed luciferin sulfonation for bioluminescence. These firefly orthologs of *ASKSA* are the only members amongst their paralogs to contain a PTS1 (*Appendix 4—figure 8*), suggesting specialized localization to the peroxisome, the location of the luminescence reaction. This suggests that the levels of sulfoluciferin and luciferin may be actively regulated within the peroxisome of lantern cells in response to luminescence. Overall, our findings of several directly orthologous enzymes that share expression patterns in the light organs of both *P. pyralis* and *A. lateralis* suggests that the enzymatic physiology and/or the gene expression patterns of the photocytes were already fixed in the Luciolinae-Lampyrinae ancestor.

We also performed a similar expression analysis for genes not annotated as enzymes, yielding several genes with predicted lysosomal function (*Appendix 4—table 6*; Appendix 4.4). This suggests that the abundant but as yet unidentified 'differentiated zone granule' organelles of the firefly light organ (*Ghiradella and Schmidt, 2004*) could be lysosomes. Interestingly, we found a HE (TPM value ~300) and DE opsin, *Rh7*, in the light organ of *A. lateralis*, but not *P. pyralis* (*Appendix 4—*

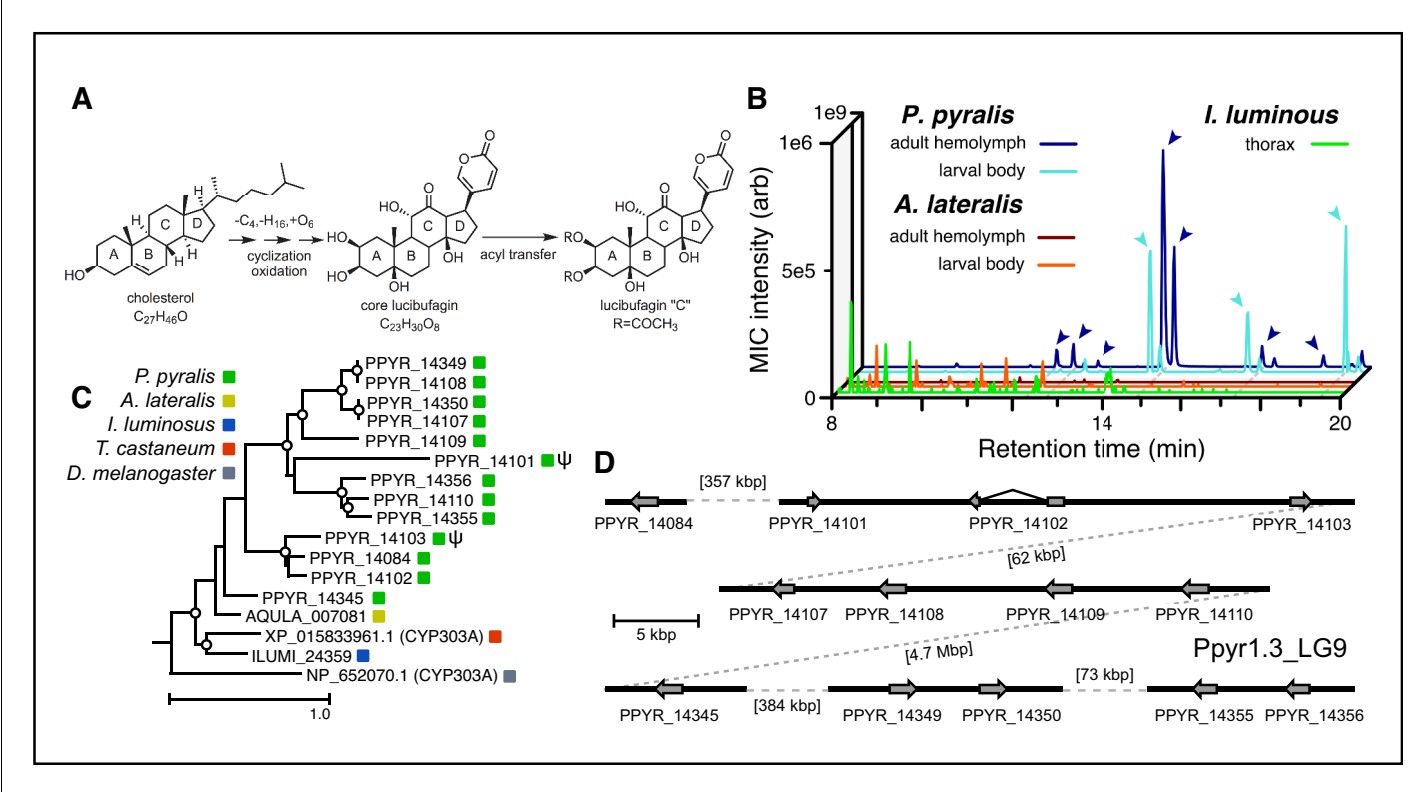

**Figure 6.** An expansion in the CYP303-P450 family correlates with lucibufagin content. (**A**) Hypothesized lucibufagin biosynthetic pathway, starting from cholesterol. (**B**) LC-HRAM-MS multi-ion-chromatograms (MIC) showing the summation of exact mass traces for the [M + H]$^+$ of 11 lucibufagin chemical formulas ± 5 ppm, calibrated for run-specific systematic *m/z* error (***Appendix 4—table 9***). Y-axis upper limit for *P. pyralis* adult hemolymph and larval body extract is 1000x larger than other traces. Arrows (blue/teal) indicate features with high MS$^2$ spectral similarity to known lucibufagins. Sporadic peaks in *A. lateralis* body, and *I. luminosus* thorax traces are not abundant, preventing MS$^2$ spectral acquisition and comparison, but do not match the *m/z* and RT of *P. pyralis* lucibufagins (Appendix 4.6). (**C**) Maximum likelihood tree of CYP303 family cytochrome P450 enzymes from *P. pyralis*, *A. lateralis*, *T. castaneum*, and *D. melanogaster*. *P. pyralis* shows a unique CYP303 family expansion, whereas the other species only have a single CYP303. Circles represent node bootstrap support >60%. Branch length measures substitutions per site. Pseudogenes are annotated with the greek letter Ψ (Appendix 1.10.1; 4.2.4). (**D**) Genomic loci for *P. pyralis* CYP303 family genes. These genes are found in multiple gene clusters on LG9, supporting origin via tandem duplication. Introns >4 kbp are shown.
DOI: https://doi.org/10.7554/eLife.36495.016

The following source data is available for figure 6:

**Source data 1.** CYP303 multiple sequence alignment and gene tree for ***Figure 6C***.
DOI: https://doi.org/10.7554/eLife.36495.017

*figure 9*; Appendix 4.5), suggesting a potential light perception role for *Rh7* in the *A. lateralis* lantern, akin to the light perception role described for *Drosophila Rh7* (***Ni et al., 2017***).

## Genomic insights into firefly chemical defense

Firefly bioluminescence is postulated to have first evolved as an aposematic warning of larval chemical defenses (***Branham and Wenzel, 2003***). Lucibufagins are abundant unpalatable defense steroids described from certain North American firefly species, most notably in the genera *Photinus* (***Meinwald et al., 1979***), *Lucidota* (***Gronquist et al., 2005***), and *Ellychnia* (***Smedley et al., 2017***), and hence are candidates for ancestral firefly defense compounds. To test whether lucibufagins are widespread among bioluminescent beetles, we assessed the presence of lucibufagins in *P. pyralis*, *A. lateralis,* and *I. luminosus* by liquid-chromatography high-resolution accurate-mass mass-spectrometry (LC-HRAM-MS). While lucibufagins were found in high abundance in *P. pyralis* adult hemolymph, they were not observed in *A. lateralis* adult hemolymph, nor in *I. luminosus* metathorax extract (***Figure 6B***; Appendix 4.6). Since chemical defense is presumably most critical in the long-

lived larval stage, we next tested whether lucibufagins are present in all firefly larvae even if they are not present in the adults of certain species. We found lucibufagins in *P. pyralis* larval extracts; however, they were not observed in *A. lateralis* larval extracts (*Figure 6B*; Appendix 4.6). Together, these results suggest that the lucibufagin biosynthetic pathway is either a derived trait only found in particular firefly taxa (e.g. subfamily: Lampyrinae), or that lucibufagin biosynthesis was an ancestral trait that was lost in *A. lateralis*. Consistent with the former hypothesis, the presence of lucibufagins in non-North-American Lampyrinae has been previously reported (*Tyler et al., 2008*), but to date there are no reports of lucibufagins in the Luciolinae.

The lucibufagin biosynthetic pathway is currently unknown. However, their chemical structure suggests a biosynthetic origin from cholesterol followed by a series of hydroxylations, -OH acetylations, and the side-chain oxidative pyrone formation (*Figure 6A*) (*Meinwald et al., 1979*). We hypothesized that cytochrome P450s, an enzyme family widely involved in metabolic diversification of organic substrates (*Hamberger and Bak, 2013*), could underlie several oxidative reactions in the proposed lucibufagin biosynthetic pathway. We therefore inferred the P450 phylogeny among our three bioluminescent beetle genomes to identify any lineage-specific genes correlated with lucibufagin presence. Our analysis revealed a unique expansion of one P450 family, the CYP303 family, in *P. pyralis*. While 94/97 of currently sequenced winged-insect genomes on OrthoDB (*Zdobnov et al., 2017*), as well as the *A. lateralis* and *I. luminosus* genomes, contain only a single *CYP303* family gene, the *P. pyralis* genome contains 11 *CYP303* genes and two pseudogenes (*Figure 6C*), which expanded via tandem duplication on the same linkage group (*Figure 6D*). The CYP303 ortholog of *D. melanogaster*, CYP303A1, has been shown to play a role in mechanosensory bristle development (*Willingham and Keil, 2004*). Although the exact biochemical function and substrate of *D. melanogaster* CYP303A1 is unknown, its closely related P450 families operate on an insect steroid hormone ecdysone (*Willingham and Keil, 2004*). As ecdysone and lucibufagins are structurally similar, CYP303 may operate on steroid-like compounds. Therefore, the lineage-specific expansion of the CYP303 family in *P. pyralis* is a compelling candidate in the metabolic evolution of lucibufagins as chemical defenses associated with the aposematic role of bioluminescence. Alternatively, this CYP303 expansion in *P. pyralis* may be associated with other lineage-specific chemical traits, such as pheromone production.

## Symbionts of bioluminescent beetles

Given the increasingly recognized contributions of symbionts to host metabolism (*Newman and Cragg, 2015*), we characterized the hologenome of all three beetles as potential contributors to metabolic processes related to bioluminescence. Whole genome sequencing of our wild-caught and laboratory reared fireflies revealed a rich microbiome. Amongst our firefly genomes, we found various bacterial genomes, viral genomes, and the complete mtDNA for a phorid parasitoid fly, *Apocephalus antennatus*, the first mtDNA reported for genus *Apocephalus*. This mtDNA was inadvertently included in the *P. pyralis* PacBio library via undetected parasitization of the initial specimens, and was assembled via a metagenomic approach (Appendix 5.2). Independent collection of *A. antennatus* which emerged from field-collected *P. pyralis* adults and targeted COI sequencing later confirmed the taxonomic origin of this mtDNA (Appendix 5.3). We also sequenced and metagenomically assembled the complete circular genome (1.29 Mbp, GC: 29.7%; ~50x coverage) for a *P. pyralis*-associated mollicute (Phylum: Tenericutes), *Entomoplasma luminosum* subsp. pyralis (Appendix 5.1). *Entomoplasma* spp. were first isolated from the guts of North American fireflies (*Hackett et al., 1992*) and our assembly provides the first complete genomic assembly of any *Entomoplasma* species. Broad read coverage for the *E. luminosus* subsp. pyralis genome was detected in 5/6 of our *P. pyralis* DNA libraries, suggesting that *Entomplasma* is a highly prevalent, possibly vertically inherited, *P. pyralis* symbiont. It has been hypothesized that these *Entomoplasma* mollicutes could play a role in firefly metabolism, specifically via contributing to cholesterol metabolism and lucibufagin biosynthesis (*Smedley et al., 2017*).

Within our unfiltered *A. lateralis* genomic assembly (Alat1.2), we also found 43 scaffolds (2.3 Mbp; GC:29.8%, ~64x coverage), whose taxonomic annotation corresponded to the Tenericutes (Appendix 2.5.2), suggesting that *A. lateralis* may also harbor a mollicute symbiont. Alat1.2 also contains 2119 scaffolds (13.0 Mbp, GC:63.7%, ~25x coverage) annotated as of Proteobacterial origin. Limited Proteobacterial symbionts were detected in the *I. luminosus* assembly (0.4 Mbp; GC:30–65% ~10x coverage) (Appendix 3.5.2), suggesting no stable symbiont is present in adult *I. luminosus*.

Lastly, we detected two species of novel orthomyxoviridae-like ssRNA viruses, which we dub *Photinus pyralis* orthomyxo-like virus 1 and 2 (PpyrOMLV1/2), that were highly prevalent across our *P. pyralis* RNA-Seq datasets, and showed multi-generational transovarial transmission in the laboratory (Appendix 5.4). We also found several endogenous viral elements (EVEs) for PpyrOMLV1/2 in *P. pyralis* (Appendix 5.5). These viruses are the first reported in any firefly species, and represent only the second report of transgenerational transfer of any *Orthomyxoviridae* virus (*Marshall et al., 2014*), and the second report of *Orthomyxoviridae* derived EVEs (*Katzourakis and Gifford, 2010*). Together, these genomes from the firefly holobiont provide valuable resources for the continued inquiry of the symbiotic associates of fireflies and their biological and ecological significance.

## Discussion

Here, we generated genome assembles, diverse tissue and life-stage RNA-Seq data, and LC/MS data for three evolutionarily informative and historically well-studied bioluminescent beetles, and used a series of comparative analyses to illuminate long-standing questions on the origins and evolution of beetle bioluminescence. By analyzing the genomic synteny and molecular evolution of the beetle luciferases and their extant and inferred-ancestral homologs, we found strong support for the independent origins of luciferase, and therefore bioluminescence, between fireflies and click beetles. Our approaches and analyses lend molecular evidence to the previous morphology-phylogeny based hypotheses of parallel gain proposed by Darwin and others (*Darwin, 1872*; *Branham and Wenzel, 2003*; *Costa, 1975*; *Sagegami-Oba et al., 2007*; *Bocakova et al., 2007*; *Oba, 2009*; *Day, 2013*). While our elaterid luciferase selection analysis strongly supports an independent gain, we did not perform an analogous selection analysis of luciferase homologs across all bioluminescent beetles, due to the lack of genomic data from key related beetle families. Additional genomic information from early-diverged firefly lineages, other luminous beetle taxa (e.g. Phengodidae and Rhagophthalmidae), and non-luminous elateroid taxa (e.g. Cantharidae and Lycidae), will be useful to further develop and test models of luciferase evolution, including the hypothesis that bioluminescence also originated independently in the Phengodidae and/or Rhagophthalmidae. As some phylogenetic relationships of fireflies and other lineages of superfamily Elateroidea remain uncertain, continued efforts to produce reference phylogeny for these taxa are required (*Martin et al., 2017*; *Bocak et al., 2018*). Toward this goal, the recently published *Pyrocoelia pectoralis* Lampyrinae firefly genome is an important advance which will contribute to future phylogenetic and evolutionary studies (*Fu et al., 2017*).

The independent origins of the firefly and click beetle luciferases provide an exemplary natural model system to understand enzyme evolution through parallel mutational trajectories and the evolution of complex metabolic traits generally. The abundance of gene duplication events of PACSs and ACSs at the ancestral luciferase locus in both fireflies and *I. luminosus* suggests that ancestral promiscuous enzymatic activities served as raw materials for the selection of new adaptive catalytic functions (*Weng, 2014*). But while parallel evolution of luciferase implies evolutionary independence of bioluminescence overall, the reality may be more complex, and the other subtraits of bioluminescence amongst the bioluminescent beetles likely possess different evolutionary histories from luciferase. While subtraits presumably dependent on an efficient luciferase, such as specialized tissues and neural control, almost certainly arose well after luciferase specialization, and thus can be inferred to also have independent origins between fireflies and click beetles, luciferin, which was presumably a prerequisite to luciferase neofunctionalization, may have been present in their common ancestor. Microbial endosymbionts, such as the tenericutes detected in our *P. pyralis* and *A. lateralis* datasets, are intriguing candidate contributors to luciferin metabolism and biosynthesis. Alternatively, recent reports have shown that firefly luciferin is readily produced non-enzymatically by mixing benzoquinone and cysteine (*Kanie et al., 2016*), and that a compound resulting from the spontaneous coupling of benzoquinone and cysteine acts as a luciferin biosynthetic intermediate in *A. lateralis* (*Kanie et al., 2018*). Benzoquinone is known to be a defense compound of distantly related beetles (*Dettner, 1987*) and other arthropods (e.g. millipedes) (*Shear, 2015*). Therefore, the evolutionary role of sporadic low-level luciferin synthesis through spontaneous chemical reactions, either in the ancestral bioluminescent taxa themselves, or in non-bioluminescent taxa, and dietary acquisition of luciferin by either the ancestral or modern bioluminescent taxa, should be considered. To decipher between these alternative evolutionary possibilities, the discovery of genes involved in luciferin

metabolism in fireflies and other bioluminescent beetles will be essential. Here, as a first step toward that goal, we identified conserved, enriched and highly expressed enzymes of the firefly lantern that are strong candidates in luciferin metabolism and the elusive luciferin *de novo* biosynthetic pathway. Ultimately focused experimentation will be needed to decipher the biochemical function of these enzymes.

The early evolution of firefly bioluminescence was likely associated with an aposematic role. The adaptive light production of the primordial firefly (or alternatively, a primordial bioluminescent cantharoid beetle) that enabled the selection and neofunctionalization of luciferase was perhaps linked to a response to predators by a primitive whole-body oxygen-gated luminescence, where a startle-response mediated increase in hemolymph oxygenation through spiracle opening and escape locomotion caused a concomitant increase in luminescence (*Buck and Case, 2002*; *Case, 2004*). Alternatively, an early role for firefly luminescence in mate attraction has not been ruled out (*Buck and Case, 2002*). The presence of particular unpalatable defense compounds in all extant fireflies would be consistent with an ancestral role and the former hypothesis, and the chemical analysis of tissues across species and life stages presented in this work provides new insights into the evolutionary occurrence of lucibufagins, the most well-studied defense compounds associated with fireflies. Our results reject lucibufagins as ancestral defense compounds of fireflies, but rather suggest them as a derived metabolic trait associated with Lampyrinae. Additional chemical analyses across more lineages of fireflies are needed, however, to further support or falsify this hypothesis. Toward this goal, the high sensitivity of our LC-HRAM-MS and MS$^2$ molecular networking-based lucibufagin identification approach is particularly well suited to broadened sampling in the future, including those of rare taxa and possibly museum specimens. Combined with genomic data showing a concomitant expansion of the CYP303 gene family in *P. pyralis*, we present a promising path toward elucidating the biosynthetic mechanism underlying these potent firefly toxins.

Overall, the resources and analyses generated in this study shed valuable light on the evolutionary questions Darwin first pondered, and will enable future studies of the ecology, behavior, and evolution of bioluminescent beetles. These resources will also accelerate the discovery of new enzymes from bioluminescent beetles that could enhance biotechnological applications of bioluminescence. Finally, we hope that the genomic resources shared here will facilitate the development of effective population genomic tools to monitor and protect wild bioluminescent beetle populations in the face of changing climate and habitats.

## Materials and methods

Detailed materials and methods are available in the Appendices. Methods relating to *P. pyralis* are given in Appendix 1, while methods relating to *A. lateralis* and *I. luminosus* are given in Appendix 2 and Appendix 3, respectively. Methods for comparative genomic analyses are given in Appendix 4, while methods for microbiome characterization are given in Appendix 5. References to relevant sections of the Appendices are placed in-line throughout the maintext.

### Data and materials availability

Genomic assemblies (Ppyr1.3, Alat1.3, and Ilumi1.2), associated official geneset data, a Sequence-Server (*Priyam et al., 2015*) BLAST server, and a JBrowse (*Skinner et al., 2009*) genome browser are available at www.fireflybase.org. Raw genomic and RNA-Seq reads for *P. pyralis*, *A. lateralis*, and *I. luminosus*, are available under the NCBI/EBI/DDBJ BioProjects PRJNA378805, PRJDB6460, and PRJNA418169 respectively. Raw WGBS reads can be found on the NCBI Gene Expression Omnibus (GSE107177). Mitochondrial genomes for *P. pyralis* and *I. luminosus* and *A. antennatus* are available on NCBI GenBank with accessions KY778696, MG242621, and MG546669. The complete genome of *Entomoplasma luminosum* subsp. pyralis is available on NCBI GenBank with accession CP027019. The viral genomes for *Photinus pyralis* orthomyxo-like virus 1 and 2 are available on NCBI Genbank with accessions MG972985-MG972994. LC-MS data is available on MetaboLights (Accession MTBLS698). Other supporting datasets are available on FigShare (Appendix 6.1).

## Acknowledgements

We thank Dr. Fu-Shuang Li, Mr. Geoffrey Liou, Dr. Tomáš Pluskal, Ms. Aska Pluskal, Dr. Tina Udhwani, and Dr. Rob Unckless for their assistance with collection of *P. pyralis*. We thank Dr. David Jenkins for collection of *I. luminosus*. Collection permits and permission from the Mercer County Parks Commission, the National Park Service, and the Arlington Virginia Parks and Resources division are gratefully acknowledged. We thank members of the Weng Laboratory, anonymous reviewers on the PubPeer.com platform, and Dr. Stephen Deyrup for commenting on the manuscript. We thank the Whitehead Institute Genome Technology Core, Whitehead Institute high-performance-computing support, MIT BioMicroCenter, University of Rochester Center for Integrated Research Computing, and the Georgia Advanced Computing Resource Center (GACRC) for sequencing and computation resources. We thank Mr. Nick A Rohr for MethylC-Seq library preparation and the Georgia Genomics Facility (GGF) for sequencing. We thank Beijing Genomics Institute for gratis library prep and BGI-SEQ-500 RNA-Seq of *I. luminosus*. We thank Dr. Katsushi Yamaguchi for *A. lateralis* sequencing and DDBJ data submission. We thank Mr. Haruyoshi Ikeya (Toin Gakuen High School, Yokohama, Japan) for providing *A. lateralis* specimens and advice on laboratory culture. We thank, Dr. Kota Ogawa, Ryosuke Nakai, and Takahiro Bino for determining *A. lateralis* genome size. We thank Dr. D Winston Bellot and RN for useful discussions.

## Additional information

### Funding

| Funder | Grant reference number | Author |
|---|---|---|
| Arnold and Mabel Beckman Foundation | | Jing-Ke Weng |
| Kinship Foundation | | Jing-Ke Weng |
| National Science Foundation | Graduate Student Fellowship | Sarah E Lower |
| Pew Charitable Trusts | Pew Scholar Program in the Biomedical Sciences | Jing-Ke Weng |
| National Institute for Basic Biology | Cooperative Research Program, 12-202 | Shuji Shigenobu |
| National Institute of General Medical Sciences | 5R35-GM119515-03 | Amanda M Larracuente |
| Georgia Research Alliance | Lars G. Ljungdahl Distinguished Investigator | Robert J Schmitz |
| Experiment.com | https://experiment.com/projects/illuminating-the-firefly-genome | Timothy R Fallon Sarah E Lower Gavin J Martin Megan Behringer Sara M Lewis Seth M Bybee Amanda M Larracuente Jing-Ke Weng |
| National Science Foundation | MCB-1818132 | Jing-Ke Weng |

The funders had no role in study design, data collection and interpretation, or the decision to submit the work for publication.

### Author contributions

Timothy R Fallon, Conceptualization, Data curation, Formal analysis, Investigation, Methodology, Writing—original draft, Writing—review and editing, Performed *P. pyralis* PacBio and Hi-C sequencing, performed *I. luminosus*, mitochondrial, and non-viral symbiont genome assemblies; Sarah E Lower, Conceptualization, Data curation, Formal analysis, Investigation, Methodology, Writing—original draft, Writing—review and editing, Performed *P. pyralis* Illumina sequencing; Ching-Ho Chang, Resources, Data curation, Formal analysis, Investigation, Performed *P. pyralis* genome

assembly; Manabu Bessho-Uehara, Conceptualization, Data curation, Formal analysis, Investigation, Methodology, Writing—review and editing, Performed A. lateralis RNA-Seq, luciferase phylogenetic analysis, and Rh7 phylogenetic analysis; Gavin J Martin, Formal analysis, Methodology, Writing—review and editing; Adam J Bewick, Data curation, Formal analysis, Investigation, Methodology, Performed methylation analysis; Megan Behringer, Data curation, Formal analysis, Investigation, Methodology, Performed bacterial symbiont annotation and analysis; Humberto J Debat, Data curation, Formal analysis, Investigation, Methodology, Writing—review and editing, Performed viral genome assembly and analysis; Isaac Wong, Formal analysis, Methodology, Performed in situ hybridizations; John C Day, Data curation, Formal analysis, Investigation; Anton Suvorov, Data curation, Formal analysis, Investigation, Methodology; Christian J Silva, Data curation, Formal analysis, Investigation, Performed repeat analysis; Kathrin F Stanger-Hall, Supervision, Writing—review and editing; David W Hall, Supervision; Robert J Schmitz, Supervision, Investigation; David R Nelson, Formal analysis, Performed manual annotation of P450s; Sara M Lewis, Conceptualization, Writing—review and editing; Shuji Shigenobu, Conceptualization, Data curation, Formal analysis, Investigation, Methodology, Performed A. lateralis genome assembly; Seth M Bybee, Conceptualization, Supervision, Writing—review and editing; Amanda M Larracuente, Conceptualization, Data curation, Formal analysis, Supervision, Investigation, Methodology, Writing—review and editing, Performed repeat analysis; Yuichi Oba, Conceptualization, Formal analysis, Supervision, Writing—review and editing; Jing-Ke Weng, Conceptualization, Formal analysis, Supervision, Funding acquisition, Investigation, Writing—original draft, Writing—review and editing

### Author ORCIDs
Timothy R Fallon (iD) http://orcid.org/0000-0002-3048-7679
Sarah E Lower (iD) https://orcid.org/0000-0003-0889-3344
Ching-Ho Chang (iD) https://orcid.org/0000-0001-9361-1190
Manabu Bessho-Uehara (iD) https://orcid.org/0000-0002-7388-464X
Gavin J Martin (iD) http://orcid.org/0000-0002-8989-2601
Humberto J Debat (iD) https://orcid.org/0000-0003-3056-3739
Isaac Wong (iD) https://orcid.org/0000-0003-4877-5748
John C Day (iD) http://orcid.org/0000-0002-5483-4487
Anton Suvorov (iD) http://orcid.org/0000-0003-3898-9195
Christian J Silva (iD) https://orcid.org/0000-0002-9803-5249
David W Hall (iD) http://orcid.org/0000-0001-7708-6656
Robert J Schmitz (iD) http://orcid.org/0000-0001-7538-6663
David R Nelson (iD) https://orcid.org/0000-0003-0583-5421
Shuji Shigenobu (iD) https://orcid.org/0000-0003-4640-2323
Amanda M Larracuente (iD) https://orcid.org/0000-0001-5944-5686
Yuichi Oba (iD) http://orcid.org/0000-0003-2108-4947
Jing-Ke Weng (iD) http://orcid.org/0000-0003-3059-0075

### Decision letter and Author response
Decision letter https://doi.org/10.7554/eLife.36495.175
Author response https://doi.org/10.7554/eLife.36495.176

## Additional files

### Supplementary files
• Transparent reporting form
DOI: https://doi.org/10.7554/eLife.36495.018

### Data availability
Genomic assemblies (Ppyr1.3, Alat1.3, and Ilumi1.2), associated official geneset data, a Sequence-Server BLAST server, and a JBrowse genome browser are available at www.fireflybase.org (see Appendix 6.2). Raw genomic and RNA-Seq reads for P. pyralis, A. lateralis, and I. luminosus, are available under the NCBI/EBI/DDBJ BioProjects PRJNA378805, PRJDB6460, and PRJNA418169

respectively. Raw WGBS reads can be found on the NCBI Gene Expression Omnibus (GSE107177). Mitochondrial genomes for P. pyralis and I. luminosus and A. antennatus are available on NCBI GenBank with accessions KY778696, MG242621, and MG546669. The complete genome of Entomoplasma luminosum subsp. pyralis is available on NCBI GenBank with accession CP027019. The viral genomes for Photinus pyralis orthomyxo-like virus 1 & 2 are available on NCBI Genbank with accessions MG972985-MG972994. LC-MS data is available on MetaboLights (Accession MTBLS698). Other supporting datasets are available on FigShare (Appendix 6).

The following datasets were generated:

| Author(s) | Year | Dataset title | Dataset URL | Database, license, and accessibility information |
|---|---|---|---|---|
| Timothy R Fallon, Sarah E Lower, Ching-Ho Chang, Manabu Bessho-Uehara, Gavin J Martin, Adam J Bewick, Megan Behringer, Humberto J Debat, Isaac Wong, John C Day, Anton Suvorov, Christian J Silva, Kathrin F Stanger-Hall, David W Hall, Robert J Schmitz, David R Nelson, Sara M Lewis, Shuji Shigenobu, Seth M Bybee, Amanda M Larracuente, Yuichi Oba, Jing-Ke Weng | 2018 | Photinus pyralis orthomyxo-like virus 2 HA gene, complete cds | https://www.ncbi.nlm.nih.gov/nuccore/MG972990.1 | Publicly available at the NCBI/EBI/DDBJ Genbank database (accession no: MG972990.1) |
| Timothy R Fallon, Sarah E Lower, Megan Behringer, Jing-Ke Weng | 2018 | Entomoplasma luminosum isolate NJ-2016 chromosome, complete genome | https://www.ncbi.nlm.nih.gov/nuccore/CP027019 | Publicly available at the NCBI/EBI/DDBJ Genbank database (accession no: CP027019) |
| Timothy R Fallon, Sarah E Lower, Ching-Ho Chang, Manabu Bessho-Uehara, Gavin J Martin, Adam J Bewick, Megan Behringer, Humberto J Debat, Isaac Wong, John C Day, Anton Suvorov, Christian J Silva, Kathrin F Stanger-Hall, David W Hall, Robert J Schmitz, David R Nelson, Sara M Lewis, Shuji Shigenobu, Seth M Bybee, Amanda M Larracuente, Yuichi Oba, Jing-Ke Weng | 2018 | Photinus pyralis orthomyxo-like virus 1 HA gene, complete cds | https://www.ncbi.nlm.nih.gov/nuccore/MG972985.1 | Publicly available at the NCBI/EBI/DDBJ Genbank database (accession no: MG972985.1) |
| Timothy R Fallon, Sarah E Lower, Ching-Ho Chang, Manabu Bessho-Uehara, Gavin J Martin, Adam J Bewick, Megan Behringer, Humberto J Debat, Isaac Wong, John C Day, Anton | 2018 | Photinus pyralis orthomyxo-like virus 2 NP gene, complete cds | https://www.ncbi.nlm.nih.gov/nuccore/MG972991.1 | Publicly available at the NCBI/EBI/DDBJ Genbank database (accession no: MG972991.1) |

| | | | | |
|---|---|---|---|---|
| Suvorov, Christian J Silva, Kathrin F Stanger-Hall, David W Hall, Robert J Schmitz, David R Nelson, Sara M Lewis, Shuji Shigenobu, Seth M Bybee, Amanda M Larracuente, Yuichi Oba, Jing-Ke Weng | | | | |
| Timothy R Fallon, Sarah E Lower, Ching-Ho Chang, Manabu Bessho-Uehara, Gavin J Martin, Adam J Bewick, Megan Behringer, Humberto J Debat, Isaac Wong, John C Day, Anton Suvorov, Christian J Silva, Kathrin F Stanger-Hall, David W Hall, Robert J Schmitz, David R Nelson, Sara M Lewis, Shuji Shigenobu, Seth M Bybee, Amanda M Larracuente, Yuichi Oba, Jing-Ke Weng | 2018 | Photinus pyralis orthomyxo-like virus 2 PA gene, complete cds | https://www.ncbi.nlm.nih.gov/nuccore/MG972992.1 | Publicly available at the NCBI/EBI/DDBJ Genbank database (accession no: MG972992.1) |
| Fallon TR, Lower SE, Chang C-H, Bessho-Uehara M, Martin GJ, Bewick AJ, Behringer M, Debat HJ, Wong I, Day JC, Suvorov A, Silva CJ, Stanger-Hall KF, Hall DW, Schmitz RJ, Nelson DR, Lewis SM, Shigenobu S, Bybee SM, Larracuente AM, Oba Y, Weng J-K | 2018 | Sequencing and de novo assembly of the Photinus pyralis genome | https://www.ncbi.nlm.nih.gov/bioproject/?term=PRJNA378805 | Publicly available at the NCBI/EBI/DDBJ BioProjects database (accession no: PRJNA378805) |
| Fallon TR, Lower SE, Chang C-H, Bessho-Uehara M, Martin GJ, Bewick AJ, Behringer M, Debat HJ, Wong I, Day JC, Suvorov A, Silva CJ, Stanger-Hall KF, Hall DW, Schmitz RJ, Nelson DR, Lewis SM, Shigenobu S, Bybee SM, Larracuente AM, Oba Y, Weng J-K | 2018 | Japanese Firefly Aquatica lateralis Genome Project | https://www.ncbi.nlm.nih.gov/bioproject/?term=PRJDB6460 | Publicly available at the NCBI/EBI/DDBJ BioProjects database (accession no: PRJDB6460) |
| Fallon TR, Lower SE, Chang C-H, Bessho-Uehara M, Martin GJ, Bewick AJ, Behringer M, Debat HJ, Wong I, Day JC, Suvorov A, Silva CJ, Stanger-Hall KF, Hall DW, Schmitz RJ, Nelson | 2018 | Ignelater luminosus Genome sequencing and assembly | https://www.ncbi.nlm.nih.gov/bioproject/?term=PRJNA418169 | Publicly available at the NCBI/EBI/DDBJ BioProjects database (accession no: PRJNA418169) |

| | | | | |
|---|---|---|---|---|
| DR, Lewis SM, Shigenobu S, Bybee SM, Larracuente AM, Oba Y, Weng J-K | | | | |
| Fallon TR, Lower SE, Chang C-H, Bessho-Uehara M, Martin GJ, Bewick AJ, Behringer M, Debat HJ, Wong I, Day JC, Suvorov A, Silva CJ, Stanger-Hall KF, Hall DW, Schmitz RJ, Nelson DR, Lewis SM, Shigenobu S, Bybee SM, Larracuente AM, Oba Y, Weng J-K | 2018 | Whole-genome bisulfite sequencing methylation analysis of P. pyralis | https://www.ncbi.nlm.nih.gov/geo/query/acc.cgi?acc=GSE107177 | Publicly available at the NCBI Gene Expression Omnibus database (accession no: GSE107177) |
| Fallon TR, Lower SE, Chang C-H, Bessho-Uehara M, Martin GJ, Bewick AJ, Behringer M, Debat HJ, Wong I, Day JC, Suvorov A, Silva CJ, Stanger-Hall KF, Hall DW, Schmitz RJ, Nelson DR, Lewis SM, Shigenobu S, Bybee SM, Larracuente AM, Oba Y, Weng J-K | 2018 | Photinus pyralis mitochondrion, complete genome | https://www.ncbi.nlm.nih.gov/nuccore/KY778696 | Publicly available at the NCBI/EBI/DDBJ Genbank database (accession no: KY778696.1) |
| Fallon TR, Lower SE, Chang C-H, Bessho-Uehara M, Martin GJ, Bewick AJ, Behringer M, Debat HJ, Wong I, Day JC, Suvorov A, Silva CJ, Stanger-Hall KF, Hall DW, Schmitz RJ, Nelson DR, Lewis SM, Shigenobu S, Bybee SM, Larracuente AM, Oba Y, Weng J-K | 2018 | Ignelater luminosus mitochondrion, complete genome | https://www.ncbi.nlm.nih.gov/nuccore/MG242621 | Publicly available at the NCBI/EBI/DDBJ Genbank database (accession no: MG242621.1) |
| Fallon TR, Lower SE, Chang C-H, Bessho-Uehara M, Martin GJ, Bewick AJ, Behringer M, Debat HJ, Wong I, Day JC, Suvorov A, Silva CJ, Stanger-Hall KF, Hall DW, Schmitz RJ, Nelson DR, Lewis SM, Shigenobu S, Bybee SM, Larracuente AM, Oba Y, Weng J-K | 2018 | Apocephalus antennatus mitochondrion, complete genome | https://www.ncbi.nlm.nih.gov/nuccore/MG546669.1 | Publicly available at the NCBI/EBI/DDBJ Genbank database (accession no: MG546669.1) |
| Fallon TR, Lower SE, Chang C-H, Bessho-Uehara M, Martin GJ, Bewick AJ, Behringer M, Debat HJ, Wong I, | 2018 | Photinus pyralis orthomyxo-like virus 1 PB2 gene, complete cds | https://www.ncbi.nlm.nih.gov/nuccore/MG972989.1 | Publicly available at the NCBI/EBI/DDBJ Genbank database (accession no:-MG972989.1) |

| | | | | |
|---|---|---|---|---|
| Day JC, Suvorov A, Silva CJ, Stanger-Hall KF, Hall DW, Schmitz RJ, Nelson DR, Lewis SM, Shigenobu S, Bybee SM, Larracuente AM, Oba Y, Weng J-K | | | | |
| Fallon TR, Lower SE, Chang C-H, Bessho-Uehara M, Martin GJ, Bewick AJ, Behringer M, Debat HJ, Wong I, Day JC, Suvorov A, Silva CJ, Stanger-Hall KF, Hall DW, Schmitz RJ, Nelson DR, Lewis SM, Shigenobu S, Bybee SM, Larracuente AM, Oba Y, Weng J-K | 2018 | Photinus pyralis orthomyxo-like virus 2 PB2 gene, complete cds | https://www.ncbi.nlm.nih.gov/nuccore/MG972994.1 | Publicly available at the NCBI/EBI/DDBJ Genbank database (accession no: MG972994.1) |
| Fallon TR, Lower SE, Chang C-H, Bessho-Uehara M, Martin GJ, Bewick AJ, Behringer M, Debat HJ, Wong I, Day JC, Suvorov A, Silva CJ, Stanger-Hall KF, Hall DW, Schmitz RJ, Nelson DR, Lewis SM, Shigenobu S, Bybee SM, Larracuente AM, Oba Y, Weng J-K | 2018 | LC-HRAM-MS of bioluminescent beetles | https://www.ebi.ac.uk/metabolights/MTBLS698 | Publicly available at the EBI Metabolights database (accession no: MTBLS698) |
| Fallon TR, Lower SE, Chang C-H, Bessho-Uehara M, Martin GJ, Bewick AJ, Behringer M, Debat HJ, Wong I, Day JC, Suvorov A, Silva CJ, Stanger-Hall KF, Hall DW, Schmitz RJ, Nelson DR, Lewis SM, Shigenobu S, Bybee SM, Larracuente AM, Oba Y, Weng J-K | 2018 | Photinus pyralis sighting records (Excel spreadsheet) | https://figshare.com/articles/_/5688826 | Publicly available at FigShare (DOI: 10.6084/m9.figshare.5688826) |
| Fallon TR, Lower SE, Chang C-H, Bessho-Uehara M, Martin GJ, Bewick AJ, Behringer M, Debat HJ, Wong I, Day JC, Suvorov A, Silva CJ, Stanger-Hall KF, Hall DW, Schmitz RJ, Nelson DR, Lewis SM, Shigenobu S, Bybee SM, Larracuente AM, Oba Y, Weng J-K | 2018 | Ilumi1.0 Blobtools results | https://figshare.com/articles/_/5688952 | Publicly available at FigShare (DOI: 10.6084/m9.figshare.5688952) |
| Fallon TR, Lower SE, Chang C-H, | 2018 | Alat1.2 Blobtools results | https://figshare.com/articles/_/5688928 | Publicly available at FigShare (DOI: 10.60 |

| | | | | |
|---|---|---|---|---|
| Bessho-Uehara M, Martin GJ, Bewick AJ, Behringer M, Debat HJ, Wong I, Day JC, Suvorov A, Silva CJ, Stanger-Hall KF, Hall DW, Schmitz RJ, Nelson DR, Lewis SM, Shigenobu S, Bybee SM, Larracuente AM, Oba Y, Weng J-K | | | | 84/m9.figshare. 5688928) |
| Fallon TR, Lower SE, Chang C-H, Bessho-Uehara M, Martin GJ, Bewick AJ, Behringer M, Debat HJ, Wong I, Day JC, Suvorov A, Silva CJ, Stanger-Hall KF, Hall DW, Schmitz RJ, Nelson DR, Lewis SM, Shigenobu S, Bybee SM, Larracuente AM, Oba Y, Weng J-K | 2018 | Ppyr1.2 Blobtools results | https://figshare.com/articles/_/5688982 | Publicly available at FigShare (DOI: 10.6084/m9.figshare. 5688982) |
| Fallon TR, Lower SE, Chang C-H, Bessho-Uehara M, Martin GJ, Bewick AJ, Behringer M, Debat HJ, Wong I, Day JC, Suvorov A, Silva CJ, Stanger-Hall KF, Hall DW, Schmitz RJ, Nelson DR, Lewis SM, Shigenobu S, Bybee SM, Larracuente AM, Oba Y, Weng J-K | 2018 | Protein multiple sequence alignment for P450 tree - Appendix 1-figure 13 | https://figshare.com/articles/_/5697643 | Publicly available at FigShare (DOI: 10.6084/m9.figshare. 5697643) |
| Fallon TR, Lower SE, Chang C-H, Bessho-Uehara M, Martin GJ, Bewick AJ, Behringer M, Debat HJ, Wong I, Day JC, Suvorov A, Silva CJ, Stanger-Hall KF, Hall DW, Schmitz RJ, Nelson DR, Lewis SM, Shigenobu S, Bybee SM, Larracuente AM, Oba Y, Weng J-K | 2018 | Photinus pyralis orthomyxo-like virus 1 sequence and annotation | https://figshare.com/articles/_/5714806 | Publicly available at FigShare (DOI: 10.6084/m9.figshare. 5714806) |
| Fallon TR, Lower SE, Chang C-H, Bessho-Uehara M, Martin GJ, Bewick AJ, Behringer M, Debat HJ, Wong I, Day JC, Suvorov A, Silva CJ, Stanger-Hall KF, Hall DW, Schmitz RJ, Nelson DR, Lewis SM, Shigenobu S, Bybee SM, Larracuente AM, Oba Y, Weng | 2018 | Photinus pyralis orthomyxo-like virus 2 sequence and annotation | https://figshare.com/articles/_/5714812 | Publicly available at FigShare (DOI: 10.6084/m9.figshare. 5714812) |

J-K

| | | | | |
|---|---|---|---|---|
| Fallon TR, Lower SE, Chang C-H, Bessho-Uehara M, Martin GJ, Bewick AJ, Behringer M, Debat HJ, Wong I, Day JC, Suvorov A, Silva CJ, Stanger-Hall KF, Hall DW, Schmitz RJ, Nelson DR, Lewis SM, Shigenobu S, Bybee SM, Larracuente AM, Oba Y, Weng J-K | 2018 | OrthoFinder protein clustering analysis (Orthogroups) | https://figshare.com/articles/_/5715136 | Publicly available at FigShare (DOI: 10.6084/m9.figshare.5715136) |
| Fallon TR, Lower SE, Chang C-H, Bessho-Uehara M, Martin GJ, Bewick AJ, Behringer M, Debat HJ, Wong I, Day JC, Suvorov A, Silva CJ, Stanger-Hall KF, Hall DW, Schmitz RJ, Nelson DR, Lewis SM, Shigenobu S, Bybee SM, Larracuente AM, Oba Y, Weng J-K | 2018 | PPYR_OGS1.1 kallisto RNA-Seq expression quantification (TPM) | https://figshare.com/articles/_/5715139 | Publicly available at FigShare (DOI: 10.6084/m9.figshare.5715139) |
| Fallon TR, Lower SE, Chang C-H, Bessho-Uehara M, Martin GJ, Bewick AJ, Behringer M, Debat HJ, Wong I, Day JC, Suvorov A, Silva CJ, Stanger-Hall KF, Hall DW, Schmitz RJ, Nelson DR, Lewis SM, Shigenobu S, Bybee SM, Larracuente AM, Oba Y, Weng J-K | 2018 | AQULA_OGS1.0 kallisto RNA-Seq expression quantification (TPM) | https://figshare.com/articles/_/5715142 | Publicly available at FigShare (DOI: 10.6084/m9.figshare.5715142) |
| Fallon TR, Lower SE, Chang C-H, Bessho-Uehara M, Martin GJ, Bewick AJ, Behringer M, Debat HJ, Wong I, Day JC, Suvorov A, Silva CJ, Stanger-Hall KF, Hall DW, Schmitz RJ, Nelson DR, Lewis SM, Shigenobu S, Bybee SM, Larracuente AM, Oba Y, Weng J-K | 2018 | Figure 5. PPYR_OGS1.1 + AQULA_OGS1.0 Sleuth / differential expression Venn diagram analysis (BSN-TPM) | https://figshare.com/articles/_/5715151 | Publicly available at FigShare (DOI: 10.6084/m9.figshare.5715151) |
| Fallon TR, Lower SE, Chang C-H, Bessho-Uehara M, Martin GJ, Bewick AJ, Behringer M, Debat HJ, Wong I, Day JC, Suvorov A, Silva CJ, Stanger-Hall KF, Hall DW, Schmitz RJ, Nelson | 2018 | Ilumi_OGS1.2 kallisto RNA-Seq expression quantification (TPM) | https://figshare.com/articles/_/5715157 | Publicly available at FigShare (DOI: 10.6084/m9.figshare.10.6084/m9.figshare.5715157) |

| | | | | |
|---|---|---|---|---|
| DR, Lewis SM, Shigenobu S, Bybee SM, Larracuente AM, Oba Y, Weng J-K | | | | |
| Fallon TR, Lower SE, Chang C-H, Bessho-Uehara M, Martin GJ, Bewick AJ, Behringer M, Debat HJ, Wong I, Day JC, Suvorov A, Silva CJ, Stanger-Hall KF, Hall DW, Schmitz RJ, Nelson DR, Lewis SM, Shigenobu S, Bybee SM, Larracuente AM, Oba Y, Weng J-K | 2018 | Appendix 4-figure 2: DNA and tRNA methyltransferase gene phylogeny | https://figshare.com/articles/_/6531311 | Publicly available at FigShare (DOI: 10.6084/m9.figshare.6531311) |
| Fallon TR, Lower SE, Chang C-H, Bessho-Uehara M, Martin GJ, Bewick AJ, Behringer M, Debat HJ, Wong I, Day JC, Suvorov A, Silva CJ, Stanger-Hall KF, Hall DW, Schmitz RJ, Nelson DR, Lewis SM, Shigenobu S, Bybee SM, Larracuente AM, Oba Y, Weng J-K | 2018 | Appendix 4-figure 6 Preliminary maximum likelihood phylogeny of luciferase homologs | https://figshare.com/articles/_/6687086 | Publicly available at FigShare (DOI: 10.6084/m9.figshare.6687086) |
| Fallon TR, Lower SE, Chang C-H, Bessho-Uehara M, Martin GJ, Bewick AJ, Behringer M, Debat HJ, Wong I, Day JC, Suvorov A, Silva CJ, Stanger-Hall KF, Hall DW, Schmitz RJ, Nelson DR, Lewis SM, Shigenobu S, Bybee SM, Larracuente AM, Oba Y, Weng J-K | 2018 | Appendix 4-figure 9A Opsin gene tree | https://figshare.com/articles/_/5723005 | Publicly available at FigShare (DOI: 10.6084/m9.figshare.5723005) |
| Fallon TR, Lower SE, Chang C-H, Bessho-Uehara M, Martin GJ, Bewick AJ, Behringer M, Debat HJ, Wong I, Day JC, Suvorov A, Silva CJ, Stanger-Hall KF, Hall DW, Schmitz RJ, Nelson DR, Lewis SM, Shigenobu S, Bybee SM, Larracuente AM, Oba Y, Weng J-K | 2018 | Appendix 4.3.4: MEME selected site analysis | https://figshare.com/articles/_/6626651 | Publicly available at FigShare (DOI: 10.6084/m9.figshare.6626651) |
| Fallon TR, Lower SE, Chang C-H, Bessho-Uehara M, Martin GJ, Bewick AJ, Behringer M, Debat HJ, Wong I, | 2018 | Appendix 4.3.4: PAML-BEB selected site analysis | https://figshare.com/articles/_/6725081 | Publicly available at FigShare (DOI: 10.6084/m9.figshare.6725081) |

| | | | | |
|---|---|---|---|---|
| Day JC, Suvorov A, Silva CJ, Stanger-Hall KF, Hall DW, Schmitz RJ, Nelson DR, Lewis SM, Shigenobu S, Bybee SM, Larracuente AM, Oba Y, Weng J-K | | | | |
| Timothy R Fallon, Sarah E Lower, Ching-Ho Chang, Manabu Bessho-Uehara, Gavin J Martin, Humberto J Debat, Adam J Bewick, Megan Behringer, Isaac Wong, John C Day, Anton Suvorov, Christian J Silva, Kathrin F Stanger-Hall, David W Hall, Robert J Schmitz, David R Nelson, Sara M Lewis, Shuji Shigenobu, Seth M Bybee, Amanda M Larracuente, Yuichi Oba, Jing-Ke Weng | 2018 | Photinus pyralis orthomyxo-like virus 2 PB1 gene, complete cds | https://www.ncbi.nlm.nih.gov/nuccore/MG972993.1 | Publicly available at the NCBI/EBI/DDBJ Genbank database (accession no: MG972993.1) |
| Timothy R Fallon, Sarah E Lower, Ching-Ho Chang, Manabu Bessho-Uehara, Gavin J Martin, Adam J Bewick, Megan Behringer, Humberto J Debat, Isaac Wong, John C Day, Anton Suvorov, Christian J Silva, Kathrin F Stanger-Hall, David W Hall, Robert J Schmitz, David R Nelson, Sara M Lewis, Shuji Shigenobu, Seth M Bybee, Amanda M Larracuente, Yuichi Oba, Jing-Ke Weng | 2018 | Photinus pyralis orthomyxo-like virus 1 NP gene, complete cds | https://www.ncbi.nlm.nih.gov/nuccore/MG972986 | Publicly available at the NCBI/EBI/DDBJ Genbank database (accession no:-MG972986) |
| Timothy R Fallon, Sarah E Lower, Ching-Ho Chang, Manabu Bessho-Uehara, Gavin J Martin, Adam J Bewick, Megan Behringer, Humberto J Debat, Isaac Wong, John C Day, Anton Suvorov, Christian J Silva, Kathrin F Stanger-Hall, David W Hall, Robert J Schmitz, David R Nelson, Sara M Lewis, Seth M Bybee, Amanda M Larracuente, Yuichi Oba, Jing-Ke Weng | 2018 | Photinus pyralis orthomyxo-like virus 1 PA gene, complete cds | https://www.ncbi.nlm.nih.gov/nuccore/MG972987.1 | Publicly available at the NCBI/EBI/DDBJ Genbank database (accession no:-MG972987.1) |
| Timothy R Fallon, | 2018 | Photinus pyralis orthomyxo-like | https://www.ncbi.nlm. | Publicly available at |

| Sarah E Lower, Ching-Ho Chang, Manabu Bessho-Uehara, Gavin J Martin, Megan Behringer, Humberto J Debat, Isaac Wong, John C Day, Anton Suvorov, Christian J Silva, Kathrin F Stanger-Hall, David W Hall, Robert J Schmitz, David R Nelson, Sara M Lewis, Shuji Shigenobu, Seth M Bybee, Amanda M Larracuente, Yuichi Oba, Jing-Ke Weng | virus 1 PB1 gene, complete cds | nih.gov/nuccore/MG972988.1 | the NCBI/EBI/DDBJ Genbank database (accession no:- MG972988.1) |

The following previously published datasets were used:

| Author(s) | Year | Dataset title | Dataset URL | Database, license, and accessibility information |
|-----------|------|---------------|-------------|--------------------------------------------------|
| Timothy R Fallon, Fu-Shuang Li, Maria A. Vicent, Jing-Ke Weng | 2016 | Photinus pyralis posterior abdomen de novo transcriptome assembly | https://www.ncbi.nlm.nih.gov/bioproject/?term=PRJNA321737 | Publicly available at NCBI BioProject (accession no. PRJNA321737) |
| Sarah E Lower, David W Hall | 2015 | Transcriptomes of 10 North American firefly species | https://www.ncbi.nlm.nih.gov/bioproject/?term=PRJNA289908 | Publicly available at NCBI BioProject (accession no. PRJNA289908) |
| Nooria Al-Wathiqui, Timothy R. Fallon, Adam South, Jing-Ke Weng | 2016 | Molecular composition of the male nuptial gift in the firefly Photinus pyralis | https://www.ncbi.nlm.nih.gov/bioproject/?term=PRJNA328865 | Publicly available at NCBI BioProject (accession no. PRJNA328865) |

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

## Appendix 1

DOI: https://doi.org/10.7554/eLife.36495.019

### *Photinus pyralis* additional information

#### 1.1 Taxonomy, biology, and life history

*Photinus pyralis* (Linnaeus, 1767) is amongst the most widespread and abundant of all U.S. fireflies (*Lloyd, 1966*; *Lloyd, 2008*). It inspired extensive work on the biochemistry and physiology of firefly bioluminescence in the early 20th century, and the first luciferase gene was cloned from this species (*de Wet et al., 1985*). A habitat generalist, *P. pyralis* occurs in fields, meadows, suburban lawns, forests, and woodland edges, and even urban environments. For example, the authors have observed *P. pyralis* flashing in urban New York City and Washington D.C. Adults rest on vegetation during the day and signaling begins as early as 20 min before sunset (*Lloyd, 1966*). Male flashing is cued by ambient light levels, thus shaded or unshaded habitats can show up to a 30 min difference in the initiation of male flashing (*Lloyd, 1966*). Males can be cued to flash outside of true twilight if exposed to light intensities simulating twilight (*Case, 2004*). *P. pyralis* were also reported to flash during totality of the total solar eclipse of 2017 (Personal communication: L.F. Faust, M.A. Branham). Courtship activity lasts for 30–45 min and both sexes participate in a bioluminescent flash dialog, as is typical for *Photinus* fireflies.

Males initiate courtship by flying low above the ground while repeating a single ~300 ms patrol flash at ~5–10 s intervals (*Case, 2004*). Males emit their patrol flash while dipping down and then ascending vertically, creating a distinctive J-shaped flash gesture (*Lloyd, 1966*; *Case, 2004*) (*Figure 1A*). During courtship, females perch on vegetation and respond to a male patrol flash by twisting their abdomen toward the source of the flash and giving a single response flash given after a 2–3 s delay (*Video 1*). Receptive females will readily respond to simulated male flashes, such as those produced by an investigator's penlight. Females have fully developed wings and are capable of flight. Both sexes are capable of mating several times during their adult lives. During mating, males transfer to females a fitness-enhancing nuptial gift consisting of a spermatophore manufactured by multiple accessory glands (*van der Reijden et al., 1997*); the molecular composition of this nuptial gift has recently been elucidated for *P. pyralis* (*Al-Wathiqui et al., 2016*). In other *Photinus* species, male gift size decreases across sequential matings (*Cratsley et al., 2003*), and multiple matings are associated with increased female fecundity (*Rooney and Lewis, 2002*).

Adult *P. pyralis* live 2–3 weeks, and although these adults are typically considered non-feeding, both sexes have been reported drinking nectar from the flowers of the milkweed *Asclepias syriaca* (*Faust and Faust, 2014*). Mated females store sperm and lay ~30–50 eggs over the course of a few days on moss or in moist soil. The eggs take 2–3 weeks to hatch. Larval bioluminescence is thought to be universal for the Lampyridae, where it appears to function as an aposematic warning signal. Like other *Photinus*, *P. pyralis* larvae are predatory, live on and beneath the soil, and appear to be earthworm specialists (*Hess, 1920*). In the northern parts of its range, slower development likely requires *P. pyralis* to overwinter at least twice, most likely as larvae. Farther south, *P. pyralis* may complete development within several months, achieving two generations per year (*Faust, 2017*), which may be possibly be observed in the South as a 'second wave' of signalling *P. pyralis* in September-October.

Anti-predator chemical defenses of male *P. pyralis* include several bufadienolides, known as lucibufagins, that circulate in the hemolymph (*Meinwald et al., 1979*). Pterins have also been reported to be abundant in *P. pyralis* (*Goetz et al., 1981*); however, the potential defense role of these compounds has never been tested (Personal communication: J. Meinwald). When attacked, *P. pyralis* males release copious amounts of rapidly coagulating hemolymph and such 'reflex-bleeding' may also provide physical protection against small predators (*Blum and Sannasi, 1974*; *Faust et al., 2012*).

## 1.2 Species distribution

Although *Photinus pyralis* is widely distributed in the Eastern United States, published descriptions of its range are limited, with the notable exception of Lloyd's 1966 monograph (*Lloyd, 1966*) which addresses the range of many *Photinus* species. We therefore sought to characterize the current distribution of *P. pyralis* in order to produce an updated map to inform our experimental design and enable future population genetic studies. Four sources of data were used to produce the presented range map of *P. pyralis*: (i) Field surveys by the authors (ii) Published (*Lloyd, 1966*; *Luk et al., 2011*) and unpublished sightings of *P. pyralis* at county level resolution, provided by Dr. J. Lloyd (University of Florida), (iii) coordinates and dates of *P. pyralis* sightings, obtained by targeted e-mail surveys to firefly field biologists, (iv) citizen scientist reports of *P. pyralis* through the iNaturalist platform (*iNaturalist, 2017*). iNaturalist sightings were manually curated to only include reports which could be unambiguously identified as *P. pyralis* from the photos, and also that also included GPS geotagging to <100 m accuracy. A spreadsheet of these sightings is available on FigShare (DOI: 10.6084/m9.figshare.5688826).

QGIS (v2.18.9, *OSG Foundation, 2017*) was used for data viewing and figure creation. A custom Python script (*Fallon, 2018e*; copy archived at https://github.com/elifesciences-publications/2017_misc_scripts) within QGIS was used to link *P. pyralis* sightings to counties from the US census shapefile (*United States Census Bureau, 2017*). Outlying points that were located in Desert Ecoregions of the World Wildlife Fund (WWF) Terrestrial Ecoregions shapefile (*Olson et al., 2001*; *World Wildlife Fund, 2017*) or the westernmost edge of the range were manually removed, as they are likely isolated populations not representative of the contiguous range. For *Figure 1B*, these points were converted to a polygonal range map using the 'Concave hull' QGIS plugin ('nearest neighbors = 19') followed by smoothing with the Generalizer QGIS plugin with Chaiken's algorithm (Level = 10, and Weight = 3.00). Below (*Appendix 1—figure 1*), red circles indicate county-centroided presence records.

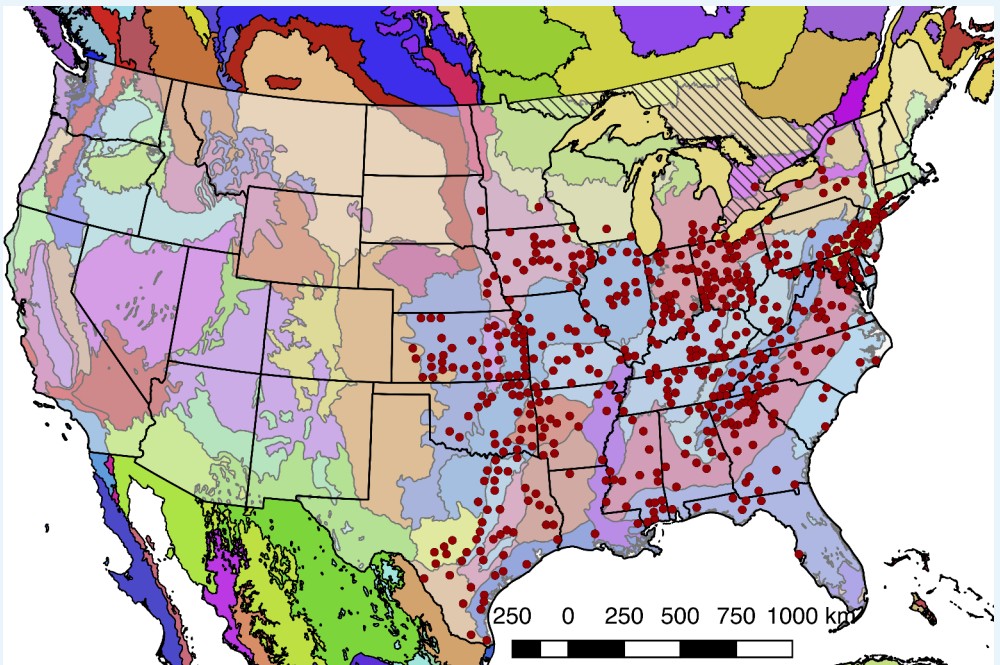

**Appendix 1—figure 1.** Detailed geographic distribution map for *P. pyralis*. *P. pyralis* sightings (red circles show county centroided reports) in the United States and Ontario, Canada (diagonal hashes). The World Wildlife Fund Terrestrial Ecoregions (*Olson et al., 2001*; *World Wildlife Fund, 2017*) are also shown (colored shapes). The *P. pyralis* sighting dataset shown is identical to that used to prepare *Figure 1B*.

DOI: https://doi.org/10.7554/eLife.36495.020

In our field surveys, we found that the range of *P. pyralis* was notably extended from the range reported by Lloyd, specifically we found *P. pyralis* in abundance to the west of the Mill river in Connecticut. *P. pyralis* is found with confidence roughly from Connecticut to Texas, and possibly as far south as Guatemala (Personal communication: A. Catalán). These possible southern populations require further study.

## 1.3 Specimen collection and identification

Adult male *P. pyralis* specimens for Illumina short-insert and mate-pair sequencing were collected at sunset on June 13th, 2011 near the Visitor's Center at Great Smoky Mountains National Park (permit to Dr. Kathrin Stanger-Hall). Specimens were identified to species and sex via morphology (**Green, 1956**), flash pattern and behavior (**Lloyd, 1966**), and *cytochrome-oxidase I* (*COI*) similarity (partial sequence: primers HCO, LCO [**Stanger-Hall and Lloyd, 2015**]) when blasted against an in-house database of firefly *COI* nucleotide sequences. Collected fireflies were stored in 95% ethanol at −80°C until DNA extraction.

Adult male *P. pyralis* specimens for Pacific Biosciences (PacBio) RSII sequencing were captured during flight at sunset on June 9th, 2016, from Mercer Meadows in Lawrenceville, NJ (40.3065 N 74.74831 W), on the basis of the characteristic 'rising J' flash pattern of *P. pyralis* (permit to TRF via Mercer County Parks Commission). Collected fireflies were sorted, briefly checked to be likely *P. pyralis* by the presence of the margin of ventral unpigmented abdominal tissue anterior to the lanterns, flash frozen with liquid N$_2$, lyophilized, and stored at −80°C until DNA extraction. A single aedeagus (male genitalia) was dissected from the stored specimens and confirmed to match the *P. pyralis* taxonomic key (**Green, 1956**) (**Appendix 1—figure 2**).

**A** 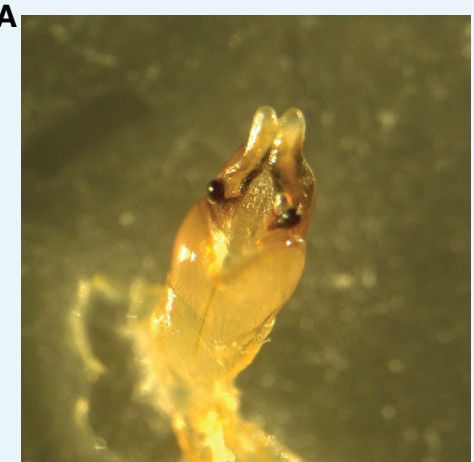 **B** 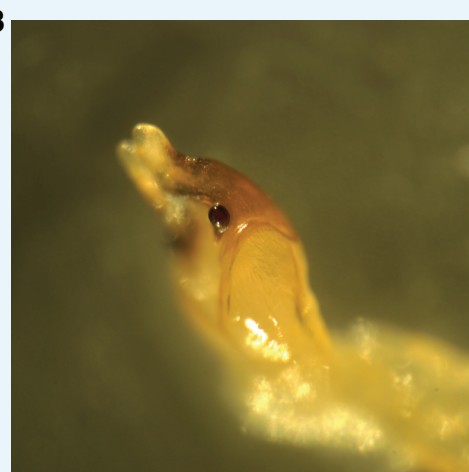

**Appendix 1—figure 2.** *P. pyralis* aedeagus (male genitalia). (**A**) Ventral and (**B**) side view of a *P. pyralis* aedeagus dissected from specimens collected on the same date and locality as those used for PacBio sequencing. Note the strongly sclerotized paired ventro-basal processes ('mickey mouse ears') emerging from the median process, characteristic of *P. pyralis* (**Green, 1956**).

DOI: https://doi.org/10.7554/eLife.36495.021

## 1.3.2 Collection and rearing of *P. pyralis* larvae

We intended to survey the lucibufagin content of *P. pyralis* larvae (**Figure 6B**; Appendix 4.6), and as well as the transovarial transmission of Photinus pyralis orthomyxo-like viruses from parent to larvae (Appendix 5.4), but as *P. pyralis* larvae are subterranean and extremely difficult to collect from the wild, we reared *P. pyralis* larvae from eggs laid from mated pairs. It

is important to note that these *P. pyralis* larval rearing experiments were unexpectedly successful. Although there has been some success in laboratory rearing and domestication of Asian *Aquatica* spp. (**Chiang and Yang, 2010**), including the *A. lateralis* Ikeya-Y90 strain described in this manuscript, rearing of North American fireflies is considered extremely difficult with numerous unpublished failures for unclear reasons (**Lloyd, 1996**), and limited reports of successful rearing of mostly non-Photinus genera, including *Photuris* sp. (**McLean et al., 1972**), *Pyractomena* angulata (**Buschman, 1988**), and *Pyractomena borealis* (Personal communication: Scott Smedley). The below protocol for *P. pyralis* larval rearing is presented in the context of disclosure of the methods of this manuscript, and should be considered a preliminary, unoptimized rearing protocol. A full description of the *P. pyralis* larvae and it's life history and behavior will be presented in a separate manuscript.

Four adult female *P. pyralis* were collected from the Bluemont Junction Trail in Arlington, VA from June 12th through June 18th 2017 (collection permission obtained by TRF from Arlington County Parks and Recreation department). The females were mated to *P. pyralis* males collected either from the same locality and date, or to males collected from Kansas in late June. Mating was performed by housing one to two males and one female in small plastic containers for ~1–3 days with a wet kimwipe to maintain humidity. Mating pairs were periodically checked for active mating, which in *Photinus* fireflies takes several hours. Successfully mated females were transferred to Magenta GA-7 plastic boxes (Sigma-Aldrich, USA), and provided a ~4 cm x 4 cm piece of locally collected moss (species diverse and unknown) as egg deposition substrate, and allowed to deposit eggs until their death in ~1–4 days. Deceased females were removed, artificial freshwater (AFW; 1:1000 diluted 32 PSU artificial seawater) was sprayed into the box to maintain high humidity, and eggs were kept for 2–3 weeks at room temperature and periodically checked until hatching. Like other firefly eggs, the eggs of *P. pyralis* were observed to be faintly luminescent imaging using a cooled CCD camera (**Appendix 1—figure 3**); however, this luminescence was not visible to the dark-adapted eye, indicating that this luminescence is less intense than other firefly species such as *Luciola cruciata* (**Harvey, 1952**).

Upon hatching, first instar larvae were mainly fed ~1 cm cut pieces of Canadian Nightcrawler earthworms (*Lumbricus terrestris*; Windsor Wholesale Bait, Ontario, Canada), and occasional live White Worms (*Enchytraeus albidus;* Angels Plus, Olean, NY). Although *P. pyralis* first instar larvae were observed to attack live *Enchytraeus albidus,* an experiment to determine if this would be suitable as a single food source was not performed. Uneaten and putrefying earthworm pieces were removed after 1 day, and the container cleaned. Once the larvae had been manually fed for ~2 weeks and deemed sufficiently strong, they were transferred to plastic shoeboxes (P/N: S-15402, ULINE, USA) which were intended to mimic a soil ecosystem. In personal discussions of unpublished firefly rearing attempts by various firefly researchers, we noted that a common theme was the difficulty of preventing the uneaten prey of these predatory larvae from putrifying. Therefore, we sought to create ecologically inspired 'eco-shoeboxes', where fireflies would prey on live organisms, and other organisms would assist in cleanup of uneaten or partially eaten prey that had been fed to the firefly larvae, to prevent the growth of pathogenic microorganisms on uneaten prey.

First, these shoeboxes were filled with 1L of mixed 50% (v/v) potting soil, and 50% coarse sand (Quikrete, USA) that had been washed several times with distilled water to remove silt and dust. The soil-sand mix was wet well with AFW, and live *Enchytraeus albidus* (50+), temperate springtails (50+; *Folsomia candida*; Ready Reptile Feeders, USA), and dwarf isopods (50+; *Trichorhina tomentosa*; Ready Reptile Feeders, USA) were added to the box, and several types of moss, coconut husk, and decaying leaves were sparingly added to the corners of the box. The non-firefly organisms were included to mimic a primitive detritivore (*Enchytraeus albidus* and *Trichorhina tomentosa*) and fungivore (*Folsomia candida*) system. About 50 firefly larvae were included per box. No interactions between the *P. pyralis* larvae and the additional organisms were observed. Predation on *Enchytraeus albidus* seems likely, but careful observations were not made. Distilled water was sprayed into the box every ~2 days to maintain a high humidity. Throughout this period, live *Lumbricus terrestris* (~10–15 cm) were added to the box every 2–3 days as food. These earthworms were first prepared by

washing with distilled water several times to remove attached soil, weakened and stimulated to secrete coelemic fluid and gut contents by spraying with 95% ethanol, washed several times in distilled water, and left overnight in ~2 cm depth distilled water at 4°C. Anecdotally this pre-cleaning and preparation process reduced the rate and degree that dead earthworms putrefied. Young *P. pyralis* larvae were observed to successfully kill and gregariously feed on these live earthworms (*Appendix 1—figure 4*). The possibility that firefly larvae possess a paralytic venom used to stun or kill prey has been noted by other researchers (*Hess, 1920*; *Williams, 1917*). In our observations, an earthworm would immediately react to the bite from a single *P. pyralis* larvae, thrashing about for several minutes, but would then become seemingly paralyzed over time, supporting the role of a potent, possibly neurotoxic, firefly venom. The *P. pyralis* larvae would then begin extra-oral digestion and gregarious feeding on the liquified earthworm. Once the earthworm had been killed and broken apart by firefly larvae, *Enchytraeus albidus* would enter through gaps in the cuticle and begin to feed in large numbers throughout the interior of the earthworm. The other detritivores were observed at later stages of feeding. Between the combined action of the *P. pyralis* larvae, and the other detritivores, the live earthworm was completely consumed within 1–2 days, and no manual cleanup was required.

Compared to the initial manual feeding and cleaning protocol for *P. pyralis* 1st instar larvae, the 'eco-shoebox' rearing method was low-input and convenient for large numbers of larvae. The feeding and cleanup process was efficient for ~2 months (July through September), leading to a large number of healthy 3-4th instar larvae (*Appendix 1—figure 5*). However, after that point, *P. pyralis* larvae, possibly in preparation for a winter hibernation, seemingly became quiescent, and were less frequently seen patrolling throughout the box. At the same time, the *Enchytraeus albidus* earthworms were observed to become less abundant, either due to continual predation by *P. pyralis*, or due to population collapse from insufficient fulfillment of nutritional requirements from feeding of *Enchytraeus albidus* on *Lumbricus terrestris* alone.

At this point, earthworms were not consumed within 1–2 days, and became putrid, and *P. pyralis* which had been feeding on these earthworms were frequently found dead nearby, and themselves quickly putrefied. Generally after this point *P. pyralis* larvae were more frequently found dead and partially decayed, indicating the possibility of pathogenesis from microorganisms from putrefying earthworms. At this stage, it was observed that mites (Acari), probably from the soil contained in the guts of the fed earthworms, became abundant, and were observed to act as ectoparasitic on *P. pyralis* larvae. An attempt to simulate hibernation of *P. pyralis* larvae was made by storing them at 4°C for ~3 weeks, however a large proportion (~30%) of larvae died during this hibernation to a seeming fungal infection. Other larvae revived quickly when returned to room temperature, but all *Trichorhina tomentosa* were killed by even transient exposure to 4°C. To date, a smaller number of fifth and sixth instar *P. larvae* have been obtained, but pupation in the laboratory has not occured. The lack of pupation is unsurprising as it is likely occurs in the wild after 1–2 years of growth, is likely under temperature and photoperiodic control, and may require a licensing stage of cold temperature hibernation for several weeks. Overall, manual feeding of first1 st instar larvae followed by the 'eco-shoebox' method was unexpectedly successful approach for the maintenance and growth of *P. pyralis* larvae.

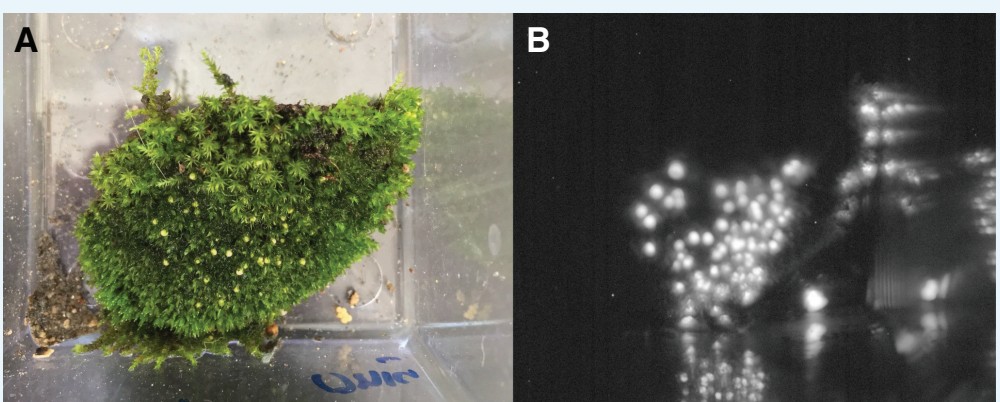

**Appendix 1—figure 3.** Luminescence of *P. pyralis* eggs. (**A**) Photograph under ambient light of ~1 day post-deposition *P. pyralis* eggs. (**B**) Photograph of self-luminescence of ~1 day post-deposition *P. pyralis* eggs. Both photographs taken with a NightOwl LB98 cooled CCD luminescence imager (Berthold Technologies, USA). Luminescence was not visible to the dark-adapted eye.
DOI: https://doi.org/10.7554/eLife.36495.022

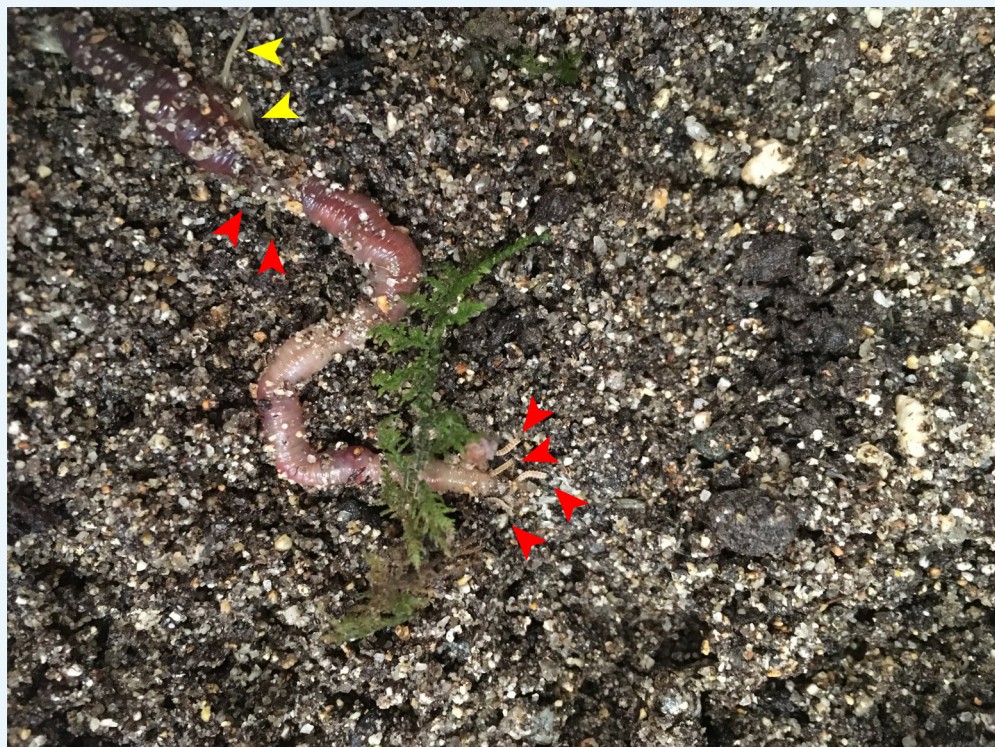

**Appendix 1—figure 4.** Gregarious predation of young *P. pyralis* larvae on a live *Lumbricus terrestris*. Both *P. pyralis* larvae (red arrows), and *Enchytraeus albidus* (yellow arrows), were observed to feed on the paralyzed earthworms.
DOI: https://doi.org/10.7554/eLife.36495.023

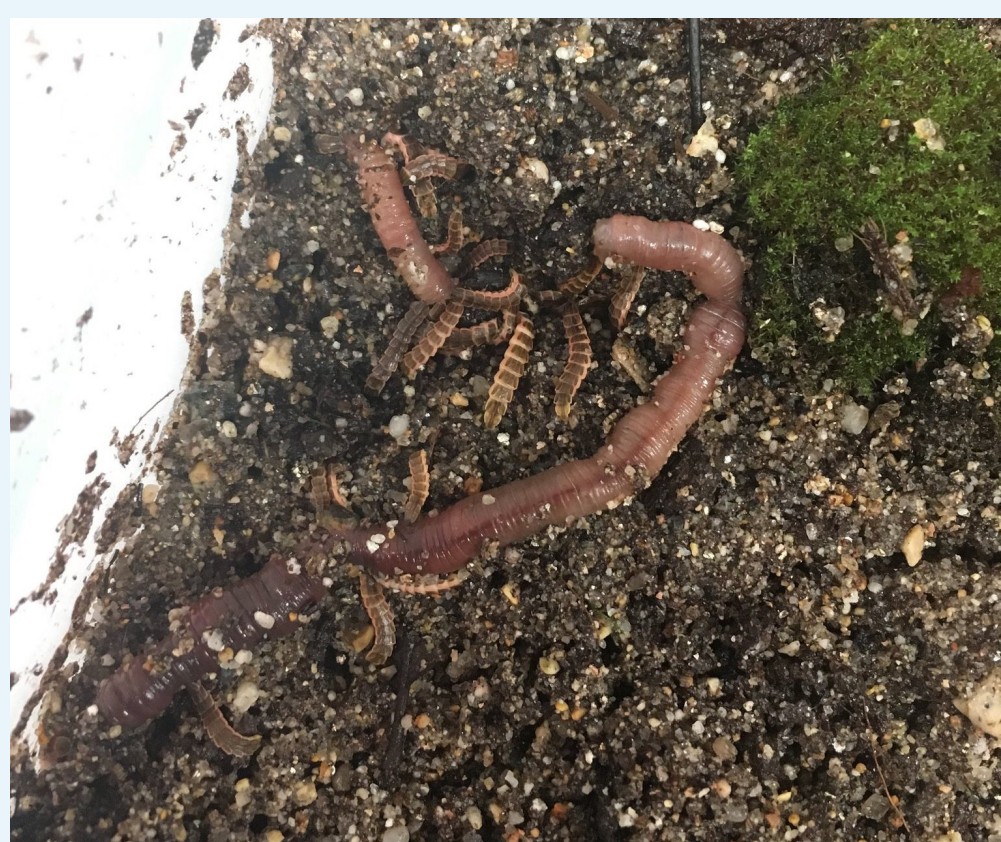

**Appendix 1—figure 5.** Gregarious predation of 3rd-4th instar *P. pyralis* larvae on a live *Lumbricus terrestris*.
DOI: https://doi.org/10.7554/eLife.36495.024

## 1.4 Karyotype and genome size

The karyotype of *P. pyralis* was previously reported to be 2n = 20 with XO sex determination (male, 18A + XO; female, 18A + XX) (**Wasserman and Ehrman, 1986**). The genome sizes of four *P. pyralis* adult males were previously determined to be 422 ± 9 Mbp (SEM, n = 4), whereas the genome sizes of five *P. pyralis* adult females were determined to be 448 ± 7 (SEM, n = 5) by nuclear flow cytometry analysis (**Lower et al., 2017**). From these analyses, the size of the X-chromosome is inferred to be ~26 Mbp. Genome size inference via kmer spectral analysis of the *P. pyralis* short-insert Illumina data from a single adult *P. pyralis* male estimated a genome size of 343 Mbp (**Appendix 1—figure 6**).

## 1.5 Library preparation and sequencing

See **Appendix 4—table 1** for a overview of all sequence libraries. Library specific construction methods are detailed below.

### 1.5.1 Illumina

DNA was extracted from sterile-water-washed thorax of Great Smoky Mountains National Park collected specimens using phenol-chloroform extraction with RNAse digestion, checked for quality via gel electrophoresis, and quantified by Nanodrop or Qubit (Thermo Scientific, USA). To obtain sufficient DNA for both short insert and mate-pair library construction, libraries were constructed separately from DNA from each of two individual males and pooled DNA of three males, all from the same population. Males were selected for sequencing as they are more easily found in the field than females. In addition, as *P. pyralis* males are XO (**Dias et al., 2007**), differences in sequencing coverage could inform localization of scaffolds to the X

chromosome. Illumina TruSeq short insert (average insert size: 300 bp) and Nextera mate-pair libraries (insert size: 3 Kbp, 6 Kbp) were constructed at the Georgia Genomics Facility (Athens, GA) and subsequently sequenced on two lanes of Illumina HiSeq2000 100 × 100 bp PE reads (University of Texas; *Appendix 4—table 1*).

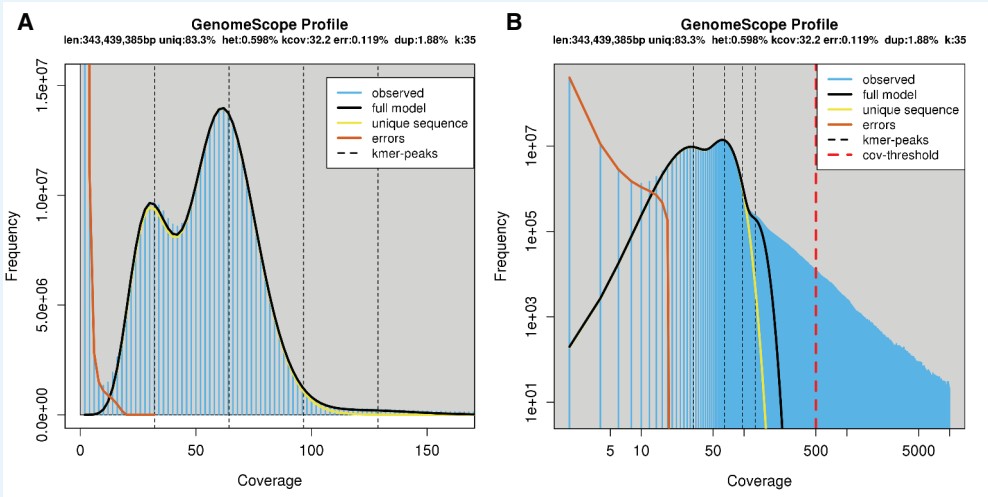

**Appendix 1—figure 6.** Genome scope kmer analysis of the *P. pyralis* short read library. (**A**) Linear and (**B**) log plot of a kmer spectral genome composition analysis of the '8369' *P. pyralis* Illumina short-read library from a single *P. pyralis* XO adult male (Appendix 1.5.1; *Appendix 4—table 1*) with jellyfish (v2.2.9; parameters: -C -k 35) (*Marçais and Kingsford, 2011*) and GenomeScope (v1.0; parameters: Kmer length = 35, Read length = 100, Max kmer coverage = 1000) (*Vurture et al., 2017*). len = inferred haploid genome length, uniq = percentage non-repetitive sequence, het = overall rate of genome heterozygosity, kcov = mean kmer coverage for heterozygous bases, err = error rate of the reads, dup: average rate of read duplications. These results are consistent with the genome size of a XO male, when possible systematic error of kmer spectral analysis and flow cytometry genome size estimates is considered. The heterozygosity is somewhat low when compared to some other arthropods.

DOI: https://doi.org/10.7554/eLife.36495.025

## 1.5.2 PacBio

High-molecular-weight DNA (HMW DNA) was extracted from four pooled lyophilized adult male *P. pyralis* (dry mass 90.8 mg) from the MMNJ field site. These specimens were first externally washed using 95% ethanol, after which DNA extraction proceeded with a 100/G Genomic Tip plus Genomic Buffers kit (Qiagen, USA). DNA extraction followed the manufacturer's protocol, with the exception of the final precipitation step, where HMW DNA was pelleted with 40 µg RNA grade glycogen (Thermo Scientific, USA) and centrifugation (3000 x g, 30 min, 4 °C) instead of spooling on a glass rod. Although increased genomic heterozygosity from four pooled males and a resulting more complicated genome assembly was a concern for a wild population like *P. pyralis*, four males were used in order to extract enough DNA for workable coverage using 15 Kbp+ size selected PacBio RSII sequencing. All extracted DNA was used for library preparation, and all of the final library was used for sequencing. Adult males, being XO, were chosen over the preferable XX females, as adult males are much more easily captured because they signal during flight, whereas females are typically found in the brush below and generally only flash in response to authentic male signals.

Precipitated HMW DNA was redissolved in 80 µL Qiagen QLE buffer (10 mM Tris-Cl, 0.1 mM EDTA, pH 8.5) yielding 17.1 µg of DNA (214 ng/µL) and glycogen (500 ng/µL). Final DNA concentration was measured with a Qubit fluorometer (Thermo Scientific) using the Qubit

Broad Range kit. Manipulations hereafter, including HMW DNA size QC, fragmentation, size selection, library construction, and PacBio RSII sequencing, were performed by the Broad Technology Labs of the Broad Institute (Cambridge, MA).

First, the size distribution of the HMW DNA was confirmed by pulsed-field-gel-electrophoresis (PFGE). In brief, 100 ng of HMW DNA was run on a 1% agarose gel (in 0.5x TBE) with the BioRad CHEF DRIII system. The sample was run out for 16 hr at six volts/cm with an angle of 120 degrees with a running temperature of 14 ˚C. The gel was stained with SYBRgreen dye (Thermo Scientific - Part No. S75683). 1 µg of 5 Kbp ladder (BioRad, part no 170–3624) was used as a standard. These results demonstrated the HMW DNA had a mean size of >48 Kbp (*Appendix 1—figure 7*). This pool of HMW DNA is designated 1611_PpyrPB1 (NCBI BioSample SAMN08132578).

Next, HMW DNA (17.1 µg) was sheared to a targeted average size of 20–30 Kbp by centrifugation in a Covaris g-Tube (part no. 520079) at 2500 x g for 2 min. SMRTbell libraries for sequencing on the PacBio platform were constructed according to the manufacturer's recommended protocol for 20 Kbp inserts, which includes size selection of library constructs larger than 15 Kbp using the BluePippin system (Sage Science, Beverly, MA). Two separate cassettes were run. In each cassette, two lanes were used in which there was 1362 ng/lane (PAC20kb kit). Constructs 15 Kbp and above were eluted over a period of 4 hr. An additional damage repair step was carried out post size-selection. Insert size range for the final library was determined using the Fragment Analyzer System (Advanced Analytical, Ankeny, IA). The size-selected SMRTbell library was then sequenced over 61 SMRT cells on a PacBio RSII instrument of the Broad Technology Labs (Cambridge, MA), using the P6 v.2 polymerase and the v.4 DNA Sequencing Reagent (P6-C4 chemistry; part numbers 100-372-700, 100-612-400). PacBio sequencing data is available on the NCBI Sequence Read Archive (Bioproject PRJNA378805).

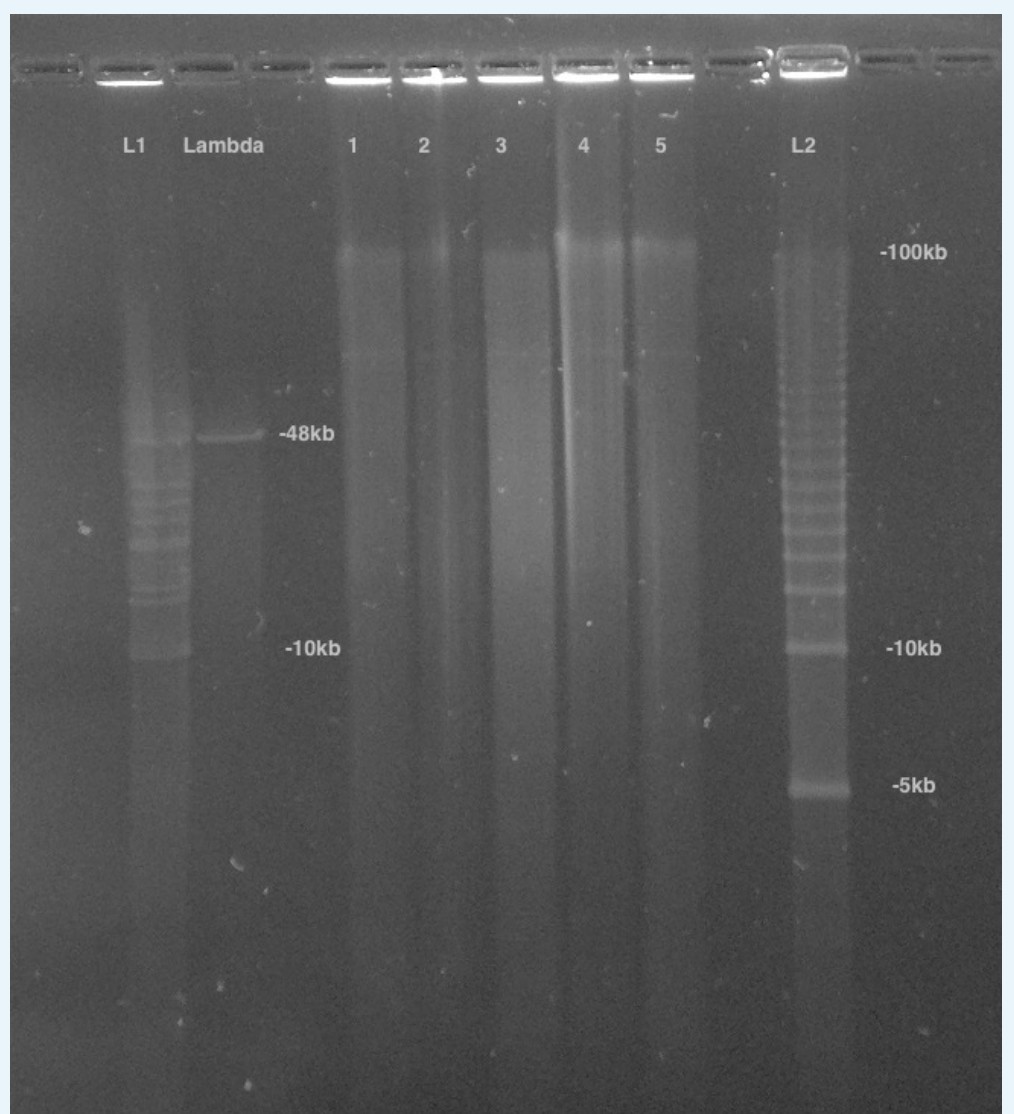

**Appendix 1—figure 7.** PFGE of *P. pyralis* HMW DNA used for PacBio sequencing. Lane 1 was used for further library prep and sequencing, Lanes 2–5 represent separate batches of *P. pyralis* HMW DNA that was not used for PacBio sequencing. Lane 1 was used as it had the highest DNA yield, and an equivalent DNA size distribution to the other samples.
DOI: https://doi.org/10.7554/eLife.36495.026

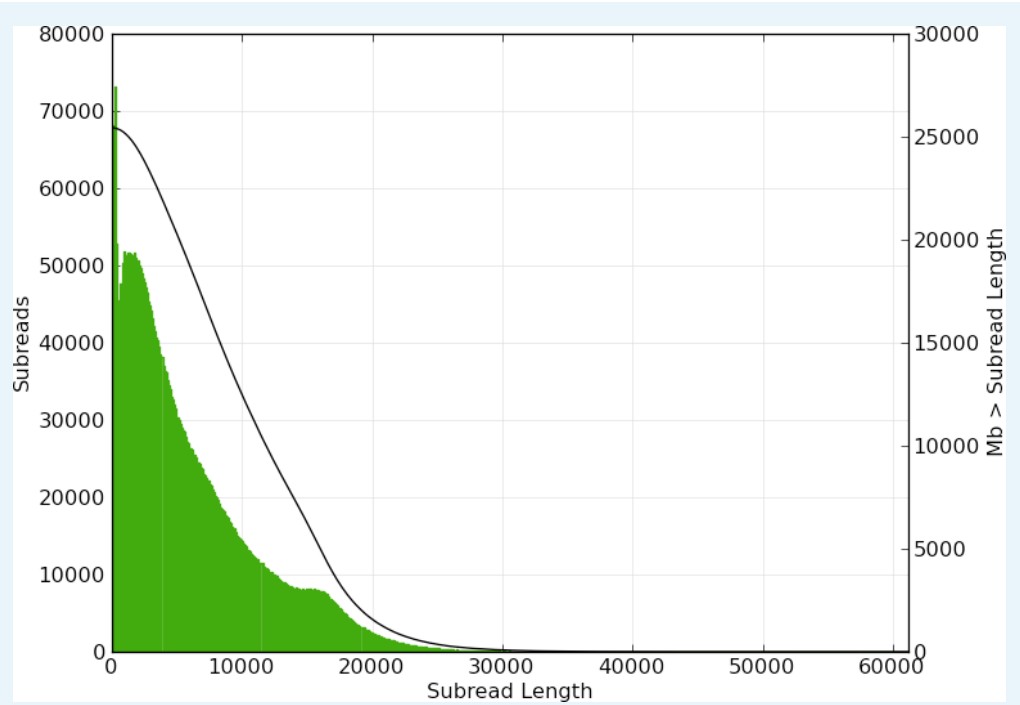

**Appendix 1—figure 8.** Subread length distribution for *P. pyralis* PacBio RSII sequencing. Figure produced with SMRTPortal (v2.3.0.140936, *Pacific Biosciences, 2017*) by aligning all PacBio reads from data from the 61 SMRT cells against Ppyr1.3 using the RS_Resequencing.1 protocol with default parameters. Subread length unit is basepair (bp).
DOI: https://doi.org/10.7554/eLife.36495.027

### 1.5.3 Hi-C library preparation

Two adult *P. pyralis* MMNJ males were flash frozen in liquid nitrogen, stored at −80°C, and shipped on dry-ice to Phase Genomics (Seattle, WA). Manipulations hereafter occurred at Phase Genomics, following previously published protocols (*Lieberman-Aiden et al., 2009*; *Burton et al., 2013*; *Bickhart et al., 2017*). Briefly, a streamlined version of the standard Hi-C protocol (*Lieberman-Aiden et al., 2009*) was used to perform a series of steps resulting in proximity-ligated DNA fragments, in which physically proximate sequence fragments are joined into linear chimeric molecules. First, in vivo chromatin was cross-linked with formaldehyde, fixing physically proximate loci to each other. Chromatin was then extracted from cellular material and digested with the *Sau*3AI restriction enzyme, which cuts at the GATC motif. The resulting fragments were proximity ligated with biotinylated nucleotides and pulled down with streptavidin beads. These chimeric sequences were then sequenced with 80 bp PE sequencing on the Illumina NextSeq platform, resulting in Hi-C read pairs.

## 1.6 Genome assembly

The *P. pyralis* genome assembly followed three stages: (1) a hybrid assembly using Illumina and PacBio reads, producing assembly Ppyr1.1 (Appendix 1.6.2), (2) Ppyr1.1 scaffolded using Hi-C data, producing assembly Ppyr1.2 (Appendix 1.6.3), and (3) Ppyr1.2 manually curation for proper X-chromosome assembly and removal of putative non-firefly sequences, producing Ppyr1.3 (1.6.4).

### 1.6.2 Ppyr1.1: MaSuRCA hybrid assembly

Several genome assembly approaches were evaluated with the general goal of maximizing conserved gene content and contiguity. The highest quality *P. pyralis* assembly was generated by a hybrid assembly approach using a customized MaSuRCA (v3.2.1_01032017) (*Zimin et al., 2013*; *Zimin et al., 2017*) pipeline that combined both Illumina-corrected PacBio reads (Mega-

reads) and synthetic long reads constructed from short-insert reads alone (Super-reads) using a custom small overlap length (59 bp).

We first applied MaSuRCA (v3.2.1_01032017) (*Zimin et al., 2013*; *Zimin et al., 2017*) to correct our long reads (38x coverage; Library ID 1611_PpyrPB1; *Appendix 4—table 1*) using our short-insert and mate-pair reads (Libraries: 8369, 375_3K, 8375_6K, 83_3K, 83_6K; *Appendix 4—table 1*). No pre-filtering of reads was performed, as Illumina adaptors are automatically removed within the MaSuRCA pipeline. We modified the pipeline to assemble the genome using both corrected long reads (Mega-reads) and synthetic long reads (Super-reads) with a custom smaller overlap length (59 bp). All reads (short-insert, mate-pair and PacBio) were then used within the MaSuRCA pipeline to call a genomic consensus.

To scaffold the contigs, we first filtered Illumina short-reads from the mate-pair libraries (Libraries 8375_3K, 8375_6K, 83_3K, 83_6K) with Nxtrim (v0.4.1) (*O'Connell et al., 2015*) with parameters '–separate –rf –justmp'. We then manually integrated the MaSuRCA assembly by replacing the incomplete mitochondrial contigs with complete mitochondrial assemblies from *P. pyralis* and *Apocephalus antennatus* (Appendix 5.2). We scaffolded and gap-filled the assembly using the Illumina short-insert and filtered mate-pair reads (Libraries: 8369, 8375_3K, 8375_6K, 83_3K, 83_6K) via Redundans (v0.13a) (*Pryszcz and Gabaldón, 2016*) with default settings. After scaffolding with our Illumina data, redundant sequences were removed by the MaSuRCA 'deduplicate_contigs.sh' script. We then applied PBjelly (v15.8.24) (*English et al., 2012*) and PacBio reads to scaffold and gap-fill the assembly, and redundancy reduction with 'deduplicate_contigs.sh' script was run again. Finally, we replaced mitochondrial sequences which had been artificially extended by the scaffolding, gap-filling and sequence extension process with the proper sequences. The resultant assembly was dubbed Ppyr1.1.

### 1.6.3 Ppyr1.2: Scaffolding with Hi-C

The Hi-C read pairs were applied in a manner similar to that originally described here (*Burton et al., 2013*) and later expanded upon (*Bickhart et al., 2017*). Briefly, Hi-C reads were mapped to Ppyr1.1 with BWA (v1.7.13) (*Li and Durbin, 2009*), requiring perfect, unique mapping locations for a read pair to be considered usable. The number of read pairs joining a given pair of contigs is referred to as the 'link frequency' between those contigs, and when normalized by the number of restriction sites in the pair of contigs, is referred to as the 'link density' between those contigs.

A three-stage scaffolding process was used to create the final scaffolds, with each stage based upon previously described analysis of link density (*Burton et al., 2013*; *Bickhart et al., 2017*). First, contigs were placed into chromosomal groups. Second, contigs within each chromosomal group were placed into a linear order. Third, the orientation of each contig is determined. Each scaffolding stage was performed many times in order to optimize the scaffolds relative to expected Hi-C linkage characteristics.

In keeping with previously described methods (*Burton et al., 2013*; *Bickhart et al., 2017*), the number of chromosomal scaffolds to create–10–was an *a priori* input to the scaffolding process derived from the previously published chromosome count of *P. pyralis* (*Wasserman and Ehrman, 1986*). However, to verify the correctness of this assumption, scaffolds were created for haploid chromosome numbers ranging from 5 to 15. A scaffold number of 10 was found to be optimal for containing the largest proportion of Hi-C linkages within scaffolds, which is an expected characteristic of actual Hi-C data.

### 1.6.4 Ppyr1.3: Manual curation and taxonomic annotation filtering

#### 1.6.4.1 Defining the X chromosome

Hi-C data was mapped and converted to the 'hic' file format with the juicer pipeline (v1.5.6) (*Durand et al., 2016b*), and then visualized using juicebox (v1.5.2) (*Durand et al., 2016a*). This visualization revealed a clear breakpoint in Hi-C linkage density on LG3 at ~22,220,000 bp. Mapping of Illumina short-insert and PacBio reads with Bowtie2 (v2.3.1) (*Langmead and Salzberg, 2012*) and SMRTPortal (v2.3.0.140893) with the 'RS_Resequencing.1' protocol, followed by visualization with Qualimap (v2.2.1) (*Okonechnikov et al., 2016*), revealed that the first section of LG3 (1–22,220,000 bp), here termed LG3a, was present at roughly half the coverage of LG3b (22,220,001–50,884,892 bp) in both the Illumina and PacBio libraries.

Mapping of *Tribolium castaneum* X chromosome proteins (NCBI Tcas 5.2) to the Ppyr1.2 assembly using both tblastn (v2.6.0) (*Camacho et al., 2009*) and Exonerate(v2.2.0) (*Slater and Birney, 2005*) based 'protein2genome' alignment through the MAKER pipeline revealed a relative enrichment on LG3a only. Taken together, this data suggested that the half-coverage section of LG3 (LG3a) corresponded to the X-chromosome of *P. pyralis*, and that it was misassembled onto an autosome. Therefore, we manually split LG3 into LG3a and LG3b in the final assembly.

#### 1.6.4.2 Taxonomic annotation filtering

Given the recognized importance of filtering genome assemblies to avoid misinterpretation of the data (*Koutsovoulos et al., 2016*), we sought to systematically remove assembled non-firefly contaminant sequence from Ppyr1.2. Using the blobtools toolset (v1.0.1) (*Laetsch and Blaxter, 2017*), we taxonomically annotated our scaffolds by performing a blastn (v2.6.0+) nucleotide sequence similarity search against the NCBI nt database, and a diamond (v0.9.10.111) (*Buchfink et al., 2015*) translated nucleotide sequence similarity search against the of Uniprot reference proteomes (July 2017). Using this similarity information, we taxonomically annotated the scaffolds with blobtools using parameters '-x bestsumorder –rank phylum'. A tab delimited text file containing the results of this blobtools annotation are available on FigShare (DOI: 10.6084/m9.figshare.5688982). We then generated the final genome assembly by retaining scaffolds that either contained annotated features (genes or non-simple/low-complexity repeats), had coverage >10.0 in both the Illumina (*Appendix 1—figure 9*) and PacBio libraries (*Appendix 1—figure 10*), and if the taxonomic phylum was annotated as 'Arthropod' or 'no-hit' by the blobtools pipeline (*Appendix 1—figure 11*). This approach removed 374 scaffolds (2.1 Mbp), representing 15% of the scaffold number and 0.4% of the nucleotides of Ppyr1.2. Notably, four tenericute scaffolds, likely corresponding to a partially assembled *Entomoplasma sp.* genome, distinct from the *Entomoplasma luminosus var. pyralis* assembled from the PacBio library (Appendix 5) were removed. Furthermore, we removed two contigs representing the mitochondrial genome of *P. pyralis* (complete mtDNA available via Genbank: KY778696). The final filtered assembly, Ppyr1.3, is available at www.fireflybase.org.

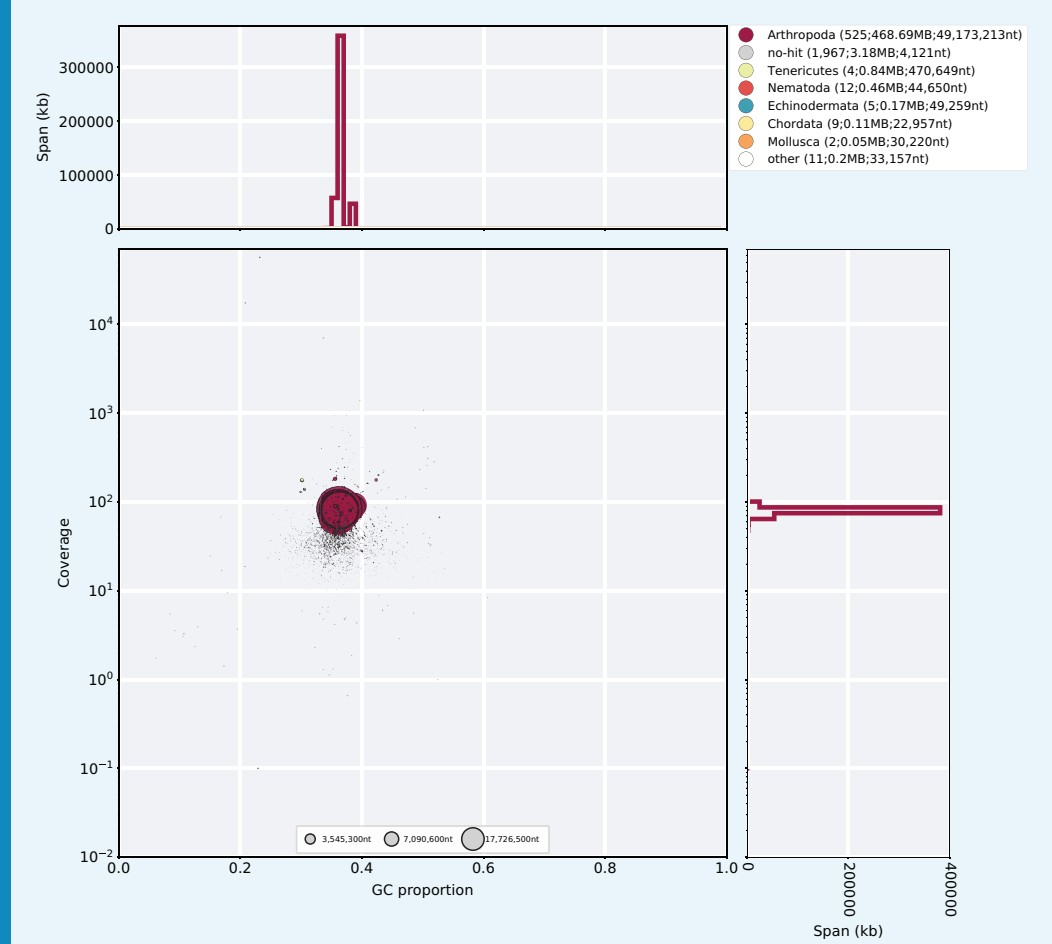

**Appendix 1—figure 9.** Blobplot of Illumina short-insert reads aligned against the Ppyr1.2 reference. Coverage shown represents mean coverage of reads from the Illumina short-insert library (Sample name 8369; *Appendix 4—table 1*), aligned against Ppyr1.2 using Bowtie2 with parameters (–local). Scaffolds were taxonomically annotated as described in Appendix 1.6.4.2.
DOI: https://doi.org/10.7554/eLife.36495.028

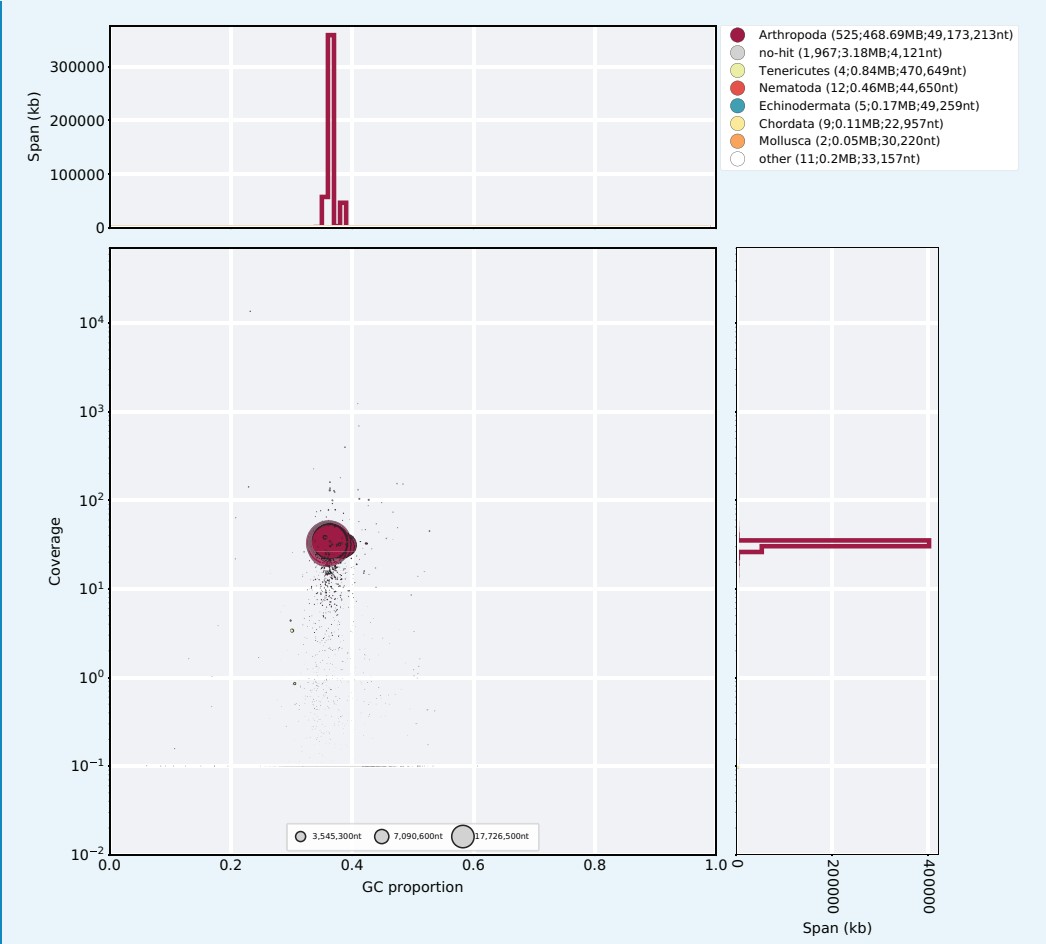

**Appendix 1—figure 10.** Blobplot of *P. pyralis* PacBio reads aligned against Ppyr1.2. Coverage shows represents mean coverage of reads from the PacBio library (Sample name 1611; *Appendix 4—table 1*). The reads were aligned using SMRTPortal v2.3.0.140893 with the 'RS_Resequencing.1' protocol with default parameters. Scaffolds were taxonomically annotated as described in Appendix 1.6.4.2.

DOI: https://doi.org/10.7554/eLife.36495.029

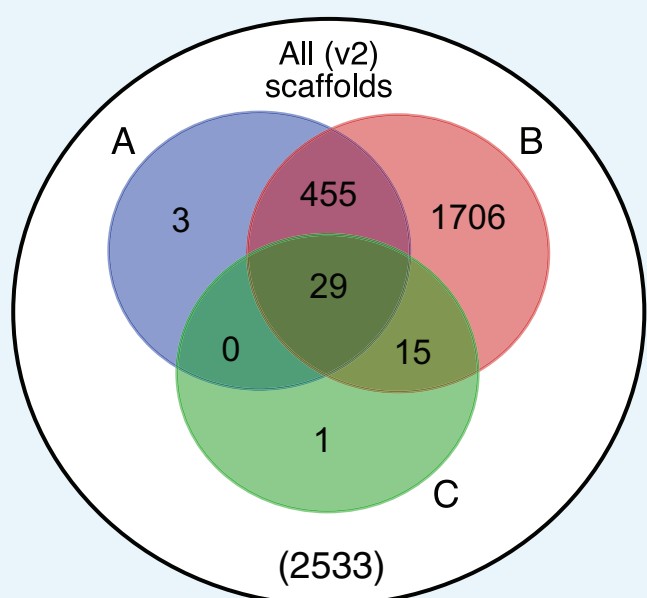

**Appendix 1—figure 11.** Venn diagram representation of blobtools taxonomic annotation filtering approach for Ppyr1.2 scaffolds. (**A**) The blue set represents scaffolds which have >10.0 coverage in both Illumina and PacBio libraries. (**B**) The red set represents scaffolds which had either genes on repeats (non simple or low-complexity) annotated. (**C**) The green set represents scaffolds with suspicious taxonomic assignment (Non 'Arthropod' or 'no-hit'). Outside A, B, and C, represents low-coverage, unannotated scaffolds. Ppyr1.3 consists of the intersection of A and B, minus the intersection of C. All linkage groups (LG1-LG10) were annotated as 'Arthropod' by blobtools, and captured in the intersection between A and B but not set C.

DOI: https://doi.org/10.7554/eLife.36495.030

## 1.7 Ppyr0.1-PB: PacBio only genome assembly

In addition to our finalized genome assembly (Ppyr1.3), we sought to better understand the symbiont composition that varied between our *P. pyralis* PacBio and Illumina libraries. Therefore, we produced a long-read only assembly of our PacBio data to assemble the sequence that might be unique to this library. To achieve this, we first filtered the HDF5 data from the 61 sequence SMRT cells to. FASTQ format subreads using SMRTPortal (v2.3.0.140893) (*Pacific Biosciences, 2017*) with the 'RS_Subreads.1' protocol with default parameters. These subreads were then input into Canu (Github commit 28ecea5/v1.6) (*Koren et al., 2017*) with parameters 'genomeSize = 450 m corOutCoverage = 200 ovlErrorRate = 0.15 obtErrorRate = 0.15 -pacbio-raw'. The unpolished contigs from this produced genome assembly are dubbed Ppyr0.1-PB.

## 1.8 Mitochondrial genome assembly and annotation

To achieve a full length mitochondrial genome (mtDNA) assembly of *P. pyralis*, sequences were assembled separately from the nuclear genome. Short insert Illumina reads from a single GSMNP individual (Sample 8369; *Appendix 4—table 1*) were mapped to the known mtDNA of the closest available relative, *Pyrocoelia rufa* (NC_003970.1 [*Bae et al., 2004*]) using bowtie2 v2.3.1 (parameters: –very-sensitive-local). All concordant read pairs were input to SPAdes (v3.8.0) (*Nurk et al., 2013*) (parameters: –plasmid –only-assembler -k35,55,77,90) for assembly. The resulting contigs were then combined with the *P. rufa* mitochondrial reference genome for a second round of read mapping and assembly. The longest resulting contig aligned well to the *P. rufa* mitochondrial genome; however, it was ~1 Kbp shorter than expected, with the unresolved region appearing to be the tandem repetitive region (TRU)

(*Bae et al., 2004*), previously described in the *P. rufa* mitochondrial genome. To resolve this, all PacBio reads were mapped to the draft mitochondrial genome, and a single high-quality PacBio circular-consensus-sequencing (CCS) read that spanned the unresolved region was selected using manual inspection and manually assembled with the contiguous sequence from the Illumina sequencing to produce a complete circular assembly. The full assembly was confirmed by re-mapping the Illumina short-read data using bowtie2 followed by consensus calling with Pilon v1.21(*Walker et al., 2014*). Re-mapped PacBio long-read data also confirmed the structure of the mtDNA, and indicated variability in the repeat unit copy number of the TRU amongst the four sequenced *P. pyralis* individuals (Sample 1611_PpyrPB1; *Appendix 4—table 1*). The *P. pyralis* mtDNA was then 'restarted' using seqkit(*Shen et al., 2016*), such that the FASTA record break occurred in the AT-rich region, and annotated using the MITOS2 annotation server (*Bernt et al., 2017*). Low confidence and duplicate gene predictions were manually removed from the MITOS2 annotation. The final *P. pyralis* mtDNA with annotations is available on GenBank (KY778696).

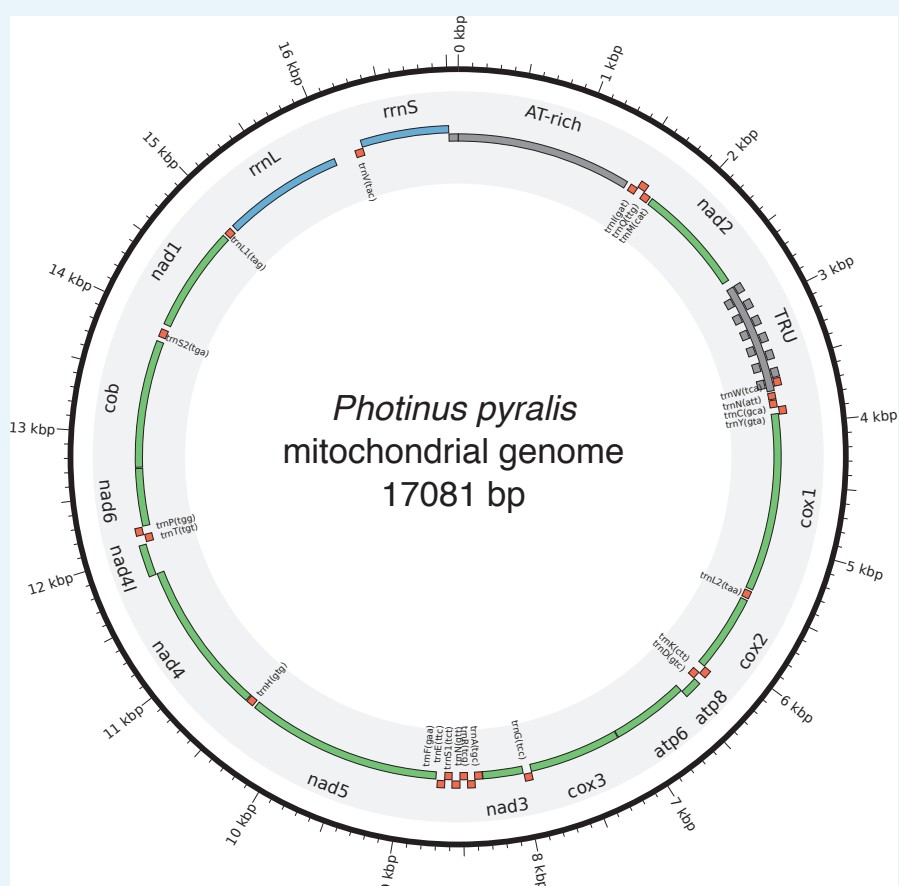

**Appendix 1—figure 12.** Mitochondrial genome of *P. pyralis*. The mitochondrial genome of *P. pyralis* was assembled and annotated as described. Note the firefly specific tandem-repeat-unit (TRU) region. Figure produced with Circos (*Krzywinski et al., 2009*).
DOI: https://doi.org/10.7554/eLife.36495.031

## 1.9 Transcriptome analysis

### 1.9.1 RNA-extraction, library preparation and sequencing

In order to capture expression from diverse life stages, stranded RNA-Seq libraries were prepared from whole bodies of four life stages/sexes (eggs, 1 st instar larvae, adult male, and adult female; *Appendix 1—table 1*). Eggs and larvae were derived from a laboratory mating of *P. pyralis* (Collected MMNJ, July 2016). Briefly, live adult *P. pyralis* were transported to the

lab and allowed to mate in a plastic container over several days. The female, later sequenced, was observed mating with two independent males on two separate nights. The female was then transferred to a plastic container with moss, and allowed to oviposit over several days. Once no more oviposition was observed, the female was removed, flash frozen with liquid N$_2$, and stored at −80˚C for RNA extraction. Resulting eggs were washed 3x with dilute bleach/H$_2$O and reared in aggregate in plastic containers on moist Whatman paper. ~13 days after the start of egg oviposition, a subset of eggs were flash frozen for RNA extraction. The remaining eggs were allowed to hatch and larvae were flash frozen the day after emergence (first instar). Total RNA was extracted from a single stored adult male (non-paternal to eggs/larvae), the adult female (maternal to eggs/larvae), seven pooled eggs, and four pooled larvae using the RNeasy Lipid Tissue Mini Kit (QIAGEN) with the optional on-column DNase treatment. Illumina sequencing libraries were prepared by the Whitehead Genome Technology Core (WI-GTC) using the TruSeq Stranded mRNA library prep kit (Illumina) and following the manufacturer's instructions with modification to select for larger insert sizes (~300–350 bp). These samples were multiplexed with unrelated plant RNA-Seq samples and sequenced 150 × 150 nt on one rapid mode flowcell (two lanes) of a HiSeq2500 (WI-GTC), to a depth of ~30M paired reads per library.

To examine gene expression in adult light organs, we generated non-strand specific sequencing of polyA pulldown enriched mRNA from dissected photophore tissue (**Appendix 1—table 1**). Photophores were dissected from the abdomens of adult *P. pyralis* males (Collected MMNJ, July 2015) by Dr. Adam South (Harvard School of Public Health), using three individuals per biological replicate. These tissues and libraries were co-prepared and sequenced with other previously published libraries (full library preparation and sequencing details here [*Al-Wathiqui et al., 2016*]) at a depth of ~10M paired reads per library.

To examine gene expression in larval light organs, we performed RNA-seq on dissected larval light organs. We first extracted total RNA from a pool of six dissected larval photophores from three individuals using the RNeasy Lipid Tissue Mini Kit (QIAGEN) with the optional on-column DNase treatment. The larvae were the same larvae described in Appendix 1.3.2. The total RNA was enriched to mRNA via polyA pulldown and prepared into a paired unstranded Illumina sequencing using the Kapa HyperPrep kit (Kapa Biosystems, USA), and sequenced to a depth of 43M 100 × 100 paired reads on a HiSeq2500 sequencer (Illumina, USA).

All these data were combined with previously published tissue, sex, and stage-specific libraries (**Appendix 1—table 1**) for reference-guided transcriptome assembly (Appendix 1.9.3). Strand-specific data was used for *de novo* transcriptome assembly (Appendix 1.9.2).

**Appendix 1—table 1.** *P. pyralis* RNA sequencing libraries. N: number of individuals pooled for sequencing; **Sex/stage**: M = male, F = female, A = adult, L = larva, L1 = larva 1 st instar, L4 = larvae fourth instar, E13 = 13 days post fertilization eggs; **Tissue**: H = head, PA = lantern abdominal segments, FB = abdominal fat body, T = thorax, OAG = other accessory glands, SD = spermatophore digesting gland/bursa, SG = spiral gland, SC = spermatheca, p=dissected photophore, E = egg, WB = whole body.

| Library name | Source* | SRA ID | N | Sex/stage | Tissue | Library type |
|---|---|---|---|---|---|---|
| 8175 *Photinus pyralis* male head (adult) transcriptome | SRA1 | SRR2103848 | 1 | M/A | H | |
| 8176 *Photinus pyralis* male light organ (adult) transcriptome | SRA1 | SRR2103849 | 1 | M/A | PA | |
| 8819 *Photinus pyralis* light organ (larval) transcriptome | SRA1 | SRR2103867 | 1 | L | PA | |
| 9_Photinus_sp_1_lantern | SRA2 | SRR3521424 | 1 | M/A | PA | Strand-specific. Ribo-zero |

*Appendix 1—table 1 continued on next page*

Appendix 1—table 1 continued

| Library name | Source* | SRA ID | N | Sex/stage | Tissue | Library type |
|---|---|---|---|---|---|---|
| Ppyr_FatBody_1 | SRA3 | SRR3883756 | 6 | M/A | FB | |
| Ppyr_FatBody_2 | SRA3 | SRR3883757 | 6 | M/A | FB | |
| Ppyr_FatBody_3 | SRA3 | SRR3883766 | 6 | M/A | FB | |
| Ppyr_FatBody_Mated | SRA3 | SRR3883767 | 4 | M/A | FB | |
| Ppyr_FThorax | SRA3 | SRR3883768 | 3 | F/A | T | |
| Ppyr_MThorax_1 | SRA3 | SRR3883769 | 6 | M/A | T | |
| Ppyr_MThorax_2 | SRA3 | SRR3883770 | 6 | M/A | T | |
| Ppyr_MThorax_3 | SRA3 | SRR3883771 | 6 | M/A | T | |
| Ppyr_OAG_1A | SRA3 | SRR3883772 | 6 | M/A | AG | |
| Ppyr_OAG_1B | SRA3 | SRR3883773 | 6 | M/A | AG | |
| Ppyr_OAG_2 | SRA3 | SRR3883758 | 6 | M/A | AG | |
| Ppyr_OAG_Mated | SRA3 | SRR3883759 | 4 | M/A | AG | |
| Ppyr_SDGBursa | SRA3 | SRR3883760 | 3 | F/A | SD | |
| Ppyr_SG_Mated | SRA3 | SRR3883761 | 4 | M/A | SG | |
| Ppyr_Spermatheca | SRA3 | SRR3883762 | 3 | F/A | SC | |
| Ppyr_SpiralGland_1 | SRA3 | SRR3883763 | 6 | M/A | SG | |
| Ppyr_SpiralGland_2 | SRA3 | SRR3883764 | 6 | M/A | SG | |
| Ppyr_SpiralGland_3 | SRA3 | SRR3883765 | 6 | M/A | SG | |
| Ppyr_Lantern_1A | ‡ | SRR6345453 | 6 | M/A | P | |
| Ppyr_Lantern_2 | ‡ | SRR6345454 | 6 | M/A | P | |
| Ppyr_Lantern_3 | ‡ | SRR6345446 | 6 | M/A | P | |
| Ppyr_Eggs | ‡ | SRR6345447 | 7 | E13 | E | Strand-specific |
| Ppyr_Larvae | ‡ | SRR6345445 | 4 | L1 | WB | Strand-specific |
| Ppyr_wholeFemale[†] | ‡ | SRR6345449 | 1 | F/A | WB | Strand-specific |
| Ppyr_wholeMale | ‡ | SRR6345452 | 1 | M/A | WB | Strand-specific |
| TF_VA2017_3pooled_larval_lantern | ‡ | SRR7345580 | 3 | L4 | P | |

*SRA1 = NCBI BioProject PRJNA289908 (*Sander and Hall, 2015*); SRA2 = NCBI BioProject PRJNA321737 (*Fallon et al., 2016*); SRA3 = NCBI BioProject PRJNA328865 (*Al-Wathiqui et al., 2016*).
[†]Parent of eggs and larvae with data from this study.
[‡]This study.
DOI: https://doi.org/10.7554/eLife.36495.032

## 1.9.2 *De novo* transcriptome assembly and genome alignment

One strand-specific *de novo* transcriptome was produced from all available MMNJ strand-specific reads (WholeMale, WholeFemale, eggs, larvae) and strand-specific reads from SRA (SRR3521424) (*Appendix 1—table 1*). Reads from these five libraries were pooled (158.6M paired-reads) as input for *de novo* transcriptome assembly. Transcripts were assembled using Trinity (v2.4.0) (*Grabherr et al., 2011*) with default parameters except the following: (–SS_lib_type RF –trimmomatic –min_glue 2 min_kmer_cov 2 –jaccard_clip –no_normalize_reads). Gene structures were then predicted from alignment of the *de novo* transcripts to the Ppyr1.3 genome using the PASA pipeline (v2.1.0) (*Haas et al., 2008*) with the following steps: first, poly-A tails were trimmed from transcripts using the internal seqclean component; next, transcript accessions were extracted using the accession_extractor.pl component; finally, the trimmed transcripts were aligned to the genome with modified parameters (–aligners blat,gmap –ALT_SPLICE –transcribed_is_aligned_orient –tdn tdn.accs). Using both the blat (v. 36 × 2) (*Kent, 2002*)

and gmap (v2017-09-11) (*Wu and Watanabe, 2005*) aligners was required, as an appropriate gene model for Luc2 was not correctly produced using only a single aligner. Importantly, it was also necessary to set (–NUM_BP_PERFECT_SPLICE_BOUNDARY = 0) for the validate_alignments_in_db.dbi step, to ensure transcripts with natural variation near the splice sites were not discarded. Post alignment, potentially spurious transcripts were filtered out using a custom script (*Fallon, 2017*; copy archived at https://github.com/elifesciences-publications/PASA_expression_filter_2017) that removed extremely lowly-expressed transcripts (<1% of the expression of a given PASA assembly cluster). Expression values used for filtering were calculated from the WholeMale library reads using the Trinity align_and_estimate_abundance.pl utility script. The WholeMale library was selected because it was the highest quality library - strand-specific, low contamination, and many reads - thereby increasing the reliability of the transcript quantification. Finally, the PASA pipeline was run again with this filtered transcript set to generate reliable transcript structures. Peptides were predicted from the final transcript structures using Transdecoder (v.5.0.2) (*Haas, 2018*) with default parameters. Direct coding gene models (DCGMs) were then produced with the Transdecoder 'cdna_alignment_orf_to_genome_orf.pl' utility script with the PASA assembly GFF and transdecoder predicted peptide GFF as input. The unaligned *de novo* transcriptome assembly is dubbed 'PPYR_Trinity_stranded', whereas the aligned direct coding gene models are dubbed 'Ppyr1.3_Trinity-PASA_stranded-DCGM'.

### 1.9.3 Reference guided transcriptome assembly

Two reference guided transcriptomes, one strand-specific and one non-strand-specific, were produced from all available *P. pyralis* RNA-Seq reads (*Appendix 1—table 1*) using HISAT2 (v2.0.5) (*Kim et al., 2015*) and StringTie (v1.3.3b) (*Pertea et al., 2015*). For each library, reads were first mapped to the Ppyr1.3 genome assembly with HISAT2 (parameters: -X 2000 –dta –fr) and then assembled using StringTie with default parameters except use of '–rf' for the strand-specific libraries. The resulting library-specific assemblies were then merged into a final assembly using StringTie (–merge), one for the strand-specific and one for the non-strand specific libraries, producing two final assemblies. For each final assembly, a transcript fasta file was produced and peptides predicted using Transdecoder with default parameters. Then, the StringTie. GTFs were converted to GFF format with the Transdecoder 'gtf_to_alignment_gff3.pl' utility script and direct coding gene models (DCGMs) were produced with the Transdecoder 'cdna_alignment_orf_to_genome_orf.pl' utility script, with the StringTie GFF and transdecoder predicted peptide GFF as input. The final GFFs were validated and sorted with genometools (v1.5.9) with parameters (parameters: gff3 -tidy -sort -retainids), and then sorted again for IGV format with igvtools (parameters: sort). The aligned direct coding gene models for the stranded and unstranded reference guided transcriptomes are dubbed 'Ppyr1.3_Stringtie_stranded-DCGM' and 'Ppyr1.3_Stringtie_unstranded-DCGM'.

### 1.9.4 Transcript expression analysis

*P. pyralis* RNA-Seq reads (*Appendix 1—table 1*) were pseudoaligned to the PPYR_OGS1.1 geneset CDS sequences using Kallisto (v0.44.0) (*Bray et al., 2016*) with 100 bootstraps (-b 100), producing transcripts-per-million reads (TPM). Kallisto expression quantification analysis results are available on FigShare (DOI: 10.6084/m9.figshare.5715139).

## 1.10 Official coding geneset annotation (PPYR_OGS1.1)

We annotated the coding gene structure of *P. pyralis* by integrating direct coding gene models produced from the *de novo* transcriptome (Appendix 1.9.2) and reference guided transcriptome (Appendix 1.9.3), with a lower weighted contribution of *ab initio* gene predictions, using the Evidence Modeler (EVM) algorithm (v1.1.1) (*Haas et al., 2008*). First, Augustus (v3.2.2) (*Stanke et al., 2006*) was trained against Ppyr1.2 with BUSCO (parameters: -l endopterygota_odb9 –long –species tribolium2012). Next, preliminary gene models for prediction training were produced by the alignment of the *P. pyralis de novo* transcriptome to Ppyr1.2 with the MAKER pipeline (v3.0.0β) (*Holt and Yandell, 2011*) in 'est2genome'

mode. Preliminary gene models were used to train SNAP (v2006-07-28) (**Korf, 2004**) following the MAKER instructions (**MAKER Tutorial for GMOD Online Training, 2014**). Augustus and SNAP gene predictions of Ppyr1.3 were then produced through the MAKER pipeline, with hints derived from MAKER blastx/exonerate mediated protein alignments of peptides from *Drosophila melanogaster* (NCBI GCF_000001215.4_Release_6_plus_ISO1_MT_protein.faa), *Tribolium castaneum* (NCBI GCF_000002335.3_Tcas5.2_protein), and *Aquatica lateralis* (AlatOGS1.0; this report), and MAKER blastn/exonerate transcript alignments of the *P. pyralis de novo* transcriptome. These *ab initio* coding gene models are dubbed 'Ppyr1.3_abinitio_Augustus-SNAP-MAKER-GMs.gff3'

We then integrated the *ab initio* predictions with our *de novo* and reference guided direct coding gene models, using EVM. A variety of evidence sources, and EVM evidence weights were empirically tested and evaluated using a combination of inspection of known gene models (e.g. Luc1/Luc2), and the BUSCO score of the geneset. In the final version, six sources of evidence were used for EVM: *de novo* transcriptome direct coding gene models (Ppyr1.3_Trinity-PASA_stranded-DCGM; weight = 11), protein alignments (*D. melanogaster*, *T. castaneum, A. lateralis;* weight = 8), GMAP and BLAT alignments of *de novo* transcriptome (via PASA; weight = 5), reference guided transcriptome direct coding gene models (Ppyr1.3_Stringtie_stranded-DCGM; weight = 3), Augustus and SNAP *ab initio* gene models (via MAKER; weight = 2). A custom script (**Fallon, 2018a**; copy archived at https://github.com/elifesciences-publications/maker_gff_to_evm_gff_2017) was necessary to convert MAKER GFF format to an EVM compatible GFF format.

Lastly, gene models for luciferase homologs, P450s (Appendix 1.10.1), and *de novo* methyltransferases (DNMTs) which were fragmented or were incorrect (e.g. fusions of adjacent genes) were manually corrected based on the evidence of the *de novo* and reference guided direct coding gene models. Manual correction was performed by performing TBLASTN searches with known good genes from these gene families within SequencerServer(v1.10.11) (**Priyam et al., 2015**), converting the TBLASTN results to gff3 format with a custom script (**Fallon, 2018b**; copy archived at https://github.com/elifesciences-publications/firefly_genomes_general_scripts), and viewing these alignments alongside the alternative direct coding gene models (Appendix 1.9.2; 1.9.3) in Integrative Genomics Viewer(v2.4.8) (**Thorvaldsdóttir et al., 2013**). The official gene set models gff3 file was manually modified in accordance with the evidence from the direct gene models. Different revision numbers of the official geneset (e.g. PPYR_OGS1.0, PPYR_OGS1.1) represent the improvement of the geneset over time due to these continuing manual gene annotations.

## 1.10.1 P450 annotation

Translated *de novo* transcripts were formatted to be BLAST searchable with NCBI's standalone software. The peptides were searched with 58 representative insect P450s in a batch BLAST (evalue = 10). The query set was chosen to cover the diversity of insect P450s. The top 100 hits from each search were retained. The resulting 5837 hit IDs were filtered to remove duplicates, leaving 472 unique hits. To reduce redundancy due to different isoforms, the Trinity transcript IDs (style DNXXX_cX_gX_iX) were filtered down to the 'DN' level, resulting in 136 unique IDs. All peptides with these IDs were retrieved and clustered with CD-Hit (v4.5.4) (**Li and Godzik, 2006**) to 99% identity to remove short overlapping peptides. These 535 protein sequences were batch BLAST compared to a database of all named insect P450s to identify best hits. False positives were removed and about 30 fungal sequences were removed. These fungal sequences could potentially be from endosymbiotic fungi in the gut. Overlapping sequences were combined and the transcriptome sequences were BLAST searched against the *P. pyralis* genome assembly to fill gaps and extend the sequences to the ends of the genes were possible. This approach was very helpful with the CYP4G gene cluster, allowing fragments to be assembled into whole sequences. When a new genome assembly and geneset became available, the P450s were compared to the integrated gene models in PPYR_OGS1.0. Some hybrid sequences were corrected. The final set contains 170 named cytochrome P450 sequences (166 genes, two pseudogenes).

The cytochrome P450s in insects belong to four established clans CYP2, CYP3, CYP4 and Mito (*Appendix 1—figure 13*). *P. pyralis* has about twice as many P450s as *Drosophila melanogaster* (86 genes, four pseudogenes) and slightly more than the red flour beetle *Tribolium castaneum* (137 genes, 10 pseudogenes). Pseudogenes were determined by a lack of conserved sites common to all P450s. The CYP3 clan is the largest, mostly due to three families: CYP9 (40 sequences), CYP6 (36 sequences) and CYP345 (18 sequences). Insects have few conserved sequences across species. These include the halloween genes for 20-hydroxyecdysone synthesis and metabolism CYP302A1, CYP306A1, CYP307A2, CYP314A1 and CYP315A1 (*Rewitz et al., 2007*) in the CYP2 and Mito clans. The CYP4G subfamily makes a hydrocarbon waterproof coating for the exoskeleton (*Helvig et al., 2004*). Additional conserved P450s are CYP15A1 (juvenile hormone [*Helvig et al., 2004*]) and CYP18A1 (20-hydroxyecdysone degradation [*Guittard et al., 2011*]) in the CYP2 clan. Most of the other P450s are limited to a narrower phylogenetic range. Many are unique to a single genus, although this may change as more sampling is done. It is common for P450s to expand into gene blooms (*Sezutsu et al., 2013*).

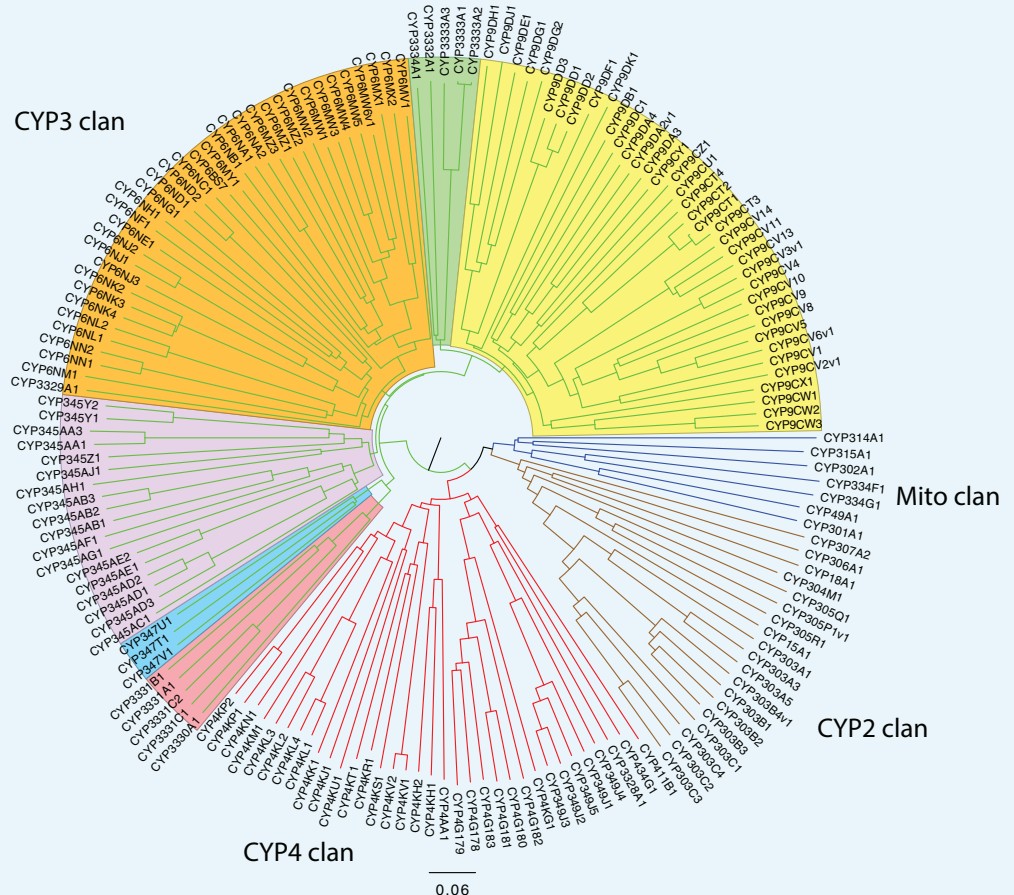

**Appendix 1—figure 13.** *P. pyralis* P450 gene phylogenetic tree. Neighbor-joining phylogenetic tree of 165 cytochrome P450s from *P. pyralis*. Four pseudogenes and one short sequence were removed. The P450 clans have colored spokes (CYP2 clan brown, CYP3 clan green, CYP4 clan red, Mito clan blue). Shading highlights different families and family clusters within the CYP3 clan. The tree was made using Clustal Omega at EBI (*European Bioinformatics Institute, 2017*) with default settings. The resulting multiple sequence alignment is available on FigShare (DOI: 10.6084/m9.figshare.5697643). The tree was drawn with FigTree v1.3.1 using midpoint rooting.
DOI: https://doi.org/10.7554/eLife.36495.033

## 1.10.2 Virus annotation and analysis

Viruses were discovered from analysis of published *P. pyralis* RNA sequencing libraries (NCBI TSA: GEZM00000000.1) and the Ppyr1.2 genome assembly (Appendix 5.4). 24 *P. pyralis* RNA sequencing libraries were downloaded from SRA (taxid: 7054, date accessed: 15th June 2017). RNA sequence reads were first *de novo* assembled using Trinity v2.4.0 (**Grabherr et al., 2011**) with default parameters. Resulting transcriptomes were assessed for similarity to known viral sequences by TBLASTN searches (max e-value = $1 \times 10^{-5}$) using as a probe the complete predicted non redundant viral Refseq proteins retrieved from NCBI (date accessed: 15th June 2017). Significant hits were explored manually and redundant contigs discarded. False-positives were eliminated by comparing candidate viral contigs to the entire non-redundant nucleotide (nt) and protein (nr) database to remove false-positives.

Candidate virus genome segment sequences were curated by iterative mapping of reads using Bowtie 2 (v2.3.2) (**Langmead and Salzberg, 2012**). Special attention was taken with the segments' terminis – an arbitrary cut off of 10x coverage was used as threshold to support terminal base calls. The complementarity and folded structure of untranslated ends, as would be expected for members of the Orthomyxoviridae, was assessed by Mfold 2.3 (**Zuker, 2003**). Further, conserved UTR sequences were identified using ClustalW2 (**Larkin et al., 2007**) (support of >65% required to call a base). To identify/rule out additional segments of no homology to the closely associated viruses we used diverse *in silico* approaches based on RNA levels including: the sequencing depth of the transcript, predicted gene product structure, or conserved genome termini, and significant co-expression with the remaining viral segments.

After these filtering steps, putative viral sequences were annotated manually. First, potential open-reading frames (ORF) were predicted by ORFfinder (**Wheeler et al., 2003**) and manually inspected by comparing predicted ORFS to those from the closest-related reference virus genome sequence. Then, translated ORFs were blasted against the non-redundant protein sequences NR database and best hits were retrieved. Predicted ORF protein sequences were also subjected to a domain-based Blast search against the Conserved Domain Database (CDD) (v3.16) (**Marchler-Bauer et al., 2017**) and integrated with SMART (**Letunic and Bork, 2018**), Pfam (**Finn et al., 2016**), and PROSITE (**Sigrist et al., 2002**) results to characterize the functional domains. Secondary structure was predicted with Garnier as implemented in EMBOSS (v6.6) (**Rice et al., 2000**), signal and membrane cues were assessed with SignalP (v4.1) (**Petersen et al., 2011**), and transmembrane topology and signal peptides were predicted by Phobius (**Käll et al., 2004**). Finally, the potential functions of predicted ORF products were explored using these annotations as well as similarity to viral proteins of known function.

To characterize *Orthomyxoviridae* viral diversity in *P. pyralis* in relation to known viruses, predicted *P. pyralis* viral proteins were used as probes in TBLASTN (max e-value = $1 \times 10^{-5}$) searches of the complete 2754 Transcriptome Shotgun Assembly (TSA) projects on NCBI (date accessed: 15th June 2017). Significant hits were retrieved and the target TSA projects further explored with the complete *Orthomyxoviridae* refseq collection to assess the presence of additional similar viral segments. Obtained transcripts were extended/curated using the SRA associated libraries for each TSA hit and then the curated virus sequences were characterized and annotated as described above.

To identify *P. pyralis* viruses to family/genus/species, amino acid sequences of the predicted viral polymerases, specifically the PB1 subunit, were used for phylogenetic analyses with viruses of known taxonomy. To do this, multiple sequence alignment were generated using MAFFT (v7.310) (**Katoh and Standley, 2013**) and unrooted maximum-likelihood phylogenetic trees were constructed using FastTree (**Price et al., 2010**) with standard parameters. FastTree accounted for variable rates of evolution across sites by assigning each site to one of 20 categories, with the rates geometrically spaced from 0.05 to 20, and set each site to its most likely rate category using a Bayesian approach with a gamma prior. Support for individual nodes was assessed using an approximate likelihood ratio test with the Shimodaira-Hasegawa-like procedure. Tree topology, support values and substitutions per site were based on 1000 tree resamples.

To facilitate taxonomic identification, we complemented BLASTP data with two levels of phylogenetic insights: (i) Trees based on the complete refseq collection of ssRNA (-) viruses which permitted a conclusive assignment at the virus family level. (ii) Phylogenetic trees based on reported, proposed, and discovered *Orthomyxoviridae* viruses that allowed tentative species demarcation and genera postulation. PB1-based trees were complemented independently with phylogenetic studies derived from amino acids of predicted nucleoproteins, hemagglutinin protein, PB2 protein, and PA protein which supported species, genera and family demarcation based on solely on PB1, the standard in *Orthomyxoviridae*. In addition, sequence similarity of concatenated gene products of International Committee on Taxonomy of Viruses (ICTV) allowed demarcation to species and firefly viruses were assessed by Circoletto diagrams(**Darzentas, 2010**) (e-value = 1e-2). Where definitive identification was not easily assessed, protein Motif signatures were determined by identification of region of high identity between divergent virus species, visualized by Sequence Logo(**Crooks et al., 2004**), and contrasted with related literature. Heterotrimeric viral polymerase 3D structure prediction was generated with the SWISS-MODEL automated protein structure homology-modeling server (**Biasini et al., 2014**) with the best fit template 4WSB: the crystal structure of Influenza A virus 4WSB. Predicted structures were visualized in UCSF Chimera (**Pettersen et al., 2004**) and Needleman-Wunsch sequence alignments from structural superposition of proteins were generated by MatchMaker and the Match->Align Chimera tool. Alternatively, 3D structures were visualized in PyMOL (v1.8.6.0; Schrodinger).

Viral RNA levels in the transcriptome sequences were also examined. Virus transcripts RNA levels were obtained by mapping the corresponding raw SRA FASTQ read pairs using either Bowtie2(**Langmead and Salzberg, 2012**) or the reference mapping tool of the Geneious 8.1.9 suite (Biomatters, Ltd.) with standard parameters. Using the mapping results and retrieving library data, absolute levels, TPMs and FPKM were calculated for each virus RNA segment. Curated genome segments and coding annotation of the identified PpyrOMLV1 and 2 are available on FigShare at (DOI: 10.6084/m9.figshare.5714806) and (DOI: 10.6084/m9.figshare.5714812) respectively, and NCBI Genbank (accessions MG972985 through MG972994)

All curation, phylogeny construction, and visualization were conducted in Geneious 8.1.9 (Biomatters, Ltd.). Animal silhouettes in *Appendix 5—figure 2* were developed based on non-copyrighted public domain images. Figure compositions were assembled using Photoshop CS5 (Adobe). Bar graphs were generated with Excel 2007 software (Microsoft). RNA levels normalized as mapped transcripts per million per library were visualized using Shinyheatmap (**Khomtchouk et al., 2017**).

Finally, to identify endogenous viral-like elements, tentative virus detections and the viral refseq collection were contrasted to the *P. pyralis* genome assembly Ppyr1.2 by BLASTX searches (e-value = 1e-6) and inspected by hand. Then 15 Kbp genome flanking regions were retrieved and annotated. Lastly, transposable elements (TEs) were determined by the presence of characteristic conserved domains (e.g. RNASE_H, RETROTRANSPOSON, INTEGRASE) on predicted gene products and/or significant best BLASTP hits to reported TEs (e-value <1e-10).

## 1.11 Repeat annotation

Repeat prediction for *P. pyralis* was performed *de novo* using RepeatModeler (v1.0.9) (**Smit and Hubley, 2017**) and MITE-Hunter (v11-2011) (**Han and Wessler, 2010**). RepeatModeler uses RECON (**Bao and Eddy, 2002**) and RepeatScout (**Price et al., 2005**) to predict interspersed repeats, and then refines and classifies the consensus repeat models to build a repeat library. MITE-Hunter detects candidate MITEs (miniature inverted-repeat transposable elements) by scanning the assembly for terminal inverted repeats and target site duplications < 2 kb apart. To identify tandem repeats, we also ran Tandem Repeat Finder (v4.09; parameters: 2 7 7 80 10) (**Benson, 1999**), and added repeats whose repeat block length was >5 kb to the repeat library annotated as 'complex tandem repeat'. The RepeatModeler and MITE-Hunter libraries were combined and classified using

RepeatClassifier (RepeatModeler 1.0.9 distribution) (*Smit and Hubley, 2017*). The complex repeats identified by Tandem Repeat Finder were added to this classified list to create the final library of 3118 repeats. This repeat library is dubbed the *P. pyralis* Official Repeat Library 1.0 (PPYR_ORL1.0).

**Appendix 1—table 2.** Annotated repetitive elements in *P. pyralis*.

| Repeat class | Family | Counts | Bases | % of assembly |
|---|---|---|---|---|
| DNA | All | 122551 | 38364685 | 8.14 |
| | Helitrons | 35068 | 9308100 | 1.97 |
| LTR | All | 28860 | 11401648 | 2.42 |
| Non-LTR | All | 52107 | 17744320 | 3.76 |
| | LINE | 48983 | 16763499 | 3.56 |
| | SINE | 1241 | 139637 | 0.03 |
| Unknown interspersed | | 696511 | 141970977 | 30.1 |
| Complex tandem repeats | | 10395 | 2352796 | 0.50 |
| Simple repeat | | 48224 | 2372183 | 0.50 |
| rRNA | | 449 | 161517 | 0.034 |

DOI: https://doi.org/10.7554/eLife.36495.034

## 1.12 *P. pyralis* methylation analysis

MethylC-seq libraries were prepared from HMW DNA prepared from four *P. pyralis* MMNJ males using a previously published protocol (*Urich et al., 2015*), and sequenced to ~36x expected depth on an Illumina NextSeq500. Methylation analysis was performed using methylpy (*Schultz et al., 2015*) Methylpy calls programs for read processing and aligning: (i) reads were trimmed of sequencing adapters using Cutadapt (*Martin, 2011*), (ii) processed reads were mapped to both a converted forward strand (cytosines to thymines) and converted reverse strand (guanines to adenines) using bowtie (flags: -S, -k 1, -m 1, – chunkmbs 3072, –best, –strata, -o 4, -e 80, -l 20, -n 0 [*Langmead et al., 2009*]), and (iii) PCR duplicates were removed using Picard (*Picard Tools, 2017*). In total, 49.4M reads were mapped corresponding to an actual sequencing depth of ~16x. A sodium bisulfite non-conversion rate of 0.17% was estimated from Lambda phage genomic DNA. Raw WGBS data can be found on the NCBI Gene Expression Omnibus (GSE107177). Previously published whole genome bisulfite sequencing (WGBS)/MethylC-seq libraries for *Apis mellifera* (*Herb et al., 2012*), *Bombyx mori* (*Xiang et al., 2010*), *Nicrophorus vespilloides* (*Cunningham et al., 2015*), and *Zootermopsis nevadensis* (*Glastad et al., 2016*) were downloaded from the Short Read Archive (SRA) using accessions SRR445803–4, SRR027157–9, SRR2017555, and SRR3139749, respectively. Libraries were subjected to identical methylation analysis as *P. pyralis*.

Weighted DNA methylation was calculated for CG sites by dividing the total number of aligned methylated reads by the total number of methylated plus un-methylated reads (*Schultz et al., 2012*). For genic metaplots, the gene body (start to stop codon), 1000 base pairs (bp) upstream, and 1000 bp downstream was divided into 20 windows proportional windows based on sequence length (bp). Weighted DNA methylation was calculated for each window and then plotted in R (v3.2.4) (*R Development Core Team, 2013*).

## 1.13 Telomere FISH analysis

We synthesized a 5' fluorescein-tagged (TTAGG)$_5$ oligo probe (FAM; Integrated DNA Technologies) for fluorescence in situ hybridization (FISH). We conducted FISH on squashed larval tissues according to previously published methods (*Larracuente and Ferree, 2015*), with some modification. Briefly, we dissected larvae in 1X PBS and treated tissues with a hypotonic solution (0.5% Sodium citrate) for 7 min. We transferred treated larval tissues to

45% acetic acid for 30 s, fixed in 2.5% paraformaldehyde in 45% acetic acid for 10 min, squashed, and dehydrated in 100% ethanol. We treated dehydrated slides with detergent (1% SDS), dehydrated again in ethanol, and then stored until hybridization. We hybridized slides with probe overnight at 30°C, washed in 4X SSCT and 0.1X SSC at 30°C for 15 min per wash. Slides were mounted in VectaShield with DAPI (Vector Laboratories), visualized on a Leica DM5500 upright fluorescence microscope at 100X, imaged with a Hamamatsu Orca R2 CCD camera. Images were captured and analyzed using Leica's LAX software.

**Appendix 1—table 3.** *Photinus pyralis* genome Experiment.com crowdfunding donors (https://experiment.com/projects/illuminating-the-firefly-genome).

| Liliana Bachrach | Doug Fambrough | Benjamin Lower | Luis Cunha | Joshua Guerriero |
|---|---|---|---|---|
| Atsuko Fish | Tom Alar | Noreen Huefner | David Esopi | John Skarha |
| Rutong Xie | Richard Hall | Zachary Michel | Jack Hynes | Keith Guerin |
| Nathan Shaner | Joe Doggett | Joe T. Bamberg | Michael McGurk | Pureum Kim |
| Sara Lewis | Mark Lewis | Lauren Solomon | Peter Berx | Milo Grika |
| Jing-Ke Weng | Sarah Sander | Dr. Husni Elbahesh | Matt Grommes | Daniel Zinshteyn |
| Peter Rodenbeck | Daniel Bear | Kathryn Larra-cuente | Colette Dedyn | Tom Brekke |
| Larry Fish | Don Salvatore | Matthew Cichocki | Florencia Schlamp | Edoardo Gianni |
| Amanda Larra-cuente | Emily Davenport | Marcel Bruchez | Marie Lower | Cindy Wu |
| Hunter Lower | Ted Sharpe | Robert Unckless | Michael R. McKain | Christina Tran |
| Allan Kleinman | David Plunkett | Arvid Ågren | Ben Pfeiffer | Eric Damon Walters |
| Misha Koksharov | Tim Fallon | Margaret S Butler | Kathryn Keho | Geoffrey Giller |
| Sarah Shekher | Edward Garrity | Yasir Ahmed-Brai-mah | Jenny Wayfarer | Fahd Butt |
| Jared Lee | Huaping Mo | Ruth Ann Grissom | Darby Thomas | Christophe Man-dy |
| Raphael De Cock | TimG | Tomáš Pluskal | Emily Hatas | |
| Linds Fallon | Jan Thys | Genome Galaxy | Richard Casey | |
| Grace Li | Francisco Martinez Gasco | Dustin Greiner | William Nicholls | |

DOI: https://doi.org/10.7554/eLife.36495.035

## Appendix 2

DOI: https://doi.org/10.7554/eLife.36495.036

### *Aquatica lateralis* additional information

#### 2.1 Taxonomy, biology, and life history

*Aquatica lateralis* (Motschulsky, 1860) (Japanese name, Heike-botaru / ヘイケボタル) is one of the most common and popular luminous insects in mainland Japan. This species is a member of the subfamily Luciolinae and had long belonged in the genus *Luciola*, but was recently moved to the new genus *Aquatica* with some other Asian aquatic fireflies (*Fu et al., 2010*).

The life cycle of *A. lateralis* is usually 1 year. Aquatic larva possesses a pair of outer gills on each abdominal segment and live in still or slow streams near rice paddies, wetlands and ponds. Larvae mainly feed on freshwater snails. They pupate in a mud cocoon under the soil near the water. Adults emerge in early to end of summer. While both males and females are full-winged and can fly, there is sexual dimorphism in adult size: the body length is about 9 mm in males and 12 mm in females (*Ohba, 2004*).

Like other firefly larvae, *A. lateralis* larvae are bioluminescent. Larvae possess a pair of lanterns at the dorsal margin of the abdominal segment 8. Adults are also luminescent and possess lanterns at true abdominal segments 6 and 7 in males and at segment six in females (*Branham and Wenzel, 2003*; *Ohba, 2004*; *Kanda, 1935*). The adult is dusk active. Male adults flash yellow-green for about 1.0 s in duration every 0.5–1.0 s while flying ~1 m above the ground. Female adults, located on low grass, respond to the male signal with flashes of 1–2 s in duration every 3–6 s. Males immediately approach females and copulate on the grass (*Ohba, 2004*; *Ohba, 1983*). Like many other fireflies, *A. lateralis* is likely toxic: both adults and larvae emit an unpleasant smell when disturbed and both invertebrate (dragonfly) and vertebrate (goby) predators vomit up the larva after biting (*Ohba and Hidaka, 2002*). *A. lateralis* larvae have eversible glands on each of the eight abdominal segments (*Fu et al., 2010*). The contents of the eversible glands is perhaps similar to that reported for *A. leii* (*Fu et al., 2007*).

#### 2.2 Species distribution

The geographical range of *A. lateralis* includes Siberia, Northeast China, Kuril Isls, Korea, and Japan (Hokkaido, Honshu, Shikoku, Kyushu, Tsushima Isls.) (*Kawashima et al., 2003*). Natural habitats of these Japanese fireflies have been gradually destroyed through human activity, and currently these species can be regarded as 'flagship species' for conservation (*Higuchi, 1996*). For example, in 2017, Japanese Ministry of Environment began efforts to protect the population of *A. lateralis* in the Imperial Palace, Tokyo, where 3000 larvae cultured in an aquarium were released in the pond beside the Palace (*Imperial Palace Outer Garden Management Office, 2017*).

#### 2.3 Specimen collection

Individuals used for genome sequencing, RNA sequencing, and LC-HRAM-MS were derived from a small population of laboratory-reared fireflies. This population was established from a few individuals collected from rice paddy in Kanagawa Prefecture of Japan in 1989 and 1990 (*Ikeya, 2016*) by Mr. Haruyoshi Ikeya, a highschool teacher in Yokohama, Japan. Mr. Ikeya collected adult *A. lateralis* specimens from their natural habitat in Yokohama and has propagated them for over 25 years (~25 generations) in a laboratory aquarium without any addition of wild individuals. This population has since been propagated in the laboratory of YO and JKW, and is dubbed the 'Ikeya-Y90' cultivar. Because of the small number of individuals used to establish the population and the number of generations of propagation, this population likely represents a partially inbred strain. Larvae were kept in aquarium at 19–

21°C and fed using freshwater snails (*Physella acuta* and *Indoplanorbis exustus*). Under laboratory rearing conditions, the life cycle is reduced to 7–8 months. The original habitat of this strain has been destroyed and the wild population which led to the laboratory strain is now extinct.

## 2.4 Karyotype and genome size

Unlike *P. pyralis*, the karyotype of *A. lateralis* is reported to be 2n = 16 with XY sex determination (male, 14A + XY; female, 14A + XX) (**Inoue and Yamamoto, 1987**). The Y chromosome is much smaller than X chromosome, and the typical behaviors of XY chromosomes, such as partial conjugation of X/Y at the first meiotic metaphase and a separation delay of X/Y at the first meiotic anaphase, were observed in testis cells (**Inoue and Yamamoto, 1987**).

We determined the genome size of *A. lateralis* using flow cytometry-mediated calibrated-fluorimetry of DNA content with propidium iodide stained nuclei. First, the head+prothorax of a single pupal female (gender identified by morphological differences in abdominal segment VIII) was homogenized in 100 µL PBS. These tissues were chosen to avoid the ovary tissue. Once homogenized, 900 µL PBS, 1 µL Triton X-100 (Sigma-Aldrich), and 4 µL 100 mg/mL RNase A (QIAGEN) were added. The homogenate was incubated at 4°C for 15 min, filtered with a 30 µm Cell Tries filter (Sysmex), and further diluted with 1 mL PBS. 20 µL of 0.5 mg/mL propidium iodide was added to the mixture and then average fluorescence of the 2C nuclei determined with a SH-800 flow cytometer (Sony, Japan). Three technical replicates of this sample were performed. Independent runs for extracted Aphid nuclei (*Acyrthosiphon pisum*; 517 Mbp), and fruit fly nuclei (*Drosophila melanogaster*; 175 Mbp) were performed as calibration standards. Genome size was estimated at 940 Mbp ±1.4 (S.D.; technical replicates = 3).

Genome size inference via Kmer spectral analysis estimated a genome size of 772 Mbp (**Appendix 2—figure 1**).

## 2.5 Genomic sequencing and assembly

Genomic DNA was extracted from the whole body of a single laboratory-reared *A. lateralis* adult female (c.v. Ikeya-Y90) using the QIAamp Kit (Qiagen). Purified DNA was fragmented with a Covaris S2 sonicator (Covaris, Woburn, MA), size-selected with a Pippin Prep (Sage Science, Beverly, MA), and then used to create two paired-end libraries using the TruSeq Nano Sample Preparation Kit (Illumina) with insert sizes of ~200 and~800 bp. These libraries were sequenced on an Illumina HiSeq1500 using a 125 × 125 paired-end sequencing protocol. Mate-pair libraries of 2–20 Kb with a peak at ~5 Kb were created from the same genomic DNA using the Nextera Mate Pair Sample Preparation Kit (FC-132–1001, Illumina), and sequenced on HiSeq 1500 using a 100 × 100 paired-end sequencing protocol at the NIBB Functional Genomics Facility (Aichi, Japan). In total, 133.3 Gb of sequence (159x) was generated.

Reads were assembled using ALLPATHS-LG (build# 48546) (**Gnerre et al., 2011**), with default parameters and the 'HAPLOIDIFY = True' option. Scaffolds were filtered to remove non-firefly contaminant sequences using blobtools (**Laetsch and Blaxter, 2017**), resulting in the final assembly (Alat1.3). The final assembly (Alat1.3) consists of 5388 scaffolds totaling 908.5 Gbp with an N50 length of 693.0 Kbp, corresponding to 96.6% of the predicted genome size of 940 Mbp based on flow cytometry (Appendix 2.4). Genome sequencing library statistics are available in **Appendix 4—table 1**.

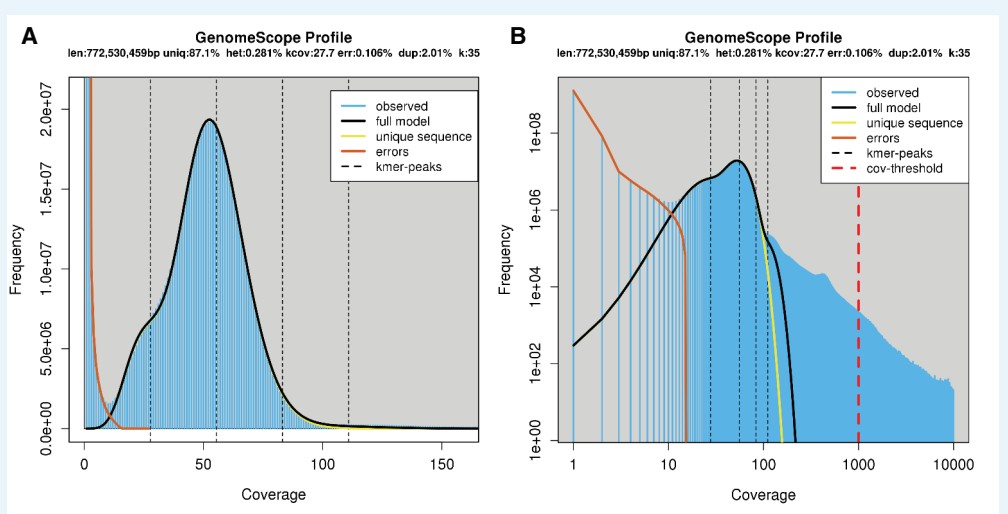

**Appendix 2—figure 1.** Genome scope kmer analysis of the *A. lateralis* short-insert genomic library. (**A**) Linear and (**B**) log plot of a kmer spectral genome composition analysis of the 'FFGPE_PE200' *A. lateralis* Illumina short-insert library (Appendix 2.5; *Appendix 4—table 1*) with jellyfish (v2.2.9; parameters: -C -k 35) (*Marçais and Kingsford, 2011*) and GenomeScope (v1.0; parameters: Kmer length = 35, Read length = 100, Max kmer coverage = 1000) (*Vurture et al., 2017*). len = inferred haploid genome length, uniq = percentage non-repetitive sequence, het = overall rate of genome heterozygosity, kcov = mean kmer coverage for heterozygous bases, err = error rate of the reads, dup: average rate of read duplications. These results are consistent when considering the possible systematic error of kmer spectral analysis and flow cytometry genome size estimates. The heterozygosity is lower than that measured for *P. pyralis*, possibly reflecting the long-term laboratory rearing in reduced population sizes of *A. lateralis* strain Ikeya-Y90.
DOI: https://doi.org/10.7554/eLife.36495.037

## 2.5.2 Taxonomic annotation filtering

Potential contaminants in Alat1.2 were identified using the blobtools toolset (v1.0) (*Laetsch and Blaxter, 2017*). First, scaffolds were compared to known sequences by performing a blastn (v2.5.0+) nucleotide sequence similarity search against the NCBI nt database and a diamond (v0.9.10) (*Buchfink et al., 2015*) translated nucleotide sequence similarity search against the of Uniprot reference proteomes (July 2017). Using this similarity information, scaffolds were annotated with blobtools (parameters '-x bestsumorder'). We also inspected the read coverage by mapping the paired-end reads (FFGPE_PE200) on the genome using bowtie2. A tab delimited text file containing the results of this blobtools annotation are available on FigShare (DOI: 10.6084/m9.figshare.5688928). The contigs derived from potential contaminants and/or poor quality contigs were then removed: contigs with higher %GC (>50%) with bacterial hits or no database hits and showing low read coverage (<30 x) (see *Appendix 2—figure 2*). This process removed 1925 scaffolds (1.17 Mbp), representing 26.3% of the scaffold number and 1.3% of the nucleotides of Alat1.2, producing the final filtered assembly, dubbed Alat1.3.

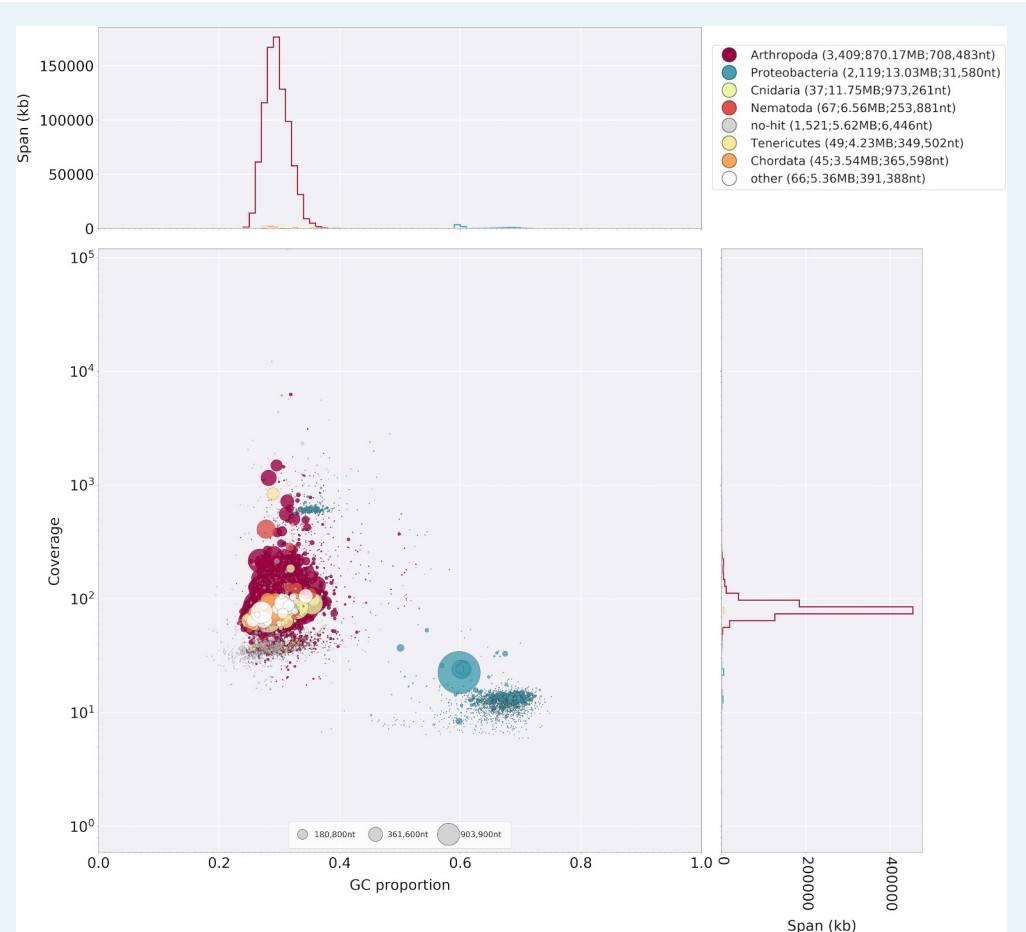

**Appendix 2—figure 2.** Blobplot of *A. lateralis* Illumina reads aligned against Alat1.2. Coverage shown represents mean coverage of reads from the Illumina short-insert library (Sample name FFGPE_PE200; *Appendix 4—table 1*), aligned against Alat1.2 using Bowtie2. Scaffolds were taxonomically annotated as described in Appendix 2.5.2.
DOI: https://doi.org/10.7554/eLife.36495.038

## 2.6 RNA-extraction, library preparation and sequencing

In order to capture transcripts from diverse life-stages and tissues, non-stranded RNA-Seq libraries were prepared from fresh specimens of nine life stages/sexes/tissues (eggs, fifth (the last) instar larvae, both sex of pupae, adult male head, male abdomen (prothorax-to-fifth segment), male lantern, adult female head, and female lantern (*Appendix 2—table 1*). Live specimens were anesthetized on ice and dissected during the day. The lantern tissue was dissected from the abdomen and contains the cuticle, photocyte layer and reflector layer. For eggs, larvae, and pupae, total RNA was extracted using the RNeasy Mini Kit (QIAGEN) with the optional on-column DNase treatment. For adult specimens, total RNA was extracted using TRIzol reagent (Invitrogen) to avoid contamination of pigments and uric acid. These were then treated with DNase in solution and then cleaned using a RNeasy Mini kit.

cDNA libraries were generated from purified Total RNA (500 ng from each sample) using a TruSeq RNA Sample Preparation Kit v2 (Illumina) according to the manufacturer's protocol (Low-Throughput Protocol), except that all reactions were carried out at half scale. The fragmentation of mRNA was performed for 4 min. The enrichment PCR was done using six cycles. A subset of nine libraries (BdM1, HeF1, HeM1, LtF1, LtM1, Egg1, Lrv1, PpEF, PpLM; *Appendix 2—table 1*) were multiplexed and sequenced in a single lane of Hiseq1500 101 × 101 bp paired-end reads. The remaining 23 libraries (BdM2, BdM3, HeF2, HeF3, HeM2, HeM3, LtF2, LtF3, LtM2, LtM3, WAF1, WAF2, WAF3, WAM1, WAM2, WAM3, Egg2, Lrv2,

Lrv3, PpEM, PpLF, PpMF, PpMM) were multiplexed and sequenced in two lanes of Hiseq1500 66 bp single-end reads. Sequence quality was inspected with FastQC (*Andrews, 2017*).

**Appendix 2—table 1.** *Aquatica lateralis* RNA sequencing. **N**: number of individuals pooled for sequencing; **Sex/stage**: M = male, F = female, A = adult, L = larva, L = larvae, E = Eggs, p=Pupae, P-E = Pupae early, P-M = Pupae middle, P-L = Pupae late; **Tissue**: H = head, La = dissected lantern containing cuticle, photocyte layer and reflector layer, H = head, B = Thorax, plus abdomen excluding lantern containing segments. W = whole specimen. AEL = After egg laying.

| Library name | Label | SRA ID | N | Sex/ Stage | Tissue | Library type |
|---|---|---|---|---|---|---|
| R102L6_idx13 | BdM1 | DRR119264 | 1 | M/A | B | Illumina paired-end, non-stranded specific, PolyA |
| R128L1_idx25 | BdM2 | DRR119265 | 1 | M/A | B | Illumina single-end, non-stranded specific, PolyA |
| R128L2_idx27 | BdM3 | DRR119266 | 1 | M/A | B | Illumina single-end, non-stranded specific, PolyA |
| R102L6_idx15 | HeF1 | DRR119267 | 3 | F/A | H | Illumina paired-end, non-stranded specific, PolyA |
| R128L1_idx22 | HeF2 | DRR119268 | 3 | F/A | H | Illumina single-end, non-stranded specific, PolyA |
| R128L2_idx23 | HeF3 | DRR119269 | 3 | F/A | H | Illumina single-end, non-stranded specific, PolyA |
| R102L6_idx12 | HeM1 | DRR119270 | 2 | M/A | H | Illumina paired-end, non-stranded specific, PolyA |
| R128L1_idx20 | HeM2 | DRR119271 | 2 | M/A | H | Illumina single-end, non-stranded specific, PolyA |
| R128L2_idx21 | HeM3 | DRR119272 | 2 | M/A | H | Illumina single-end, non-stranded specific, PolyA |
| R102L6_idx16 | LtF1 | DRR119273 | 5 | F/A | La | Illumina paired-end, non-stranded specific, PolyA |
| R128L1_idx06 | LtF2 | DRR119274 | 5 | F/A | La | Illumina single-end, non-stranded specific, PolyA |
| R128L2_idx12 | LtF3 | DRR119275 | 5 | F/A | La | Illumina single-end, non-stranded specific, PolyA |
| R102L6_idx14 | LtM1 | DRR119276 | 5 | M/A | La | Illumina paired-end, non-stranded specific, PolyA |
| R128L1_idx05 | LtM2 | DRR119277 | 5 | M/A | La | Illumina single-end, non-stranded specific, PolyA |
| R128L2_idx19 | LtM3 | DRR119278 | 5 | M/A | La | Illumina single-end, non-stranded specific, PolyA |
| R128L2_idx15 | WAF1 | DRR119279 | 1 | F/A | W | Illumina single-end, non-stranded specific, PolyA |
| R128L1_idx16 | WAF2 | DRR119280 | 1 | F/A | W | Illumina single-end, non-stranded specific, PolyA |
| R128L2_idx18 | WAF3 | DRR119281 | 1 | F/A | W | Illumina single-end, non-stranded specific, PolyA |
| R128L1_idx11 | WAM1 | DRR119282 | 1 | M/A | W | Illumina single-end, non-stranded specific, PolyA |
| R128L2_idx13 | WAM2 | DRR119283 | 1 | M/A | W | Illumina single-end, non-stranded specific, PolyA |
| R128L1_idx14 | WAM3 | DRR119284 | 1 | M/A | W | Illumina single-end, non-stranded specific, PolyA |

*Appendix 2—table 1 continued on next page*

*Appendix 2—table 1 continued*

| Library name | Label | SRA ID | N | Sex/ Stage | Tissue | Library type |
|---|---|---|---|---|---|---|
| R102L6_idx4 | Egg1 | DRR119285 | 19.6 mg (~30–50) | E ~6 hr AEL | W | Illumina paired-end, non-stranded specific, PolyA |
| R128L1_idx01 | Egg2 | DRR119286 | 21.6 mg (~30–50) | E ~7 d AEL | W | Illumina single-end, non-stranded specific, PolyA |
| R102L6_idx5 | Lrv1 | DRR119287 | 1 | L | W | Illumina paired-end, non-stranded specific, PolyA |
| R128L1_idx03 | Lrv2 | DRR119288 | 1 | L | W | Illumina single-end, non-stranded specific, PolyA |
| R128L2_idx04 | Lrv3 | DRR119289 | 1 | L | W | Illumina single-end, non-stranded specific, PolyA |
| R128L1_idx07 | PpEM | DRR119290 | 1 | M/P-E | W | Illumina single-end, non-stranded specific, PolyA |
| R128L2_idx10 | PpLF | DRR119291 | 1 | F/P-L | W | Illumina single-end, non-stranded specific, PolyA |
| R128L1_idx09 | PpMF | DRR119292 | 1 | F/P-M | W | Illumina single-end, non-stranded specific, PolyA |
| R128L2_idx08 | PpMM | DRR119293 | 1 | M/P-M | W | Illumina single-end, non-stranded specific, PolyA |
| R102L6_idx7 | PpEF | DRR119294 | 1 | F/P-E | W | Illumina paired-end, non-stranded specific, PolyA |
| R102L6_idx6 | PpLM | DRR119295 | 1 | M/P-L | W | Illumina paired-end, non-stranded specific, PolyA |

DOI: https://doi.org/10.7554/eLife.36495.039

## 2.7 Transcriptome analysis

### 2.7.1 *De novo* transcriptome assembly and alignment

To build a comprehensive set of reference transcript sequences, reads derived from the pool of nine libraries (BdM1, HeF1, HeM1, LtF1, LtM1, Egg1, Lrv1, PpEF, PpLM; *Appendix 2—table 1*) were pooled. These represent RNA prepared from various tissues (head, thorax + abdomen, lantern) and stages (egg, pupae, adult) of both sexes. A non strand-specific *de novo* transcriptome assembly was produced with Trinity (v2.6.6) (*Grabherr et al., 2011*) using default parameters exception the following: (–min_glue 2 min_kmer_cov 2 –jaccard_clip –no_normalize_reads –trimmomatic). Peptides were predicted from the *de novo* transcripts via Transdecoder (v5.3.0; default parameters). *De novo* transcripts were then aligned to the *A. lateralis* genome (Alat1.3) using the PASA pipeline with blat (v36 × 2) and gmap (v2018-05-03) (–aligners blat,gmap), parameters for alternative splice analysis and strand specificity (–ALT_SPLICE –transcribed_is_aligned_orient), and input of the previously extracted Trinity accessions (–tdn tdn.accs). Importantly, it was necessary to set (–NUM_BP_PERFECT_SPLICE_BOUNDARY = 0) for the validate_alignments_in_db.dbi step, to ensure transcripts with natural variation near the splice sites were not discarded. Direct coding gene models (DCGMs) were then produced with the Transdecoder 'cdna_alignment_orf_to_genome_orf.pl' utility script, with the PASA assembly GFF and transdecoder predicted peptide GFF as input. The unaligned *de novo* transcriptome assembly is dubbed 'AQULA_Trinity_unstranded', whereas the aligned direct coding gene models are dubbed 'Alat1.3_Trinity_unstranded-DCGM'.

### 2.7.2 Reference guided transcriptome alignment and assembly

A reference guided transcriptome was produced from all available *A.lateralis* RNA-seq reads (*Appendix 2—table 1*) using HISAT2 (v2.1.0) (*Kim et al., 2015*) and StringTie (v1.3.3b) (*Pertea et al., 2015*). Reads were first mapped to the *A. lateralis* genome (Alat1.3) with HISAT2 (parameters: -X 2000 –dta –fr). Then StringTie assemblies were performed on each separate bam file corresponding to the original libraries using default parameters. Finally, the produced. GTF files were merged using StringTie (–merge). A transcript fasta file was produced from the StringTie GTF file with the transdecoder 'gtf_genome_to_cdna_fasta.pl' utility script, and peptides were predicted for these transcripts using Transdecoder (v5.3.0) with default parameters. The StringTie GTF was converted to GFF format with the Transdecoder 'gtf_to_alignment_gff3.pl' utility script, and direct coding gene models (DCGMs) were then produced with the Transdecoder 'cdna_alignment_orf_to_genome_orf. pl' utility script, with the StringTie-provided GFF and transdecoder predicted peptide GFF as input. The reference guided transcriptome assembly was dubbed 'AQULA_Stringtie_unstranded', whereas the aligned direct coding gene models were dubbed 'Alat1.3_Stringtie_unstranded-DCGM'.

### 2.7.3 Transcript expression analysis

*A. lateralis* RNA-Seq reads (*Appendix 2—table 1*) were pseudoaligned to the AQULA_OGS1.0 geneset mRNAs using Kallisto (v0.43.1) (*Bray et al., 2016*) with 100 bootstraps (-b 100), producing transcripts-per-million reads (TPM). Kallisto expression quantification analysis results are available on FigShare (DOI: 10.6084/m9.figshare.5715139).

## 2.8 Official coding geneset annotation (AQULA_OGS1.0)

A protein-coding gene reference set for *A. lateralis* was generated by Evidence Modeler (v1.1.1) using both aligned transcripts and aligned proteins. For transcripts, we combined reference guided and *de novo* transcriptome assembly approaches. Notably, these reference guided and *de novo* transcriptome assembly approaches differed from the current *de novo* (Appendix 2.7.1) and reference guided (Appendix 2.7.2) transcriptome assembly approaches. In the reference-guided approach applied here, RNA-Seq reads were mapped to the genome assembly with TopHat and assembled into transcripts with Cufflinks (parameters: –min-intron-length 30) (*Trapnell et al., 2010*). The Cufflinks transcripts were subjected to the TransDecoder program to extract ORFs. In the *de novo* transcriptome approach applied here, RNA-seq reads were assembled *de novo* by Trinity and ORFs were predicted using TransDecoder. We used CD-HIT-EST (*Li and Godzik, 2006*) to reduce the redundancy of the predicted ORFs. The ORF sequences were mapped to the genome using Exonerate in est2genome mode for splice-aware alignment. We processed homology evidence at the protein level using the reference proteomes of *D. melanogaster* and *T. castaneum*. These reference proteins were split-mapped to the *A. lateralis* genome in two steps: first with BLASTX to find approximate loci, and then with Exonerate in protein2genome mode to obtain more refined alignments. These gene models derived from multiple evidence were merged by the EVM program to obtain the reference annotation for the genomes. We also predicted *ab initio* gene models using Augustus, but we didn't include Augustus models for the EVM integration because our preliminary analysis showed the *ab initio* gene models had no positive impact on gene prediction.

Lastly, gene models for luciferase homologs, P450s, and *de novo* methyltransferases (DNMTs) which were fragmented or were incorrect (e.g. fusions of adjacent genes) were manually corrected based on the evidence of the *de novo* and reference guided direct coding gene models. Manual correction was performed by performing TBLASTN searches with known good genes from these gene families within SequencerServer(v1.10.11) (*Priyam et al., 2015*), converting the TBLASTN results to gff3 format with a custom script (*Fallon, 2018b*), and viewing these alignments alongside the alternative direct coding gene models (Appendix 2.7.1; 2.7.2) in Integrative Genomics Viewer(v2.4.8) (*Thorvaldsdóttir et al., 2013*). The official gene set. gff3 file was manually modified in accordance with the alternative gene models. Different revision numbers of the official

geneset (e.g. AQULA_OGS1.0, AQULA_OGS1.1) represent the improvement of the geneset over time due to these continuing manual gene annotations.

## 2.9 Repeat annotation

A *de novo* species-specific repeat library for *A. lateralis* was constructed using RepeatModeler (v1.0.9), and Tandem Repeat Finder (v4.09; settings: 2 7 7 80 10) (**Benson, 1999**). Only tandem repeats from Tandem Repeat Finder with a repeat block length >5 kb (annotated as 'complex tandem repeat') were added to the RepeatModeler library. This process yielded a final library of 1695 interspersed repeats. We then used this library and RepeatMasker (v4.0.5) (**Smit et al., 2015**) to identify and mask interspersed and tandem repeats in the genome assembly. This repeat library is dubbed the *Aquatica lateralis* Official Repeat Library 1.0 (AQULA_ORL1.0).

**Appendix 2—table 2.** Annotated repetitive elements in *A. lateralis*.

| Repeat class | Family | Counts | Bases | % of assembly |
|---|---|---|---|---|
| DNA | All | 229064 | 73263593 | 8.06 |
| | Helitrons | 930 | 466679 | 0.051 |
| LTR | All | 59499 | 23391956 | 2.57 |
| Non-LTR | All | 151788 | 50394853 | 5.55 |
| | LINE | 151788 | 50394853 | 5.55 |
| | SINE | 0 | 0 | 0 |
| Unknown interspersed | | 450934 | 99998958 | 11.01 |
| Complex tandem repeats | | 295 | 33237 | 0.004 |
| Simple repeat | | 155265 | 6656757 | 0.73 |
| rRNA | | 0 | 0 | 0 |

DOI: https://doi.org/10.7554/eLife.36495.040

## Appendix 3

DOI: https://doi.org/10.7554/eLife.36495.041

### *Ignelater luminosus* additional information

#### 3.1 Taxonomy, biology, and life history

*Ignelater luminosus* is a member of the beetle family Elateridae ('click beetles'), related to Lampyridae within the superfamily Elateroidea. The Elateridae includes about 10,000 species (*Slipinski et al., 2011*) (17 subfamilies) (*Costa et al., 2010*), which are widespread throughout the globe. Unlike in fireflies, where bioluminescence is universal, only ~200 described elaterid species are luminous. These luminous species are recorded only from tropical and subtropical regions of Americas and some small Melanesian islands, such as Fiji and Vanuatu (*Costa, 1975*; *Costa et al., 2010*). For instance, the tropical American *Pyrophorus noctilucus* is considered the largest (~30 mm) and brightest bioluminescent insect (*Harvey and Stevens, 1928*; *Levy, 1998*). All luminous species are closely related - luminous click beetles belong to the tribes Pyrophorini and Euplinthini (*Costa, 1975*; *Arias-Bohart, 2015*) of the subfamily Agrypninae, with the single exception of *Campyloxenus pyrothorax* (Chile) in the related subfamily Campyloxeninae (*Stibick, 1979*). The luminescence of a pair of pronotal 'light organs' of the adult *Balgus schnusei* (*Costa, 1984*), a species that has now been assigned to the Thylacosterninae of the Elateridae (*Costa et al., 2010*), has not been confirmed by later observation. This near-monophyly of bioluminescent elaterid taxa is supported by both morphological (*Douglas, 2011*) and molecular phylogenetic analysis (*Sagegami-Oba et al., 2007*; *Oba and Sagegami-Oba, 2007*; *Kundrata and Bocak, 2011*), although early morphological phylogenies were inconsistent (*Stibick, 1979*; *Hyslop, 1917*; *Ohira, 1962*; *Dolin, 1978*; *Ohira, 2013*). This suggests a single origin of bioluminescence in this family.

The genus *Ignelater* was established by Costa in 1975 and *I. luminosus* was included in this genus (*Costa, 1975*). Often this species is called *Pyrophorus luminosus* as an 'auctorum', a name used to describe a variety of taxa (*Johnson, 2002*). This use of 'Pyrophorus' as an auctorum may be due to the heightened difficulty of classifying Elateridae (*Costa, 1975*). The genus *Ignelater* is characterized by the presence of both dorsal and ventral photophores (*Costa, 1975*; *Rosa, 2007*). An unreviewed report suggested that the adult *I. luminosus* has a ventral light organ only in males (*Reyes and Lee, 2010*). Phylogenetic analyses based on the morphological characters suggested that the genera *Ignelater* and *Photophorus* (which contain only two species from Fiji and Vanuatu) are the most closely related genera in the tribe Pyrophorini (*Rosa, 2007*). The earliest fossil of an Elateridae species was recorded from the Middle Jurassic of Inner Mongolia, China (*Chang et al., 2009*). McKenna and Farrell suggested that, based on molecular analyses, the family Elateridae originated in the Early Cretaceous (130 Mya) (*McKenna and Farrell, 2009*). It is expected that many recent genera in Elateroidea were established by the Early Tertiary (<65 Mya) (*Grimaldi and Engel, 2005*).

The exact function of bioluminescence across different life stages remains unknown for many luminous elaterid species. Bioluminescent elaterid beetles typically have two paired lanterns on the dorsal surface of the prothorax, and a single lantern on the ventral abdomen, which is only exposed during flight. Several bioluminescent Elateridae produce different colored luminescence from their prothorax and abdominal lanterns (*Oba et al., 2010a*; *Feder and Velez, 2009*). Harvey reported that there was not a marked difference in the luminescence color of the dorsal and ventral lanterns of Puerto Rican *I. luminosus* (*Harvey, 1952*). Like fireflies, elaterid larvae often produce light, with the glowing termite mounds of Brazil that contain the predatory larvae of *Pyrearinus termitilluminans* being a striking example (*Costa and Vanin, 2010*). A description of the anatomy of the larval light organ of *Pyrophorus* is provided by (*Harvey, 1952*), and a more modern photograph of the larval light organ is provided by (*Bechara and Stevani, 2018*). Like other bioluminescent elaterid larvae, *I. luminosus* larvae produce a diffuse light from their prothorax, however they are only luminous when disturbed (*Wolcott, 1948*). *I. luminosus* larvae are subterranean predators and are an enthusiastic predator of the white grub (*Ancylonycha* spp.), reportedly

consuming 50 + to reach full size (**Wolcott, 1950**). Adult *I. luminosus* are luminescent and a bioluminescent courtship behavior was described in an unreviewed study (**Kretsch, 2000**). Reportedly, males search during flight with their prothorax lanterns illuminated steadily, while females stay on the ground modulating the intensity of their prothorax lanterns in ~2 s intervals. Once a female is observed, the prothorax lanterns of the male go dark, the ventral lantern becomes illuminated, and the male approaches the female via a circular search pattern. Mating is brief, reportedly taking only 5 seconds.

Unlike fireflies, bioluminescent elaterid species are not known to have potent chemical defenses. For example, the Jamaican bioluminescent elaterid beetle *Pyrophorus plagiophthalmus*, does not appear to be strongly unpalatable, as bats were observed to regularly capture the beetles during their flying bioluminescent displays (**Vélez, 2006**). A defense role for *I. luminosus* luminescence to startle predators is possible.

## 3.2 Species distribution

*I. luminosus* is often considered to be endemic to Puerto Rico (**Virkki et al., 1984**); however, the genus *Ignelater* is reported in Florida (USA), Vera Cruz (Mexico), the Bahamas, Cuba, Isla de la Juventud, Hispaniola (Haiti + Dominican Republic), Puerto Rico, and the Lesser Antilles (**Costa, 1975**). Similarly, *I. luminosus* itself has been reported on the island of Hispaniola (**Kretsch, 2000**; **Perez-Gelabert, 2008**), indicating *I. luminosus* is not restricted to Puerto Rico. This geographic distribution of *Ignelater* suggests that Puerto Rico may contain multiple *Ignelater* species and, given the difficulty of distinguishing different species of bioluminescent Elateridae by morphological characters, a definitive species distribution for *I. luminosus* cannot be stated, other than this species is seemingly not strictly endemic to Puerto Rico.

## 3.3 Collection

*I. luminosus* (Illiger, 1807) adult specimens were collected from private land in Mayagüez, Puerto Rico (18° 13' 12.1974' N, 67° 6' 31.6866' W) with permission of the landowner by Dr. David Jenkins (USDA-ARS). Individuals were captured at night on April 20th and April 28th 2015 during flight on the basis of light production. The *I. luminosus* specimens were frozen in a −80°C freezer, lyophilized, shipped to the laboratory (MIT) on dry ice, and stored at −80 °C. Full collection metadata is available from the NCBI BioSample records of these specimens (NCBI Bioproject PRJNA418169). Identification to species was performed by comparing antenna and dissected genitalia morphology to published keys (**Costa, 1975**; **Rosa, 2007**; **Rosa, 2010**) (**Appendix 3—figure 1**). All inspected specimens were male (3/3). Specimens collected at the same time, but not those used for genitalial dissection, were used for sequencing. Although the genitalia morphology of the sequenced specimens was not inspected to confirm their sex, sequenced specimens were inferred to be male, based on the fact that female bioluminescent elaterid beetles are rarely seen in flight (Personal communication: S. Velez) and the dissected specimens collected in the same batch as the sequenced specimens were confirmed to be male.

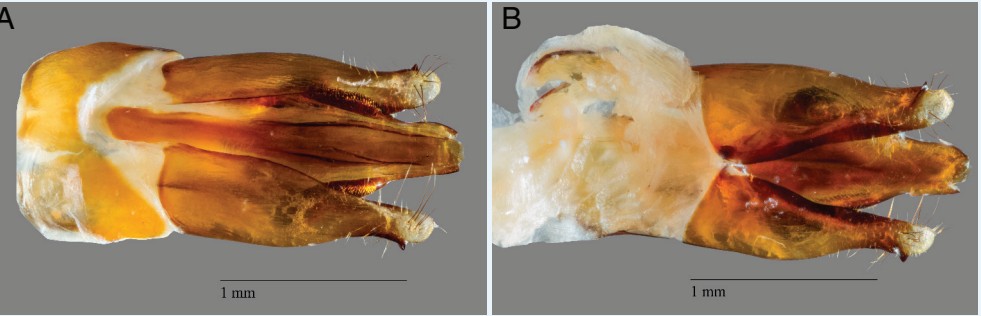

**Appendix 3—figure 1.** *I. luminosus* aedeagus (male genitalia). (**A**) Dorsal and (**B**) ventral view of

an *Ignelater luminosus* aedeagus, dissected from the same batch of specimens used for linked-read sequencing and genome assembly. The species identity of this specimen was confirmed as *I. luminosus* by comparison of the aedeagus to the keys of Costa and Rosa (**Costa, 1975**; **Rosa, 2007**; **Rosa, 2010**).

DOI: https://doi.org/10.7554/eLife.36495.042

## 3.4 Karyotype and genome size

The karyotype of male Puerto Rican *I. luminosus* (as *Pyrophorus luminosus*) was reported as $2n = 14A + X_1X_2Y$ (**Virkki et al., 1984**). The genome sizes of 5 male *I. luminosus* were determined by flow cytometry-mediated calibrated-fluorimetry of DNA content with propidium iodide stained nuclei by Dr. J. Spencer Johnston (Texas A&M University). The frozen head of each individual was placed into 1 mL of cold Galbraith buffer in a 1 mL Kontes Dounce Tissue Grinder along with the head of a female *Drosophila virilis* standard (1C = 328 Mbp). The nuclei from the sample and standard were released with 15 strokes of the 'B' (loose) pestle, filtered through 40 μm Nylon mesh, and stained with 25 mg/mL Propidium Iodide (PI). After a minimum of 30 min staining in the dark and cold, the average fluorescence channel number for the PI (red) fluorescence of the 2C (diploid) nuclei of the sample and standard were determined using a CytoFlex Flow Cytometer (Beckman-Coulter). The 1C amount of DNA in each sample was determined as the ratio of the 2C channel number of the sample and standard times 328 Mbp. The genome size of these *I. luminosus* males was determined to be 764 ± 7 Mbp (SEM, n = 5). Genome size inference via Kmer spectral analysis of the *I. luminosus* linked-read data estimated a genome size of 841 Mbp (**Appendix 3—figure 2**).

## 3.5 Genomic sequencing and assembly

HMW DNA (25 μg) was extracted from a single male specimen of *I. luminosus* using a 100/G Genomic Tip with the Genomic buffers kit (Qiagen, USA). The *I. luminosus* specimen was first washed with 95% ethanol, and DNA was extracted following the manufacturer's protocol, with the exception of the final precipitation step, where HMW DNA was pelleted with 40 μg RNA grade glycogen (Thermo Scientific, USA) and centrifugation (3000 x g, 30 min, 4°C) instead of spooling on a glass rod. HMW DNA was sent on dry-ice to the Hudson Alpha Institute of Biotechnology Genomic Services Lab (HAIB-GSL), where pulsed-field-gel-electrophoresis (PFGE) quality control and 10x Genomics Chromium Genome v1 library construction was performed. PFGE quality control indicated the mean size of the input DNA was >35 kbp+. The resulting library was then sequenced on one HiSeqX lane. 408,838,927 paired reads (150 × 150 PE) were produced, corresponding to a genomic coverage of 153x. To evaluate the effect of different Ilumina instruments on data and assembly quality, the library was also sequenced on one HiSeq2500 lane, where 145,250,480 reads (150 × 150 PE) were produced, corresponding to a genomic coverage of 54x. A summary of the library statistics for the genomic sequencing is available in **Appendix 4—table 1**. The draft genome of *I. luminosus* (Ilumi1.0) was assembled from the obtained HiSeqX genomic sequencing reads using the Supernova assembler (v1.1.1) (**Weisenfeld et al., 2017**), on a 40 core 1 TB RAM server at the Whitehead Institute for Biomedical Research. The reported mean molecule size was 12.23 kbp. The assembly was exported to FASTA format using Supernova mkoutput (parameters: –style=pseudohap), and modified by taxonomic annotation filtering (Appendix 3.5.2) and polishing (Appendix 3.5.3) to form Ilumi1.1. A Supernova (v2.0.0) assembly was also produced from combined HiSeqX and HiSeq2500 reads, but on a brief inspection the quality was equivalent to Ilumi1.1, so the new assembly was not used for further analyses. Manual long-read based scaffolding was then applied to produce a final assembly Ilumi1.2 (Appendix 3.5.4).

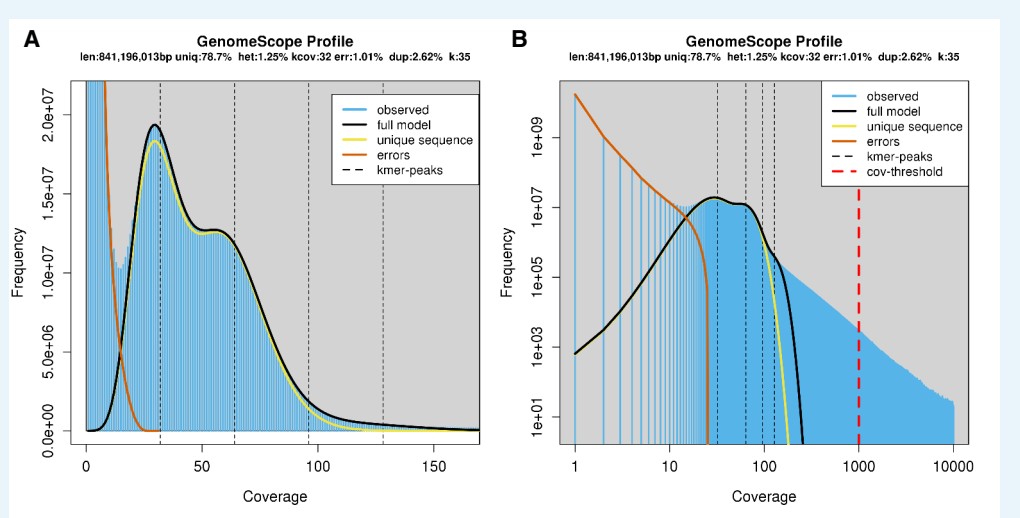

**Appendix 3—figure 2.** Genome scope kmer analysis of the *I. luminosus* linked-read genomic library. (**A**) Linear and (**B**) log plot of a kmer spectral genome composition analysis of the '1610_IlumiHiSeqX' *I. luminosus* Illumina linked-read library (Appendix 2.5; *Appendix 4—table 1*) with jellyfish (v2.2.9; parameters: -C -k 35) (*Marçais and Kingsford, 2011*) and GenomeScope (v1.0; parameters: Kmer length = 35, Read length = 138, Max kmer coverage = 1000) (*Vurture et al., 2017*). Before analysis, 10x Chromium barcodes were trimmed off Read1 using cutadapt (v1.8; parameters: -u 23) (*Martin, 2011*). vlen = inferred haploid genome length, uniq = percentage non-repetitive sequence, het = overall rate of genome heterozygosity, kcov = mean kmer coverage for heterozygous bases, err = error rate of the reads, dup: average rate of read duplications. These results are consistent when considering the possible systematic error of kmer spectral analysis and flow cytometry genome size estimates. The heterozygosity is higher than that measured for *P. pyralis* and *A. lateralis*. The read error rate for this library is also significantly higher than the *P. pyralis* and *A. lateralis* results, possibly highlighting the difference in raw read error rate between HiSeq2500 and HiSeqX sequencing, or is possibly an artifact of the Chromium library.
DOI: https://doi.org/10.7554/eLife.36495.043

## 3.5.2 Taxonomic annotation filtering

We sought to systematically remove assembled non-elaterid contaminant sequence from Ilumi1.0. Using the blobtools toolset (v1.0.1), (*Laetsch and Blaxter, 2017*), we taxonomically annotated our scaffolds by performing a blastn (v2.6.0+) nucleotide sequence similarity search against the NCBI nt database, and a diamond (v0.9.10.111) (*Buchfink et al., 2015*) translated nucleotide sequence similarity search against the of Uniprot reference proteomes (July 2017). Using this similarity information, we taxonomically annotated the scaffolds with blobtools using parameters '-x bestsumorder –rank phylum' (*Appendix 3—figure 3*). A tab delimited text file containing the results of this blobtools annotation is available on FigShare (DOI: 10.6084/m9.figshare.5688952). We then generated the final genome assembly by retaining scaffolds that had coverage >10.0 in the 1610_IlumiHiSeqX library, and did not have a high scoring (score >5000) taxonomic assignment for 'Proteobacteria', followed by polishing indels and gap-filling with Pilon (Appendix 3.5.3). This approach removed 235 scaffolds (330 Kbp), representing 0.2% of the scaffold number and 0.03% of the nucleotides of Ilumi1.0. While filtering the Ilumi1.0 assembly, we noted a large contribution of scaffolds taxonomically annotated as Platyhelminthes (1740 scaffolds; 119.56 Mbp). Upon closer inspection, we found conflicting information as to the most likely taxonomic source of these scaffolds. Diamond searches of these scaffolds had hits in Coleoptera, whereas blastn searches showed these scaffold had confident hits (nucleotide identity >90%, evalue = 0) against the Rat Tapeworm *Hymenolepis diminuta* genome (NCBI BioProject PRJEB507). Removal of these scaffolds decreased the endopterygota BUSCO score, from C:97% D:1.3% to C:76.0% D:1.1%. This loss

of the endopterygota BUSCOs led us to conclude that the Platyhelminthes annotated scaffolds were authentic scaffolds of *I. luminosus*, but sequences of *Hymenolepis* sp. may have been transferred into the *I. luminosus* genome via horizontal-gene-transfer (HGT). Although *Hymenolepis diminuta* infects mammals, it also spends a period of its life cycle in intermediate insect hosts, including beetles, as cysticercoids (*Center for Disease Control and Prevention, 2017*; *Sheiman et al., 2006*). For a beetle like *I. luminosus*, which has a extended predatory larval stage, the accidental ingestion and harboring of a *Hymenolepis* sp. is plausible, potentially enabling HGT between *Hymenolepis* sp. and *I. luminosus* over evolutionary timescales.

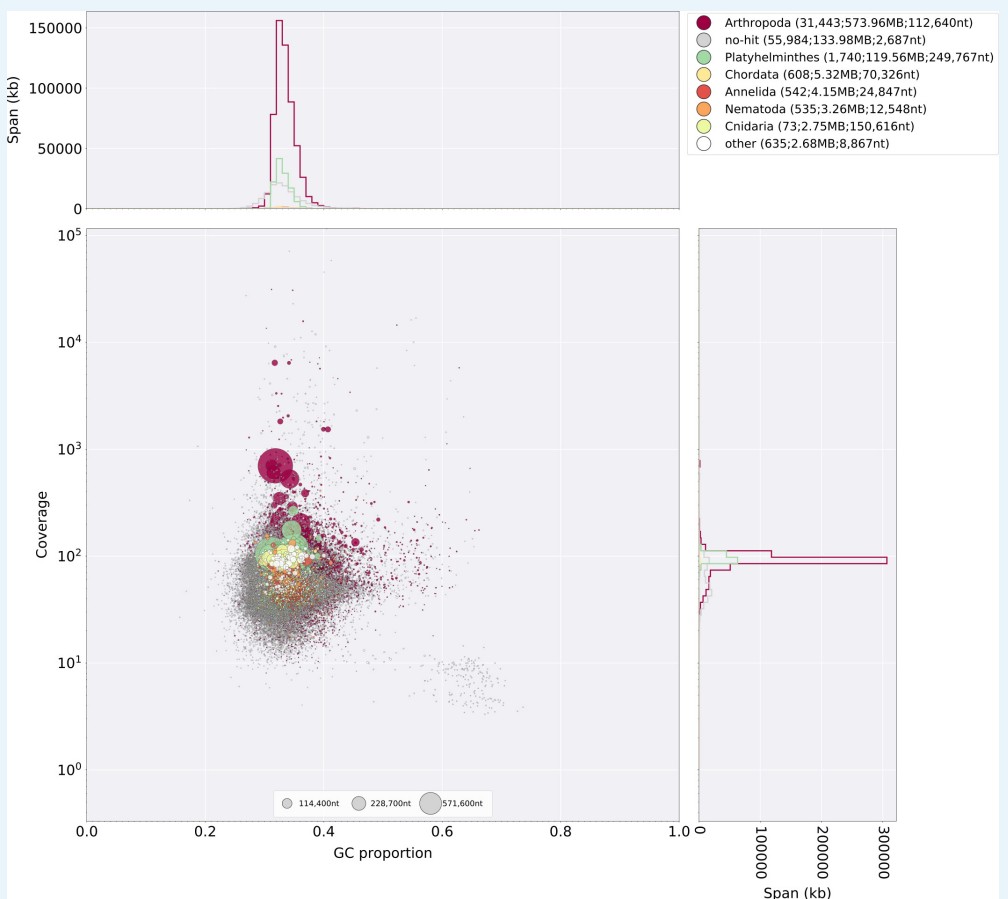

**Appendix 3—figure 3.** Blobtools plot of Ilumi1.0. Coverage shown represents mean coverage of reads from the HiSeqX Chromium library sequencing (Sample name 1610_IlumiHiSeqX; *Appendix 4—table 1*), aligned against Ilumi1.0 using Bowtie2 with parameters (–local). Scaffolds were taxonomically annotated as described in Appendix 3.5.2.

DOI: https://doi.org/10.7554/eLife.36495.044

### 3.5.3 Ilumi1.1: Indel polishing

Manual inspection of the initial gene-models for Ilumi1.0 revealed a key luciferase homolog had an unlikely frameshift occurring after a polynucleotide run. Mapping of the 1610_IlumiHiSeqX and 1706_IlumiHiSeq2500 reads (*Appendix 4—table 1*) with Bowtie2 using parameters (–local), revealed that this indel was not supported by the majority of the data, and that indels were present at a notable frequency after polynucleotide runs. As a greatly increased indel rate after polynucleotide runs (~10% error) is a known systematic error of Illumina sequencing, and has been noted as the major error type in Supernova assemblies (*Weisenfeld et al., 2017*), we therefore sought to correct these errors globally through the use of Pilon (v1.2.2) (*Walker et al., 2014*). In order to run Pilon efficiently, we split the

taxonomically filtered Ilumi1.0 reference (dubbed Ilumi1.0b; Appendix 3.5.2) using Kirill Kryukov's fasta_splitter.pl script (v0.2.6) (*Kryukov, 2017*), partitioned the previously mapped 1610_IlumiHiSeqX paired-end reads to these references using samtools, and ran Pilon in parallel on the partitioned reads and records with parameters (–fix gaps,indels –changes –vcf –diploid). The final consensus FASTAs produced by Pilon were merged to produce the polished assembly (Ilumi1.1). Ilumi1.1 (842,900,589 nt; 91,325 scaffolds) was slightly smaller than Ilumi1.0b (845,332,796 nt; 91,325 scaffolds), indicating the gaps filled by Pilon were smaller than their predicted size. The BUSCO score increased modestly after polishing (C:93.3% to C:94.8%), suggesting that indel polishing and gap filling had a net positive effect.

### 3.5.4 Ilumi1.2: Manual long-read scaffolding

We determined via manual gene-model annotation of Ilumi1.1 (Appendix 3.8), that the second through seventh exon of IlumPACS4 (ILUMI_06433 PA) were present on Ilumi1.1_Scaffold13255, but that the first exon was missing from this scaffold. Targeted tblastn using PangPACS (AB479114.1) (*Oba et al., 2010a*), the most closely related gene sequence to IlumPACS4, indicated that the most similar region in the *I. luminosus* genome to the predicted PangPACS first exon was a right-pointing region on Ilumi1.1_Scaffold11560, not captured in any gene model, but downstream of the existing luciferase homolog genes IlumPACS1 and IlumPACS2. We surmised that this region was the correct first exon for IlumPACS4, and that the IlumPACS4 gene model spanned Ilumi1.1_Scaffold13255 and Ilumi1.1_Scaffold11560, and thus that the right edge of Ilumi1.1_Scaffold13255 and the left edge of the reverse complement of Ilumi1.1_Scaffold11560 should be joined. To substantiate this, we performed long-read Oxford Nanopore MinION sequencing at the MIT BioMicroCenter. The HMW DNA used was the same DNA used for Chromium library prep, and had been stored at −80°C since extraction. Thawing of DNA and size distribution QC on a FEMTO Pulse capillary electrophoresis instrument (Advanced Analytical Technologies Inc, USA) indicated the DNA had a mean size distribution peak of ~17 kbp. A 1D Nanopore library was prepared from this DNA using the standard kit and protocol (Part #: SQK-LSK108). The resulting library was sequenced for 48 hr on a MinION sequencer using a R9.4 flow cell (Part #:FLO-MIN106). Raw trace data was basecalled live within the MinKNOW software (v18.01.6). 824,248 reads (2.4 Gbp; ~1–2x of the *I. luminosus* genome) were obtained. Reads were mapped to Ilumi1.1 with minimap2 (v2.8-r686-dirty) (*Li, 2018*) using parameters (-ax map-ont). Inspection of mapped reads with Integrative Genomics Viewer(v2.4.8) (*Thorvaldsdóttir et al., 2013*) revealed a 17.6 kbp read with seven kbp antiparallel alignment to the right edge of Scaffold13255. Inspection of the extension of this read off Scaffold13255 revealed it contained 10 Kbp+ of a non-palindromic complex tandem repeat DNA with an ~100 bp repeat unit (*Appendix 3—figure 4*). The repeat unit of this complex tandem repeat DNA (*Appendix 3—table 1*) is annotated in our *de novo* repeat library construction as 'Ilumi.complex.repeat.1' (Appendix 3.9), and via blastn is clearly interspersed at low copy numbers throughout the Ilumi1.1 genome assembly. Notably, this repeat unit was present the right edge of Ilumi1.1_Scaffold13255, while the reverse complement of this repeat unit was present on the right edge of Ilumi1.1_Scaffold11560, supporting that these scaffolds were adjacent to one another, but the assembly had been broken by this large stretch of tandem repetitive DNA. Although our Nanopore sequencing did not unambiguously span this repetitive element and bridge the two scaffolds, we surmised that this information was sufficient to manually merge these scaffolds (*Appendix 3—figure 5*). The long Ilumi1.1_Scaffold13255 extending read was adaptor trimmed with porechop (v0.2.3) (*Wick, 2018*), removing 35 bp from the start of the read. Next, the 3' end of the read which aligned up to the last nucleotide of Ilumi1.1_Scaffold13255 was trimmed. Finally, the remaining read was reverse complemented, and concatenated to the right edge of Ilumi1.1_Scaffold13255. 1337 Ns were concatenated to the right edge of the extended Ilumi1.1_Scaffold13255 to indicate an uncertainty in the repeat copy number, and Ilumi1.1_Scaffold11560 was reverse complemented and concatenated to Ilumi1.1_Scaffold13255 to produce the final version of Ilumi1.2_Scaffold13255 (*Appendix 3—figure 5*). Further whole genome scaffolding using this Nanopore data and the LINKS pipeline (v1.8.5) (*Warren et al., 2015*) with parameters (-d 4000,8000,10000,14000,16000,20000 t

2,3,5,9 | 2 -a 0.75) was attempted, but only a single additional pair of scaffolds was merged, so this whole-genome scaffolding was not used further.

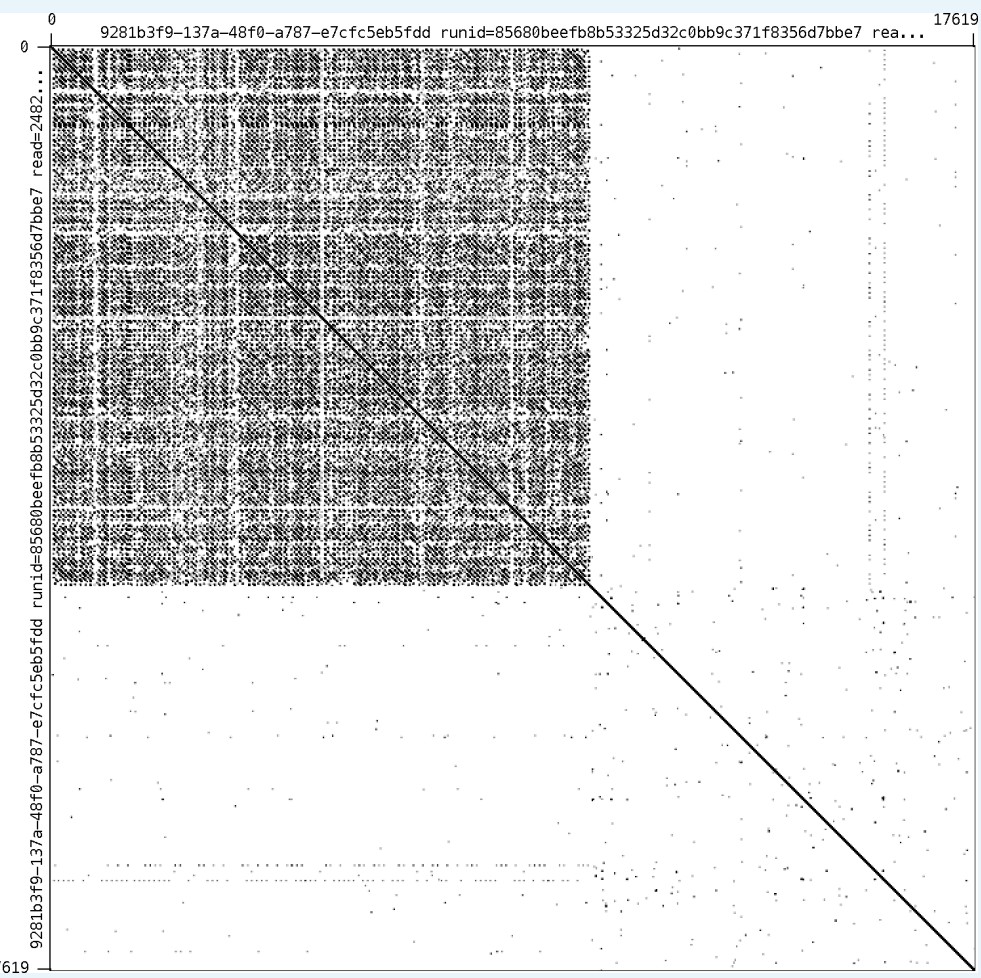

**Appendix 3—figure 4.** Self alignment of the Ilumi1.1_Scaffold13255 right-edge extending long MinION read. Alignment performed in in Gepard (*Krumsiek et al., 2007*). Note the large (10 kbp+) tandem repetitive region.

DOI: https://doi.org/10.7554/eLife.36495.045

**Appendix 3—table 1.** Sequence of the *I. luminosus* luciferase cluster splitting complex tandem repeat.

| Repeat name | Repeat unit length | Repeat unit sequence |
|---|---|---|
| Ilumi.complex.repeat.1 | ~100 bp | TGGTACGAACTATACACGTATACTCAAATCTAATT GTGATACAGCAAAGTAATAATGCAGCATTGTTTGCC GCTCTATACTGCGATTTTATAGTGGT |

DOI: https://doi.org/10.7554/eLife.36495.046

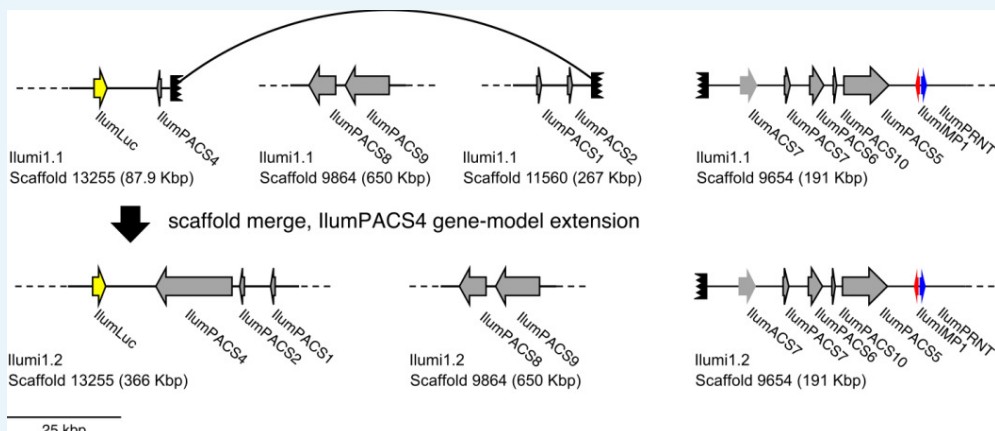

**Appendix 3—figure 5.** Diagram of manual scaffold merges between Ilumi1.1 and Ilumi1.2. Diagram of the manual merge of Ilumi1.1_Scaffold13255 with Ilumi1.1_Scaffold11560 between *I. luminosus* genome assembly versions Ilumi1.1 and Ilumi1.2. This merge was supported by: (1) The putative missing first exon of IlumPACS4 being present on the right edge of Ilumi1.2_Scaffold11560. (2) The right edge of Ilumi1.1_Scaffold13255, and the right edge of Ilumi1.1_Scaffold11560, having anti-parallel versions of a homologous complex tandem repeat. See *Figure 3* in the maintext for explanation of presented genes.
DOI: https://doi.org/10.7554/eLife.36495.047

## 3.6 RNA extraction, library prep, and sequencing

### 3.6.1 HiSeq2500

Total RNA was extracted from the head + prothorax of an *I. luminosus* presumed male using the RNeasy Lipid Tissue Mini Kit (Qiagen, USA). Illumina sequencing libraries were prepared from total RNA enriched to mRNA with a polyA pulldown using the TruSeq RNA Library Prep Kit v2 (Illumina, San Diego, CA). The library was sequenced at the Whitehead Institute Genome Technology Core (Cambridge, MA) on two lanes of an Illumina HiSeq 2500 using rapid mode 100 × 100 bp PE. This library was multiplexed with the *P. pyralis* RNA-Seq libraries of Al-Wathiqui and colleagues (*Al-Wathiqui et al., 2016*), and thus, *P. pyralis* reads arising from index misassignment were present in this library which necessitated downstream filtering to avoid misinterpretation.

### 3.6.2 BGISEQ-500

Total RNA was extracted from the head + prothorax, mesothorax + metathorax, and abdomen of adult presumed *I. luminosus* males using the RNeasy Lipid Tissue Mini Kit (Qiagen, USA), and sent on dry-ice to Beijing Genomics Institute (BGI, China). Transcriptome libraries for RNA each sample were prepared from total RNA using the BGISEQ-500 (BGI, China) RNA sample prep protocol. Briefly, poly-A mRNA was purified using oligo (dT) primed magnetic beads and chemically fragmented into smaller pieces. Cleaved fragments were converted to double-stranded cDNA by using N6 primers. After gel purification and end-

repair, an 'A' base was added at the 3'-end of each strand. The Ad153-2B adapters with barcode was ligated to both ends of the end repaired/dA tailed DNA fragments, then amplification by ligation-mediated PCR. Following this, a single strand DNA was separated at a high temperature and then a Splint oligo sequence was used as bridge for DNA cyclization to obtain the final library. Then rolling circle amplification (RCA) was performed to produce DNA Nanoballs (DNBs). The qualified DNBs were loaded into the patterned nanoarrays and the libraries were sequenced as $50 \times 50$ bp (PE-50) read through on the BGISEQ-500 platform. Sequencing-derived raw image files were processed by BGISEQ-500 base-calling software with the default parameters, generating the 'raw data' for each sample stored in FASTQ format. This library preparation and sequencing was provided free of charge as an evaluation of the BGISEQ-500 platform.

**Appendix 3—table 2.** *I. luminosus* RNA-Seq libraries.

| Library name | SRA ID | N | Sex | Tissue | Notes |
|---|---|---|---|---|---|
| Pyrophorus_luminosus_head | SRR6339835 | 1 | M* | Prothorax and head (lantern containing) | Illumina RNA-Seq |
| Prothorax_A3 | SRR6339834 | 1 | M* | Prothorax and head (lantern containing) | BGISEQ-500 RNA-Seq |
| Thorax_A3 | SRR6339833 | 1 | M* | Mesothorax and metathorax | BGISEQ-500 RNA-Seq |
| Abdomen_A3 | SRR6339832 | 1 | M* | Abdomen (lantern containing) | BGISEQ-500 RNA-Seq |
| Prothorax_A4 | SRR6339831 | 1 | M* | Prothorax and head (lantern containing) | BGISEQ-500 RNA-Seq |
| Thorax_A4 | SRR6339830 | 1 | M* | Mesothorax and metathorax | BGISEQ-500 RNA-Seq |
| Abdomen_A4 | SRR6339838 | 1 | M* | Abdomen (lantern containing) | BGISEQ-500 RNA-Seq |

*Gender inferred. See Appendix 3.3 for a discussion on this inference.
DOI: https://doi.org/10.7554/eLife.36495.048

### 3.7 Transcriptome analysis

Both *de novo* (Appendix 3.7.1) and reference guided (Appendix 3.7.2) transcriptome assembly approaches using Trinity and Stringtie were used, respectively.

### 3.7.1 *De novo* transcriptome assembly and alignment

For the *de novo* transcriptome approach, all available *I. luminosus* RNA-Seq reads (head + prothorax,metathorax + mesothorax, abdomen - both Illumina and BGISEQ-500) were pooled and input into Trinity. A non-strand-specific *de novo* transcriptome assembly was produced with Trinity (v2.4.0) (*Grabherr et al., 2011*) using default parameters exception the following: (–min_glue 2 min_kmer_cov 2 –jaccard_clip –no_normalize_reads –trimmomatic). Peptides were predicted from the *de novo* transcripts via Transdecoder (v5.0.2; default parameters). *De novo* transcripts were then aligned to the *I. luminosus* genome (Ilumi1.1) using the PASA pipeline with blat (v36 × 2) and gmap (v2017-09-11) (–aligners blat,gmap), parameters for alternative splice analysis and strand specificity (–ALT_SPLICE – transcribed_is_aligned_orient), and input of the previously extracted Trinity accessions (–tdn tdn.accs). Importantly, it was necessary to set (– NUM_BP_PERFECT_SPLICE_BOUNDARY = 0) for the validate_alignments_in_db.dbi step, to ensure transcripts with natural variation near the splice sites were not discarded. Direct coding gene models (DCGMs) were then produced with the Transdecoder 'cdna_alignment_orf_to_genome_orf.pl' utility script, with the PASA assembly GFF and transdecoder predicted peptide GFF as input. The resulting DCGM GFF3 file was manually lifted over to the Ilumi1.2 assembly. The unaligned *de novo* transcriptome assembly is

dubbed 'ILUMI_Trinity_unstranded', whereas the aligned direct coding gene models are dubbed 'Ilumi1.2_Trinity_unstranded-DCGM'.

### 3.7.2 Reference guided transcriptome alignment and assembly

A reference guided transcriptome was produced from all available *I. luminosus* RNA-seq reads (head + prothorax, mesothorax + metathorax, abdomen - both Illumina and BGISEQ-500) using HISAT2 (v2.0.5) (*Kim et al., 2015*) and StringTie (v1.3.3b) (*Pertea et al., 2015*). Reads were first mapped to the *I. luminosus* draft genome with HISAT2 (parameters: -X 2000 –dta –fr). Then StringTie assemblies were performed on each separate bam file corresponding to the original libraries using default parameters. Finally, the produced GTF files were merged using StringTie (–merge). A transcript fasta file was produced from the StringTie GTF file with the transdecoder 'gtf_genome_to_cdna_fasta.pl' utility script, and peptides were predicted for these transcripts using Transdecoder (v5.0.2) with default parameters. The StringTie GTF was converted to GFF format with the Transdecoder 'gtf_to_alignment_gff3.pl' utility script, and direct coding gene models (DCGMs) were then produced with the Transdecoder 'cdna_alignment_orf_to_genome_orf.pl' utility script, with the StringTie-provided GFF and transdecoder predicted peptide GFF as input. The resulting DCGM GFF3 file was manually lifted over to the Ilumi1.2 assembly. The reference guided transcriptome assembled was dubbed 'ILUMI_Stringtie_unstranded', whereas the aligned direct coding gene models were dubbed 'Ilumi1.2_Stringtie_unstranded-DCGM'

### 3.7.3 Transcript expression analysis

*I. luminosus* RNA-Seq reads (*Appendix 3—table 2*) were pseudoaligned to the ILUMI_OGS1.2 geneset CDS sequences using Kallisto (v0.44.0) (*Bray et al., 2016*) with 100 bootstraps (-b 100), producing transcripts-per-million reads (TPM). Kallisto expression quantification analysis results are available on FigShare (DOI: 10.6084/m9.figshare.5715139).

## 3.8 Official coding geneset annotation (ILUMI_OGS1.2)

We annotated the coding gene structure of *I. luminosus* by integrating direct coding gene models produced from the *de novo* transcriptome (Appendix 3.7.1) and reference guided transcriptome (Appendix 3.7.2), with a lower weighted contribution of *ab initio* gene predictions, using the Evidence Modeler (EVM) algorithm (v1.1.1) (*Haas et al., 2008*). First, Augustus (v3.2.2) (*Stanke et al., 2006*) was trained against Ilumi1.0 with BUSCO (parameters: -l endopterygota_odb9

–long –species tribolium2012). Augustus predictions of Ilumi1.0 were then produced through the MAKER pipeline, with hints derived from MAKER blastx/exonerate mediated protein alignments of peptides from *Drosophila melanogaster* (NCBI GCF_000001215.4_Release_6_plus_ISO1_MT_protein.faa), *Tribolium castaneum* (NCBI GCF_000002335.3_Tcas5.2_protein), *Photinus pyralis* (PPYR_OGS1.0; this report), *Aquatica lateralis* (AlatOGS1.0; this report), the *I. luminosus de novo* transcriptome translated peptides, and MAKER blastn/exonerate transcript alignments of the *I. luminosus de novo* transcriptome transcripts.

We then integrated the *ab initio* predictions with our *de novo* and reference guided direct coding gene models, using EVM. In the final version, eight sources of evidence were used for EVM: *de novo* transcriptome direct coding gene models (Ilumi1.1_Trinity_unstranded-DCGM; weight = 8), reference guided transcriptome direct coding gene models (Ilumi1.1_Stringtie_unstranded-DCGM; weight = 4), MAKER/Augustus *ab initio* predictions (Ilumi1.1_maker_augustus_ab-initio; weight = 1), protein alignments (*P. pyralis*, *A. lateralis*, *D. melanogaster*, *T. castaneum, I. luminosus*; weight = 1 each). A custom script (*Fallon, 2018a*) was used to convert the input MAKER GFF to an EVM compatible GFF format.

Lastly, gene models for luciferase homologs, P450s, and *de novo* methyltransferases (DNMTs) which were fragmented or were incorrectly assembled (e.g. adjacent gene fusions) were manually corrected based on the evidence of the *de novo* and reference guided direct coding gene models (Appendix 3.7.1; 3.7.2). Manual correction was performed by

performing TBLASTN searches with known good genes from these gene families within SequencerServer(v1.10.11) (**Priyam et al., 2015**), converting the TBLASTN results to gff3 format with a custom script (**Fallon, 2018b**), and viewing these TBLASTN alignments alongside the alternative direct coding gene models and the official geneset in Integrative Genomics Viewer (v2.4.8) (**Thorvaldsdóttir et al., 2013**). The official gene set models gff3 file was then manually modified based on the observed evidence. Different revision numbers of the official geneset (e.g. ILUMI_OGS1.0, ILUMI_OGS1.1) represent the improvement of the geneset over time due to these continuing manual gene annotations.

### 3.9 Repeat annotation

A *de novo* species-specific repeat library for *I. luminosus* was constructed using RepeatModeler (v1.0.9), and Tandem Repeat Finder (v4.09; settings: 2 7 7 80 10) (**Benson, 1999**). Only tandem repeats from Tandem Repeat Finder with a repeat block length >5 kb (annotated as 'complex tandem repeat') were added to the RepeatModeler library. This process yielded a final library of 2259 interspersed repeats. We then used this library and RepeatMasker (v4.0.5) (**Smit et al., 2015**) to identify and mask interspersed and tandem repeats in the genome assembly. This repeat library is dubbed the *Ignelater luminosus* Official Repeat Library 1.0 (ILUMI_ORL1.0).

**Appendix 3—table 3.** Annotated repetitive elements in *I. luminosus*.

| Repeat class | Family | Counts | Bases | % of assembly |
|---|---|---|---|---|
| DNA | All | 158853 | 71221843 | 8.45 |
| | Helitrons | 344 | 139863 | 0.016 |
| LTR | All | 23433 | 11341577 | 1.35 |
| Non-LTR | All | 151788 | 50394853 | 4.75 |
| | LINE | 97703 | 40052840 | 4.75 |
| | SINE | 0 | 0 | 0 |
| Unknown interspersed | | 757206 | 159587269 | 18.93 |
| Complex tandem repeats | | 4976 | 848992 | 0.1 |
| Simple repeat | | 108914 | 4439967 | 0.52 |
| rRNA | | 0 | 0 | 0 |

DOI: https://doi.org/10.7554/eLife.36495.049

### 3.10 Mitochondrial genome assembly and annotation

The mitochondrial genome sequence of *I. luminosus* was assembled by a targeted sub-assembly approach. First, Chromium linked-reads were mapped to the previously sequenced mitochondrial genome of the Brazilian elaterid beetle *Pyrophorus divergens* (NCBI ID: NC_009964.1) (**Arnoldi et al., 2007**), using Bowtie2 (v2.3.1; parameters: –very-sensitive-local ) (**Langmead et al., 2009**). Although these reads still contain the 16 bp Chromium library barcode on read 1 (R1), Bowtie2 in local mapping mode can accurately map these reads. Mitochondrial mapping R1 reads with a mapping read 2 (R2) pair were extracted with 'samtools view -bh -F 4 f 8', whereas mapping R2 reads with a mapping R1 pair were extracted with 'samtools view -bh -F 8 f 4'. R1 and R2 singleton mapping reads were extracted with 'samtools view -bh -F 12' for diagnostic purposes, but were not used further in the assembly. The R1, R2, and singleton reads in. BAM format were merged, sorted, and converted to FASTQ format with samtools and 'bedtools bamtofastq', respectively. The resultant R1 and R2 FASTQ files containing only the paired mapped reads (995523 pairs, 298 Mbp) were assembled with SPAdes (**Nurk et al., 2013**) without error correction and with the plasmidSPAdes module (**Antipov et al., 2016**) enabled (parameters: -t 16 –plasmid -k55,127 –cov-cutoff 1000 –only-assembler). The resulting 'assembly_graph.fastg' file was viewed in Bandage (**Wick et al., 2015**), revealing a 16,088 bp node with 1119x average coverage that

circularized through two possible paths: a 246 bp node with 252x average coverage, or a 245 bp node with 1690x coverage. The lower coverage path was observed to differ only in a 'T' insertion after a 10-nucleotide poly-T stretch when compared to the higher coverage path. Given that increased levels of insertions after polynucleotide stretches are a known systematic error of Illumina sequencing, it was concluded that the lower coverage path represented technical error rather than an authentic genetic variant and was deleted. This produced a single 16,070 bp circular contig. This contig was 'restarted' with seqkit (v0.7.0) (**Shen et al., 2016**) to place the FASTA record break in the AT-rich region, and was submitted to the MITOSv2 mitochondrial genome annotation web server. Small mis-annotations (e.g. low scoring additional predictions of already annotated mitochondrial genes) were manually inspected and removed. This annotation indicated that all expected features were present on the contig, including subunits of the NAD$^+$ dehydrogenase complex (NAD1, NAD2, NAD3, NAD4, NAD4l, NAD5, NAD6), the large and small ribosomal RNAs (rrnL, rrnS), subunits of the cytochrome c oxidase complex (COX1, COX2, COX3), cytochrome b oxidase (COB), ATP synthase (atp6, atp8), and tRNAs. BLASTN of the *Ignelater luminosus* mitochondrial genome against published complete mitochondrial genomes from beetles indicated 96–89% alignment with 86–73% nucleotide identity, with poor or no sequence level alignment in the A-T rich region. Like other reported elaterid beetle genomes, the *I. luminosus* mitochondrial genome does not contain the tandem repeat unit (TRU) previously reported in Lampyridae (**Bae et al., 2004**).

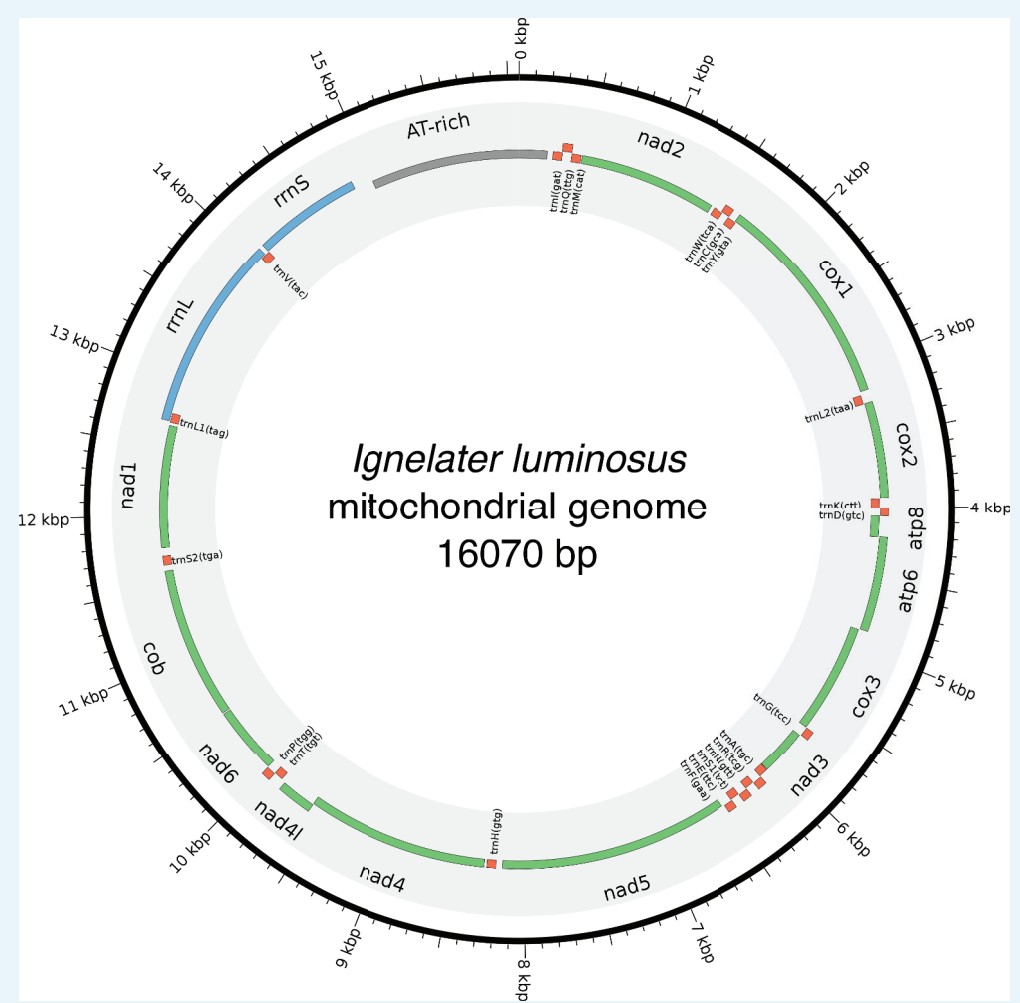

**Appendix 3—figure 6.** Mitochondrial genome of *I. luminosus*. The mitochondrial genome of *I.*

*luminosus* was assembled and annotated as described. in the Appendix 3.10. Figure produced with Circos (*Krzywinski et al., 2009*).

DOI: https://doi.org/10.7554/eLife.36495.050

# Appendix 4

DOI: https://doi.org/10.7554/eLife.36495.051

## Comparative analyses

### 4.1 Assembly statistics and comparisons

The level of non-eukaryote contamination of the raw read data for each *P. pyralis* library was assessed using kraken v1.0 (**Wood and Salzberg, 2014**) using a dust-masked minikraken database to eliminate comparison with repetitive sequences. Overall contamination levels were low (**Appendix 4—table 1**), in agreement with a low level of contamination in our final assembly (**Appendix 1—figure 9**, **Appendix 2—figure 2**, **Appendix 3—figure 3**). On average, contamination was 3.5% in the PacBio reads (whole body) and 1.6% in the Illumina reads (only thorax) (**Appendix 4—table 1**). There was no support for Wolbachia in any of the *P. pyralis* libraries, with the exception of a single read from a single library which had a kraken hit to Wolbachia. QUAST version 4.3 (**Gurevich et al., 2013**) was used to calculate genome quality statistics for comparison and optimization of assembly methods (**Appendix 4—table 2**). BUSCO (v3.0.2) (**Simão et al., 2015**) was used to estimate the percentage of expected single copy conserved orthologs captured in our assemblies and a subset of previously published beetle genome assemblies (**Appendix 4—table 3**). The endopterygota_odb9 (metamorphosing insects) BUSCO set was used. The bacteria_odb9 gene set was used to identify potential contaminants by screening contigs and scaffolds for conserved bacterial genes. For genome predictions from beetles, the parameter '–species tribolium2012' was used to improve the BUSCO internal Augustus gene predictions. For *Drosophila melanogaster* BUSCO genome predictions (**Appendix 4—table 3**) '–species=fly' was used.

### 4.2 Comparative analyses

#### 4.2.1 Protein orthogroup clustering

Orthologs were identified by clustering the *P. pyralis*, *A. lateralis*, and *I. luminosus* geneset peptides with the *D. melanogaster* (UP000007266) and *T. castaneum* (UP000000803) reference Uniprot protein genesets using the OrthoFinder (v2.2.6) (**Emms and Kelly, 2015**) pipeline with parameters '-M msa -A mafft -T fasttree -I 1.5'. The pipeline was executed with NCBI blastp + v0.2.7.1, mafft 7.313, and FastTree v2.1.10 with Double precision (No SSE3). The Uniprot reference proteomes were first filtered using a custom script to remove multiple isoforms-per-gene using a custom script (**Fallon, 2018c**; copy archived at https://github.com/elifesciences-publications/filter_uniprot_to_best_isoform), which utilized blastp evidence against either the *Drosophila melanogaster* or *Tribolium castaneum* NCBI datasets (whichever species was not being filtered), and the *Apis mellifera*, *Bombyx mori*, *Caenorhabditis elegans*, *Anopheles gambiae* NCBI peptide genesets. Not all redundant isoforms are removed as there may not have been sufficient evidence to support a particular isoform as the canonical isoform, or there were unusual annotation situations (alternative splice variants annotated as separate genes). OrthoFinder clustering results are available on FigShare (DOI: 10.6084/m9.figshare.5715136). Overlaps of number of shared orthogroups across species are shown in **Appendix 4—figure 1**. Overlaps on a gene-basis (only *P. pyralis*, *A. lateralis*, *I. luminosus*, and *T. castaneum*) are shown in **Figure 2E**.

**Appendix 4—table 1.** Genomic sequencing library statistics.

**ID:** NCBI BioProject or Gene Expression Omnibus (GEO) ID. **N:** Number of individuals used for sequencing. **Date:** collection date for wild-caught individuals. **Locality:** GSMNP: Great Smoky Mountains National Park, TN; MMNJ: Mercer Meadows, Lawrenceville, NJ; IY90: laboratory strain Ikeya-Y90; MAPR: Mayagüez, Puerto Rico. **Tissue:** Thr: thorax; WB: whole-body; **Type:** SI: Illumina short insert; MP: Illumina mate pair; PB: Pacific Biosciences, RSII P6-C4; HC: Hi-C; BS: Bisulfite; CH: 10x Chromium; ONT: Oxford Nanopore MiniION R9.4. **Reads:** PE: paired-end, CLR: continuous long read. **Number:** number of reads. **Cov:** Mode of autosomal coverage (mode of putative X chromosome, LG3a, coverage), determined from mapped reads with QualiMap (v2.2). ND: Not Determined. **Insert size:** Mode of insert size after alignment (orientation: FR: forward, RF: reverse), determined from mapped reads with QualiMap. **Contamination:** Percent contamination as estimated by kraken v1.0.

| Library | SRA ID | N | Date | Locality | Sex | Tissue | Type | Reads | Number | Cov | Insert size (Ori) | Contamination |
|---|---|---|---|---|---|---|---|---|---|---|---|---|
| *Photinus pyralis* | | | | | | | | | | | | |
| 8369* | SRR6345451 / SRR2127932 | 1 | 6/13/11 | GSMNP | M | Thr | SI | 101 × 101 PE | 203,074,230 | 98 (49) | 354 bp (FR) | 0.28 |
| 8375_3 K[†] | SRR6345448 | 1 | 6/13/11 | GSMNP | M | Thr | MP | 101 × 101 PE | 101,624,630 | 21 | 2155 bp (RF) | 2.63 |
| 8375_6 K[†] | SRR6345457 | 1 | 6/13/11 | GSMNP | M | Thr | MP | 101 × 101 PE | 23,564,456 | 5 | 4889 bp (RF) | 3.36 |
| 83_3 K[†] | SRR6345450 | 3 | 6/13/11 | GSMNP | M | Thr | MP | 101 × 101 PE | 121,757,858 | 13 | 2247 bp (RF) | 0.79 |
| 83_6 K[†] | SRR6345455 | 3 | 6/13/11 | GSMNP | M | Thr | MP | 101 × 101 PE | 17,905,700 | 1 | 4877 bp (RF) | 1.38 |
| 1611_PpyrPB1 | SRX3444870 | 4 | 7/9/16 | MMNJ | M | WB | PB | CLR-PB | 3,558,201 | 38 (21) | 7 Kbp[‡] | 3.5 |
| 1704 | SRR6345456 | 2 | 7/9/16 | MMNJ | M | WB | HC | 80 × 80 PE | 93,850,923 | ND | ND | ND |
| 1705 | GSE107177 | 1 | 7/9/16 | MMNJ | M | WB | BS | 150 SE | 113,761,746 | ~16x[§] | ND | ND |
| *Aquatica lateralis* | | | | | | | | | | | | |
| FFGPE_PE200 | DRR119296 | 1 | N/A | IY90 | F | WB | SI | 126 × 126 PE | 561,450,686 | 72 | 180 bp (FR) | ND |
| FFGPE_PE800 | DRR119297 | | | | | WB | SI | 126 × 126 PE | 218,830,950 | 20 | 476 bp (FR) | ND |
| FFGMP_MPGF | DRR119298 | | | | | WB | MP | 101 × 101 PE | 358,601,808 | 31 | 2300 bp (RF) | ND |
| *Ignelater luminosus* | | | | | | | | | | | | |

*Appendix 4—table 1 continued on next page*

Appendix 4—table 1 continued

| Library | SRA ID | N | Date | Locality | Sex | Tissue | Type | Reads | Number | Cov | Insert size (Ori) | Contamination |
|---|---|---|---|---|---|---|---|---|---|---|---|---|
| 1610_Ilumi HiSeqX[#] | SRR6339837 | 1 | | MAPR | M[¶] | WB | CH | 151 × 151 PE | 408,838,927 | 99 | 339 bp (FR) | ND |
| 1706_Ilumi HiSeq2500[#] | SRR6339836 | | | | | WB | CH | 150 × 150 PE | 145,250,480 | 48 | 334 bp (FR) | ND |
| 18_lib1 | SRR6760567 | | | | | | ONT | CLR | 824,248 | ~2x | 2984[‡] | |

*Mean of three sequencing lanes
[†]Mean of two sequencing lanes
[‡]Mean subread (PacBio) or read (Oxford Nanopore) length after alignment
[§]Estimate from quantity of mapped reads
[#]Same library, different instruments
[¶]Inferred from specimens collected at the same time and locality

DOI: https://doi.org/10.7554/eLife.36495.052

**Appendix 4—table 2.** Assembly statistics

| Assembly | Libraries | Assembly scheme | Assembly*/measured** genome size (Gbp) | Scaffold/Contig (#) | Contig NG50*** (Kbp) | Scaffold NG50*** (Kbp) | BUSCO statistics |
|---|---|---|---|---|---|---|---|
| Ppyr0.1-PB | PacBio (61 RSII SMRT cells) | Canu (no polishing) | 721/422 | 25986/25986 | 86 | 86 | C:93.8%[S:65.2%, D:28.6%], F:3.3%, M:2.9% |
| Ppyr1.1 | Short read Mate Pair PacBio | MaSuRCA + redundancy reduction | 473/422 | 8065/8285 | 193.4 | 202 | C:97.2% [S:88.8%, D:8.4%], F:1.9%, M:0.9% |
| Ppyr1.2 | Short — PacBio — Hi-C | Ppyr1.1+ Phase Genomics scaffolder (in-house) | 473/422 | 2535/7823 | 193.4 | 50,607 | C:97.2% [S:88.8% , D:8.4%], F:1.9%, M:0.9% |
| Ppyr1.3 | Short read Mate Pair PacBio | Ppyr1.2 +Blobtools + manual filtering | 472/422 | 2160/7533 | 192.5 | 49,173 | C:97.2% [S:88.8%, D:8.4%], F:1.9%, M:0.9% |
| Alat1.2 | Short read Mate Pair | ALLPATHS-LG | 920/940 | 7313/36467 | 38 | 673 | C:97.4% [S:96.2%, D:1.2%], F:1.8%, M:0.8% |
| Alat1.3 | Short read Mate Pair | Alat1.2+Blobtools + manual filtering | 909/940 | 5388/34298 | 38 | 670 | C:97.4% [S:96.2%, D:1.2%], F:1.8%, M:0.8% |
| Ilumi1.0 | Linked-read | Supernova | 845/764 | 91560/105589 | 31.6 | 116.5 | C:93.7% [S:92.3%, D:1.4%, F:4.3%, M:2.0%, |

*Appendix 4—table 2 continued on next page*

*Appendix 4—table 2 continued*

| Assembly | Libraries | Assembly scheme | Assembly*/measured** genome size (Gbp) | Scaffold/Contig (#) | Contig NG50*** (Kbp) | Scaffold NG50*** (Kbp) | BUSCO statistics |
|---|---|---|---|---|---|---|---|
| IIumi1.2 | Linked read+nanopore | IIumi1.0+Blobtools+Pilon indel and gap polishing. Manual scaffolding | 842/764 | 91305/105262 | 34.5 | 115.8 | C:94.8% [S:93.4%, D:1.4%], F:3.5%, M:1.7% |

*Calculated from genome assembly file with 'seqkit stat'

**Measured via flow cytometry of propidium iodide stained nuclei. See Appendix 1.4, 2.4, 3.4.

***Calculated with QUAST (v4.5) (*Gurevich et al., 2013*), parameters '-e –scaffolds –est-ref-size X –min-contig 0' and the measured genome size for 'est-ref-size'

DOI: https://doi.org/10.7554/eLife.36495.053

**Appendix 4—table 3.** Comparison of BUSCO conserved gene content with other insect genome assemblies

| Species | Genome version (NCBI assemblies) | Note | Genome BUSCO (endopterygota_odb9) | Protein geneset BUSCO (endopterygota_odb9)** |
|---|---|---|---|---|
| Drosophila melanogaster | GCA_000001215.4 Release 6 | Model insect | C:99.4%[S:98.7%,D:0.7%],F:0.4%,M:0.2%,n:2442 | C:99.6%[S:92.8%,D:6.8%],F:0.3%,M:0.1%,n:2442 |
| Tribolium castaneum | GCF_000002335.3 Release 5.2 | Model beetle | C:98.4%[S:97.9%,D:0.5%],F:1.2%,M:0.4%,n:2442 | C:98.0%[S:95.8%,D:2.2%],F:1.6%,M:0.4%,n:2442 |
| Photinus pyralis* | Ppyr1.3* | North American firefly | C:97.2%[S:88.8%,D:8.4%],F:1.8%,M:1.0%,n:2442 | C:94.2%[S:84.0%,D:10.2%],F:1.2%,M:4.6%,n:2442 |
| Aquatica lateralis* | Alat1.3* | Japanese firefly | C:97.4%[S:96.2%,D:1.2%],F:1.8%,M:0.8% | C:90.0%[S:89.1%,D:0.9%],F:3.2%,M:6.8%,n:2442 |
| Nicrophorus vespilloides (Cunningham et al., 2015) | GCF_001412225.1 Release 1.0 | Burying beetle | C:96.8%[S:95.3%,D:1.5%],F:2.1%,M:1.1%,n:2442 | C:98.7%[S:69.4%,D:29.3%],F:0.8%,M:0.5%,n:2442 |
| Agrilus planipennis (Poelchau et al., 2015) | GCF_000699045.1 Release 1.0 | Emerald Ash Borer beetle | C:92.7%[S:91.8%,D:0.9%],F:4.6%,M:2.7%,n:2442 | C:92.1%[S:64.1%,D:28.0%],F:4.5%,M:3.4%,n:2442 |
| Ignelater luminosus* | Ilumi1.2 | Puerto Rican bioluminescent click beetle | C:94.8%[S:93.4%,D:1.4%],F:3.5%,M:1.7%,n:2442 | C:91.8%[S:89.8%,D:2.0%],F:4.4%,M:3.8%,n:2442 |

*=This report, **=Protein genesets downloaded from the NCBI Genome resource associated with the mentioned assembly in the 2nd column, or in the case of D. melanogaster, and T. casta-neum, protein genesets were produced from Uniprot Reference Proteomes which had been heuristically filtered down to 'canonical' isoforms with a custom script and BLASTP against the D. melanogaster, T. castaneum, Apis mellifera, Bombyx mori, Caenorhabditis elegans, and Anopheles gambiae protein genesets associated with their more recent genome assembly on NCBI. See Appendix 4.2.1 for more detail.

DOI: https://doi.org/10.7554/eLife.36495.054

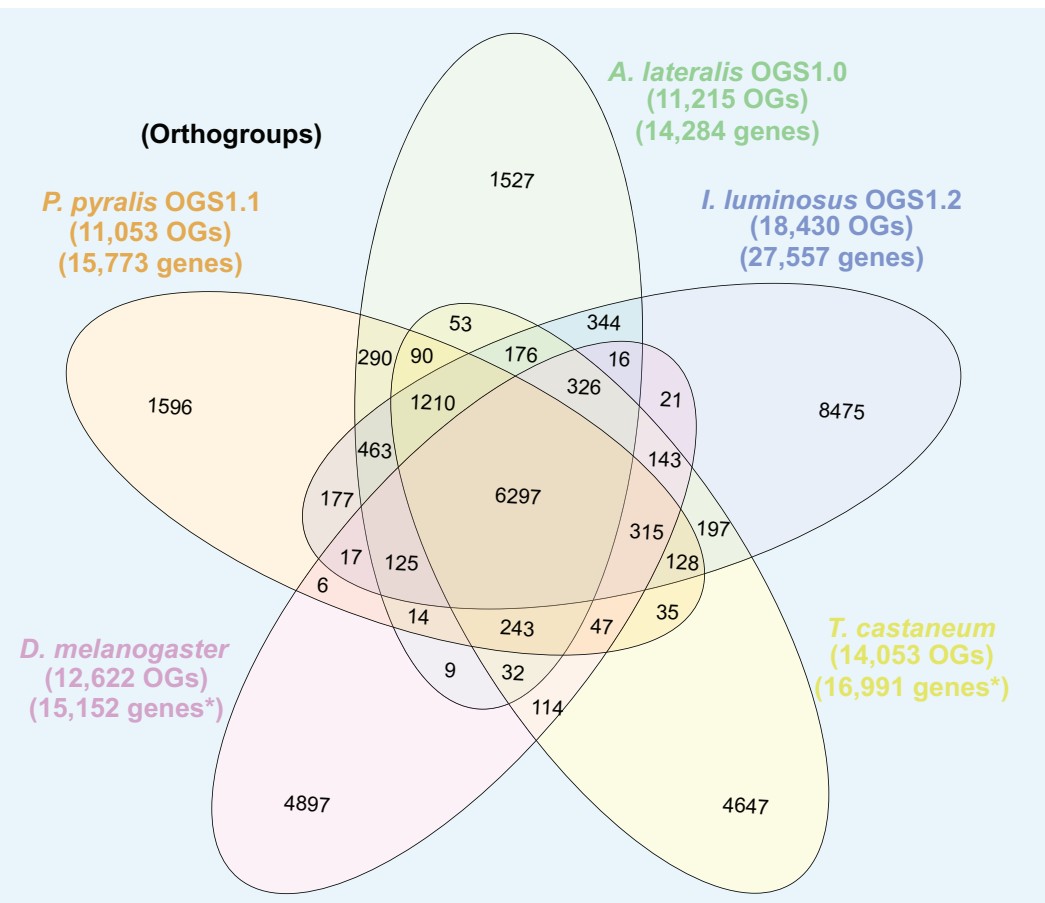

**(Orthogroups)**

**Appendix 4—figure 1.** Venn diagram of *P. pyralis*, *A. lateralis*, *I. luminosus*, *T. castaneum*, and *D. melanogaster* orthogroup relationships. Orthogroups were calculated between the PPYR_OGS1.1, AQULA_OGS1.0, ILUMI_OGS1.2, genesets, and the *T. casteneum* and *D. melanogaster* filtered Uniprot reference proteomes using OrthoFinder(***Emms and Kelly, 2015***). See Appendix 4.2.1 for description of clustering method. OGs = Orthogroups, OGS = Official gene set, *=Not completely filtered to single peptide per gene. Figure produced with InteractiVenn (***Heberle et al., 2015***). Intermediate scripts and species specific overlaps are available as ***Figure 2—source data 1***.
DOI: https://doi.org/10.7554/eLife.36495.055

## 4.2.2 Comparative RNA-Seq differential expression analysis (***Figure 5***)

For differential expression testing, Kallisto transcript expression results for *P. pyralis* (Appendix 1.9.4) and *A. lateralis* (Appendix 2.7.3) were independently between-sample normalized using Sleuth (v0.30.0) (***Pimentel et al., 2017***) with default parameters, producing between-sample-normalized transcripts-per-million reads (BSN-TPM). Differential expression (DE) tests for *P. pyralis* (adult male dissected fatbody vs. adult male dissected lantern - three biological replicates per condition), and for *A. lateralis* (adult male thorax + abdominal segments 1–5 vs. adult male dissected lantern - three biological replicates per condition), were performed using the Wald test within Sleuth. Genes whose mean BSN-TPM across bioreplicates was above the 90th percentile were annotated as 'highly expressed' (HE). Genes with a Sleuth DE q-value <0.05 were annotated as 'differentially expressed.' (DE). Enzyme encoding (E/NotE) genes were predicted from the InterProScan functional annotations using a custom script (***Fallon, 2018d***; copy archived at https://github.com/elifesciences-publications/ interproscan_to_enzyme_go) and GOAtools (***Tang et al., 2018***), with the modification that the enzymatic activity GO term was manually added to select InterPro annotations: IPR029058, IPR036291, and IPR001279. These enzyme lists are available as supporting files associated with the official geneset filesets. Orthogroup membership was determined from the

OrthoFinder analysis (Appendix 4.2.1). The enzyme HE/DE/E + NotE gene filtering and overlaps (*Figure 5*) were performed using custom scripts. These custom scripts and results of the differential expression testing are available on FigShare (10.6084/m9.figshare.5715151).

## 4.2.3 Comparative methylation analyses

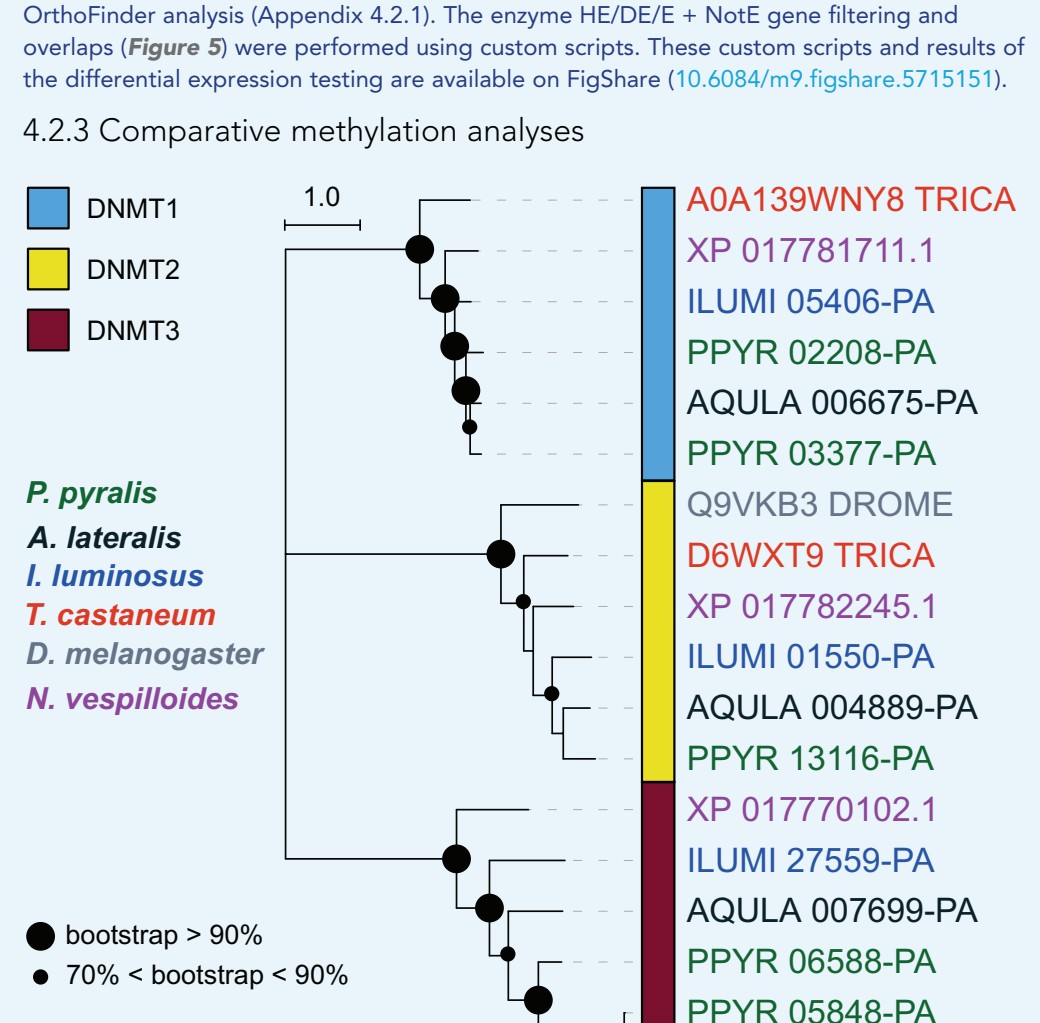

**Appendix 4—figure 2.** DNA and tRNA methyltransferase gene phylogeny. Levels and patterns of mCG in *P. pyralis* are corroborated by the presence of *de novo* and maintenance DNMTs (DNMT3 and DNMT1, respectively). Notably, *P. pyralis* possesses two copies of DNMT1, and 3 copies of DNMT3, in contrast to a single copy of DNMT1 and DNMT3 in the firefly *Aquatica lateralis*. The evolutionary history was inferred by using the Maximum Likelihood method with the LG + G (five gamma categories) (*Le and Gascuel, 2008*). Evolutionary analyses were conducted in MEGA7 (*Kumar et al., 2016*). Size of circles at nodes corresponds to bootstrap support (100 bootstrap replicates). Branch lengths are in amino acid substitutions per site. *T. castaneum = Tribolium castaneum*, *D. melanogaster = Drosophila melanogaster*, *N. vespilloides = Nicrophorus vespilloides*. The multiple sequence alignment and phylogenetic topology are available on FigShare (10.6084/m9.figshare.6531311). DOI: https://doi.org/10.7554/eLife.36495.056

### 4.2.3.2 $CpG_{[O/E]}$ methylation analysis

$CpG_{[O/E]}$ is a non-bisulfite sequencing metric that captures spontaneous deamination of methylated cytosines (*Suzuki et al., 2007*), and confidently recovers the presence/absence of DNA methylation in insects (*Bewick et al., 2017*). In a mixture of loci that are DNA methylated and low to un-methylated, a bimodal distribution of $CpG_{[O/E]}$ values is expected. Conversely, a unimodal distribution is suggestive of a set of loci that are mostly low to un-methylated.

$CpG_{[O/E]}$ was estimated for each annotated gene in the official gene set of *A. lateralis*, *I. luminosus*, and *P. pyralis*. Additionally, $CpG_{[O/E]}$ was estimated for each annotated gene for a true positive and negative coleopteran (*Nicrophorus vespilloides* [https://i5k.nal.usda.gov/

nicrophorus-vespilloides] and *Tribolium castaneum* [https://i5k.nal.usda.gov/tribolium-castaneum], respectively), and a true negative dipteran (*Drosophila melanogaster* [http://flybase.org/]).

The modality of $CpG_{[O/E]}$ distributions was tested using Gaussian mixture modeling in R (https://www.r-project.org/: mclust v5.4 and mixtools v1.0.4). Two modes were modeled for each $CpG_{[O/E]}$ distribution, and the subsequent means and 95% confidence interval (CI) of the means were compared with overlapping or nonoverlapping CI's signifying unimodality or bimodality, respectively.

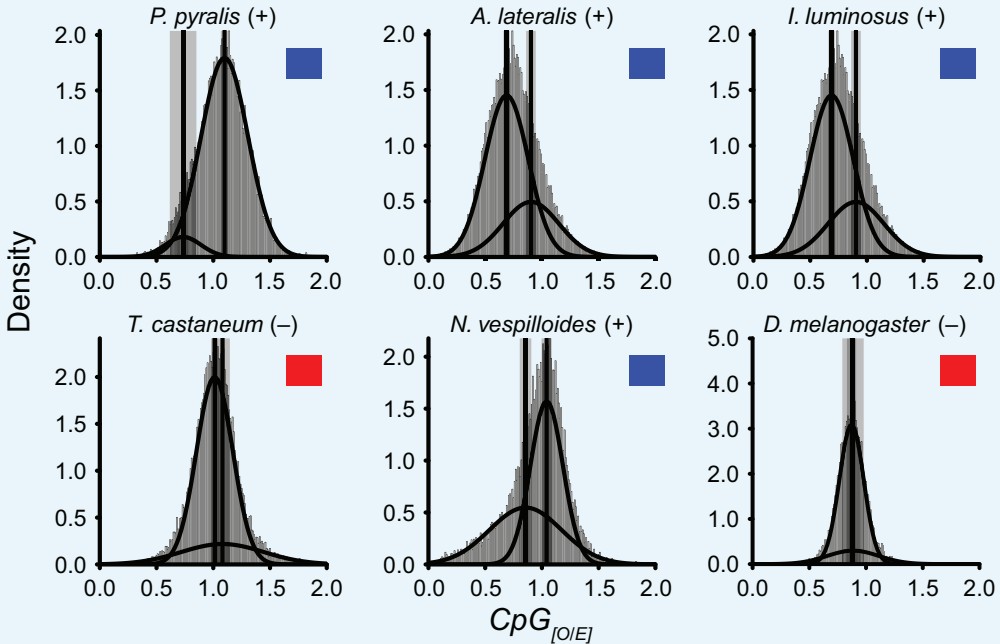

**Appendix 4—figure 3.** Detection of DNA methylation using $CpG_{[O/E]}$.
DOI: https://doi.org/10.7554/eLife.36495.057

Distributions of $CpG_{[O/E]}$ ($CpG_{[O/E]}$ methylation analysis) within sequenced species (*P. pyralis*, *A. lateralis*, and *I. luminosus*), other coleopterans (*N. vespilloides* and *T. castaneum*), and the dipteran *D. melanogaster*. Curves represent two independently modeled Gaussian distributions, and the solid vertical lines and shaded areas represent the mean and 95% confidence interval (CI) of the mean of each distribution. Modality of the distributions accurately predicts presence (+)/blue square or absence (–)/red square of DNA methylation in each species.

### 4.2.4 CYP303 evolutionary analysis (*Figure 6C*)

Candidate P450s were identified using BLASTP (e-value: $1 \times 10^{-20}$) of a *P. pyralis* CYP303 family member (PPYR_OGS1.0: PPYR_14345-PA) against the *P. pyralis*, *A. lateralis*, and *I. luminosus* reference set of peptides, and the *D. melanogaster* (NCBI GCF_000001215.4) and *T. castaneum* (NCBI GCF_000002335.3) geneset peptides. Resulting hits were merged, aligned with MAFFT E-INS-i (v7.243) (*Katoh and Standley, 2013*), and a preliminary neighbor-joining (NJ) tree was generated using MEGA7 (*Kumar et al., 2016*). Genes descending from the common ancestor of the *CYP303* and *CYP304* genes were selected from this NJ tree, and the peptides within this subset re-aligned with MAFFT using the L-INS-i algorithm. Then the maximum likelihood evolutionary history of these genes was inferred within MEGA7 using the LG + G model (five gamma categories (+G, parameter = 2.4805). Initial tree(s) for the heuristic search were obtained automatically by applying Neighbor-Join and BioNJ algorithms to a matrix of pairwise distances estimated using a JTT model, and then selecting the topology with the best log likelihood value. The resulting tree was rooted using *D. melanogaster* Cyp6a17 (NP_652018.1). The tree shown in *Figure 6C* was truncated in Dendroscope

(v3.5.9) (*Huson and Scornavacca, 2012*) to display only the *CYP303* clade. The multiple sequence alignment FASTA files and newick files of the full and truncated tree are available in *Figure 6—source data 1*.

## 4.3 Luciferase evolution analyses

### 4.3.1 Luciferase genetics overview

The gene for firefly luciferase was first isolated from the North American firefly *P. pyralis* (*de Wet et al., 1985*; *Wood et al., 1984*; *de Wet et al., 1987*) and then identified from the Japanese fireflies *Luciola cruciata* (*Masuda et al., 1989*) and *Aquatica lateralis* (*Tatsumi et al., 1992*). To date, firefly luciferase genes have been isolated from more than 30 lampyrid species in the world. Two different types of luciferase genes, *Luc1* and *Luc2*, have been reported from *Photuris pennsylvanica* (*Ye et al., 1997*) (Photurinae), *L. cruciata* (*Oba et al., 2010b*) (Luciolinae), *A. lateralis* (*Oba et al., 2013a*) (Luciolinae), *Luciola parvula* (*Bessho-Uehara and Oba, 2017*) (Luciolinae), and *Pyrocoelia atripennis* (*Bessho-Uehara et al., 2017*) (Lampyrinae).

Luciferase genes have also been isolated from members of the other luminous beetles families: Phengodidae, Rhagophthalmidae, and Elateridae (*Wood et al., 1989*; *Viviani et al., 1999a*; *Viviani et al., 1999b*; *Ohmiya et al., 2000*) with amino acid identities to firefly luciferases at >48% (*Oba, 2014*). The chemical structures of the substrates for these enzymes are identical to firefly luciferin. These results that the bioluminescence systems of luminous beetles are essentially the same, supports a single origin of the bioluminescence in elateroid beetles. Recent molecular analyses based on the mitochondrial genome sequences strongly support a sister relationship between the three luminous families: Lampyridae, Phengodidae, and Rhagophthalmidae (*Timmermans et al., 2010*; *Timmermans and Vogler, 2012*), suggesting the monophyly of Elateroidea and a single origin of the luminescence in the ancestor of these three lineages (*Oba, 2014*). However, ambiguity in the evolutionary relationships among luminous beetles, including luminous Elaterids, does not yet exclude multiple origins.

Molecular analyses have suggested that the origin of Lampyridae was dated back to late Jurassic (*McKenna and Farrell, 2009*) or mid-Cretaceous periods (*Mckenna et al., 2015*). Luciolinae and Lampyrinae was diverged at the basal position of the Lampyridae (*Martin et al., 2017*) and the fossil of the Luciolinae firefly dated at Cretaceous period was discovered in Burmese amber (*Shi et al., 2012*; *Kazantsev, 2015*). Taken together, the divergence of Luciola and Lampyridae is dated back at least 100 Mya.

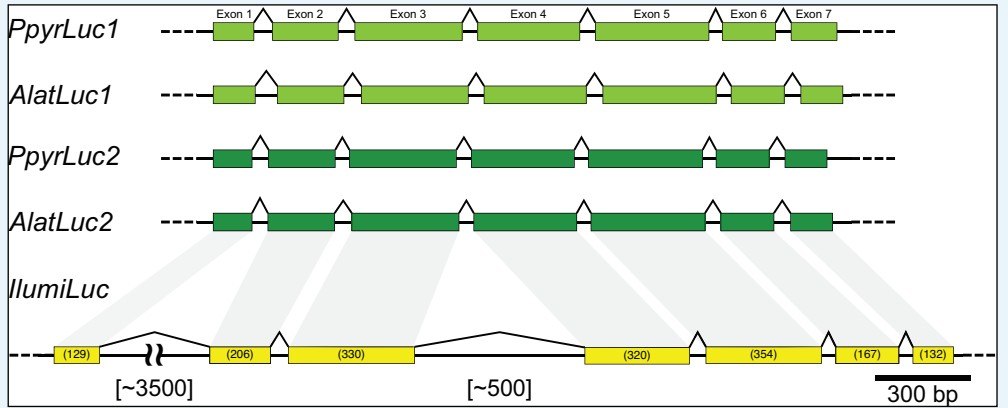

**Appendix 4—figure 4.** Intron-exon structure of beetle luciferases. (A) Intron-exon structure of *P. pyralis* and *A. lateralis Luc1* and *Luc2* from Ppyr1.3 and Alat1.3, and *IlumLuc* from Ilumi1.2. Between fireflies and click-beetles, the structure of the luciferase genes are globally similar, with seven exons, similar intron lengths, and identical splice junction locations (*Appendix 4—figure 5*). The intron-exon structure of *IlumLuc* is consistent with the reported intron-exon structure of *Pyrophorus plagiophthalamus* luciferase (*Velez and Feder, 2006*).

DOI: https://doi.org/10.7554/eLife.36495.058

**Appendix 4—figure 5.** Multiple sequence alignment of firefly luciferase genes. MAFFT (*Katoh and Standley, 2013*) L-INS-i multiple sequence alignment of luciferase gene nucleotide sequences from PpyrOGS1.1 and AlatOGS1.0 demonstrates the location of intron-exon junctions (bolded blue text) is completely conserved amongst the four luciferases. Exonic sequence is capitalized, whereas intronic sequence is lowercase.

DOI: https://doi.org/10.7554/eLife.36495.059

## 4.3.2 Luciferase homolog gene tree (*Figure 3C*)

From our reference genesets, a protein BLAST search detected 24, 20, 32, and two luciferase homologs (E-value $<1 \times 10^{-60}$) to *P. pyralis* luciferase (PpyrLuc1; Genbank accession AAA29795) from the *P. pyralis, A. lateralis, I. luminosus* genesets, and *Drosophila melanogaster*, respectively. We defined the luciferase co-orthology as followings: (1) shows an BLASTP E-value lower than $1.0 \times 10^{-60}$ toward *DmelPACS* (CG6178), (2) phylogenetically sister to *DmelPACS*, which is the most similar gene to firefly luciferase in *D. melanogaster*, based on a preliminary maximum likelihood (ML) phylogenetic reconstruction (*Appendix 4—figure 6*). Preliminary ML phylogenetic reconstruction was performed as follows: The sequences of luciferase homologs from *Mengenilla moldrzyki, Pediculus humanus, Limnephilus lunatus, Ladona fulva, Frankliniella occidentalis, Zootermopsis nevadensis, Onthophagus taurus, Anoplophora glabripennis, Agrilus planipennis, Harpegnathos saltator, Blattella germanica, Acyrthosiphon pisum, Tribolium castaneum, Bombyx mori, Anopheles gambiae,*

*Apis mellifera, Leptinotarsa decemlineata,* and *Dendroctonus ponderosae* were obtained from OrthoDB (https://www.orthodb.org) (*Zdobnov et al., 2017*). The sequences which show 99% similarity were filtered by CD-HIT (v4.7) (*Fu et al., 2012*). The resulting sequences and beetle luciferases were aligned using (MAFFT v7.309) (*Katoh and Standley, 2013*) using the BLOSUM62 matrix and filtered for spurious sequences and poorly aligned regions using trimAl (v.1.2rev59) (*Capella-Gutiérrez et al., 2009*) (parameters: -strict). The final alignment was 385 blocks and 264 sequences. Then, the best fit amino acid substitution model, LG + F Gamma, was estimated by Aminosan (v1.0.2016.11.07) (*Tanabe, 2011*) using the Akaike Information Criterion. Finally, a maximum likelihood gene phylogeny was estimated using RAxML (v8.2.9; 100 bootstrap replicates) (*Stamatakis, 2006*). Supporting files such as multiple sequence alignment, gene accession numbers, and other annotations are available on FigShare (DOI: 10.6084/m9.figshare.6687086).

To more closely examine luciferase evolution, an independent maximum likelihood gene tree was constructed for luciferase co-orthologous genes defined above (highlighted clade as grey in *Appendix 4—figure 6*) with well important genes: non-luminescent luciferase homolog from two model insect *D. melanogaster* (DmelPACS and DmelACS as outgroup) and *T. castaneum* (TcasPACSs and TcasACSs), biochemically characterized non-luminescent PACS (LcruPACS1 and LcruPACS2 from *Luciola cruciata,* DmelPACS, and PangPACS from *Pyrophorus angustus*) and biochemically characterized luciferases from Lampyrinae (PatrLuc1 and 2: *Pyrocoelia atripennis*), Ototoretinae (DaxiLuc1 and SazuLuc1: *Drilaster axillaris* and *Stenocladius azumai),* Phausis (PretLuc1: *Phausis reticulata*) from Lampyridae, Rhagophthalmidae (RohbLuc: *Rhagophthalmus ohbai*), Phengodeidae (PhirLucG and R: *Phrixothrix hirtus*), and Elateridae (PangLucD and V: *P. angustus*). Then co-orthologous genes were confirmed to be phylogenetically sister to *DmelPACS* (CG6178) and their evolution examined using a maximum likelihood (ML) gene phylogeny approach. First, amino acid sequences were aligned using (MAFFT v7.308) (*Katoh and Standley, 2013*) using the BLOSUM62 matrix (parameters: gap open penalty = 1.53, offset value = 0.123) and filtered for spurious sequences and poorly aligned regions using trimAl (*Capella-Gutiérrez et al., 2009*) (parameters: gt = 0.8). The final alignment was 533 blocks and 67 sequences. Then, the best fit amino acid substitution model, LG + F Gamma, was estimated by Aminosan (v1.0.2016.11.07) (*Tanabe, 2011*) using the Akaike Information Criterion. Finally, a maximum likelihood gene phylogeny was estimated using RAxML (v8.2.9; 100 bootstrap replicates) (*Stamatakis, 2006*). The tree was rooted using *DmelACS* as an outgroup. The peroxisomal targeting signal 1 (PST1) was predicted using the regular expressions provided by the Eukaryotic Linear Motif database (*Dinkel et al., 2012*) and verified using the mendel PTS1 prediction server (*Neuberger et al., 2003*; *Neuberger et al., 2017*). Supporting files such as multiple sequence alignment, gene accession numbers, and other annotation and expression values are available as *Figure 3—source data 1*.

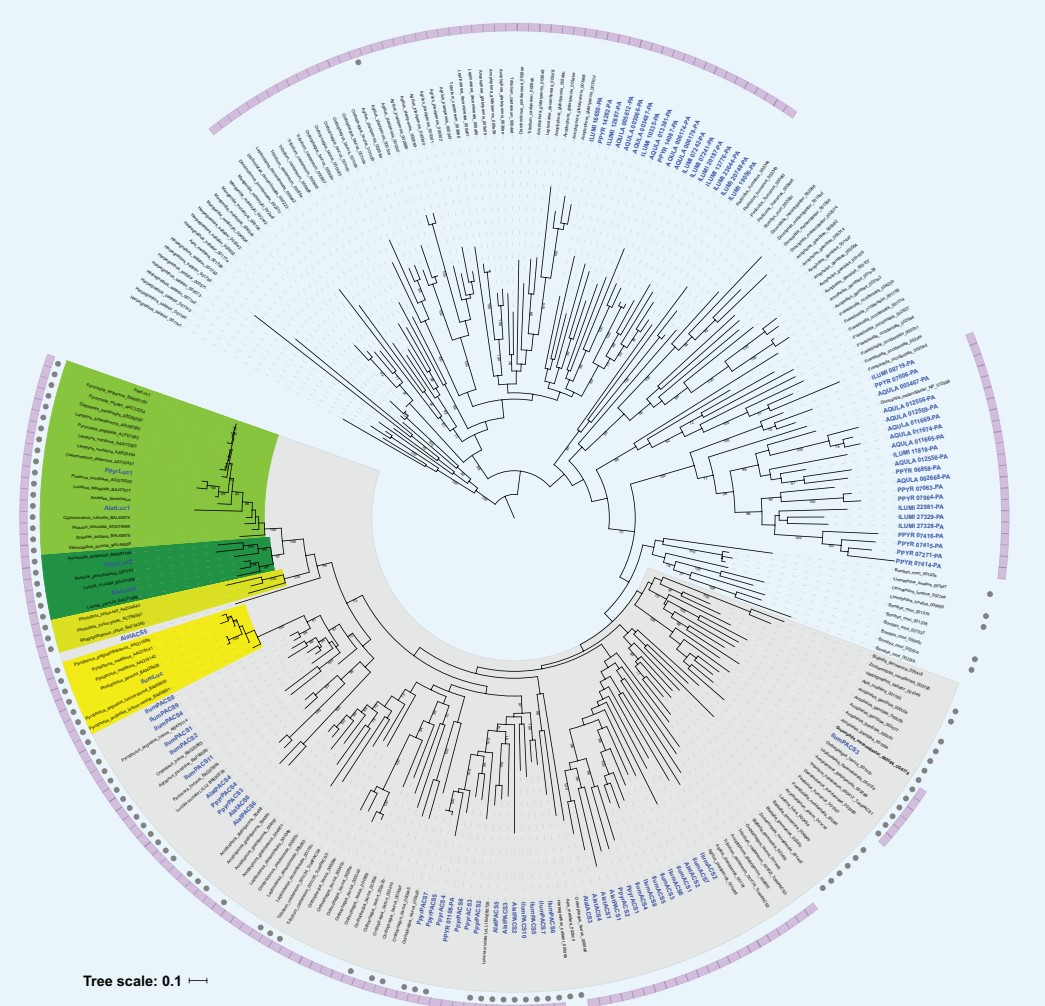

**Appendix 4—figure 6.** Preliminary maximum likelihood phylogeny of luciferase homologs. A preliminary maximum likelihood tree was reconstructed from a 385 amino acid multiple sequence alignment, generated via a BLASTP and orthoDB search using *P. pyralis* luciferase as query (e-value: $1.0 \times 10^{-60}$). Members of the clade that includes both known firefly luciferase and CG6178 of *D. melanogaster* (bold) are defined as luciferase co-orthologous genes (highlighted in gray), and were selected and used for the independent maximum likelihood analysis in *Figure 3C* (Appendix 4.3.2). Branch length represents substitutions per site. Genes found from this study are indicated in blue. Lampyridae Luc1-type and Luc2-type luciferases are highlighted in yellow-green and green. Rhagophthalmidae and Phengodidae luciferases are highlighted in lime-green. Elateridae luciferases are highlighted in yellow. Genbank accession numbers of luciferase orthologs genes are indicated after the species name. OrthoDB taxon and protein IDs of luciferase co-orthologs are indicated after species name. Bootstrap values are indicated on the nodes. The genes from Coleoptera are indicated as purple strip. Grey closed circles indicate genes that have PTS1.
DOI: https://doi.org/10.7554/eLife.36495.060

## 4.3.3 Ancestral state reconstruction of luciferase activity (*Figure 4A*)

We performed an ancestral character state reconstruction of luciferase activity on the luciferase homolog gene tree within Mesquite (v3.31) (*Maddison and Maddison, 2017*), using an unordered parsimony analysis, and maximum likelihood (ML) analyses. First, the gene tree from *Figure 3C* in Newick format was filtered using Dendroscope(v3.5.9) (*Huson and Scornavacca, 2012*) to include only the clade descending from the common ancestor of TcasPACS4 and PpyrLuc1. TcasPACS4 was set as the rooting outgroup. Luciferase activity of

these extant genes was coded as a character state within Mesquite with: (0 = no luciferase activity, 1 = luciferase activity, ?=undetermined). A gene was given the 1-state if it had been previously characterized as having luciferase activity, or was directly orthologous to a gene with previously characterized luciferase activity against firefly D-luciferin. A gene was given the 0-state if it had been previously characterized as a non-luciferase, or was directly orthologous to a gene previously characterized to not have luciferase activity towards firefly D-luciferin. The non-luciferase activity determination for TcasPACS4 was inferred via orthology to the previously characterized non-luciferase *Tenebrio molitor* enzyme Tm-LL2 (*Oba et al., 2006b*). The non-luciferase activity of AlatPACS4 (AQULA_005073-PA) was inferred via orthology to the non-luciferase enzyme LcruPACS2 (*Oba et al., 2006a*). The non-luciferase activity of IlumPACS4 (ILUMI_06433-PA) was inferred via orthology to the non-luciferase *Pyrophorus angustus* enzyme PangPACS (*Oba et al., 2010a*; *Mofford et al., 2017*). IlumLuc luciferase activity was inferred via orthology to the *P. angustus* dorsal and ventral luciferases (*Oba et al., 2010a*). The luciferase activity of PpyrLuc2 (PPYR_00002-PA) was inferred via orthology to other Luc2s, e.g. *A. lateralis* Luc2 (*Oba et al., 2013a*). The luciferase activity of the included phengodid (*Viviani et al., 1999a*; *Arnoldi et al., 2010*; *Amaral et al., 2017*), rhagophthalmid (*Ohmiya et al., 2000*; *Li et al., 2007*), and firefly luciferases (*Oba et al., 2012*; *Viviani et al., 2011*; *Branchini et al., 2017*) were annotated from the literature. We then reconstructed the ancestral luciferase activity character state over the tree, using an unordered parsimony model, and a maximum likelihood (ML) model. ML analyses were performed under the AsymmMk model with default parameters (i.e. Root State Frequencies Same as Equilibrium). NEXUS files with presented parsimony and ML reconstructions are available as *Figure 4— source data 1*.

### 4.3.4 Testing for ancestral selection of elaterid ancestral luciferase (*Figure 4B*)

**Discovery**

Peptide sequences for elaterid luciferase homologs descending from the putative common ancestor of firefly and elaterid luciferase as determined by a preliminary maximum likelihood molecular evolution analysis of luciferase homologs (not shown), were selected from Uniprot, whereas their respective CDS sequences were selected from the European Nucleotide Archive (ENA) or National Center for Biotechnology Information (NCBI). These sequences include: The dorsal (PangLucD; ENA ID = BAI66600.1) and ventral (PangLucV; ENA ID = BAI66601.1) luciferases, and a luciferase-like homolog without luciferase-activity (PangPACS; ENA ID = BAI66602.1) from *Pyrophorus angustus* (*Oba et al., 2010a*), and two unpublished but database deposited luciferase homologs without luciferase-activity (data not shown) from *Cryptalaus berus* (CberPACS; ENA ID = BAQ25863.1) and *Pectocera fortunei fortunei* (PffPACS; ENA ID = BAQ25864.1). The peptide and CDS sequence of the *Pyrearinus termitilluminans* luciferase (PtermLuc) were manually transcribed from the literature (*Viviani et al., 1999b*), as these sequences were seemingly never deposited in a publically accessible sequence database. The dorsal (PmeLucD; NCBI ID = AF545854.1) and ventral (PmeLucV; NCBI ID = AF545853.1) luciferases of *Pyrophorus mellifluus* (*Stolz et al., 2003*). The dorsal (AF543412.1) and ventral (AF543401.1) luciferase alleles of *Pyrophorus plagiophthalmus* (*Stolz et al., 2003*), which were most similar to that of *Pyrophorus mellifluus* in a maximum likelihood analysis (data not shown). The CDS sequence of the complete *I. luminosus* luciferase (IlumLuc; ILUMI_00001-PA), two closely related paralogs (IlumPACS9: ILUMI_26849-PA, IlumPACS8: ILUMI_26848-PA), and two other paralogs (IlumPACS2: ILUMI_02534-PA; IlumPACS1: ILUMI_06433-PA), and the CDS for *Photinus pyralis* luciferase (PpyrLuc1: PPYR_00001-PA) were added as an outgroup sequence.

**Alignment and Gene Phylogeny**

The 20 merged CDS sequences were multiple-sequenced-aligned with MUSCLE (*Edgar, 2004*) in 'codon' mode within MEGA7 (*Kumar et al., 2016*), using parameters (Gap Open = −0.2.9; Gap Extend = 0; Hydrophobicity Multiplier 1.2, Clustering Method = UPGMB, Min Diag Length (lambda) = 24, Genetic Code = Standard), producing a nucleotide multiple-sequence-alignment (MSA). A maximum likelihood gene tree was produced from the nucleotide MSA within MEGA7 using the General Time Reversible model (*Nei and Kumar,*

*2000*), with five gamma categories (+G, parameter = 0.8692). The analysis involved 20 nucleotide sequences. Codon positions included were 1st + 2nd + 3rd + Noncoding. There were a total of 1659 positions in the final dataset. Initial tree(s) for the heuristic search were obtained automatically by applying Neighbor-Join and BioNJ algorithms to a matrix of pairwise distances estimated using the Maximum Composite Likelihood (MCL) approach, and then selecting the topology with the superior log likelihood value. The tree with the highest log likelihood ($-16392.22$) was selected. 1000 bootstrap replicates were performed to evaluate the topology, and the percentage of trees in which the associated taxa clustered together is shown next to the branches in *Figure 4B*.

### Tests of selection: aBSREL

An adaptive branch-site REL test for episodic diversification was performed on the previously mentioned gene-tree and nucleotide MSA using the adaptive branch-site REL test for episodic diversification (aBSREL) method (*Smith et al., 2015*) within the HyPhy program (v2.3.11) (*Pond et al., 2005*). The input MSA contained 20 sequences with 553 sites (codons). All 37 branches of the gene phylogeny were formally tested for diversifying selection. The aBSREL analysis found evidence of episodic diversifying selection on 3 out of 37 branches in the phylogeny. Significance was assessed using the Likelihood Ratio Test at a threshold of $p \leq 0.01$, after the Holm-Bonferroni correction for multiple hypothesis testing. The intermediate files and results of this analysis, including the nucleotide MSA, GTR based gene-tree, and aBSREL produced adaptive rate class model gene tree are available as *Figure 4—source data 2*.

### Tests of selection: MEME

After identification of the selected branch via the aBSREL method, we turned to the MEME method within the HyPhy program (v2.3.11) (*Pond et al., 2005*), to identify those sites which may have adaptively evolved. We tested the branch leading to EAncLuc, which was previously identified as under selection in the aBSREL analysis. A single partition was recovered with 28 sites under episodic diversifying positive selection at $p<=0.1$ (*Appendix 4—table 5*). Input files and full results are available on FigShare (10.6084/m9.figshare.6626651).

### Tests of selection: PAML-BEB

To validate our findings from aBSREL and MEME using a different method, we applied Phylogenetic Analysis by Maximum Likelihood (PAML) branch by site analysis to the luciferase sequences. We tested the alternative hypothesis, that there is a class of sites under selection ($\omega > 1$) on the EAncLuc ancestral branch identified as under selection in the aBSREL analysis, against the null hypotheses, that all classes of sites on all branches are evolving either under constraint ($\omega < 1$) or neutrality ($\omega = 1$). A likelihood ratio test supported the alternative hypothesis, that 13% of sites in luciferase were in a positively selected class ($\omega = 3.25$). Subsequent Bayes Empirical Bayes (BEB) estimation identified 31 sites with evidence of selection on these branches, 5 of which were significant. Full results are available on FigShare (10.6084/m9.figshare.6725081).

### Tests of selection: Overlap

Nineteen of the overall sites were shared between the MEME analysis, and are shown in *Appendix 4—table 5*. The frequency of extant amino acids at these sites are shown in *Appendix 4—figure 7*.

**Appendix 4—table 4.** Results of PAML branch x sites analysis. Proportion indicates the proportion of sites in each site class (0, 1, 2a, 2b). Site classes 0 and 1 are those in the constrained and neutral classes, respectively. 2a are sites that were constrained on the background branches, but are either neutral (H0) or in the selective class (HA) on the foreground branches. 2b are sites that were neutral on the background branches, but are either neutral (H0) or in the selective class (HA) on the foreground branches.

| Hypothesis | Site class: | 0 | 1 | 2a | 2b | lnL |
|---|---|---|---|---|---|---|
| H0: no selection | proportion | 0.62 | 0.14 | 0.18 | 0.04 | −15888.16 |
| | background $\omega$ | 0.12 | 1 | 0.12 | 1 | |
| | foreground $\omega$ | 0.12 | 1 | 1 | 1 | |
| HA: selection | proportion | 0.71 | 0.15 | 0.11 | 0.02 | −15833.50* |
| | background $\omega$ | 0.12 | 1 | 0.12 | 1 | |
| | foreground $\omega$ | 0.12 | 1 | 3.25 | 3.25 | |

*significant (LRT: 9.32, df = 1)
DOI: https://doi.org/10.7554/eLife.36495.061

**Appendix 4—table 5.** Sites identified as under selection on foreground branches using both Bayes Empirical Bayes (BEB) and Mixed Effects Model of Evolution (MEME).

| Site numbering | | | MEME[2] | | | | | PAML-BEB | |
|---|---|---|---|---|---|---|---|---|---|
| MSA | IlumLuc | IlumLuc site AA[1] | $\alpha$ | $\beta+$ | LRT | Episodic selection p-value | # branches | BEB site class probability | BEB significance |
| 28 | 28 | M | | | | | | 0.986 | * |
| 34 | 34 | K | 0.47 | 23.5 | 4.1 | 0.0603 | 0 | | |
| 41 | 41 | Q | | | | | | 0.5 | |
| 46 | 44 | V | 0 | 3 | 4.5 | 0.0485 | 0 | | |
| 49 | 47 | I | 0.93 | 792.4 | 3.8 | 0.0692 | 0 | | |
| 50 | 48 | G | 0.57 | 3332.3 | 4.8 | 0.0427 | 0 | 0.836 | |
| 72 | 70 | N | 0.55 | 3333.1 | 3.1 | 0.0998 | 0 | 0.776 | |
| 77 | 75 | M | | | | | | 0.964 | * |
| 85 | 83 | A | | | | | | 0.962 | * |
| 89 | 87 | K | | | | | | 0.958 | * |
| 99 | 97 | W | | | | | | 0.598 | |
| 105 | 103 | V | 0.44 | 6.8 | 4.3 | 0.0549 | 0 | 0.768 | |
| 118 | 116 | C | 0.3 | 3333.1 | 7.4 | 0.0109 | 1 | | |
| 122 | 120 | G | | | | | | 0.82 | |
| 146 | 144 | L | 0.34 | 12.8 | 4.9 | 0.039 | 0 | | |
| 147 | 145 | G | 0.75 | 3333.6 | 5.9 | 0.0236 | 0 | | |
| 172 | 170 | A | | | | | | 0.698 | |
| 189 | 185 | F | | | | | | 0.534 | |
| 223 | 219 | L | | | | | | 0.507 | |
| 226 | 222 | T | 1.44 | 29.6 | 4.8 | 0.0427 | 0 | 0.889 | |
| 234 | 230 | I | 1.13 | 9.6 | 3.1 | 0.0991 | 0 | 0.613 | |
| 279 | 275 | A | | | | | | 0.559 | |
| 290 | 286 | N | 0.92 | 3333 | 4 | 0.064 | 0 | | |
| 315 | 311 | L | 0.69 | 29.5 | 5.1 | 0.0362 | 0 | 0.884 | |
| 329 | 325 | L | | | | | | 0.766 | |

*Appendix 4—table 5 continued*

| Site numbering | | | MEME[2] | | | | | PAML-BEB | |
|---|---|---|---|---|---|---|---|---|---|
| MSA | IlumLuc | IlumLuc site AA[1] | α | β+ | LRT | Episodic selection p-value | # branches | BEB site class probability | BEB significance |
| 337 | 333 | P | 0.26 | 13.3 | 6.3 | 0.0198 | 0 | | |
| 341 | 337 | C | | | | | | 0.812 | |
| 365 | 361 | L | 0.58 | 7.6 | 4.4 | 0.052 | 0 | 0.912 | |
| 369 | 365 | T | 0.21 | 6.8 | 6.6 | 0.0169 | 0 | 0.843 | |
| 379 | 375 | R | | | | | | 0.932 | |
| 383 | 379 | E | 0 | 2.8 | 4.1 | 0.0594 | 0 | | |
| 389 | 385 | Q | | | | | | 0.792 | |
| 398 | 394 | P | 0.96 | 1999.2 | 4.5 | 0.05 | 0 | 0.951 | * |
| 401 | 397 | S | | | | | | 0.617 | |
| 406 | 402 | N | 0.58 | 5.5 | 3.7 | 0.0745 | 0 | 0.949 | |
| 423 | 419 | S | 0.67 | 1574.6 | 4.7 | 0.043 | 0 | 0.569 | |
| 432 | 428 | E | 0 | 2.9 | 3.1 | 0.0999 | 1 | | |
| 441 | 437 | Y | 1.43 | 39.3 | 4.2 | 0.0573 | 0 | 0.912 | |
| 478 | 474 | V | 0 | 10.3 | 6.9 | 0.0139 | 1 | 0.646 | |
| 502 | 498 | Y | 0.5 | 1790.4 | 4.9 | 0.0393 | 0 | 0.583 | |
| 508 | 504 | R | | | | | | 0.519 | |
| 528 | 524 | N | 0 | 2.2 | 3.6 | 0.0772 | 0 | | |
| 541 | 537 | Q | 0 | 1999.2 | 10.4 | 0.0024 | 1 | | |
| 542 | 538 | L | 0.56 | 68 | 6.3 | 0.0197 | 0 | | |
| 550 | 542 | T | 0.74 | 3332.9 | 4.3 | 0.0541 | 0 | | |

[1] = amino acid. [2] =All recovered sites in a single partition with a p+ value of 1.000.

DOI: https://doi.org/10.7554/eLife.36495.062

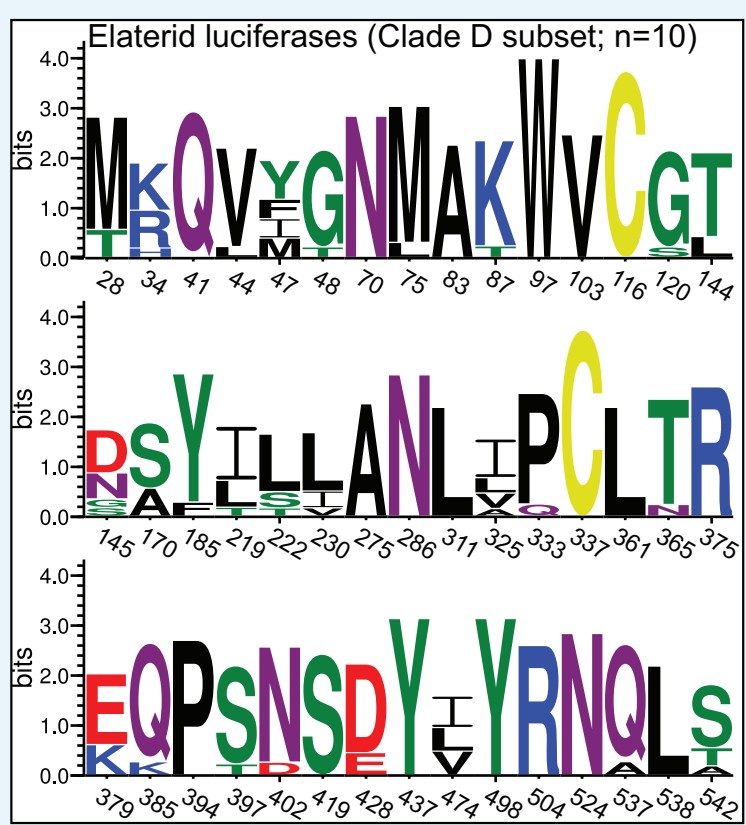

**Appendix 4—figure 7.** Amino acid variation at sites recovered in selection analysis. Amino acid variation of extant Elaterid luciferases (Clade D 'Eluc' subset; *Figure 3*) at all sites recovered via both the MEME and PAML-BEB selection analysis (*Appendix 4—table 5*). Site numbering relative to *IlumLuc*. Figure produced with seqkit (*Shen et al., 2016*) and WebLogo(v3.6.0) (*Crooks et al., 2004*).
DOI: https://doi.org/10.7554/eLife.36495.063

## 4.4 Non-enzyme highly and differentially expressed genes of the firefly lantern

PPYR_04589, a predicted fatty acid binding protein is almost certainly orthologous to the light organ fatty acid binding protein reported from *Luciola cerata* (*Goh and Li, 2011*). This fatty acid binding protein was previously reported to bind strongly to fatty acids, and weakly to luciferin. Notably, PPYR_04589 is the most highly expressed gene in the *P. pyralis* adult lantern, ahead of firefly luciferase. Three G-coupled protein receptors (GCPRs) with similarity to annotated octopamine/tyramine receptors were also detected to be highly and differentially expressed in the *P. pyralis* light organ (PPYR_11673-PA, PPYR_11364-PA, PPYR_12266-PA). Octopamine is known to be the key effector neurotransmitter of the adult and larval firefly lantern and this identified GPCR likely serves as the upstream receptor of octopamine activated adenylate cyclase, previously reported as abundant in *P. pyralis* lanterns (*Nathanson et al., 1989*).

The neurobiology of flash control, including regulation of flash pattern and intensity, is a fascinating area of behavioral research. Our data generate new hypotheses regarding the molecular players in flash control. A particularly interesting highly and differentially expressed gene in both *P. pyralis* and *A. lateralis* is the full length 'octopamine binding secreted hemocyanin'(PPYR_14966; AQULA_008529; *Appendix 4—table 6*) previously identified from *P. pyralis* light organ extracts via photoaffinity labeling with an octopamine analog and partial N-terminal Edman degradation (*Nathanson et al., 1989*). This protein is intriguing as hemocyanins are typically thought to be oxygen binding. We speculate that this octopamine binding secreted hemocyanin, previous demonstrated to be abundant,

**Appendix 4—table 6.** Highly expressed (HE), differentially expressed (DE), non-enzyme annotated (NotE), lantern genes whose closest relative in the opposite species is also HE, DE, NotE. BSN-TPM = between sample normalized TPM.

| P. pyralis ID (OGS1.1) | Predicted function | Ppyr expression rank | Ppyr BSN-TPM | Orthogroup | Alat expression rank | Alat BSN-TPM | A. lateralis ID (OGS1.0) |
|---|---|---|---|---|---|---|---|
| PPYR_04589 | Fatty-acid binding protein | 1 | 70912 | OG0000524 | 2 | 31943 | AQULA_005253 |
| PPYR_04589 | Fatty-acid binding protein | 1 | 70912 | OG0000524 | 8 | 10464 | AQULA_005257 |
| PPYR_04589 | Fatty-acid binding protein | 1 | 70912 | OG0000524 | 10 | 8520 | AQULA_005259 |
| PPYR_05098 | Peroxisomal biogenesis factor 11 (PEX11) | 15 | 4005 | OG0001490 | 26 | 3294 | AQULA_005466 |
| PPYR_14966 | Octopamine binding secreted hemocyanin | 34 | 2353 | OG0001369 | 21 | 3658 | AQULA_008529 |
| PPYR_11733 | MFS transporter superfamily | 42 | 1853 | OG0000980 | 84 | 1335 | AQULA_012209 |
| PPYR_07633 | Reticulon | 56 | 1556 | OG0004764 | 109 | 1123 | AQULA_005090 |
| PPYR_09394 | lysosomal Cystine Transporter | 87 | 1098 | OG0000847 | 69 | 1494 | AQULA_009474 |
| PPYR_08979 | PF03670 Uncharacterised protein family | 114 | 860 | OG0003009 | 340 | 411 | AQULA_012099 |
| PPYR_05852 | Vacuolar ATP synthase 16 kDa subunit | 118 | 836 | OG0001039 | 287 | 475 | AQULA_001418 |
| PPYR_11443 | RNA-binding domain superfamily | 134 | 782 | OG0004268 | 1221 | 108 | AQULA_003174 |
| PPYR_02465 | Peroxin 13 | 189 | 581 | OG0001667 | 196 | 710 | AQULA_010288 |
| PPYR_06160 | V-type ATPase, V0 complex | 209 | 543 | OG0000381 | 541 | 251 | AQULA_000400 |
| PPYR_11300 | Mitochondrial outer membrane translocase complex | 232 | 509 | OG0004557 | 402 | 349 | AQULA_004355 |
| PPYR_08174 | PF03650 Uncharacterised protein family | 249 | 475 | OG0000647 | 163 | 836 | AQULA_009867 |
| PPYR_04602 | Leucine-rich repeat domain superfamily | 262 | 459 | OG0004508 | 378 | 373 | AQULA_004134 |
| PPYR_01678 | MFS transporter superfamily | 264 | 458 | OG0000347 | 455 | 302 | AQULA_002485 |
| PPYR_08192 | PF03650 Uncharacterised protein family | 271 | 453 | OG0000647 | 163 | 836 | AQULA_009867 |
| PPYR_13497 | Mitochondrial substrate/solute carrier | 285 | 438 | OG0004402 | 379 | 372 | AQULA_003680 |
| PPYR_08917 | LysM domain superfamily | 315 | 398 | OG0002035 | 483 | 278 | AQULA_002396 |
| PPYR_04424 | Domain of unknown function (DUF4782) | 332 | 379 | OG0007447 | 1296 | 101 | AQULA_013946 |
| PPYR_08278 | Protein of unknown function DUF1151 | 348 | 365 | OG0001306 | 430 | 325 | AQULA_000628 |

*Appendix 4—table 6 continued on next page*

*Appendix 4—table 6 continued*

| P. pyralis ID (OGS1.1) | Predicted function | Ppyr expression rank | Ppyr BSN-TPM | Orthogroup | Alat expression rank | Alat BSN-TPM | A. lateralis ID (OGS1.0) |
|---|---|---|---|---|---|---|---|
| PPYR_13261 | Major facilitator superfamily | 404 | 309 | OG0000410 | 158 | 862 | AQULA_007558 |
| PPYR_14848 | Homeobox-like domain superfamily - Abdominal-B-like | 413 | 304 | OG0001849 | 737 | 186 | AQULA_000483 |
| PPYR_11623 | GNS1/SUR4 family | 446 | 281 | OG0008603 | 308 | 449 | AQULA_009341 |
| PPYR_01828 | TLDc domain | 490 | 250 | OG0002035 | 483 | 278 | AQULA_002396 |
| PPYR_03449 | Innexin | 533 | 230 | OG0000992 | 619 | 219 | AQULA_013430 |
| PPYR_05702 | Sulfate permease family | 543 | 225 | OG0007205 | 396 | 357 | AQULA_013064 |
| PPYR_05993 | V-type ATPase, V0 complex, 116 kDa subunit family | 579 | 210 | OG0000381 | 541 | 251 | AQULA_000400 |
| PPYR_04179 | Haemolymph juvenile hormone binding protein | 606 | 202 | OG0002916 | 879 | 152 | AQULA_011187 |
| PPYR_08298 | Peroxisomal membrane protein (Pex16) | 623 | 198 | OG0007339 | 395 | 358 | AQULA_013536 |
| PPYR_06294 | Homeobox-like domain superfamily - Abdominal-B-like | 627 | 197 | OG0001849 | 737 | 186 | AQULA_000483 |
| PPYR_05397 | PDZ superfamily | 773 | 164 | OG0006975 | 367 | 379 | AQULA_012321 |
| PPYR_12625 | Homeobox domain | 796 | 160 | OG0002661 | 1395 | 95 | AQULA_008665 |
| PPYR_08494 | Armadillo-type fold | 846 | 152 | OG0001600 | 986 | 133 | AQULA_008183 |
| PPYR_09217 | Haemolymph juvenile hormone binding protein | 853 | 151 | OG0001089 | 441 | 316 | AQULA_003304 |
| PPYR_01677 | MFS transporter superfamily | 1234 | 108 | OG0000347 | 455 | 302 | AQULA_002485 |

DOI: https://doi.org/10.7554/eLife.36495.064

octopamine binding, and secreted from the lantern (presumably into the hemolymph of the light organ), could be triggered to release oxygen upon octopamine binding, thereby providing a triggerable $O_2$ store within the light organ under control of neurotransmitter involved in flash control. As $O_2$ is believed to be limiting in the adult light reaction, such a release of $O_2$ could enhance flash intensity or accelerate flash kinetics. Further research is required to test this hypothesis.

## Orthogroup 698

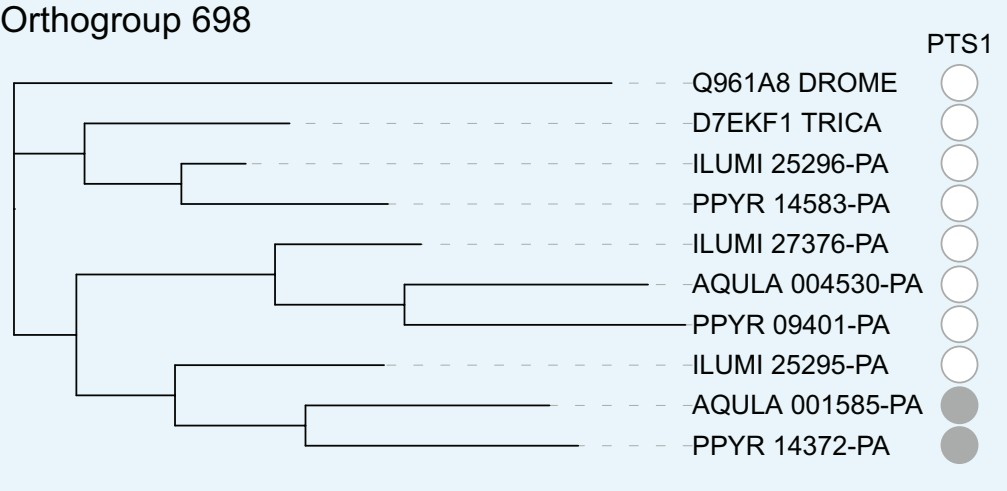

**Appendix 4—figure 8.** Maximum likelihood gene tree of the combined adenylyl-sulfate kinase and sulfate adenylyltransferase (ASKSA) orthogroup. Peptide sequences from *P. pyralis*, *A. lateralis*, *I. luminosus*, *T. castaneum*, and *D. melanogaster* were clustered (orthogroup # 698), multiple sequence aligned, and refactored into a species rooted maximum likelihood tree, via the OrthoFinder pipeline (Appendix 4.2.1). As this is a genome-wide analysis where bootstrap replicates would be computationally prohibitive, no bootstrap replicates were performed to evaluate the support of the tree topology. PTS1 sequences were predicted from the peptide sequence using the PTS1 predictor server (**Neuberger et al., 2017**). Figure produced with iTOL (**Letunic and Bork, 2016**).
DOI: https://doi.org/10.7554/eLife.36495.065

### 4.5 Opsin analysis

Opsins are G-protein-coupled receptors that, together with a bound chromophore, form visual pigments that detect light, reviewed here (**Briscoe and Chittka, 2001**). While opsin genes are known for their expression in photoreceptors and function in vision, they have also been found to be expressed in other tissues, suggesting non-visual functions in some cases. Insects generally use rhabdomeric opsins (r-opsins) for vision, while mammals generally use ciliary opsins (c-opsins) for vision, products of an ancient gene duplication (**Briscoe and Chittka, 2001**; **Porter et al., 2012**). Both insects and mammals may retain the alternate opsin type, generally in a non-visual capacity. The ancestral insect is hypothesized to have three visual opsins - one sensitive to long-wavelengths of light (LW), one to blue-wavelengths (B), and one to ultraviolet light (UV). Previously, two opsins, one with sequence similarity to other insect LW opsins and one with similarity to other insect UV opsins, were identified as highly expressed in firefly heads (**Sander and Hall, 2015**; **Martin et al., 2015**). A likely non-visual c-opsin was also detected, although not highly expressed (**Sander and Hall, 2015**; **Martin et al., 2015**).

To confirm the previously documented opsin presence and expression patterns, we collected candidate opsin genes via BLASTP searches (e-value threshold: $1 \times 10^{-20}$) of the PPYR_OGS1.0, AQULA_OGS1.0 and ILUMI_OGS1.0 reference genesets against UV opsin of *P. pyralis* (Genbank Accession: ALB48839.1), as well as collected non-firefly opsin sequences via literature searches, followed by maximum likelihood phylogenetic reconstruction

(*Appendix 4—figure 9A*), and expression analyses of the opsins (*Appendix 4—figure 9B*.B). The amino acid sequences of opsin were multiple aligned using MAFFT and trimmed using trimAL (parameters: -gt 0.5). The amino acid substitution model for ML analysis was estimated using Aminosan (v1.0.2016.11.07) (*Tanabe, 2011*). In *P. pyralis*, *A. lateralis*, and *I. luminosus*, we detected three r-opsins, including LW, UV, and an r-opsin homologous to Drosophila *Rh7* opsin, and one c-opsin. While LW and UV opsins were highly and differentially expressed in heads of both fireflies, c-opsin was lowly expressed, in *P. pyralis* head tissue only (*Appendix 4—figure 9B*). In contrast, *Rh7* was not expressed in the *P. pyralis* light organ, but was differentially expressed in the light organ of *A. lateralis* (*Appendix 4—figure 9B*). The detection of *Rh7* in our genomes is unusual in beetles (*Feuda et al., 2016*), although emerging genomic resources across the order have detected it in two taxa: *Anoplophora glabripennis* (*McKenna et al., 2016*) and *Leptinotarsa decemlineata* (*Schoville et al., 2017*). *Rh7* has an enigmatic function - a recent study in *Drosophila melanogaster* showed that *Rh7* is expressed in the brain, functions in circadian photoentrainment, and has broad UV-to-visible spectrum sensitivity (*Ni et al., 2017*; *Sakai et al., 2017*). Extraocular opsin expression has been detected in other eukaryotes: a photosensory organ is located in the genitalia at the posterior abdominal segments in butterfly (Lepidoptera) (*Arikawa and Aoki, 1982*). In the bioluminescent Ctenophore *Mnemiopsis leidyi*, three c-opsins are co-expressed with the luminous photoprotein in the photophores (*Schnitzler et al., 2012*). In the bobtail squid, *Euprymna scolopes,* one of the c-opsin isoforms is expressed in the bacterial symbiotic light organ (*Tong et al., 2009*; *Pankey et al., 2014*). Thus, it is possible that *Rh7* has a photo sensory function in the lantern of *A. lateralis*, although this putative function is seemingly not conserved in *P. pyralis*. Future study will confirm and further explore the biological, physiological, and evolutionary significance of *Rh7* expression in the light organ across firefly taxa.

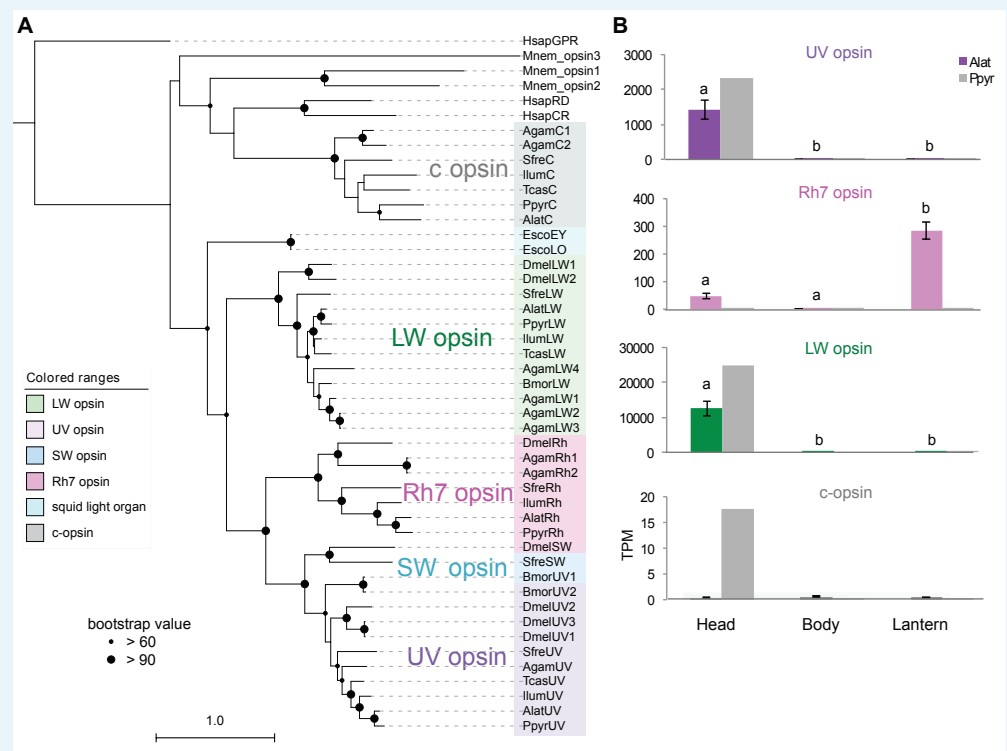

**Appendix 4—figure 9.** ML tree and gene expression levels of opsin genes.
DOI: https://doi.org/10.7554/eLife.36495.066

## 4.6 LC-HRAM-MS of lucibufagin content in *P. pyralis*, *A. lateralis*, and *I. luminosus*

We assayed the hemolymph of adult *P. pyralis* and *A. lateralis*, as well as body extracts from *P. pyralis* and *A. lateralis* larvae, and *I. luminosus* adult male thorax, for lucibufagin content using liquid-chromatography high-resolution accurate-mass mass-spectrometry (LC-HRAM-MS) and MS$^2$ spectral similarity networking approaches. We chose to analyze extracted hemolymph from both *P. pyralis*, and *A. lateralis* for lucibufagin content, as lucibufagins are known to accumulate in the adult hemolymph and hemolymph samples give less complex extracts than tissue extracts. For *P. pyralis* and *A. lateralis* larvae, and *I. luminosus* thorax, tissue extracts were sampled as we do not have a reliable hemolymph extraction protocol for these life stages and species. Specific tissues were chosen for extracts to enable a smaller quantity of tissue to go into the metabolite extraction, and to explore possible difference in compound abundance across tissues, but we expected that defense compounds like lucibufagins would be roughly equally abundant present in all tissues.

Adult male *P. pyralis* and *A. lateralis* hemolymph was extracted by the following methods: A single live adult *P. pyralis* male was placed in a 1.5 mL microcentrifuge tube with a 5-mm-glass bead underneath the specimen, and centrifuged at maximum speed (~20,000 xg) for 30 s in a benchtop centrifuge. This centrifugation crushed the specimen on top of the bead, and allowed the hemolymph to collect at the bottom of the tube. Approximately 5 µL was obtained. The extracted hemolymph was diluted with 50 µL methanol to precipitate proteins and other macromolecules. For *A. lateralis* adult hemolymph, three adult male individuals were placed in individual 1.5 mL microcentrifuge tubes with 5-mm-glass beads, and spun at 5000 RPM for 1 min in a benchtop centrifuge. The pooled extracted hemolymph (~5 µL), was diluted with 50 µL MeOH, and air dried. The *P. pyralis* extracted hemolymph was filtered through a 0.2 µm PFTE filter (Filter Vial, P/No. 15530–100, Thomson Instrument Company), whereas the *A. lateralis* hemolymph residue was redissolved in 100 µL 50% MeOH, and then filtered through the filter vial.

For extraction of *P. pyralis* larval partial body, the posterior two abdominal segments were first cut off from a single laboratory reared larvae (Appendix 1.3.2), and the remaining partial body was placed in 180 µL 50% acetonitrile, and macerated with a pipette tip. The extract was sonicated in a water bath sonicator for ~10 min, not letting the temperature of the bath go above 50°C. The extract was then centrifuged (20,000 x g for 10 min), and filtered through a 0.2 µm PFTE filter (Filter Vial, P/No. 15530–100, Thomson Instrument Company).

For extraction of *A. lateralis* larval whole body, laboratory reared *A. lateralis* larvae were flash frozen in liquid $N_2$, lyophilized, and the whole body (dry weight: 29.1 mg) was placed in 200 µL 50% methanol, and macerated with a pipette tip. The extract was sonicated in a water bath sonicator for 30 min, centrifuged (20,000xg for 10 min), and filtered through a 0.2 µm PFTE filter (Filter Vial, P/No. 15530–100, Thomson Instrument Company).

For extraction of *I. luminosus* adult thorax, the mesothorax through the two most anterior abdominal segments (ventral lantern containing segment +1 segment) of a lyophilized *I. luminosus* adult male (Appendix 3.3), was separated from the prothorax plus head and posterior three abdominal segments. This mesothorax + abdomen fragment was then placed in 0.5 mL 50% methanol, and macerated with a pipette tip. The extract was then sonicated in a water bath sonicator for ~10 min, not letting the temperature of the bath go above 50°C, centrifuged (20,000xg for 10 min), and filtered through a 0.2 µm PFTE filter (Filter Vial, P/No. 15530–100, Thomson Instrument Company).

Injections of these filtered extracts (*P. pyralis* adult male hemolymph 10 µL; *A. lateralis* adult male hemolymph 5 µL; *P. pyralis* partial larval body extract 5 µL; *A. lateralis* whole larval body 5 µL; *I. luminosus* thorax extract 20 µL) were separated and analyzed using an UltiMate 3000 liquid chromatography system (Thermo Scientific) equipped with a 150 mm C18 Column (Kinetex 2.6 µm silica core shell C18 100 Å pore, P/No. 00F-4462-Y0, Phenomenex, USA) coupled to a Q-Exactive mass spectrometer (Thermo Scientific, USA).

Two different instrument methods were used, a slow ~44 min method, and an optimized ~28 min method. Chromatographically both methods are identical up to 20 min.

*P. pyralis* hemolymph compounds were separated by the optimized method (28 min), with separation via reversed-phase chromatography on a C18 column using a gradient of Solvent A (0.1% formic acid in $H_2O$) and Solvent B (0.1% formic acid in acetonitrile); 5% B for 2 min, 5–40% B until 20 min, 40–95% B until 22 min, 95% B for 4 min, and 5% B for 5 min; flow rate 0.8 mL/min. All other sample extracts were separated by the slow (44 min) reversed-phase chromatography method, using a C18 column with a gradient of Solvent A (0.1% formic acid in $H_2O$) and Solvent B (0.1% formic acid in acetonitrile); 5% B for 2 min, 5–80% B until 40 min, 95% B for 4 min, and 5% B for 5 min; flow rate 0.8 mL/min.

The mass spectrometer was configured to perform one MS[1] scan from *m/z* 120–1250 followed by 1–3 data-dependent MS[2] scans using HCD fragmentation with a stepped collision energy of 10, 15, 25 normalized collision energy (NCE). Positive mode and negative mode MS[1] and MS[2] data were obtained in a single run via polarity switching for the optimized method, or in separate runs for the slow method. Data was collected as profile data. The instrument was always used within 7 days of the last mass accuracy calibration. The ion source parameters were as follows: spray voltage (+) at 3000 V, spray voltage (-) at 2000 V, capillary temperature at 275°C, sheath gas at 40 arb units, aux gas at 15 arb units, spare gas at one arb unit, max spray current at 100 (µA), probe heater temp at 350°C, ion source: HESI-II. The raw data in Thermo format was converted to mzML format using ProteoWizard MSConvert (**Chambers et al., 2012**). Data analysis was performed with Xcalibur (Thermo Scientific) and MZmine2 (v2.30) (**Pluskal et al., 2010**). Raw LC-MS data is available on MetaboLights (Accession: MTBLS698).

Within MZmine2, data were from all five samples on positive mode, and were first cropped to 20 min in order to compare data which was obtained with the same LC gradient parameters. Profile MS[1] data was then converted to centroid mode with the Mass detection module(Parameters: Mass Detector = Exact mass, Noise level = 1.0E4), whereas MS[2] data was converted to centroid mode with (Noise level = 1.0E1). Ions were built into chromatograms using the Chromatogram Builder module with parameters (min_time_span = 0.10,min_height = 1.0E4, *m/z* tolerance = 0.001 *m/z* or five ppm. Chromatograms were then deconvolved using the Chromatogram deconvolution module with parameters (Algorithm = Local Minimum Search, Chromatographic threshold = 5.0%, Search Minimum in RT range = 0.10 min, Minimum relative height = 1%, Minimum absolute height = 1.0E0, Min ratio of peak top/edge = 2, Peak duration range = 0.00–10.00). Isotopic peaks were annotated to their parent features with the Isotopic peaks grouper module with parameters (*m/z* tolerance = 0.001 or five ppm, Retention time tolerance = 0.2 min, Monotonic shape = yes, Maximum charge = 2, Representative isotope = Most intense). The five peaklists (*P. pyralis* hemolymph, *P. pyralis* larval partial body, *A. lateralis* adult hemolymph, *A. lateralis* larval whole body, *I. luminosus* thorax) were then joined and retention time aligned using the RANSAC algorithm with parameters (*m/z* tolerance = 0.001 or 10 ppm, RT tolerance = 1.0 min, RT tolerance after correction = 0.1 min, RANSAC iterations = 100, Minimum number of points = 5%, Threshold value = 0.5). These aligned peaklists were then gap-filled. Systematic mass accuracy error was determined with the endogenous tryptophan $[M + H]^+$ ion (m/z = 205.09, RT = 3.5–4.5 mins), and was measured to be +0.6 ppm,+9.9 ppm,+1.6 ppm,+1.1 ppm, and +0.6 ppm, for *P. pyralis* adult hemolymph, *P. pyralis* partial larval body extract, *A. lateralis* adult hemolymph, *A. lateralis* larval body extract, and *I. luminosus* thorax extract, respectively.

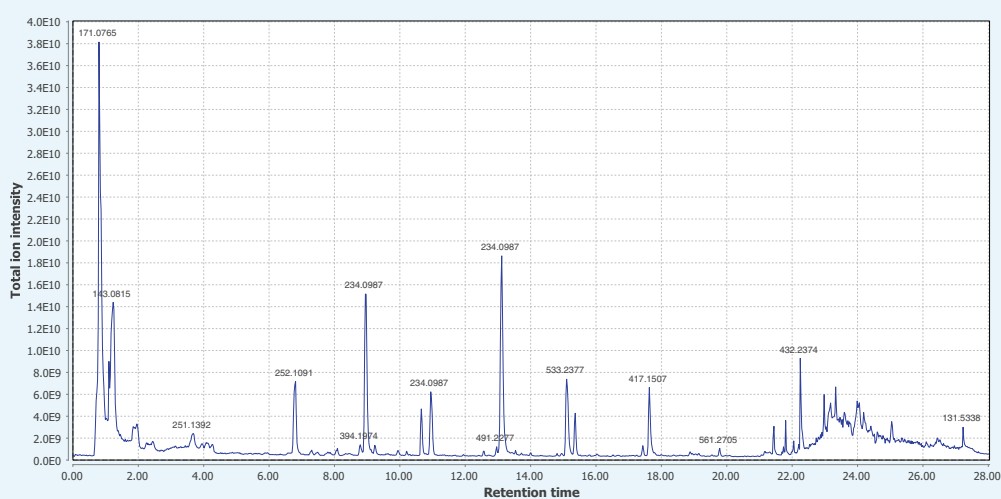

**Appendix 4—figure 10.** Positive mode MS[1] total-ion-chromatogram (TIC) of *P.pyralis* adult hemolymph LC-HRAM-MS data. Figure produced using MZmine2 (*Pluskal et al., 2010*).
DOI: https://doi.org/10.7554/eLife.36495.067

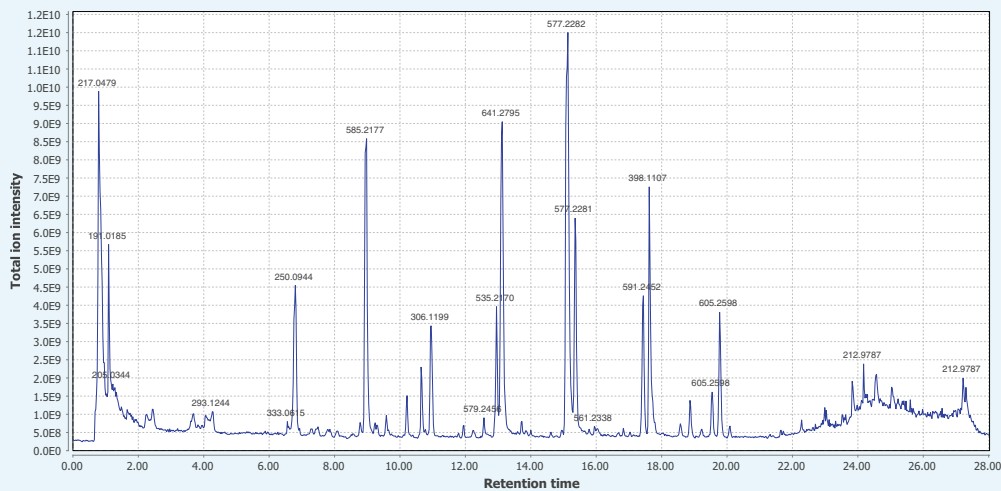

**Appendix 4—figure 11.** Negative mode MS[1] total-ion-chromatogram (TIC) of *P. pyralis* adult hemolymph LC-HRAM-MS data. Figure produced using MZmine2 (*Pluskal et al., 2010*).
DOI: https://doi.org/10.7554/eLife.36495.068

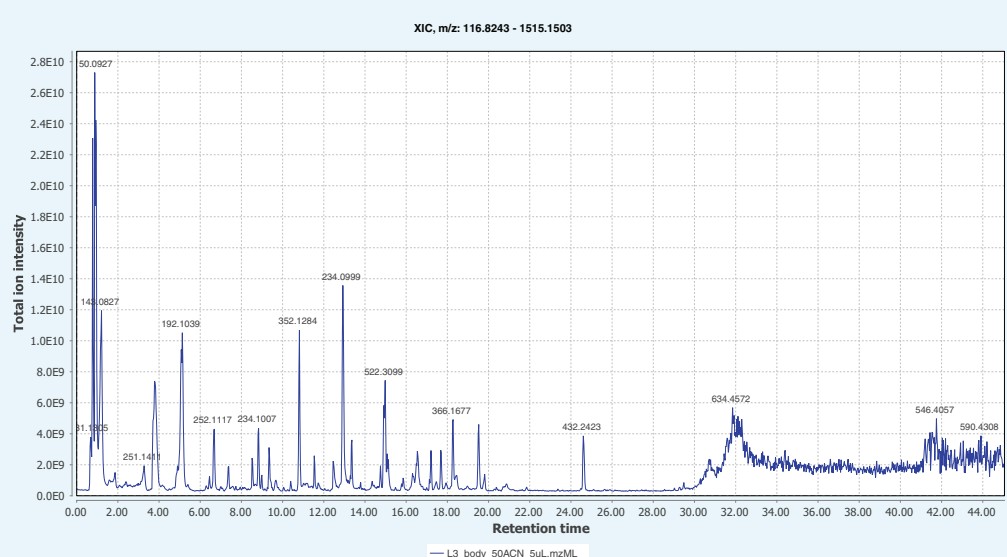

**Appendix 4—figure 12.** Positive mode MS[1] total-ion-chromatogram (TIC) of *P. pyralis* larval whole body minus two posterior segments LC-HRAM-MS data. Figure produced using MZmine2 (*Pluskal et al., 2010*).

DOI: https://doi.org/10.7554/eLife.36495.069

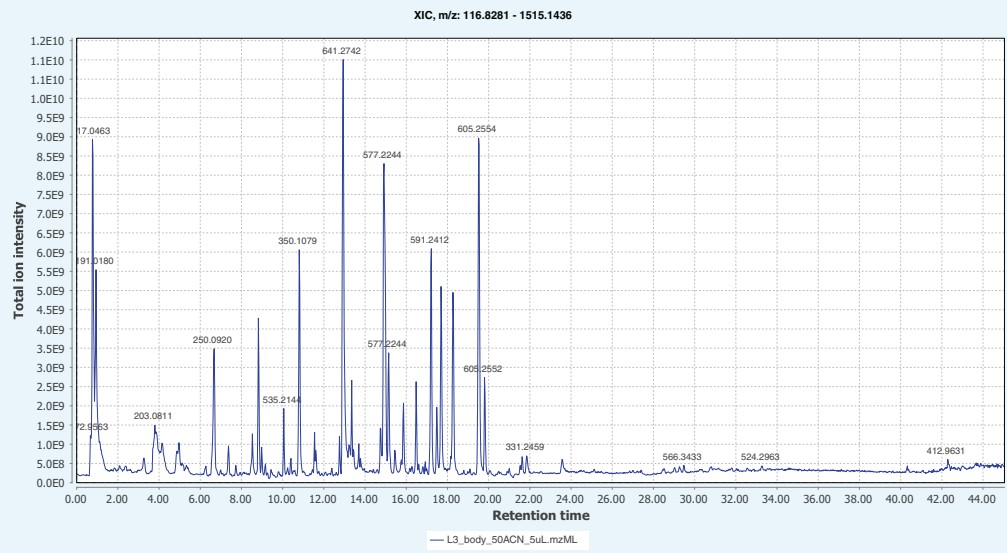

**Appendix 4—figure 13.** Negative mode MS[1] total-ion-chromatogram (TIC) of *P. pyralis* larval whole body minus two posterior segments LC-HRAM-MS data. Figure produced using MZmine2 (*Pluskal et al., 2010*).

DOI: https://doi.org/10.7554/eLife.36495.070

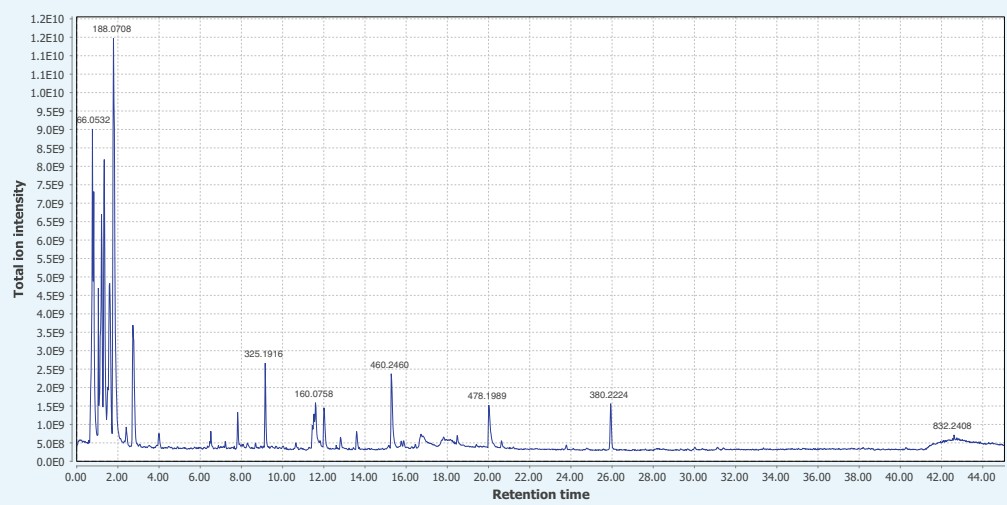

**Appendix 4—figure 14.** Positive mode MS[1] total-ion-chromatogram (TIC) of *A. lateralis* adult hemolymph LC-HRAM-MS data. Figure produced using MZmine2 (*Pluskal et al., 2010*).
DOI: https://doi.org/10.7554/eLife.36495.071

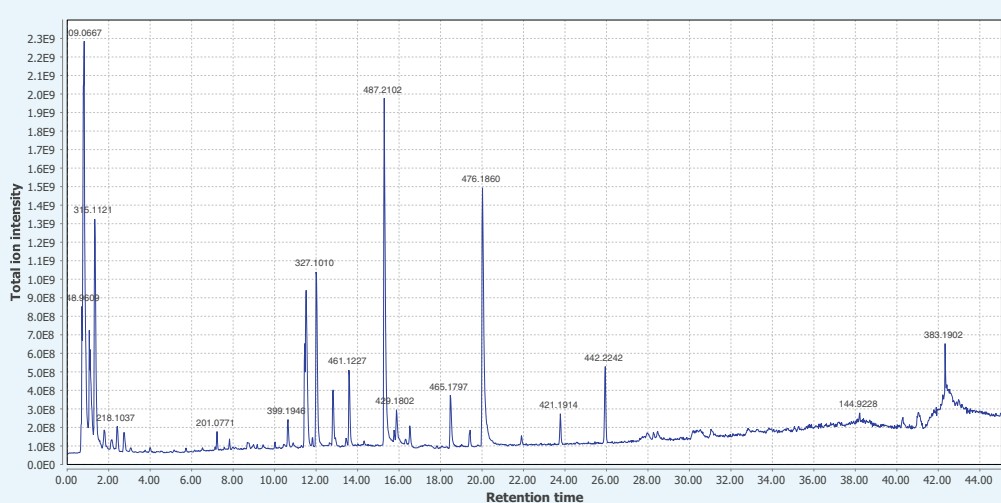

**Appendix 4—figure 15.** Negative mode MS[1] total-ion-chromatogram (TIC) of *A. lateralis* adult hemolymph LC-HRAM-MS data. Figure produced using MZmine2 (*Pluskal et al., 2010*).
DOI: https://doi.org/10.7554/eLife.36495.072

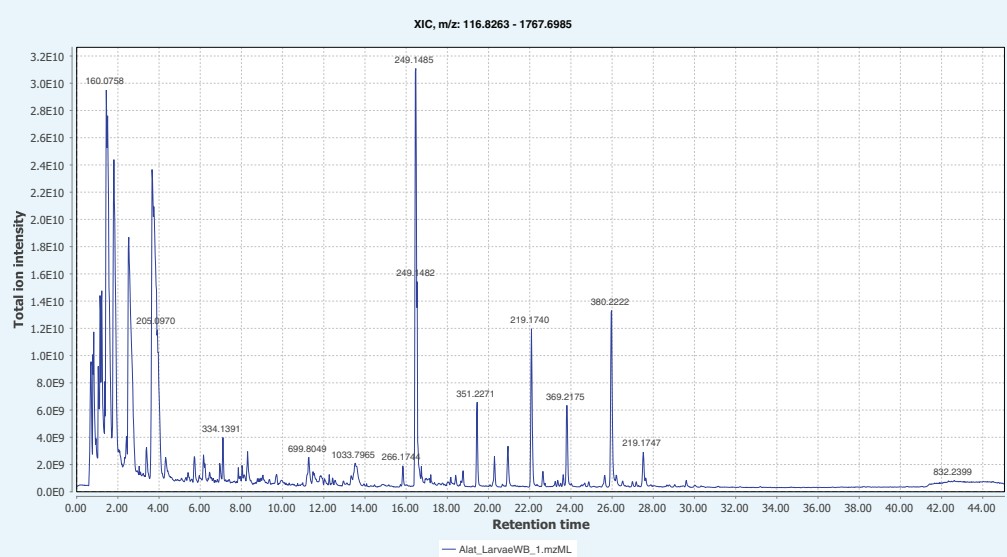

**Appendix 4—figure 16.** Positive mode MS[1] total-ion-chromatogram (TIC) of *A. lateralis* larval whole body LC-HRAM-MS data. Figure produced using MZmine2 (*Pluskal et al., 2010*).
DOI: https://doi.org/10.7554/eLife.36495.073

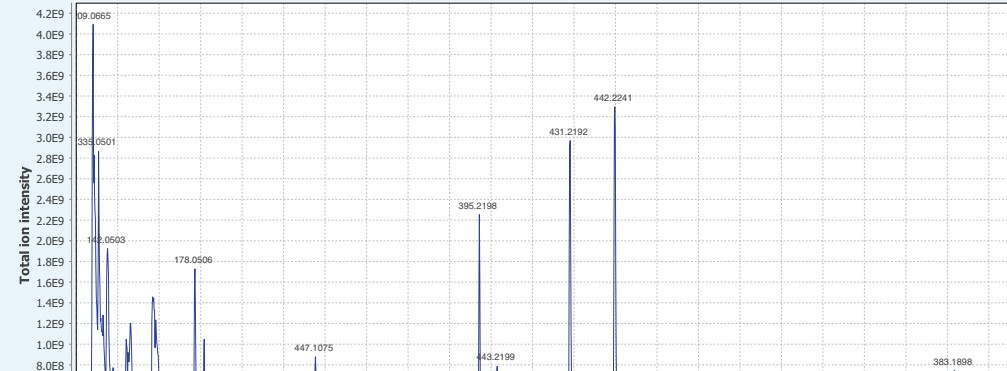

**Appendix 4—figure 17.** Negative mode MS[1] total-ion-chromatogram (TIC) of *A. lateralis* larval whole body extract LC-HRAM-MS data. Figure produced using MZmine2 (*Pluskal et al., 2010*).
DOI: https://doi.org/10.7554/eLife.36495.074

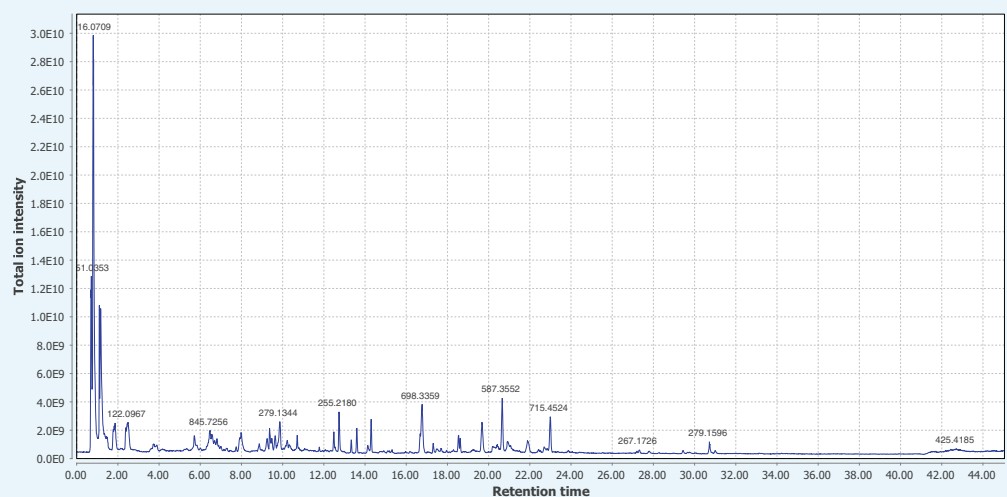

**Appendix 4—figure 18.** Positive mode MS[1] total-ion-chromatogram (TIC) of *I. luminosus* mesothorax +abdomen extract LC-HRAM-MS data. Figure produced using MZmine2 (*Pluskal et al., 2010*).

DOI: https://doi.org/10.7554/eLife.36495.075

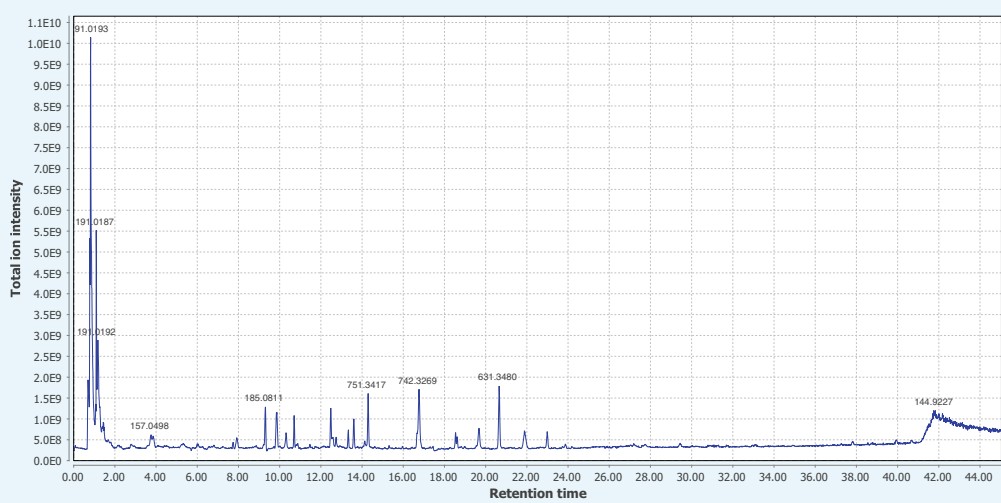

**Appendix 4—figure 19.** Negative mode MS[1] total-ion-chromatogram (TIC) of *I.luminosus* mesothorax + abdomen extract LC-HRAM-MS data. Figure produced using MZmine2 (*Pluskal et al., 2010*).

DOI: https://doi.org/10.7554/eLife.36495.076

### 4.6.5 MS[2] similarity search for *P. pyralis* lucibufagins

We first performed a MS[2] similarity search within *P. pyralis* adult hemolymph for ions that showed a similar MS[2] spectra to the MS[2] spectra arising from the diacetylated lucibufagin [M + H]+ ion from the same run ([M + H]+ *m/z* 533.2385, RT = 15.10 mins) (*Appendix 4— figure 20*). This search was performed through the MS[2] similarity search module of MZmine2 (v2.30) with parameters (*m/z* tolerance: 0.0004 *m/z* or 1 PPM; minimum # of ions to report: 3). This MS[2] similarity search revealed nine putative lucibufagin isomers with highly similar MS[2] spectra (*Appendix 4—figure 21*), which expanded to 17 putative lucibufagin isomers when considering features without MS[2] spectra, but with identical exact masses and close retention times (ΔRT <2 min) to the previously identified 9 (*Appendix 4—table 7*). Chemical formula prediction was assigned to each precursor ion using the Chemical formula search module of MZmine2, whereas chemical formula predictions for product ions was performed

within MZmine2 using SIRUIS (v3.5.1) (*Böcker et al., 2009*). The structural identity of the nine putative lucibufagins detected via the MS$^2$ spectra similarity search was easily interpreted in light that the different chemical formula represented the core lucibufagins that had undergone acetylation ($COCH_3$) or propylation ($COCH_2CH_3$), in different combinations. Notably, the most substituted isomers, dipropylated lucibufagin ([M + H]$^+$ *m/z* 561.2695, RT = 19.54 mins) were close to the edge of the cropped data (20 min), thus it may be possible that more highly substituted lucibufagins with a longer retention times are present, but not detected in the current analysis.

We then performed a MS$^2$ similarity search within *P. pyralis* partial body extract for ions that showed a MS$^2$ spectra similar to that of the dipropylated lucibufagin [M + H]$^+$ ion from the same run ([M + H]$^+$ *m/z* 561.2738, RT = 19.53). This search was performed through the MS$^2$ similarity search module of MZmine2 (v2.30) with parameters (*m/z* tolerance: 0.0004 *m/z* or 1 PPM; minimum # of ions to report: 5). This MS$^2$ similarity search revealed 14 putative lucibufagin isomers with highly similar MS$^2$ spectra (*Appendix 4—table 7*). Complexes and fragments were manually removed from the analysis. Comparison of the theoretical and observed exact mass indicated that this experimental run had an unusual degree of systematic *m/z* error, of ~+10 ppm. After manual correction m/z, chemical formula prediction revealed a several putative lucibufagins of unknown structure with nitrogen in their chemical formula, suggesting that the nitrogen containing lucibufagins reported by by Gronquist and colleagues from *Lucidota atra* (*Gronquist et al., 2005*) may be present in *P. pyralis* larvae.

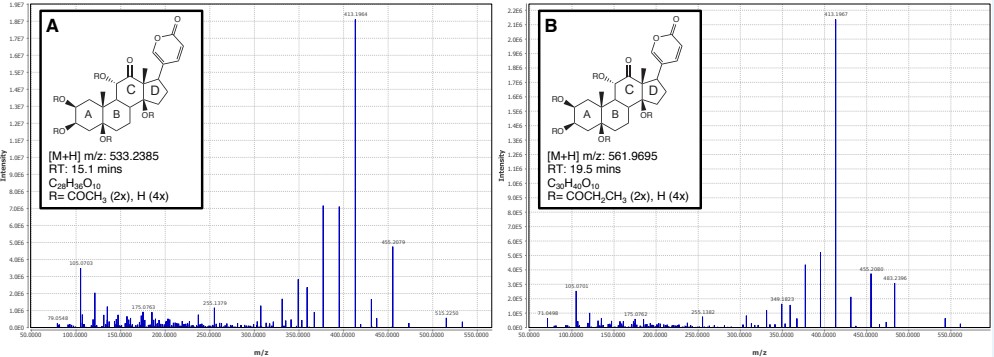

**Appendix 4—figure 20.** Positive mode MS$^2$ spectra of (**A**) diacetylated lucibufagin [M + H]$^+$ and (**B**) dipropylated lucibufagin [M + H]$^+$.
DOI: https://doi.org/10.7554/eLife.36495.077

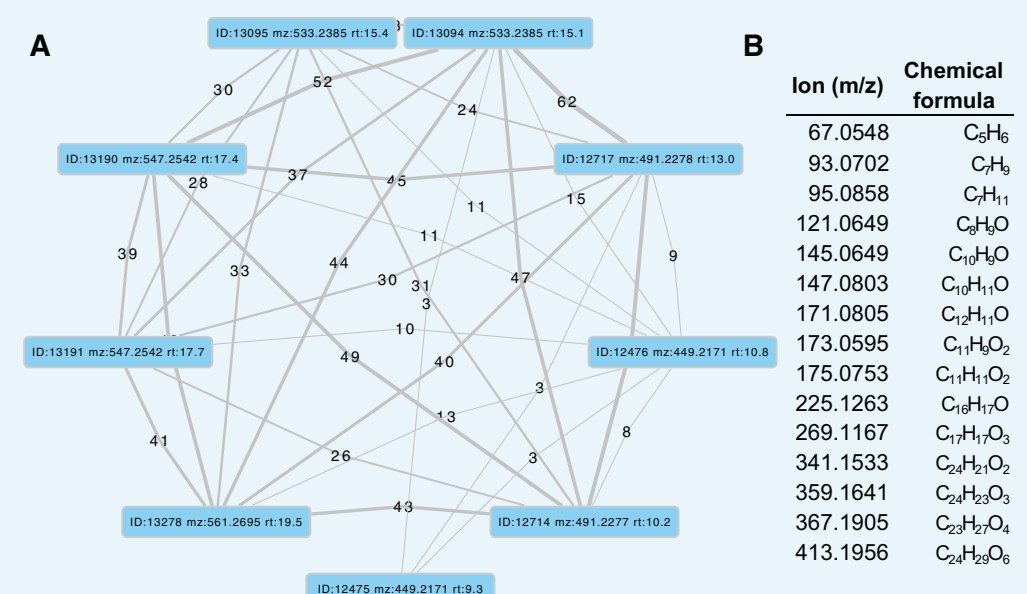

**A**

**B**

| Ion (m/z) | Chemical formula |
|-----------|------------------|
| 67.0548 | $C_5H_6$ |
| 93.0702 | $C_7H_9$ |
| 95.0858 | $C_7H_{11}$ |
| 121.0649 | $C_8H_9O$ |
| 145.0649 | $C_{10}H_9O$ |
| 147.0803 | $C_{10}H_{11}O$ |
| 171.0805 | $C_{12}H_{11}O$ |
| 173.0595 | $C_{11}H_9O_2$ |
| 175.0753 | $C_{11}H_{11}O_2$ |
| 225.1263 | $C_{16}H_{17}O$ |
| 269.1167 | $C_{17}H_{17}O_3$ |
| 341.1533 | $C_{24}H_{21}O_2$ |
| 359.1641 | $C_{24}H_{23}O_3$ |
| 367.1905 | $C_{23}H_{27}O_4$ |
| 413.1956 | $C_{24}H_{29}O_6$ |

**Appendix 4—figure 21.** $MS^2$ spectral similarity network for *P.pyralis* adult hemolymph lucibu-fagins. (**A**) $MS^2$ similarity network produced with the MZmine2 $MS^2$ similarity search module. Nodes represent $MS^2$ spectra from the initial dataset, whereas edges represent an $MS^2$ similarity match between two $MS^2$ spectra. Thickness/label of the edge represents the number of ions matched between the two $MS^2$ spectra. (**B**) Table of matched ions between diacetylated lucibufagin (*m/z*: 533.2385 RT:15.1), and core (unacetylated) lucibufagin (*m/z*: 449.2171 RT:10.8 min). $MS^1$ adducts and complexes of the presented ions were manually removed.

DOI: https://doi.org/10.7554/eLife.36495.078

**Appendix 4—table 7.** Putative lucibufagin compounds from LC-HRAM-MS of *P. pyralis* adult hemolymph. Retention time and *m/z* values are not calibrated to the other samples.

| Assigned ion identity | Ion type | Chemical formula | Expected M/z | Measured M/z | M/z error* (ppm) | Retention time (mins) | Feature area (arb) |
|---|---|---|---|---|---|---|---|
| Core lucibufagin isomer 1 | [M + H]$^+$ | $C_{24}H_{33}O_8$ | 449.2175 | 449.2171 | −0.89 | 7.9 | 6.7E + 05 |
| Core lucibufagin isomer 2 | " " | " " | " " | " " | " " | 9.3 | 1.1E + 07 |
| Monoacetylated lucibufagin isomer 1 | " " | $C_{26}H_{35}O_9$ | 491.2281 | 491.2277 | −0.81 | 10.2 | 4.2E + 07 |
| Core lucibufagin isomer 3 | " " | $C_{24}H_{33}O_8$ | 449.2175 | 449.2171 | −0.89 | 10.8 | 1.7E + 07 |
| Monoacetylated lucibufagin isomer 2 | " " | $C_{26}H_{35}O_9$ | 491.2281 | 491.2277 | −0.81 | 11.4 | 1.1E + 06 |
| Monoacetylated lucibufagin isomer 3 | " " | " " | " " | " " | " " | 11.9 | 1.8E + 07 |
| Monoacetylated lucibufagin isomer 4 | " " | " " | " " | " " | " " | 13.0 | 2.7E + 08 |
| Monoacetylated lucibufagin isomer 5 | " " | " " | " " | " " | " " | 13.2 | 6.0E + 07 |
| Monoacetylated lucibufagin isomer 6 | " " | " " | " " | " " | " " | 14.5 | 6.2E + 06 |
| Diacetylated lucibufagin isomer 1 | " " | $C_{28}H_{37}O_{10}$ | 533.2387 | 533.2385 | −0.37 | 15.1 | 4.0E + 09 |
| Diacetylated lucibufagin isomer 2 | " " | " " | " " | " " | " " | 15.4 | 1.9E + 09 |
| Monoacetylated, mono propylated lucibufagin isomer 1 | " " | $C_{29}H_{39}O_{10}$ | 547.2543 | 547.2542 | −0.18 | 17.0 | 1.5E + 07 |
| Monoacetylated, mono propylated lucibufagin isomer 2 | " " | " " | " " | " " | " " | 17.4 | 2.8E + 08 |
| Monoacetylated, mono propylated lucibufagin isomer 3 | " " | " " | " " | " " | " " | 17.7 | 1.2E + 08 |
| Dipropylated lucibufagin isomer 1 | " " | $C_{30}H_{41}O_{10}$ | 561.2700 | 561.2695 | −0.89 | 18.9 | 1.4E + 08 |
| Dipropylated lucibufagin isomer 2 | " " | " " | " " | " " | " " | 19.5 | 3.9E + 07 |
| Dipropylated lucibufagin isomer 3 | " " | " " | " " | " " | " " | 19.8 | 1.8E + 08 |

DOI: https://doi.org/10.7554/eLife.36495.079

**Appendix 4—table 8.** Putative lucibufagin compounds from LC-HRAM-MS of *P. pyralis* larval partial body extracts. Retention time and *m/z* values are not calibrated to the other samples. *=*m/z* error and expected *m/z* extrapolated from ions with similar *m/z*, and chemical formula predicted from resulting extrapolated *m/z*. **=Likely chemical formula cannot be determined due to many possible chemical formula from the expected *m/z*.

| Assigned ion identity | Ion type | Chemical formula | Expected m/z | Measured m/z | m/z error (ppm) | Retention time (mins) | Feature area (arb) |
|---|---|---|---|---|---|---|---|
| Core lucibufagin isomer 2 | [M + H]$^+$ | $C_{24}H_{33}O_8$ | 449.2175 | 449.2215 | +8.9 | 9.15 | 8.5E + 06 |
| Monoacetylated lucibufagin isomer 1 | " " | $C_{26}H_{35}O_9$ | 491.2277 | 491.2326 | +9.9 | 10.04 | 1.2E + 07 |
| Unknown | unknown | $C_{28}H_{39}O_{10}$* | 535.2543* | 535.2592 | +9.1* | 12.40 | 1.6E + 07 |
| Unknown | unknown | $C_{24}H_{38}NO_6$* | 436.2695* | 436.2735 | +9.1* | 13.30 | 2.2E + 07 |
| Unknown | unknown | $C_{27}H_{45}N_2O_8$* | 525.3173* | 525.3221 | +9.1* | 13.35 | 1.3E + 08 |
| Unknown | unknown | $C_{24}H_{40}NO_7$* | 454.2799* | 454.2840 | +9.1* | 13.73 | 1.3E + 07 |
| Diacetylated lucibufagin isomer 1 | [M + H]$^+$ | $C_{28}H_{37}O_{10}$ | 533.2387 | 533.2426 | +7.3 | 14.93 | 1.7E + 09 |
| Diacetylated lucibufagin isomer 2 | [M + H]$^+$ | " " | " " | 533.2426 | +7.3 | 15.16 | 3.5E + 08 |
| Unknown | Unknown | $C_{29}H_{46}NO_8$* | 536.3216* | 536.3256 | +7.3* | 16.57 | 4.1E + 07 |
| Unknown | Unknown | Unknown** | 563.2854* | 563.2896 | +7.3* | 16.80 | 1.3E + 07 |
| Unknown | Unknown | $C_{26}H_{31}O_7$ | 455.2056 | 455.2097 | +9.1* | 17.22 | 5.8E + 07 |
| Dipropylated lucibufagin isomer 3 | Unknown | $C_{30}H_{41}O_{10}$ | 561.2700 | 561.2738 | +6.7 | 19.53 | 2.0E + 09 |
| Dipropylated lucibufagin isomer 4 | Unknown | $C_{30}H_{41}O_{10}$ | 561.2700 | 561.2738 | +6.7 | 19.82 | 2.2E + 08 |

DOI: https://doi.org/10.7554/eLife.36495.080

**Appendix 4—table 9.** Putative lucibufagin [M + H]$^+$ exact masses adjusted for instrument run specific systematic *m/z* error (***Figure 6B***). Used for multi-ion-chromatogram (MIC) traces in ***Figure 6B***.

| Chemical formula | Predicted exact mass | Exact mass adjusted to *P. pyralis* hemolymph data (+0.6 ppm) | Exact mass adjusted to *P. pyralis* partial larval body data (+9.9 ppm) | Exact mass adjusted to *A. lateralis* hemolymph data (+1.6 ppm) | Exact mass adjusted to *A. lateralis* larval body data (+1.1 ppm) | Exact mass adjusted to *I. luminosus* thorax data (+0.6 ppm) |
|---|---|---|---|---|---|---|
| $C_{24}H_{33}O_8$ | 449.2175 | 449.2178 | 449.2219 | 449.2182 | 449.2180 | 449.2178 |
| $C_{24}H_{38}NO_6$* | 436.2699 | 436.2702 | 436.2742 | 436.2706 | 436.2704 | 436.2702 |
| $C_{24}H_{40}NO_7$* | 454.2804 | 454.2807 | 454.2849 | 454.2811 | 454.2809 | 454.2807 |
| $C_{26}H_{31}O_7$ | 455.2069 | 455.2072 | 455.2114 | 455.2076 | 455.2074 | 455.2072 |
| $C_{26}H_{35}O_9$ | 491.2281 | 491.2284 | 491.2330 | 491.2289 | 491.2286 | 491.2284 |
| $C_{27}H_{45}N_2O_8$* | 525.3175 | 525.3178 | 525.3227 | 525.3183 | 525.3181 | 525.3178 |
| $C_{28}H_{37}O_{10}$ | 533.2386 | 533.2389 | 533.2439 | 533.2395 | 533.2392 | 533.2389 |
| $C_{28}H_{39}O_{10}$* | 535.2543 | 535.2546 | 535.2596 | 535.2552 | 535.2549 | 535.2546 |
| $C_{29}H_{39}O_{10}$ | 547.2543 | 547.2546 | 547.2597 | 547.2552 | 547.2549 | 547.2546 |
| $C_{29}H_{46}NO_8$* | 536.3223 | 536.3226 | 536.3276 | 536.3232 | 536.3229 | 536.3226 |
| $C_{30}H_{41}O_{10}$ | 561.2699 | 561.2702 | 561.2755 | 561.2708 | 561.2705 | 561.2702 |

*=Chemical formula assigned for structurally unclear putative lucibufagins
DOI: https://doi.org/10.7554/eLife.36495.081

## 4.6.7 MS$^2$ similarity search for *A. lateralis* lucibufagins

Although our earlier LC-HRAM-MS analysis (***Figure 6B***; Appendix 4.6) indicated *A. lateralis* adult male hemolymph does not contain detectable quantities of the *P. pyralis* lucibufagins, this does not exclude that structurally unknown lucibufagins with chemical formula not present in *P. pyralis*, are present in *A. lateralis*. To address this, we performed a MS$^2$ similarity search against the *A. lateralis* adult male hemolymph MS$^2$ spectra, with the MS$^2$ spectra of lucibufagin C (*m/z* 533.2385, RT = 15.1) as bait, using the MZmine2 similarity search module with parameters (*m/z* tolerance = 0.001 or 10 ppm, Minimum # of matched ions = 10). After filtering to those precursors that were mostly likely to be the [M + H]$^+$ of a lucibufagin-like molecule (*m/z* 350–800, RT = 8–20 mins), 9 MS$^2$ spectra were matched (***Appendix 4—table 10***). None of these features were detected in *P. pyralis* (***Appendix 4— table 10***). Chemical formula prediction was difficult due to the high *m/z* of the ions, but in those cases where it was successful, the additions of nitrogens and/or phosphorus to the chemical formula was confident. Notably, the most confident chemical formula predictions reported ≤23 carbons, and as the core lucibufagin of *P. pyralis* contains 24 carbons, it is unlikely that these ions derive from lucibufagins. The notable degree of MS$^2$ similarity may be due to the *A. lateralis* compounds also being steroid derived compounds. That being said, the identity and role of the compound giving rise to ion 460.2462 is intriguing, as it is highly abundant in the *A. lateralis* adult hemolymph, is absent from the *P. pyralis* adult hemolymph, and is possibly a steroidal compound.

**Appendix 4—table 10.** Relative quantification of *A. lateralis* features identified by lucibufagin MS$^2$ similarity search

| Assigned identity | M/z | Chemical formula | RT (mins) | Similarity score | # of ions matched | A. lateralis feature area (arb) | P. pyralis feature area (arb) |
|---|---|---|---|---|---|---|---|
| Unknown | 460.2462 | $C_{22}H_{38}NO_7P$*; $C_{25}H_{29}N_7O_2$* | 15.27 | 4.10E + 11 | 34 | 7.04E + 08 | 0.00E + 00 |
| " " | 657.2229 | N.D. | 12.01 | 9.50E + 11 | 29 | 6.13E + 07 | " " |
| " " | 414.2043 | N.D. | 18.07 | 1.20E + 11 | 25 | 5.61E + 06 | " " |
| " " | 381.2176 | $C_{23}H_{28}N_2O_3$* | 15.77 | 3.80E + 11 | 18 | 1.22E + 08 | " " |
| " " | 476.1839 | N.D. | 15.93 | 3.80E + 11 | 16 | 9.87E + 06 | " " |
| " " | 456.2148 | N.D. | 19 | 2.30E + 11 | 14 | 5.03E + 06 | " " |
| " " | 351.228 | N.D. | 19.42 | 2.60E + 11 | 13 | 1.56E + 07 | " " |
| " " | 479.1948 | N.D. | 19.83 | 2.20E + 11 | 12 | 1.11E + 07 | " " |

*Determined with Sirius ($MS^2$ analysis), and MZmine2 (isotope pattern analysis). N.D., Not determined

DOI: https://doi.org/10.7554/eLife.36495.082

## Appendix 5

DOI: https://doi.org/10.7554/eLife.36495.083

# Microbiome analyses

## 5.1 Assembly and annotation of the complete *Entomoplasma luminosum* subsp. pyralis genome

The complete genome of the molicute (Phylum: Tenericutes) *Entomoplasma luminosum* subsp. pyralis was constructed by a long-read metagenomic sequencing and assembly approach from the *P. pyralis* PacBio data. First, BUSCO v.3 with the bacterial BUSCO set was used to identify those contigs from the PacBio only Canu assembly (Ppyr0.1-PB) which contained conserved bacterial genes. A single 1.04 Mbp contig with 73 bacterial BUSCO genes was the only contig identified with more than 1 BUSCO hit. Inspection of the Canu produced assembly graph with Bandage v0.8.1 (*Wick et al., 2015*), revealed that the contig had a circular assembly path. BLASTN alignment of the contig to the NCBI nt database indicated that this contig had a high degree of similarity to annotated Mycoplasmal genomes. Together this data suggested that this contig represented a complete Mycoplasmal genome. Polishing of the contig was performed by mapping and PacBio consensus-calling using SMRTPortal v2.3.0.140893 with the 'RS_Resequencing.1' protocol with default parameters. The median coverage was ~50x. The resulting consensus sequence was restarted with seqkit (*Shen et al., 2016*) to place the FASTA record junction 180˚ across the circular chromosome, and reentered into the polishing process to enable efficient mapping across the circular junction. This mapping, consensus calling, and rotation process was repeated three times total, after which no additional nucleotide changes occurred. The genome was 'restarted' with seqkit such that the FASTA start position began between the ribosomal RNAs, and annotation was conducted through NCBI using their prokaryotic gene annotation pipeline (PGAP). Analysis with BUSCO v.3 of the peptides produced from the aforementioned genome annotation indicated that 89.8% of expected Tenericutes single-copy conserved orthologs were captured in the annotation (C:89.8% [S:89.8%,D:0.0%], F:2.4%, M:7.8%, n:166). Comparison of the predicted 16S RNA gene sequence to the NCBI 16S RNA gene database indicated that this gene had 99% identity to the *E. luminosum* 16S sequence (ATCC 49195 - formerly *Mycoplasma luminosum;* NCBI Assembly ID ASM52685v1) (*Kyrpides et al., 2014*; *Williamson et al., 1990*), leading to our description of this genome as the genome of *Entomoplasma luminosum* subspecies (subsp.) pyralis. Protein overlap comparisons using the OrthoFinder pipeline (v1.1.10) (*Emms and Kelly, 2015*) between our predicted protein geneset for *E. luminosum* var. pyralis and the protein geneset of *Entomoplasma luminosum* (ATCC 49195 - formerly *M. luminosum;* NCBI Assembly ID ASM52685v1), indicated that 94% (670/709) of the previously annotated *E. luminosum* proteins are present in our genome of *E. luminosum* subsp. pyralis.

## 5.2 Assembly and annotation of Phorid mitochondrial genome

The complete mitochondrial genome of the dipteran parasatoid *Apocephalus antennatus,* first detected via BLASTN of mtDNAs as a concatemerized sequence in the Canu PacBio only assembly (Ppyr0.1-PB) was constructed in full by a long-read metagenomic sequencing and assembly approach. First, PacBio reads were mapped to the NCBI set of mitochondrial genomes concatenated with the *P. pyralis* mitochondrial genome assembly reported in this manuscript (NCBI accession KY778696.1), using GraphMap v0.5.2 with parameters 'align -C -t 4 -P'. Of the mitochondrially mapped reads (45949 reads), 98% (45267 reads) were partitioned to the *P. pyralis* mtDNA. The next most abundant category at 1.1% (531 reads), was partitioned to the mtDNA of the Phorid fly *Megaselia scalaris* (NCBI accession: KF974742.1). The next most abundant category at 0.11% (53 reads) was partitioned to the mitochondrion of the Red algae *Galdieria sulphuraria* (NCBI accession: NC_024666.1). The reads were then split into three partitions: *P. pyralis* mapping, *M. scalaris* mapping, and other, and input into Canu (v1.6+44) (*Koren et al., 2017*) for assembly. Each partitioned assembly by Canu produced a

single circular contig, notably the 'other' and Megaselia partitions produced highly similar sequences, whereas the *P. pyralis* partition produced a circular sequence that was highly similar to the *P. pyralis* mtDNA. We inspected the *M. scalaris* partition further as it was produced with more reads. Notably, although an inspection of the contig was circular, and showed a high degree of similarity upon blastn to the *M. scalaris* mtDNA, the contig was ~2x larger than expected (29,821 bp). An analysis of contig's self-complementarity with Gepard (v1.40) (**Krumsiek et al., 2007**), indicated that this contig had 2x tandem repetitive regions, and was duplicated overall twice. Similarly, the. GFA output of Canu noted an overlap of 29,821, indicating that the assembler was unable to determine an appropriate overlap, other than the entire contig. Manual trimming of the contig to the correct size, 180° restarting with seqkit, and polishing using SMRTPortal v2.3.0.140893 with the 'RS_Resequencing.1' protocol with default parameters, followed by 180° seqkit 'restarting', followed by another round of polishing, produced the final mtDNA (18,674 bp; **Appendix 5—figure 1**). This mtDNA was taxonomically identified in a separate analysis to originate from *A. antennatus* (Appendix 5.3). Coding regions, tRNAs, and rRNAs were predicted via the MITOSv2 mitochondrial genome annotation web server (**Bernt et al., 2017**). Small mis-annotations (e.g. low scoring additional predictions of already annotated mitochondrial genes) were manually inspected and removed. Tandem repetitive regions were manually annotated. The complete *A. antennatus* genome annotation plus assembly is available on NCBI Genbank (Accession: MG546669).

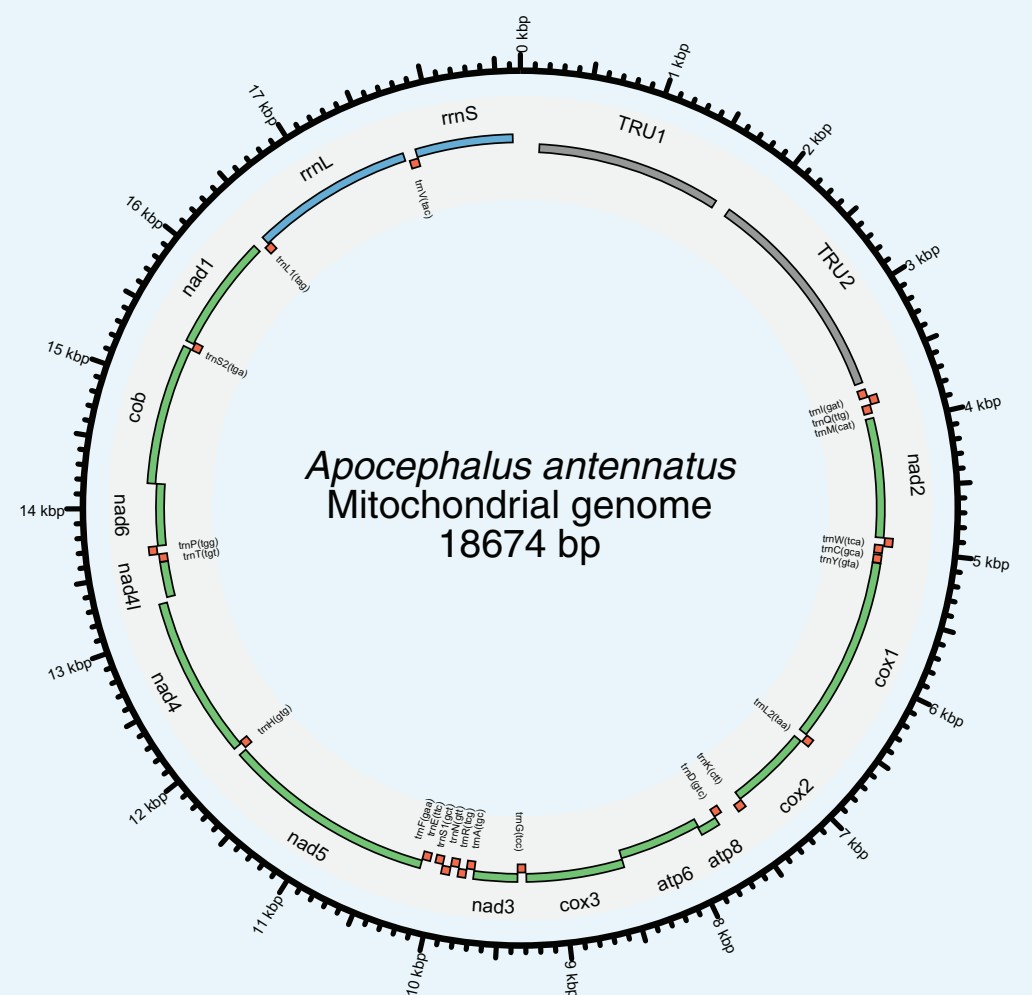

**Appendix 5—figure 1.** Mitochondrial genome of *Apocephalus antennatus*. The mitochondrial genome of *A. antennatus* was assembled and annotated as described in the Appendix 5.2,

and taxonomically identified as described in Appendix 5.3. Figure produced with Circos (*Krzywinski et al., 2009*).
DOI: https://doi.org/10.7554/eLife.36495.084

## 5.3 Taxonomic identification of Phorid mitochondrial genome origin

After the successful metagenomic assembly of the mitochondrial genome of an unknown Phorid fly species from the *P. pyralis* PacBio library (Appendix 5.2), we sought to characterize the species of origin for this mitochondrial genome. We planned to achieve this by collecting the Phorid flies which emerged from adult *P. pyralis*, taxonomically identifying them, and performing targeted mitochondrial PCR and sequencing experiments to correlate their mitochondrial genome sequence to our mtDNA assembly. We successfully obtained phorid fly larvae emerging from *P. pyralis* adult males collected from MMNJ (identical field site to PacBio collection), and Rochester, NY (RCNY), in the summer of 2017. The MMNJ phorid larvae did not successfully pupate, however we obtained five adult specimens from successful pupations of the RCNY larvae. Two adults from this batch were identified as *A. antennatus* (Malloch), by Brian V. Brown, Entomology Curator of the Natural History Museum of Los Angeles County. DNA was extracted from one of the remaining three specimens and a COI fragment was PCR-amplified and Sanger sequenced. The forward primer was 5'-TTTGATTCTTCGGCCACCCA-3', the reverse primer 5'-AGCATCGGGGTAGTCTGAGT-3'. This COI fragment from had 99% identity (558/563 nt) to the COI gene of our mitochondrial assembly. This sequenced COI fragment has been submitted to GenBank (GenBank Accession: MG517481). We conclude that this is sufficient evidence to denote that our assembled Phorid mitochondrial genome is the mitochondrial genome of *A. antennatus*. Notably, *A. antennatus* was previously reported by *Lloyd (1973)* to be a parasite of several firefly species in genera *Photuris*, *Photinus*, and *Pyractomena*, from collection sites ranging from Florida to New York. To our knowledge, this is the first report of a mitochondrial genome which was first assembled via an untargeted metagenomic approach and then later correlated to its species of origin.

## 5.4 *Photinus pyralis* orthomyxo-like viruses

We identified the first two viruses associated to *P. pyralis* and the Lampyridae family. The proposed *Photinus pyralis* orthomyxo-like virus 1 and 2 (PpyrOMLV1 and 2) present a multipartite genome conformed by five RNA segments encoding a putative nucleoprotein (NP), hemagglutinin-like glycoprotein (HA) and a heterotrimeric viral RNA polymerase (PB1, PB2 and PA). The viral genomes for Photinus pyralis orthomyxo-like virus 1 and 2 are available on NCBI Genbank with accessions MG972985-MG972994. Expression analyses on 24 RNA libraries of diverse individuals/developmental stages/tissues and geographic origins of *P. pyralis* indicate a dynamic presence, widespread prevalence, a pervasive tissue tropism, a low isolate variability, and a persistent life cycle through transovarial transmission of PpyrOMLV1 and 2. Genomic and phylogenetic studies suggest that the detected viruses correspond to a new lineage within the *Orthomyxoviridae* family (ssRNA(-)) (*Appendix 5—figure 2A-I*). The concomitant occurrence in the *P. pyralis* genome of species-specific signatures of Endogenous viral-like elements (EVEs) associated to retrotransposons linked to the identified Orthomyxoviruses, suggest a past evolutionary history of host-virus interaction (Appendix 5.5, *Appendix 5—figure 2J*). This tentative interface is correlated to low viral RNA levels, persistence and no apparent phenotypes associated with infection. We suggest that the identified viruses are potential endophytes of high prevalence as a result of potential evolutionary modulation of viral levels associated to EVEs. Photinus pyralis orthomyxo-like virus 1 and 2 (PpyrOMLV1 and PpyrOMLV2) share their genomic architecture and evolutionary clustering (*Appendix 5—figure 2A-H*, *Appendix 5—figure 3*). They are multipartite linear ssRNA negative strand viruses, conformed by five genome segments generating a ca. 10.8 Kbp total RNA genome. Genome segments one through three (ca. 2.3–2.5 Kbp long) encode a heterotrimeric viral polymerase constituted by subunit Polymerase Basic protein 1 - PB1 (PpyrOMLV1: 801 aa, 91 kDA; PpyrOMLV2: 802 aa, 91.2 kDA), Polymerase Basic protein 2 - PB2 (PpyrOMLV1: 804 aa, 92.6 kDA; PpyrOMLV2: 801 aa, 92.4 kDA) and Polymerase Acid

protein - PA (PpyrOMLV1: 754 aa, 86.6 kDA; PpyrOMLV2: 762 aa, 87.9 kDA). PpyrOMLV1 and PpyrOMLV2 PB1 present a Flu_PB1 functional domain (Pfam: pfam00602; PpyrOMLV1: interval = 49–741, e-value = 2.93e-69; PpyrOMLV2: interval = 49–763, e-value = 1.42e-62) which is the RNA-directed RNA polymerase catalytic subunit, responsible for replication and transcription of virus RNA segments, with two nucleotide-binding GTP domains. PpyrOMLV1 and PpyrOMLV2 PB2 present a typical Flu_PB2 functional domain (Pfam: pfam00604; PpyrOMLV1: interval = 26–421, e-value = 5.10e-13; PpyrOMLV2: interval = 1–692, e-value = 1.57e-11) which is involved in 5' end cap RNA structure recognition and binding to further initiate virus transcription. PpyrOMLV1 and PpyrOMLV2 PA subunits share a characteristic Flu_PA domain (Pfam: pfam00603; PpyrOMLV1: interval = 122–727, e-value = 3.73e-07; PpyrOMLV2: interval = 117–732, e-value = 5.63e-10) involved in viral endonuclease activity, necessary for the cap-snatching process (*Guilligay et al., 2014*). Genome segment four (1.6 Kbp size) encodes a Hemaglutinin protein – HA (PpyrOMLV1: 526 aa, 59.7 kDA; PpyrOMLV2: 525 aa, 58.6 kDA) presenting a Baculo_gp64 domain (Pfam: pfam03273; PpyrOMLV1: interval = 108–462, e-value = 2.16e-15; PpyrOMLV2: interval = 42–460, e-value = 1.66e-23), associated with the gp64 glycoprotein from baculovirus as well as other viruses, such as Thogotovirus (*Orthomyxoviridae* - OMV) which was postulated to be related to the arthropod-borne nature of these specific Orthomyxoviruses. In addition, HA as expected, presents an N-terminal signal domain, a C terminal transmembrane domain, and a putative glycosylation site. Lastly, genome segment five (ca. 1.8 Kbp size) encodes a putative nucleocapsid protein – NP (PpyrOMLV1: 562 aa, 62.3 kDA; PpyrOMLV2: 528 aa, 58.5 kDA) with a Flu_NP structural domain (Pfam: pfam00506; PpyrOMLV1: interval = 145–322, e-value = 1.32e-01; PpyrOMLV2: interval = 94–459, e-value = 1.47e-04) this single-strand RNA-binding protein is associated to encapsidation of the virus genome for the purposes of RNA transcription, replication and packaging (*Appendix 5—figure 2E*). Despite sharing genome architecture and structural and functional domains of their predicted proteins, PpyrOMLV1 and PpyrOMLV2 pairwise identity of ortholog gene products range between 21.4% (HA) to 49.8% (PB1), suggesting although a common evolutionary history, a strong divergence indicating separated species, borderline to be considered even members of different virus genera (*Appendix 5—figure 3*). The conserved 3' sequence termini of the viral genomic RNAs are (vgRNA ssRNA(-) 3'-end) 5'-GUUCUUACU-3' for PpyrOMLV1, and and 5'-(G/A)U(U/G)(G/U/C)(A/C/U)UACU-3'. for PpyrOMLV2. The 5' termini of the vgRNAs are partially complementary to the 3' termini, supporting a panhandle structure and a hook like structure of the 5' end by a terminal short stem loop. PpyrOMLV1 and PpyrOMLV2 genome segments present an overall high identity in their respective RNA segments ends (*Appendix 5—figure 2F*). These primary and secondary sequence cues are associated to polymerase binding and promotion of both replication and transcription. In influenza viruses, and probably every OMV, the first 10 nucleotides of the 3' end form a stem-loop or 'hook' with four base-pairs (two canonical base-pairs flanked by an A-A base-pair). This compact RNA structure conforms the promoter, which activates polymerase initiation of RNA synthesis (*Reich et al., 2017*). The presence of eventual orthologs of *OMV* additional genome segments and proteins, such as Neuraminidase (NA), Matrix (M) and Non-structural proteins (NS1, NS2) was assessed retrieving no results by TBLASTN relaxed searches, nor with *in silico* approaches involving co-expression, expression levels, or conserved terminis. Given that the presence of those additional segments varies among diverse OMV genera, and that 35 related tentative new virus species identified in TSA did not present any additional segments, we believe that these lineages of viruses are conformed by five genome segments. Further experiments based on specific virus particle purification and target sequencing could corroborate our results. Based on sequence homology to best BLASTP hits, amino acid sequence alignments, predicted proteins and domains, and phylogenetic comparisons to reported species we assigned PpyrOMLV1 and PpyrOMLV2 to the OMV virus family. These are the first viruses that have been associated with the *Lampyridae* beetle family, which includes over 2000 species. The OMV virus members share diverse structural, functional and biological characters that define and restrict the family. OMV virions are 80–120 nm in diameter, of spherical or pleomorphic morphology. The virion envelope is derived from the host cell membrane,

incorporating virus glycoproteins and eventually non-glycosylated proteins (one or two in number). Typical virion surface glycoprotein projections are 10–14 nm in length and 4–6 nm in diameter. The virus genome is multisegmented, has a helical-like symmetry, consisting of different size ribonucleoproteins (RNP), 50–150 nm in length. Influenza RNPs can perform either replication or transcription of the same template. Virions of each genus contain different numbers of linear ssRNA (-) genome segments (*King et al., 2011*). Influenza A virus (FLUAV), influenza B virus (FLUBV) and infectious salmon anemia virus (ISAV) are conformed of eight segments. Influenza C virus (FLUCV), Influenza D virus (FLUDV) and Dhori virus (DHOV) have seven segments. Thogoto virus (THOV) and Quaranfil virus (QUAV) have six segments. Johnston Atoll virus (JAV) genome is still incomplete, and only two segments have been described. Segment lengths range from 736 to 2396 nt. Genome size ranges from 10.0 to 14.6 Kbp (*King et al., 2011*). As described previously, every OMV RNA segment possess conserved and partially complementary 5′- and 3′-end sequences with promoter activity (*Hsu et al., 1987*). OMV structural proteins are tentatively common to all genera involving the three polypeptides subunits that form the viral RdRP (PA, PB1, PB2) (*Pflug et al., 2017*); a nucleoprotein (NP), which binds with each genome ssRNA segment to form RNPs; and the hemagglutinin protein (HA, HE or GP), which is a type I membrane integral glycoprotein involved in virus attachment, envelope fusion and neutralization. In addition, a non-glycosylated matrix protein (M) is present in most species. There are some species-specific divergence in some structural OMVs proteins. For instance, HA of FLUAV is acylated at the membrane-spanning region and has widespread N-linked glycans (*Eisfeld et al., 2015*). The HA protein of FLUCV, besides its hemagglutinating and envelope fusion function, has an esterase activity that induces host receptor enzymatic destruction (*King et al., 2011*). In contrast, the HA of THOV is divergent to influenzavirus HA proteins, and presents high sequence similarity to a baculovirus surface glycoprotein (*Leahy et al., 1997*). The HA protein has been described to have an important role in determining OMV host specificity. For instance, human infecting Influenza viruses selectively bind to glycolipids that contain terminal sialyl-galactosyl residues with a 2–6 linkage, in contrast, avian influenza viruses bind to sialyl-galactosyl residues with a 2–3 linkage (*King et al., 2011*). Furthermore, FLUAV and FLUBV share a neuraminidase protein (NA), which is an integral, type II envelope glycoprotein containing sialidase activity. Some OMVs possess additional small integral membrane proteins (M2, NB, BM2, or CM2) that may be glycosylated and have diverse functions. As an illustration, M2 and BM2 function during un-coating and fusion by equilibrating the intralumenal pH of the trans-Golgi apparatus and the cytoplasm. In addition, some viruses encode two nonstructural proteins (NS1, NS2) (*King et al., 2011*). OMV share replication properties, which have been studied mostly in Influenza viruses. It is important to note that gene reassortment has been described to occur during mixed OMV infections, involving viruses of the same genus, but not between viruses of different genera (*Kimble, 2013*). This is used also as a criteria for OMV genus demarcation. Influenza virus replication and transcription occurs in the cell nucleus and comprises the production of the three types of RNA species (i) genomic RNA (vRNA) which are found in virions; (ii) cRNA molecules which are complementary RNA in sequence and identical in length to vRNA; and also (iii) virus mRNA molecules which are 5′ capped by cap snatching of host RNAs and 3′ polyadenylated by polymerase stuttering on U rich stretches. These remarkable dynamic multifunction characters of OMV polymerases are associated with its complex tertiary structure, of this modular heterotrimeric replicase (*Te Velthuis and Fodor, 2016*). We explored in detail the putative polymerase subunits of the identified firefly viruses. The PB1 subunit catalyzes RNA synthesis in its internal active site opening, which is formed by the highly conserved polymerase motifs I-III. Motifs I and III (*Appendix 5—figure 2H*) present three conserved aspartates (PpyrOMLV1: Asp 346, Asp 491 and Asp 492; PpyrOMLV2: Asp 348, Asp 495 and Asp 496) which coordinate and promote nucleophilic attack of the terminal 3′ OH from the growing transcript on the alpha-phosphate of the inbound NTP (*Pflug et al., 2017*). Besides presenting, with high confidence, the putative functional domains associated with their potential replicase/transcriptase function, we assessed whether the potential spatial and functional architecture was conserved at least in part in FOML viruses. In this direction we employed the SWISS-MODEL automated protein

structure homology-modelling server to generate a 3D structure of PpyrOMLV1 heterotrimeric polymerase. The SWISS server selected as best-fit template the trimeric structure of Influenza A virus polymerase, generating a structure for each polymerase subunit of PpyrOMLV1. The generated structure shared structural cues related to its multiple role of RNA nucleotide binding, endonuclease, cap binding, and nucleotidyl transferase (*Appendix 5—figure 2G-H*). The engendered subunit structures suggest a probable conservation of PpyrOMLV1 POL, that could allow the predicted functional enzymatic activity of this multiple gene product. The overall polymerase rendered structure presents a typical U shape with two upper protrusions corresponding to the PA endonuclease and the PB2 cap-binding domain. The PB1 subunit appears to plug into the interior of the U and has the distinctive fold of related viral RNA polymerases with fingers, palm and thumb adjacent to a tentative central active site opening where RNA synthesis may occur (*Reich et al., 2017*; *Hengrung et al., 2015*). OMV Pol activity is central in the virus cycle of OMVs, which have been extensively studied. The life cycle of OMVs starts with virus entry involving the HA by receptor-mediated endocytosis. For Influenza, sialic acid bound to glycoproteins or glycolipids function as receptor determinants of endocytosis. Fusion between viral and cell membranes occurs in endosomes. The infectivity and fusion of influenza is associated to the post-translational cleavage of the virion HA. Cleavability depends on the number of basic amino acids at the target cleavage site (*King et al., 2011*). In thogotoviruses, no requirement for HA glycoprotein cleavage have been demonstrated (*Leahy et al., 1997*). Integral membrane proteins migrate through the Golgi apparatus to localized regions of the plasma membrane. New virions form by budding, incorporating matrix proteins and viral RNPs. Viral RNPs are transported to the cell nucleus where the virion polymerase complex synthesizes mRNA species (*Hara et al., 2017*). Another tentative function of the NP could be associated to the potential interference of the host immune response in the nucleus mediated by capsid proteins of some RNA virus, which could inhibit host transcription and thus liberate and direct it to viral RNA synthesis (*Wulan et al., 2015*). mRNA synthesis is primed by capped RNA fragments 10–13 nt in length that are generated by cap snatching from host nuclear RNAs which are sequestered after cap recognition by PB2 and incorporated to vRNA by PB1 and PA proteins which present viral endonuclease activity (*Sikora et al., 2017*). In contrast, thogotoviruses have capped viral mRNA without host-derived sequences at the 5′ end. Virus mRNAs are polyadenylated at the 3′ termini through iterative copying by the viral polymerase stuttering on a poly U track in the vRNA template. Some OMV mRNAs are spliced generating alternative gene products with defined functions. Protein synthesis of influenza viruses occurs in the cytoplasm. Partially complementary vRNA molecules act as templates for new viral RNA synthesis and are neither capped nor polyadenylated. These RNAs exist as RNPs in infected cells. Given the diverse hosts of OMV, biological properties of virus infection diverge between species. Influenzaviruses A infect humans and cause respiratory disease, and they have been found to infect a variety of bird species and some mammalian species. Interspecies transmission, although rare, is well documented. Influenza B virus infect humans and cause epidemics, and have been rarely found in seals. Influenzaviruses C cause limited outbreaks in humans and have been occasionally found on dogs. Influenza spreads globaly in a yearly outbreak, resulting in about three to five million cases of severe illness and about 250,000 to 500,000 human deaths (*Thompson et al., 2009*). Influenzavirus D has been recently reported and accepted and infects cows and swine (*Hause et al., 2013*). Natural transmission of influenzaviruses is by aerosol (human and non-aquatic hosts) or is water-borne (avians). In contrast, Thogoto and Dhori viruses which also infect humans, are transmitted by, and able to replicate in ticks. Thogoto virus was identified in *Rhipicephalus sp.* ticks collected from cattle in the Thogoto forest in Kenya, and Dhori virus was first isolated in India from *Hyalomma dromedarri*, a species of camel ticks (*Anderson and Casals, 1973*; *Haig et al., 1965*). Dhori virus infection in humans causes a febrile illness and encephalitis. Serological evidence suggests that cattle, camel, goats, and ducks might be also susceptible to this virus. Experimental hamster infection with THOV may be lethal. Unlike influenzaviruses, these viruses do not cause respiratory disease. The transmission of fish infecting isaviruses (ISAV) is via water, and virus infection induces the agglutination of erythrocytes of many fish species, but not avian or mammalian

erythrocytes (*Mjaaland et al., 1997*). Quaranfil and Johnston Atoll are transmitted by ticks and infect avian species (*Presti et al., 2009*).

We have limited biological data of the firefly detected viruses. Nevertheless, a significant consistency in the genomic landscape and predicted gene products of the detected viruses in comparison with accepted OMV species sufficed to suggest for PpyrOMLV1 and PpyrOMLV2 a tentative taxonomic assignment within the OMV family. Besides relying on the OMV structural and functional signatures determined by virus genome annotation, we explored the evolutionary clustering of the detected viruses by phylogenetic insights. We generated MAFFT alignments and phylogenetic trees of the predicted viral polymerase of firefly viruses and the corresponding replicases of all 493 proposed and accepted species of ssRNA(-) virus. The generated trees consistently clustered the diverse sequences to their corresponding taxonomical niche, at the level of genera. Interestingly, PpyrOMLV1 and PpyrOMLV2 replicases were placed unequivocally within the OMV family (*Appendix 5—figure 2B*). When the genetic distances of firefly viruses proteins and ICTV accepted OMV species were computed, a strong similarity was evident (*Appendix 5—figure 2B-D*). Overall similarity levels of PpyrOMLV polymerase subunits ranged between 11.03% to as high as 37.30% among recognized species, while for the more divergent accepted OMV (ISAV - *Isavirus* genus) these levels ranged only from 8.54% to 20.74%, illustrating that PpyrOMLV are within the OMV by genetic standards. Phylogenetic trees based on aa alignments of structural gene products of recognized species and PpyrOMLV supported this assignment, placing ISAV and issavirus as the most distant species and genus within the family, and clustering PpyrOMLV1 and PpyrOMLV2 in a distinctive lineage within OMV, more closely related to the *Quaranjavirus* and *Thogotovirus* genera than the *Influenza A-D* or *Isavirus* genera (*Appendix 5—figure 3*). Furthermore, it appears that virus genomic sequence data, while it has been paramount to separate species, in the case of genera, there are some contrasting data that should be taken into consideration. For instance, DHOV and THOV are both members of the *Thogotovirus* genus, sharing a 61.9% and a 34.9% identity at PB1 and PB2, respectively. However, FLUCV and FLUDV are assigned members of two different genus, *Influenzavirus C* and *Influenzavirus D*, while sharing a higher 72.2% and a 52.2% pairwise identity at PB1 and PB2, respectively (*Appendix 5—figure 3*). In addition, FLUAV and FLUBV, assigned members of two different genus, *Influenzavirus A* and *Influenzavirus D* present a comparable identity to that of DHOV and THOV thogotoviruses, sharing a 61% and a 37.9% identity at PB1 and PB2, respectively. It is worth noting that similarity thresholds and phylogenetic clustering based in genomic data have been used differently to demarcate OMV genera, hence there is a need to eventually re-evaluate a series of consensus values, which in addition to biological data, would be useful to redefine the OMV family. Perhaps, these criteria discrepancies are more related to a historical evolution of the OMV taxonomy than to pure biological or genetic standards. In contrast to FLUDV, JOV and QUAV, the other virus members of OMV have been described, proposed and assigned at least 34 years ago.

The potential prevalence, tissue/organ tropism, geographic dispersion and lifestyle of PpyrOMLV1 and 2 were assessed by the generation and analyses of 29 specific RNA-Seq libraries of *P. pyralis* (*Appendix 1—table 1*). As RNA was isolated from independent *P. pyralis* individuals of diverse origin, wild caught or lab reared, the fact that we found at least one of the PpyrOMLV present in 82% of the libraries reflects a widespread presence and potentially a high prevalence of these viruses in *P. pyralis* (*Appendix 5—figure 2J*, *Appendix 5—table 3*, S5.4.6). Wild caught individuals were collected in period spanning six years, and locations separated as much as 900 miles (New Jersey – Georgia, USA). Interestingly PpyrOMLV1 and 2 were found in individuals of both location, and the corresponding assembled isolate virus sequences presented negligible differences, with an inter-individual variability equivalent to that of isolates (0.012%). A similar result was observed for virus sequences identified in RNA libraries generated from samples collected in different years. We were not able to identified fixed mutations associated to geographical or chronological cues. Further experiments should explore the mutational landscape of PpyrOMLV1 and 2, which appears to be significantly lower than of Influenzaviruses, specifically *Influenza A virus*, which are characterized by high mutational rate (ca. one mutation per genome replication) associated to the absence of RNA

proofreading enzymes (*Pauly et al., 2017*). In addition we evaluated the presence of PpyrOMLV1 and 2 on diverse tissues and organs of *P. pyralis*. Overall virus RNA levels were generally low, with an average of 9.47 FPKM on positive samples. However, PpyrOMLV1 levels appear to be consistently higher than PpyrOMLV2, with an average of 20.50 FPKM for PpyrOMLV1 versus 4.22 FPKM for PpyrOMLV2 on positive samples. When the expression levels are scrutinized by genome segment, HA and NP encoding segments appear to be, for both viruses, at higher levels, which would be in agreement with other OMV such as Influenzaviruses, in which HA and NP proteins are the most expressed proteins, and thus viral mRNAs are consistently more expressed (*King et al., 2011*). Nevertheless, these preliminary findings related to expression levels should be taken cautiously, given the small sample size. Perhaps, the more remarkable allusion derived from the analyses of virus presence is related to tissue and organ deduced virus tropism. Strikingly, we found virus transcripts in samples exclusively obtained from light organs, complete heads, male or female thorax, female spermatheca, female spermatophore digesting glands and bursa, abdominal fat bodies, male reproductive spiral gland, and other male reproductive accessory glands (*Appendix 5—table 3*, S5.4.6), indicating a widespread tissue/organ tropism of PpyrOMLV1 and 2. This tentatively pervasive tropism of PpyrOMLV1 and 2 emerges as a differentiation character of these viruses and accepted OMV. For instance, influenza viruses present a epithelial cell-specific tropism, restricted typically to the nose, throat, and lungs of mammals, and intestines of birds. Tropism has consequences on host restriction. Human influenza viruses mainly infect ciliated cells, because attachment of all *influenza A virus* strains to cells requires sialic acids. Differential expression of sialic acid residues in diverse tissues may prevent cross-species or zoonotic transmission events of avian influenza strains to man (*Zeng et al., 2013*). Tropism has also influence in disease associated effects of OMV. Some *influenza A virus* strains are more present in tracheal and bronchial tissue which is associated with the primary lesion of tracheobronchitis observed in typical epidemic influenza. Other *influenza A virus* strains are more prevalent in type II pneumocytes and alveolar macrophages in the lower respiratory tract, which is correlated to diffuse alveolar damage with avian influenza (*Mansfield, 2007*). The presence of PpyrOMLV1 and 2 virus RNA in reproductive glands raises some potential of the involvement of sex in terms of prospective horizontal transmission. Given that most libraries corresponded to 3–6 pooled individuals samples of specific organs/tissue, direct comparisons of virus RNA levels were not always possible. However, this valuable data gives important insights into the widespread potential presence of the viruses in every analyzed organ/tissue. Importantly, RNA levels of the putative virus segments shared co-expression levels and a systematic pattern of presence/absence, supporting the suggested multipartite nature of the viruses. We observed the presence of virus RNA of both PpyrOMLV1 and 2 in eight of the RNA-Seq libraries, thus mixed infections appear to be common. Interestingly, we did not observe in any of the 24 virus positive samples evidence of reassortment. Reassortment is a common event in OMV, a process by which influenza viruses swap gene segments. Genetic exchange is possible due to the segmented nature of the OMV viral genome and may occur during mixed infections. Reassortment generates viral diversity and has been associated to host gain of Influenzavirus (*Steel and Lowen, 2014*). Reassorted Influenzavirus have been reported to occasionally cross the species barrier, into birds and some mammalian species like swine and eventually humans. These infections are usually dead ends, but sporadically, a stable lineage becomes established and may spread in an animal population (*Kimble, 2013*). Besides its evolutionary role, reassortment has been used as a criterion for species/genus demarcation, thus the lack of observed gene swap in our data supports the phylogenetic and sequence similarity insights that indicates species separation of PpyrOMLV1 and 2.

In light of the presence of virus RNA in reproductive glands, we further explored the potential life style of PpyrOMLV1 and 2 related to eventual vertical transmission. Vertical transmission is extremely exceptional for OMV, and has only been conclusively described for the *Infectious salmon anemia virus* (*Isavirus*) (*Marshall et al., 2014*). In this direction, we were able to generate a strand-specific RNA-Seq library of one *P. pyralis* adult female PpyrOMLV1 virus positive (parent), another library from seven eggs of this female at ~13 days post

fertilization, and lastly an RNA-Seq library of four 1 st instar larvae (offspring). When we analyzed the resulting RNA reads, we found as expected virus RNA transcripts of every genome segment of PpyrOMLV1 in the adult female library. Remarkably, we also found PpyrOMLV1 sequence reads of every genome segment of PpyrOMLV1 in both the eggs and larvae samples. Moreover, virus RNA levels fluctuated among the different developmental stages of the samples. The average RNA levels of the adult female were 41.10 FPKM, in contrast, the fertilized eggs sample had higher levels of virus related RNA, averaging at 61.61 FPKM and peaking at the genome segment encoding NP (104.49 FPKM). Interestingly, virus RNA levels appear to drop in first instar larvae, in the sequenced library average virus RNA levels were of 10.42 FPKM. Future experiments should focus on PpyrOMLV1 and 2 virus titers at extended developmental stages to complement these preliminary results. However, it is interesting to note that the tissue specific library corresponding to female spermatheca, where male sperm are stored prior to fertilization, presented relatively high levels of both PpyrOMLV1 and 2 virus RNAs, suggesting that perhaps during early reproductive process and during egg development virus RNAs tend to raise. This tentatively differential and variable virus RNA titers observed during development could be associated to an unknown mechanism of modulation of latent antiviral response that could be repressed in specific life cycle stages. Further studies may validate these results and unravel a mechanistic explanation of this phenomenon. Nevertheless, besides the preliminary developmental data, the consistent presence of PpyrOMLV1 in lab-reared, isolated offspring of an infected *P. pyralis* female is robust evidence demonstrating mother-to-offspring vertical transmission for this newly identified OMV.

One of many questions that remains elusive here is whether PpyrOMLV1 and 2 are associated with any potential alteration of phenotype of the infected host. We failed to unveil any specific effect of the presence of PpyrOMLV1 and 2 on fireflies. It is worth noting that subtle alterations or symptoms would be difficult to pinpoint in these insects. Future studies should enquire whether PpyrOMLV1 and 2 may have any influence in biological attributes of fireflies such as fecundity, life span or life cycle. Nevertheless, we observed in our data some hints that could be indicative of a chronic state status, cryptic or latent infection of firefly individuals: (i) virus positive individuals presented in general relatively low virus RNA levels. (ii) virus RNA was found in every assessed tissue/organ. (iii) vertical transmission of the identified viruses. The first hint is hardly conclusive, it is difficult to define what a relatively low RNA level is, and high virus RNA loads are not directly associated with disease on reported OMV. The correlation of high prevalence, prolonged host infection, and vertical transmission observed in several new mosquito viruses has resulted in their classification as 'commensal' microbes. A shared evolutionary history of viruses and host, based in strategies of immune evasion of the viruses and counter antiviral strategies of the host could occasionally result in a modulation of viral loads and a chronic but latent state of virus infection (*Hall et al., 2016*).

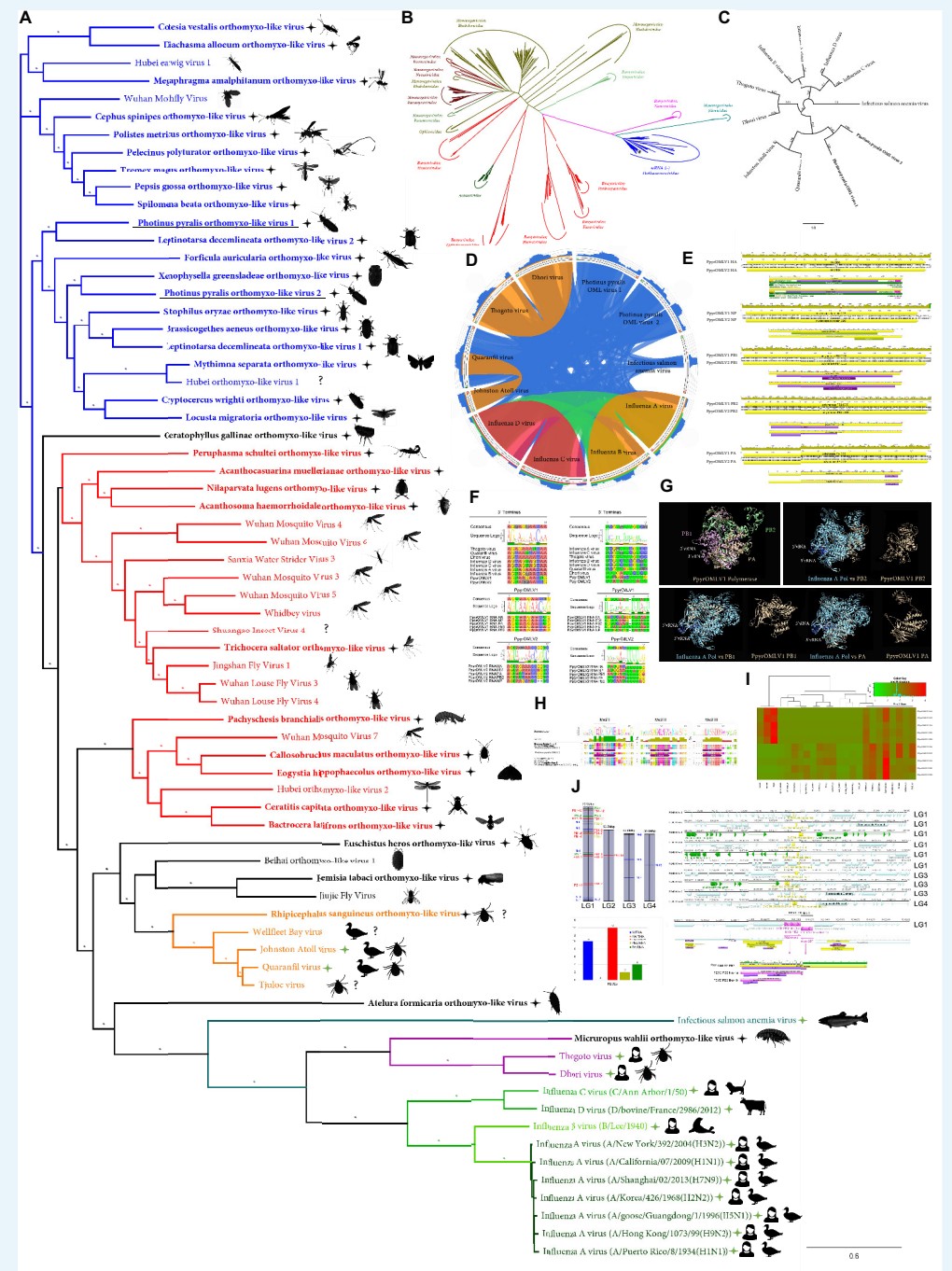

**Appendix 5—figure 2.** *Photinus pyralis* viruses and endogenous viral-like elements. (**A**) Phylogenetic tree based in MAFFT alignments of predicted replicases of *Orthomyxoviridae* (OMV) ICTV accepted viruses (green stars), new *Photinus pyralis* viruses (underlined) and tentative OMV-like virus species (black stars). ICTV recognized OMV genera: *Quaranjavirus* (orange), *Thogotovirus* (purple), *Issavirus* (turquoise), *Influenzavirus A-D* (green). Silhouettes correspond to host species. Asterisk denote FastTree consensus support >0.5. Question marks depict viruses with unidentified or unconfirmed host. (**B**) Phylogenetic tree of OMV proposed and recognized species in the context of all ssRNA (-) virus species, based on MAFFT alignments of refseq replicases. *Photinus* pyralis viruses are portrayed by black stars. (**C**) Phylogenetic tree of ICTV recognized OMV species and PpyrOMLV1 and 2. Numbers indicate FastTree consensus support. (**D**) Genetic distances of concatenated gene products of OMV depicted as circoletto diagrams. Proteins are oriented clockwise in N-HA-PB1-PB2-PA order

when available. Sequence similarity is expressed as ribbons ranging from blue (low) to red (high). (**E**) Genomic architecture, predicted gene products and structural and functional domains of PpyrOLMV1 and 2. (**F**) Virus genomic noncoding termini analyses of PpyrOLMV1 and 2 in the context of ICTV OMV. The 3' and 5' end, A and U rich respectively, partially complementary sequences are associated to tentative panhandle polymerase binding and replication activity, typical of OMV. (**G**) 3D renders of the heterotrimeric polymerase of PpyrOMLV1 based on Swiss-Expasy generated models using as template the Influenza A virus polymerase structure. Structure comparisons were made with the MatchAlign tool of the Chimera suite, and solved in PyMOL. (**H**) Conserved functional motifs of PpyrOLMV1 and 2 PB1 and related viruses. Motif I-III are essential for replicase activity of viral polymerase. (**I**) Dynamic and prevalent virus derived RNA levels of the corresponding PpyrOMLV1 and 2 genome segments, determined in 24 RNA libraries of diverse individuals/developmental stages/tissues and geographic origins. RNA levels are expressed as normalized TPM, heatmaps were generated by Shinyheatmap. Values range from low (green) to high (red). (**J**) Firefly EVEs (FEVEs) identified in the *P. pyralis* genome assembly mapped to the corresponding pseudo-molecules. A 15 Kbp region flanking nucleoprotein like FEVES are depicted, enriched in transposable elements. Representative products of a putative PB2 FEVE are aligned to the corresponding protein of PpyrOMLV 2.

DOI: https://doi.org/10.7554/eLife.36495.085

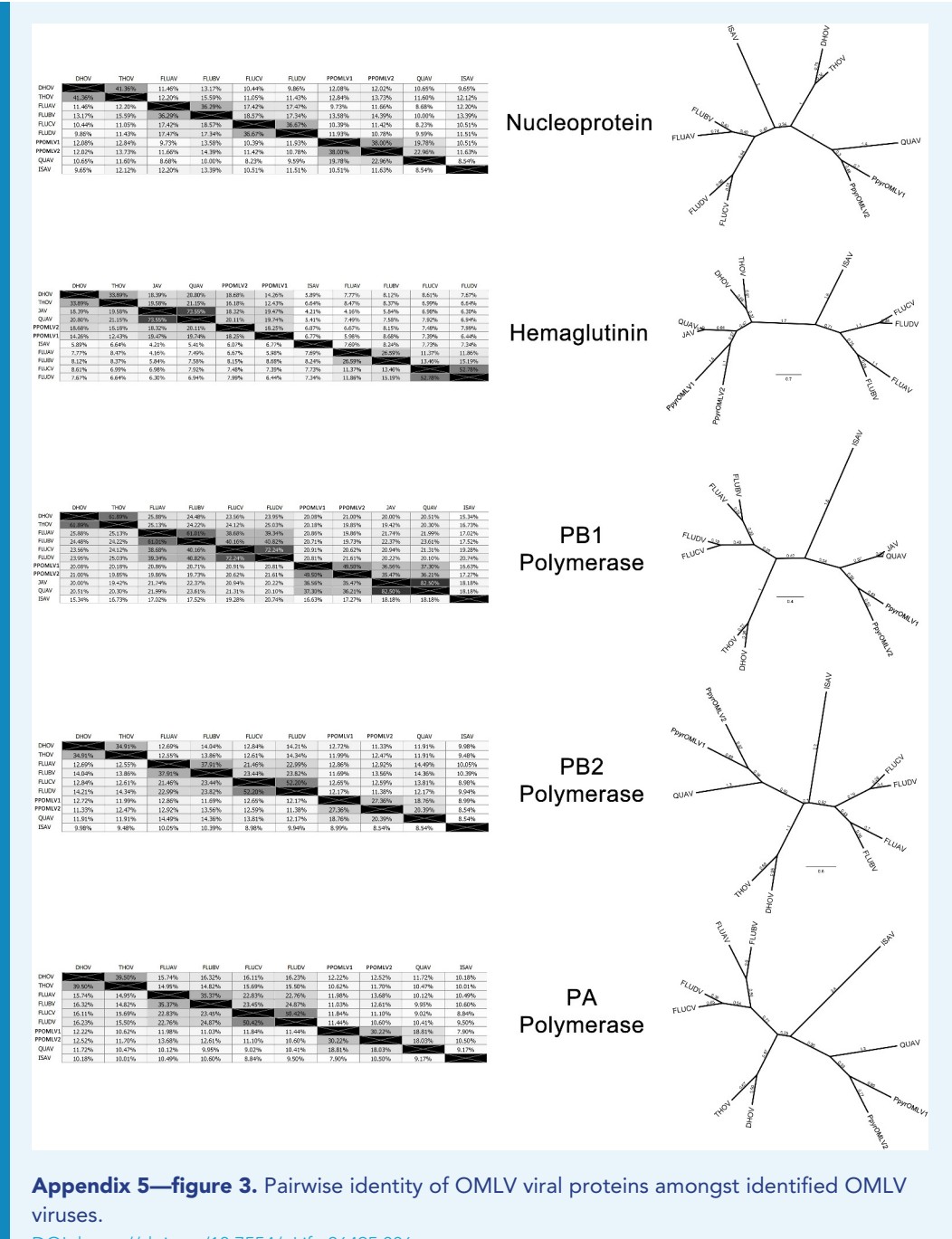

**Appendix 5—figure 3.** Pairwise identity of OMLV viral proteins amongst identified OMLV viruses.

DOI: https://doi.org/10.7554/eLife.36495.086

**Appendix 5—table 1.** Best hits from BLASTP of PpyrOMLV proteins against the NCBI database

| Genome segment | Size (nt) | Gene product (aa) | Best hit | Best hit taxonomy | Query cover | E value | Identity |
|---|---|---|---|---|---|---|---|
| PpyrOMLV1-PB1 | 2510 | 801 PB1 | Wuhan Mothfly Virus | Orthomyxoviridae | 83% | 0.0 | 51% |
| PpyrOMLV1-PA | 2346 | 754 PA | Hubei earwig virus 1 | Orthomyxoviridae | 98% | 4.00E-137 | 35% |
| PpyrOMLV1-HA | 1667 | 526 HA | Tjuloc virus | Orthomyxoviridae | 91% | 9.00E-25 | 25% |
| PpyrOMLV1-PB2 | 2517 | 804 PB2 | Hubei earwig virus 1 | Orthomyxoviridae | 91% | 3.00E-118 | 31% |
| PpyrOMLV1-N | 1835 | 562 N | Hubei earwig virus 1 | Orthomyxoviridae | 93% | 8.00E-74 | 30% |
| PpyrOMLV2-PB1 | 2495 | 802 PB1 | Hubei ortho-myxo-like virus 1 | Orthomyxoviridae | 93% | 0.0 | 48% |
| PpyrOMLV2-PA | 2349 | 762 PA | Hubei earwig virus 1 | Orthomyxoviridae | 98% | 1.00E-107 | 31% |
| PpyrOMLV2-HA | 1668 | 525 HA | Wellfleet Bay virus | Orthomyxoviridae | 82% | 3.00E-40 | 26% |
| PpyrOMLV2-PB2 | 2506 | 801 PB2 | Hubei earwig virus 1 | Orthomyxoviridae | 96% | 3.00E-86 | 27% |
| PpyrOMLV2-N | 1738 | 528 N | Hubei earwig virus 1 | Orthomyxoviridae | 95% | 6.00E-82 | 32% |

DOI: https://doi.org/10.7554/eLife.36495.087

**Appendix 5—table 2.** InterProScan domain annotation of PpyrOMLV proteins.

| Genome product | Annotation | Start | End | Length | Database | Id | InterPro ID | InterPro name |
|---|---|---|---|---|---|---|---|---|
| PpyrOMLV1-PB1 | Flu_PB1 | 48 | 752 | 705 | PFAM | PF00602 | IPR001407 | RNA_pol_PB1_influenza |
| | RDRP_SSRNA | 330 | 529 | 200 | PROSITE_PROFILES | PS50525 | IPR007099 | RNA-dir_pol_NSvirus |
| PpyrOMLV2-PB1 | Flu_PB1 | 54 | 766 | 713 | PFAM | PF00602 | IPR001407 | RNA_pol_PB1_influenza |
| | RDRP_SSRNA | 337 | 539 | 203 | PROSITE_PROFILES | PS50525 | IPR007099 | RNA-dir_pol_NSvirus |
| PpyrOMLV1-PB2 | Flu_PB2 | 13 | 421 | 409 | PFAM | PF00604 | IPR001591 | RNA_pol_PB2_orthomyxovir |
| PpyrOMLV2-PB2 | Flu_PB2 | 13 | 415 | 403 | PFAM | PF00604 | IPR001591 | RNA_pol_PB2_orthomyxovir |
| PpyrOMLV1-HA | SignalP-noTM | 1 | 19 | 19 | SIGNALP_EUK | SignalP-noTM | | Unintegrated |
| | Baculo_gp64 | 108 | 432 | 325 | PFAM | PF03273 | IPR004955 | Baculovirus_Gp64 |
| PpyrOMLV2-HA | SignalP-noTM | 1 | 21 | 21 | SIGNALP_EUK | SignalP-noTM | | Unintegrated |
| | Baculo_gp64 | 66 | 426 | 361 | PFAM | PF03273 | IPR004955 | Baculovirus_Gp64 |
| PpyrOMLV1-PA | Flu_PA | 663 | 736 | 74 | PFAM | PF00603 | IPR001009 | RNA-dir_pol_influenzavirus |

*Appendix 5—table 2 continued on next page*

*Appendix 5—table 2 continued*

| Genome product | Annotation | Start | End | Length | Database | Id | InterPro ID | InterPro name |
|---|---|---|---|---|---|---|---|---|
| PpyrOMLV2-PA | Flu_PA | 667 | 740 | 74 | PFAM | PF00603 | IPR001009 | RNA-dir_pol_influenzavirus |
| PpyrOMLV1-PB1 | flu NP-like | 94 | 459 | 366 | SUPER FAMILY | SSF161003 | | Unintegrated |
| PpyrOMLV2-PB1 | flu NP-like | 363 | 483 | 121 | SUPER FAMILY | SSF161003 | | Unintegrated |

DOI: https://doi.org/10.7554/eLife.36495.088

## 5.5 *P. pyralis* Endogenous virus-like Elements (EVEs)

To gain insights on the potential shared evolutionary history of *P. pyralis* and the IOMV PpyrOMLV1 and 2, we examined our assembly of *P. pyralis* for putative signatures or paleovirological traces (**Ballinger et al., 2014**; **Metegnier et al., 2015**; **Feschotte and Gilbert, 2012**) that would indicate ancestral integration of virus related sequences into the firefly host. Remarkably, we found Endogenous virus-like Elements (EVEs) (**Katzourakis and Gifford, 2010**), sharing significant sequence identity with most PpyrOMLV1 and 2 genome segments, spread along four *P. pyralis* linkage-groups. Virus integration into host genomes is a frequent event derived from reverse transcribing RNA viruses (*Retroviridae*). Retroviruses are the only animal viruses that depend on integration into the genome of the host cell as an obligate step in their replication strategy (**Temin, 1985**). Viral infection of germ line cells may lead to viral gene fragments or genomes becoming integrated into host chromosomes and subsequently inherited as host genes.

Animal genomes are paved by retrovirus insertions (**Bushman et al., 2005**). These insertions, which are eventually eliminated from the host gene pool within a few generations, and may, in some cases, increase in frequency, and ultimately reach fixation. This fixation in the host species can be mediated by drift or positive selection, depending on their selective value. On the other hand, genomic integration of non-retroviral viruses, such as PpyrOMLV1 and 2, is less common. Viruses with a life cycle characterized by no DNA stage, such as OMV, do not encode a reverse transcriptase or integrase, thus are not retro transcribed nor integrated into the host genome. However, exceptionally and recently, several non-retroviral sequences have been identified on animal genomes; these insertions have been usually associated with the transposable elements machinery of the host, which provided a means to genome integration (**Gilbert and Cordaux, 2017**; **Palatini et al., 2017**). Interestingly, when we screened our *P. pyralis* genome assembly Ppyr1.2 by BLASTX searches (E-value $<1e10^{-6}$) of PpyrOMLV1 and 2 genome segments, we identified several genome regions that could be defined as Firefly EVEs, which we termed FEVEs (**Appendix 5—figure 2J**; **Appendix 5—table 5**-8). We found 30 OMV related FEVEs, which were mostly found in linkage group one (LG1, 83% of pinpointed FEVEs). The majority of the detected FEVEs shared sequence identity to the PB1 encoding region of genome segment one of PpyrOMLV1 and 2 (ca. 46% of FEVEs; **Appendix 5—table 5**), followed by NP encoding genome segment five (ca. 33% of detected FEVEs; **Appendix 5—table 8**). In addition we identified four FEVEs related to genome segment three (PA region; **Appendix 5—table 7**) and two FEVEs associated to genome segment two (PB2 encoding region; **Appendix 5—table 6**). We found no evidence of FEVEs related to the hemagglutinin coding genome segment four (HA) via BLASTX. The detected *P. pyralis* FEVEs represented truncated fragments of virus like sequences, generally presenting frameshift mutations, early termination codons, lacking start codons, and sharing diverse mutations that altered the potential translation of eventual gene products. FEVEs shared sequence similarity to the coding sequence of specific genome segments of the cognate FOLMV. We generated best/longest translation products of the corresponding FEVEs, which presented an average length of ca. 21.86% of the corresponding PpyrOMLV genome segment encoding gene region (**Appendix 5—table 5-5.5.5**), and an average pairwise identity to the FOLMV virus protein of 55.08%. Nevertheless, we were able to

**Appendix 5—table 3.** Total reads mapped to PpyrOMLV genome segments from *P. pyralis* RNA-Seq datasets.

| | SRR 3883762 | SRR 3883763 | SRR 3883764 | SRR 3883765 | SRR 3883767 | SRR 3883768 | SRR 3883769 | SRR 3883770 | SRR 3883771 | SRR 3883758 | SRR 3883772 | SRR 3883773 |
|---|---|---|---|---|---|---|---|---|---|---|---|---|
| Ppyr OMLV1 HA | 2848 | 199 | 2 | 0 | 2 | 881 | 4 | 0 | 160 | 2 | 541 | 11 |
| Ppyr OMLV1 NP | 1460 | 120 | 0 | 0 | 0 | 523 | 0 | 0 | 141 | 0 | 321 | 0 |
| Ppyr OMLV1 PA | 660 | 100 | 5 | 0 | 1 | 306 | 0 | 0 | 95 | 0 | 256 | 3 |
| Ppyr OMLV1 PB1 | 1464 | 669 | 0 | 0 | 0 | 820 | 4 | 0 | 208 | 2 | 364 | 2 |
| Ppyr OMLV1 PB2 | 696 | 106 | 0 | 0 | 2 | 319 | 0 | 2 | 152 | 0 | 194 | 5 |
| Ppyr OMLV2 HA | 710 | 232 | 10 | 22 | 38 | 549 | 247 | 54 | 124 | 266 | 444 | 12 |
| Ppyr OMLV2 NP | 1067 | 274 | 57 | 205 | 24 | 653 | 299 | 66 | 144 | 275 | 526 | 29 |
| Ppyr OMLV2 PA | 838 | 50 | 8 | 15 | 18 | 97 | 204 | 40 | 72 | 216 | 88 | 12 |
| Ppyr OMLV2 PB1 | 493 | 146 | 57 | 74 | 8 | 76 | 78 | 26 | 72 | 75 | 115 | 9 |
| Ppyr OMLV2 PB2 | 728 | 173 | 72 | 85 | 22 | 110 | 131 | 47 | 67 | 57 | 50 | 5 |

| | Ppyr _eggs | Ppyr _Female | Ppyr _larvae | SRR 2103848 | SRR 2103849 | SRR 2103867 | SRR 3883766 | SRR 3883756 | SRR 3883757 | SRR 3883759 | SRR 3883760 | SRR 3883761 |
|---|---|---|---|---|---|---|---|---|---|---|---|---|
| Ppyr OMLV1 HA | 15586 | 7826 | 1664 | 0 | 0 | 0 | 0 | 867 | 6 | 2 | 578 | 0 |
| Ppyr OMLV1 NP | 6562 | 5216 | 644 | 0 | 0 | 2 | 0 | 647 | 3 | 0 | 289 | 0 |
| Ppyr OMLV1 PA | 9564 | 3692 | 1264 | 0 | 0 | 0 | 0 | 626 | 2 | 0 | 124 | 0 |

*Appendix 5—table 3 continued on next page*

Appendix 5—table 3 continued

| | SRR 3883762 | SRR 3883763 | SRR 3883764 | SRR 3883765 | SRR 3883767 | SRR 3883768 | SRR 3883769 | SRR 3883770 | SRR 3883771 | SRR 3883758 | SRR 3883772 | SRR 3883773 |
|---|---|---|---|---|---|---|---|---|---|---|---|---|
| Ppyr OMLV1 PB1 | 15952 | 7144 | 2824 | 0 | 0 | 0 | 2 | 1607 | 3 | 0 | 460 | 2 |
| Ppyr OMLV1 PB2 | 10568 | 2562 | 648 | 0 | 0 | 0 | 0 | 848 | 2 | 0 | 188 | 0 |
| Ppyr OMLV2 HA | 0 | 0 | 0 | 415 | 190 | 43 | 286 | 337 | 546 | 23 | 236 | 13 |
| Ppyr OMLV2 NP | 0 | 0 | 0 | 432 | 127 | 51 | 196 | 482 | 501 | 22 | 248 | 32 |
| Ppyr OMLV2 PA | 0 | 0 | 0 | 97 | 54 | 75 | 131 | 222 | 234 | 6 | 93 | 14 |
| Ppyr OMLV2 PB1 | 0 | 0 | 0 | 190 | 96 | 22 | 63 | 180 | 168 | 4 | 90 | 29 |
| Ppyr OMLV2 PB2 | 0 | 0 | 0 | 96 | 57 | 22 | 94 | 230 | 256 | 6 | 90 | 49 |

DOI: https://doi.org/10.7554/eLife.36495.089

**Appendix 5—table 4.** FPKM of reads mapped to PpyrOMLV genome segments from *P. pyralis* RNA-Seq datasets.

| | SRR 3883773 | SRR 3883772 | SRR 3883758 | SRR 3883771 | SRR 3883770 | SRR 3883769 | SRR 3883768 | SRR 3883767 | SRR 3883765 | SRR 3883764 | SRR 3883763 | SRR 3883762 |
|---|---|---|---|---|---|---|---|---|---|---|---|---|
| Ppyr OMLV1 HA | 19.10 | 0.32 | 0.05 | 6.46 | 0.00 | 0.11 | 30.69 | 0.05 | 0.00 | 0.08 | 4.07 | 69.54 |
| Ppyr OMLV1 NP | 10.37 | 0.00 | 0.00 | 5.21 | 0.00 | 0.00 | 16.66 | 0.00 | 0.00 | 0.00 | 2.24 | 32.61 |
| Ppyr OMLV1 PA | 6.46 | 0.06 | 0.00 | 2.74 | 0.00 | 0.00 | 7.62 | 0.02 | 0.00 | 0.13 | 1.46 | 11.52 |
| Ppyr OMLV1 PB1 | 8.53 | 0.04 | 0.04 | 5.57 | 0.00 | 0.07 | 18.95 | 0.00 | 0.00 | 0.00 | 9.07 | 23.72 |
| Ppyr OMLV1 PB2 | 4.50 | 0.10 | 0.00 | 4.03 | 0.05 | 0.00 | 7.29 | 0.03 | 0.00 | 0.00 | 1.42 | 11.16 |
| Ppyr OMLV2 HA | 16.13 | 0.36 | 7.41 | 5.15 | 2.31 | 6.80 | 19.68 | 0.90 | 1.05 | 0.39 | 4.88 | 17.84 |
| Ppyr OMLV2 NP | 17.36 | 0.79 | 6.96 | 5.44 | 2.57 | 7.48 | 21.27 | 0.52 | 8.87 | 2.01 | 5.24 | 24.36 |
| Ppyr OMLV2 PA | 2.21 | 0.25 | 4.17 | 2.07 | 1.19 | 3.89 | 2.41 | 0.30 | 0.49 | 0.21 | 0.73 | 14.58 |
| Ppyr OMLV2 PB1 | 2.73 | 0.18 | 1.37 | 1.95 | 0.73 | 1.40 | 1.78 | 0.12 | 2.30 | 1.44 | 2.01 | 8.10 |
| Ppyr OMLV2 PB2 | 1.18 | 0.10 | 1.03 | 1.81 | 1.31 | 2.34 | 2.56 | 0.34 | 2.63 | 1.81 | 2.36 | 11.88 |

| | SRR 3883761 | SRR 3883760 | SRR 3883759 | SRR 3883757 | SRR 3883756 | SRR 3883766 | SRR 2103867 | SRR 2103849 | SRR 2103848 | Ppyr_ larvae | Ppyr_ Female | Ppyr_ eggs |
|---|---|---|---|---|---|---|---|---|---|---|---|---|
| Ppyr OMLV1 HA | 0.00 | 18.29 | 0.08 | 0.21 | 23.44 | 0.00 | 0.00 | 0.00 | 0.00 | 15.89 | 74.25 | 104.49 |
| Ppyr OMLV1 NP | 0.00 | 8.37 | 0.00 | 0.09 | 16.00 | 0.00 | 0.04 | 0.00 | 0.00 | 5.62 | 45.27 | 40.24 |
| Ppyr OMLV1 PA | 0.00 | 2.81 | 0.00 | 0.05 | 12.10 | 0.00 | 0.00 | 0.00 | 0.00 | 8.63 | 25.05 | 45.85 |

*Appendix 5—table 4 continued on next page*

*Appendix 5—table 4 continued*

| | SRR 3883762 | SRR 3883763 | SRR 3883764 | SRR 3883765 | SRR 3883767 | SRR 3883768 | SRR 3883769 | SRR 3883770 | SRR 3883771 | SRR 3883758 | SRR 3883772 | SRR 3883773 |
|---|---|---|---|---|---|---|---|---|---|---|---|---|
| Ppyr OMLV1 PB1 | 70.96 | 44.97 | 17.89 | 0.00 | 0.00 | 0.00 | 0.04 | 28.83 | 0.07 | 0.00 | 9.66 | 0.04 |
| Ppyr OMLV1 PB2 | 46.51 | 15.96 | 4.06 | 0.00 | 0.00 | 0.00 | 0.00 | 15.05 | 0.05 | 0.00 | 3.91 | 0.00 |
| Ppyr OMLV2 HA | 0.00 | 0.00 | 0.00 | 8.95 | 4.94 | 1.02 | 9.74 | 9.38 | 19.30 | 0.95 | 7.68 | 0.43 |
| Ppyr OMLV2 NP | 0.00 | 0.00 | 0.00 | 8.47 | 3.00 | 1.10 | 6.07 | 12.19 | 16.09 | 0.82 | 7.34 | 0.97 |
| Ppyr OMLV2 PA | 0.00 | 0.00 | 0.00 | 1.45 | 0.97 | 1.23 | 3.09 | 4.28 | 5.73 | 0.17 | 2.10 | 0.32 |
| Ppyr OMLV2 PB1 | 0.00 | 0.00 | 0.00 | 2.68 | 1.63 | 0.34 | 1.40 | 3.27 | 3.88 | 0.11 | 1.92 | 0.63 |
| Ppyr OMLV2 PB2 | 0.00 | 0.00 | 0.00 | 1.35 | 0.96 | 0.34 | 2.08 | 4.16 | 5.88 | 0.16 | 1.90 | 1.06 |

DOI: https://doi.org/10.7554/eLife.36495.090

identify FEVEs that covered as high as ca. 60% of the corresponding gene product, and in addition, although at specific short protein regions of the putative related FOLMV, similarity values were as high as 89% pairwise identity. In addition, most of the detected FEVEs were flanked by Transposable Elements (TE) (*Appendix 5—figure 2J*) suggesting that integration followed ectopic recombination between viral RNA and transposons. We found several conserved domains associated to reverse transcriptases and integrases adjacent to the corresponding FEVEs, which supports the hypothesis that these virus-like elements could be reminiscent of an OMV-like ancestral virus that could have been integrated into the genome by occasional sequestering of viral RNAs by the TE machinery. The finding of EVEs in the *P. pyralis* genome is not trivial, OMV EVEs are extremely rare. There has been only one report of OMV like sequences integrated into animal host genomes, which is the case of *Ixodes scapularis*, the putative vector of *Quaranfil virus* and *Johnston Atoll virus* corresponding to genus *Quaranjavirus* (*Katzourakis and Gifford, 2010*). The fact that besides FEVEs, the only other OMV EVE corresponded to an Arthropod genome, given the ample studies of bird and mammal genomes, is suggestive that perhaps OMV EVEs are restricted to Arthropod hosts. Sequence similarity of FEVEs and firefly viruses suggest that these viral 'molecular fossils' could have been tightly associated to PpyrOLMV1 and 2 ancestors. Moreover, we found potential NP and PB1 EVEs in our genome of light emitting click beetle *Ignelater luminosus* (Elateridae), an evolutionary distant coleoptera. Sequence similarity levels of the corresponding EVEs averaging 52%, could not be related with evolutionary distances of the hosts. We were not able to generate conclusive phylogenetic insights of the detected EVEs, given their partial, truncated and altered nature of the virus like sequences. In specific cases such as PB1-like EVEs there appears to be a trend suggesting an indirect relation between sequence identity and evolutionary status of the firefly host, but this preceding findings should be taken cautiously until more gathered data is available. The widespread presence of DNA sequences significantly similar to OMV in the explored firefly and related genomes are an interesting and intriguing result. At this stage is prudently not to venture to suggest more likely one of the two plausible explanations of the presence of these sequences in related beetles genomes: (i) Ancestral OMV like virus sequences were retrotranscribed and incorporated to an ancient beetle, followed by speciation and eventual stabilization or lost of EVEs in diverse species. (ii) Recent and recursive integration of OMV like virus sequences in fireflies and horizontal transmission between hosts. These propositions are not mutually exclusive, and may be indistinctly applied to specific cases. Future studies should enquire in this genome dark matter to better understand this interesting phenomenon. When more data is available EVE sequences may be combined with phylogenetic data of host species to expose eventual patterns of inter-class virus transmission. Either way, more studies are needed to explore these proposals, Katzourakis and Gifford (*Katzourakis and Gifford, 2010*) suggested that EVEs could reveal novel virus diversity and indicate the likely host range of virus clades.

After identification and confirmation that firefly related EVEs are present in the host DNA genome, an obvious question follows: Are these EVEs just signatures of an evolutionary vestige of stochastic past infections; or could they be associated with an intrinsic function? It has been suggested that intensity and prevalence of infection may be a determinant of EVEs integration, and that exposure to environmental viruses may not (*Olson and Bonizzoni, 2017*). Previous reports have suggested that EVEs may firstly function as restriction factors in their hosts by conferring resistance to infection by exogenous viruses, and the eventual counter-adaptation of virus populations of EVE positive hosts, could reduce the EVE restriction mechanism to a non-functional status (*Aiewsakun and Katzourakis, 2015*). Recently, in mosquitoes, a new mechanism of antiviral immunity against RNA viruses has been proposed, relying in the production and expression of EVEs DNA (*Goic et al., 2016*). Alternatively, eventual EVE expression could lend to the production viral like truncated proteins that may compete in trans with virus proteins from infecting viruses and limit viral replication, transcription or virion assembly (*Aaskov et al., 2006*). In addition, integration and eventual modulation in the host genome may be associated with an interaction between viral RNA and the mosquito RNAi machinery (*Goic et al., 2013*). The piRNA pathway

mediates through small RNAs and Piwi-Argonaut proteins the repression of TE-derived nucleic acids based on sequence complementarity, and has also been associated to regulation of arbovirus viral-related RNA, suggesting a functional connection among resistance mechanisms against RNA viruses and TEs (*Palatini et al., 2017*; *Miesen et al., 2016*). Furthermore, arbovirus EVEs have been linked to the production of viral-derived piRNAs and virus-specific siRNA, inducing host cell immunity without limiting viral replication, supporting persistent and chronic infection (*Goic et al., 2016*). Perhaps, an EVE-dependent mechanism of modulation of virus infection could have some level of reminiscence to the paradigmatic CRISPR/Cas system which mediates bacteriophage resistance in prokaryotic hosts.

In sum, genomic studies are a great resource for the understanding of virus and host evolution. Here, we glimpsed an unexpected hidden evolutionary tale of firefly viruses and related FEVEs. Animal genomes appear to reflect as a book, with many dispersed sentences, an antique history of ancestral interaction with microbes, and EVEs functioning as virus related bookmarks. The exponential growth of genomic data would help to further understand this complex and intriguing interface, in order to advance not only in the apprehension of the phylogenomic insights of the host, but also explore a multifaceted and dynamic virome that has accompanied and even might have shifted the evolution of the host.

**Appendix 5—table 5.** FEVE hits from BLASTX of PpyrOMLV PB1.

| Scaffold | Start | End | Strand | Id with PpOMLV | E value | Coverage | FEVE |
|---|---|---|---|---|---|---|---|
| Ppyr1.2_LG1 | 12787323 | 12786796 | (-) | 56.30% | 8.22E-50 | 39.10% | EVE PB1 like-1 |
| Ppyr1.2_LG1 | 13016647 | 13016120 | (-) | 56.30% | 8.22E-50 | 39.10% | EVE PB1 like-2 |
| Ppyr1.2_LG1 | 34701480 | 34701560 | (+) | 37.00% | 2.88E-26 | 26.70% | EVE PB1 like-3 |
| Ppyr1.2_LG1 | 34701562 | 34701774 | (+) | 37.60% | 2.88E-26 | 30.20% | EVE PB1 like-3 |
| Ppyr1.2_LG1 | 34701801 | 34702214 | (+) | 45.30% | 2.88E-26 | 34.00% | EVE PB1 like-3 |
| Ppyr1.2_LG1 | 35094645 | 35095094 | (+) | 28.10% | 2.15E-10 | 9.50% | EVE PB1 like-4 |
| Ppyr1.2_LG1 | 35110084 | 35109956 | (-) | 53.50% | 2.37E-14 | 4.40% | EVE PB1 like-5 |
| Ppyr1.2_LG1 | 35110214 | 35110107 | (-) | 75.00% | 2.37E-14 | 14.70% | EVE PB1 like-5 |
| Ppyr1.2_LG1 | 35110347 | 35110213 | (-) | 42.60% | 2.37E-14 | 2.90% | EVE PB1 like-5 |
| Ppyr1.2_LG1 | 50031464 | 50031330 | (-) | 64.40% | 1.18E-09 | 10.00% | EVE PB1 like-6 |
| Ppyr1.2_LG1 | 50031498 | 50031457 | (-) | 71.40% | 1.18E-09 | 11.60% | EVE PB1 like-6 |
| Ppyr1.2_LG1 | 50613130 | 50612921 | (+) | 49.40% | 3.71E-11 | 4.90% | EVE PB1 like-7 |
| Ppyr1.2_LG1 | 50673211 | 50673621 | (+) | 38.50% | 1.03E-12 | 9.70% | EVE PB1 like-8 |
| Ppyr1.2_LG1 | 51208464 | 51207634 | (-) | 77.20% | 0 | 56.40% | EVE PB1 like-9 |
| Ppyr1.2_LG1 | 51209399 | 51208467 | (-) | 68.50% | 0 | 53.60% | EVE PB1 like-9 |
| Ppyr1.2_LG1 | 51209556 | 51209398 | (-) | 71.70% | 0 | 39.20% | EVE PB1 like-9 |
| Ppyr1.2_LG1 | 61871682 | 61872158 | (+) | 31.10% | 2.84E-23 | 36.00% | EVE PB1 like-10 |
| Ppyr1.2_LG1 | 61872158 | 61872319 | (+) | 46.30% | 2.84E-23 | 28.30% | EVE PB1 like-10 |
| Ppyr1.2_LG1 | 61872355 | 61872456 | (+) | 41.20% | 2.84E-23 | 27.00% | EVE PB1 like-10 |
| Ppyr1.2_LG1 | 61930528 | 61930205 | (-) | 38.00% | 3.58E-27 | 30.90% | EVE PB1 like-11 |
| Ppyr1.2_LG1 | 61930686 | 61930504 | (-) | 63.60% | 3.58E-27 | 35.90% | EVE PB1 like-11 |
| Ppyr1.2_LG1 | 68038999 | 68039073 | (+) | 60.00% | 7.73E-12 | 6.60% | EVE PB1 like-12 |
| Ppyr1.2_LG1 | 68039072 | 68039314 | (+) | 40.70% | 7.73E-12 | 5.00% | EVE PB1 like-12 |
| Ppyr1.2_LG1 | 68039289 | 68039330 | (+) | 64.30% | 7.73E-12 | 8.00% | EVE PB1 like-12 |
| Ppyr1.2_LG1 | 68128820 | 68129008 | (+) | 51.50% | 1.89E-06 | 4.90% | EVE PB1 like-13 |
| Ppyr1.2_LG2 | 34545814 | 34545680 | (-) | 58.70% | 3.84E-06 | 7.20% | EVE PB1 like-14 |
| Ppyr1.2_LG2 | 34546169 | 34545801 | (-) | 52.80% | 1.16E-31 | 34.10% | EVE PB1 like-14 |

DOI: https://doi.org/10.7554/eLife.36495.091

**Appendix 5—table 6.** FEVE hits from BLASTX of PpyrOMLV PB2.

| Scaffold | Start | End | Strand | Id with PpOMLV | E value | Coverage | FEVE |
|---|---|---|---|---|---|---|---|
| Ppyr1.2_LG1 | 50313869 | 50314219 | (+) | 82.10% | 6.91E-54 | 48.30% | EVE PB2 like-1 |
| Ppyr1.2_LG1 | 50314216 | 50315016 | (+) | 82.40% | 1.92E-142 | 57.90% | EVE PB2 like-1 |
| Ppyr1.2_LG1 | 50315772 | 50315002 | (-) | 89.10% | 9.97E-145 | 60.60% | EVE PB2 like-1 |
| Ppyr1.2_LG1 | 58707403 | 58706942 | (-) | 52.60% | 6.19E-42 | 35.80% | EVE PB2 like-2 |

DOI: https://doi.org/10.7554/eLife.36495.092

**Appendix 5—table 7.** FEVE hits from BLASTX of PpyrOMLV PA.

| Scaffold | Start | End | Strand | Id with PpOMLV | E value | Coverage | FEVE |
|---|---|---|---|---|---|---|---|
| Ppyr1.2_LG1 | 34977392 | 34977231 | (-) | 48.10% | 7.73E-07 | 3.50% | EVE PA like-1 |
| Ppyr1.2_LG1 | 62052289 | 62052023 | (-) | 28.70% | 8.92E-11 | 7.10% | EVE PA like-2 |
| Ppyr1.2_LG1 | 62117077 | 62116811 | (-) | 28.70% | 1.22E-10 | 7.10% | EVE PA like-3 |
| Ppyr1.2_LG1 | 62117493 | 62117101 | (-) | 26.30% | 1.22E-10 | 8.60% | EVE PA like-3 |
| Ppyr1.2_LG1 | 68122348 | 68122440 | (+) | 77.40% | 3.40E-06 | 15.70% | EVE PA like-4 |

DOI: https://doi.org/10.7554/eLife.36495.093

**Appendix 5—table 8.** FEVE hits from BLASTX of PpyrOMLV NP

| Scaffold | Start | End | Strand | Id with PpOMLV | E value | Coverage | FEVE |
|---|---|---|---|---|---|---|---|
| Ppyr1.2_LG1 | 181303 | 181404 | (+) | 79.40% | 7.01E-09 | 17.90% | EVE NP like-1 |
| Ppyr1.2_LG1 | 1029425 | 1029568 | (+) | 93.80% | 9.59E-21 | 27.40% | EVE NP like-2 |
| Ppyr1.2_LG1 | 2027860 | 2027438 | (-) | 35.50% | 3.00E-21 | 30.80% | EVE NP like-3 |
| Ppyr1.2_LG1 | 36568324 | 36568551 | (+) | 42.10% | 8.99E-11 | 7.20% | EVE NP like-4 |
| Ppyr1.2_LG1 | 52877256 | 52877086 | (-) | 68.40% | 3.87E-15 | 14.60% | EVE NP like-5 |
| Ppyr1.2_LG1 | 59927414 | 59927271 | (+) | 93.80% | 5.60E-20 | 26.40% | EVE NP like-6 |
| Ppyr1.2_LG3 | 17204346 | 17204122 | (-) | 46.70% | 7.60E-13 | 7.10% | EVE NP like-7 |
| Ppyr1.2_LG3 | 31635344 | 31635030 | (-) | 35.80% | 3.30E-08 | 10.00% | EVE NP like-8 |
| Ppyr1.2_LG3 | 50175821 | 50175922 | (+) | 79.40% | 7.01E-09 | 17.90% | EVE NP like-9 |
| Ppyr1.2_LG4 | 27811681 | 27811758 | (+) | 38.50% | 3.22E-13 | 2.50% | EVE NP like-10 |
| Ppyr1.2_LG4 | 27811853 | 27812179 | (+) | 39.00% | 3.22E-13 | 10.90% | EVE NP like-10 |

DOI: https://doi.org/10.7554/eLife.36495.094

## Appendix 6

DOI: https://doi.org/10.7554/eLife.36495.095

## Data availability

### 6.1 Files on FigShare

1. *Photinus pyralis* sighting records (Excel spreadsheet) - (10.6084/m9.figshare.5688826)
2. Ilumi1.0 Blobtools results - (10.6084/m9.figshare.5688952)
3. Alat1.2 Blobtools results - (10.6084/m9.figshare.5688928)
4. Ppyr1.2 Blobtools results - (10.6084/m9.figshare.5688982)
5. Protein multiple sequence alignment for P450 tree - *Appendix 1—figure 13* - (10.6084/m9.figshare.5697643)
6. Photinus pyralis orthomyxo-like virus 1 sequence and annotation - (10.6084/m9.figshare.5714806)
7. Photinus pyralis orthomyxo-like virus 2 sequence and annotation - (10.6084/m9.figshare.5714812)
8. OrthoFinder protein clustering analysis (Orthogroups) - (10.6084/m9.figshare.5715136)
9. PPYR_OGS1.1 kallisto RNA-Seq expression quantification (TPM) - (10.6084/m9.figshare.5715139)
10. AQULA_OGS1.0 kallisto RNA-Seq expression quantification (TPM) - (10.6084/m9.figshare.5715142)
11. *Figure 5*. PPYR_OGS1.1+AQULA_OGS1.0 Sleuth/differential expression Venn diagram analysis (BSN-TPM) - (10.6084/m9.figshare.5715151)
12. Ilumi_OGS1.2 kallisto RNA-Seq expression quantification (TPM) - (10.6084/m9.figshare.5715157)
13. *Appendix 4—figure 2*: DNA and tRNA methyltransferase gene phylogeny - (10.6084/m9.figshare.6531311)
14. *Appendix 4—figure 6* Preliminary maximum likelihood phylogeny of luciferase homologs - (10.6084/m9.figshare.6687086)
15. *Appendix 4—figure 9A* Opsin gene tree - (10.6084/m9.figshare.5723005)
16. Testing for ancestral selection of elaterid ancestral luciferase (*Figure 4B*): MEME selected site analysis - (10.6084/m9.figshare.6626651)
17. Testing for ancestral selection of elaterid ancestral luciferase (*Figure 4B*): PAML-BEB selected site analysis - (10.6084/m9.figshare.6725081)

### 6.2 Files on www.fireflybase.org/www.github.org

#### 6.2.1 *Photinus pyralis* genome and associated files

- Ppyr1.3 genome assembly - (http://www.fireflybase.org/firefly_data/Ppyr1.3.fasta.zip)
- *P. pyralis* Official Geneset (OGS) GFF3 files - (https://github.com/photocyte/PPYR_OGS; copy archived at https://github.com/elifesciences-publications/PPYR_OGS)
  - Official geneset gene-span nucleotide FASTA files
  - Official geneset mRNA nucleotide FASTA files
  - Official geneset CDS nucleotide FASTA files
  - Official geneset peptide FASTA files
- Supporting Non-OGS files - (https://github.com/photocyte/PPYR_OGS/tree/master/Supporting_non-OGS_data)
  - Trinity/PASA direct coding gene models (DCGM) GFF3 file
    - DCGM CDS FASTA file
    - DCGM peptide FASTA file
  - Stringtie stranded direct coding gene model (DCGM) GFF3 file
    - DCGM CDS FASTA file
    - DCGM peptide FASTA file
  - Stringtie unstranded direct coding gene model (DCGM) GFF3 file
    - DCGM CDS FASTA file

  - ▪ DCGM peptide FASTA file
  - ○ Expression quantification (TPM)
  - ○ InterProScan OGS functional annotation
  - ○ PTS1 OGS annotation
  - ○ Gaps GFF3 file
  - ○ Repeat library FASTA and aligned GFF3 file.
  - ○ Ab-initio gene models

### 6.2.2 *Aquatica lateralis* genome and associated files

- Alat1.3 genome assembly - (http://www.fireflybase.org/firefly_data/Alat1.3.fasta.zip)
- *A. lateralis* Official Geneset (OGS) GFF3 files - (https://github.com/photocyte/AQULA_OGS; copy archived at https://github.com/elifesciences-publications/AQULA_OGS)
  - ○ Official geneset gene-span nucleotide FASTA files
  - ○ Official geneset mRNA nucleotide FASTA files
  - ○ Official geneset CDS nucleotide FASTA files
  - ○ Official geneset peptide FASTA files
- Supporting Non-OGS files - (https://github.com/photocyte/AQULA_OGS/tree/master/Supporting_non-OGS_data)
  - ○ Trinity/PASA direct coding gene models (DCGM) GFF3 file
    - ▪ DCGM CDS FASTA file
    - ▪ DCGM peptide FASTA file
  - ○ Stringtie unstranded direct coding gene model (DCGM) GFF3 file
    - ▪ DCGM CDS FASTA file
    - ▪ DCGM peptide FASTA file
  - ○ Expression quantification (TPM)
  - ○ InterProScan OGS functional annotation
  - ○ PTS1 OGS annotation
  - ○ Gaps GFF3 file
  - ○ Repeat library FASTA and aligned GFF3 file.

### 6.2.3 *Ignelater luminosus* genome and associated files

- Ilumi1.2 genome assembly - (http://www.fireflybase.org/firefly_data/Ilumi1.2.fasta.zip)
- *I. luminosus* Official Geneset (OGS) GFF3 files - (https://github.com/photocyte/ILUMI_OGS; copy archived at https://github.com/elifesciences-publications/ILUMI_OGS)
  - ○ Official geneset gene-span nucleotide FASTA files
  - ○ Official geneset mRNA nucleotide FASTA files
  - ○ Official geneset CDS nucleotide FASTA files
  - ○ Official geneset peptide FASTA files
- Supporting Non-OGS files - (https://github.com/photocyte/ILUMI_OGS/tree/master/Supporting_non-OGS_data)
  - ○ Trinity/PASA direct coding gene models (DCGM) GFF3 file
    - ▪ DCGM CDS FASTA file
    - ▪ DCGM peptide FASTA file
  - ○ Stringtie unstranded direct coding gene model (DCGM) GFF3 file
    - ▪ DCGM CDS FASTA file
    - ▪ DCGM peptide FASTA file
  - ○ Expression quantification (TPM)
  - ○ ○ InterProScan OGS functional annotation
  - ○ ○ PTS1 OGS annotation
  - ○ Gaps GFF3 file
  - ○ Repeat library FASTA and aligned GFF3 file.
  - ○ Ab-initio gene models

### 6.3 Tracks on www.fireflybase.org JBrowse (*Skinner et al., 2009*) genome browser

For each genome:

1. Gaps

2. Repeats
3. Direct gene-models (Stringtie)
4. Direct gene-models (Trinity)
5. Official geneset gene-models

