## [Decision Letter]

Thank you for submitting your article "Firefly genomes illuminate parallel origins of bioluminescence in beetles" for consideration by *eLife*. Your article has been reviewed by four peer reviewers, one of whom is a member of our Board of Reviewing Editors, and the evaluation has been overseen by Diethard Tautz as the Senior Editor. The reviewers have opted to remain anonymous.

The reviewers have discussed the reviews with one another and the Reviewing Editor has drafted this decision to help you prepare a revised submission.

Summary:

General assessment:

The reviewers find the results presented in this manuscript compelling and generally comprehensively supported. The study's wide-ranging exploration of the evolution of bioluminescence within beetles brings together newly generated genomic, transcriptomic, and mass-spectrometry data for two firefly species and a click beetle. A key asset of this work is the integrated approach to exploring this evolutionary novelty, including discussion of which biochemical components may have been ancestral and predated independent acquisition of the associated traits required for the full manifestation of beetle bioluminescence.

Central conclusions:

1) By comparing the genomic regions that harbour the luciferase genes and reconstructing the full gene tree of the luciferases and homologous peroxisomal and non-peroxisomal fatty acyl-CoA synthetases (PACSs and ACSs) they present evidence that the firefly luciferases originated from an ancient duplication and subsequent neofunctionalisation of a PACS gene followed by a duplication of this ancestral luciferase to produce Luc1 (which remained in the same genomic neighbourhood) and Luc2 (which are elsewhere in the genomes). These analyses further suggest that the click-beetle luciferase arose from an independent duplication and subsequent neofunctionalisation of a PACS gene.

2) Complementary evolutionary analyses incorporating luciferases from several other species to perform ancestral state reconstructions further support the conclusion that the firefly and click-beetle luciferases arose through independent gene duplication events. Furthermore, molecular adaptation analysis identified a 'burst' of diversification along the branch leading to the click-beetle luciferases, consistent with neofunctionalisation of an ancestral PACS gene to acquire luciferase activity.

3) Transcriptomic analyses in the fireflies identified putative enzymes involved in the metabolism required to support bioluminescence, as well as non-enzymes that may also be important.

4) Assessing presence of lucibufagins (unpalatable defence steroids) in the three species suggests that they are only present in *P. pyralis*. The authors hypothesize that an expansion through multiple gene duplications of a cytochrome P450 gene (CYP303) in *P. pyralis*, but which remains as a single-copy gene in other species, may be linked to lucibufagin metabolism. This interestingly suggests that producing lucibufagins as a defence could in fact be a derived feature in Lampyrinae.

5) Overall, the conclusion that bioluminescence was independently acquired in the firefly and click beetle lineages is convincingly supported by phylogenetic, syntenic, and previous (anatomical) results.

6) Characterisation of the holobiomes of the three species and identified putative symbionts, a parasitoid fly, as well as bacteria and viruses.

Essential revisions:

1) Orthology analyses

Concerns were raised regarding the possibility that gene sets were not first filtered to select one protein per gene. If this is the case then these analyses will need to be performed on filtered gene sets and all downstream analyses that employ the orthology results will need to be updated, with conclusions revised accordingly.

2) Quality assessments

The assessments are comprehensive, although a few details need to be checked (see below), but concluding that all three genomes and the annotations are 'high-quality' is misleading. The best assembly (*P. pyralis*) likely contains non-collapsed haplotypes while *I. luminosus* remains fragmented (~90K scaffolds), and the annotated gene sets appear to be less complete than the assemblies suggest they should be. Reassurances of manual curation supporting analyses of specific gene families (e.g. the ACTSs, PACSs, P450s and DNMTs) should therefore be clearly presented.

3) Molecular adaptation analyses

Several points were raised regarding these analyses and the results (see below for details). (i) presenting results only for the Elateridae leads to the assumption that no positive results were found for the Lampyridae, but this is not discussed at all. (ii) reporting of some of the results leads to some confusion, e.g. the employed p-value cut-off and very high omega values. The results also leave at least two of the reviewers wondering about whether any specific sequence changes could be identified that might be linked to acquiring luciferase activity.

4) The 'contentious' hypothesis

Given that several recent studies are cited that have already lent support to the hypothesis on the independent origins of bioluminescence in beetles, including a 2016 paper from the same first author, emphasising the claimed 'contention' is unnecessary.

5) Repeat annotation

The main text only discusses repeats in *P. pyralis*, but does not consider if the repeat content could have been inflated by the inclusion of haplotypes in the assembly. A complex repeat in *I. luminosus* is discussed in the supplement, but how it relates to reconstructing the luciferase/ACSs/PACSs locus is ignored in the main text. It remains unclear whether the repeat annotations offer any insights into the rearrangement histories of these loci in the three beetle genomes.

The full reviews are appended below but you need only consider the summary and essential points above.

*Reviewer #1:*

Fallon et al. present their findings from comparative genomic analyses that focus on elucidating the origins of bioluminescence in beetles. They generate genome assemblies with hybrid approaches using multiple technologies for two fireflies and one click-beetle: a high-quality assembly with linkage group assignments for *P. pyralis*, a good-quality assembly for *A. lateralis*, and a fragmented but still relatively decent assembly for the click beetle, *I. luminosus*.

1) The fact that, for all three beetle species, the BUSCO completeness scores of the annotated gene sets are lower than for the assembly assessments suggests that despite the seemingly well-designed annotation strategies, they have failed to produce the best-quality automated annotations for these assemblies. As annotation pipelines employ a lot more evidence than BUSCO (which just uses profiles with Augustus), they generally produce gene set annotations that score better, or as good as, assessing the assemblies directly e.g. in Appendix 4—table 3 Dmela from 99.4% to 99.8%, Tcast from 98.4% to 99.0%. Having said that, the gene set completeness is still fairly good, and the main focus of the study is only on a limited set of genes, where manual curation was performed, so this should not substantially affect the main conclusions but should probably nonetheless be noted.

2) Labelling all three as 'high-quality genomes' is perhaps a bit of a stretch.

3) The very high levels of duplicated BUSCOs in the gene sets of Dmela and a Tcast in Appendix 4—table 3 seem to suggest that alternative transcripts were not first filtered from the gene sets. To obtain reliable estimates of real gene duplications the annotations must be first filtered to select one protein representative per gene. The other species may be less affected by this as they probably do not have many alternative transcripts annotated (a problem also for orthology analysis, see below). This needs to be fixed to ensure that conclusions about levels of gene duplication are based on sound and like-for-like analyses.

4) Figure 2F. The BUSCO C and D percentages do not seem to match those presented in the supplement, 93.7% vs. 94.8% and 1.3% vs. 1.4%.

5) The numbers presented as part of the orthology analyses are confusing. For example, Total numbers per species in Figure 2E do not match those reported in the text (numbers in the text are much higher). This might be explained if the figure only shows clustered genes (rather than all genes), but if so then this is not very well explained and hence the confusion. Much more worrying is the total gene count in Figure 2E for *Drosophila melanogaster*, 16991 genes – how can this possibly be when the current annotation at FlyBase has only 13931 protein-coding genes? Appendix 4—figure 8 gives a clue as to how this could be, as it shows 8 isoforms (alternative transcripts) of a single *D. melanogaster* gene. This would suggest that orthology clustering did not first select one protein per gene, a procedure that is more-or-less standard for most orthology delineation approaches. I am not familiar with the inner-workings of OrthoFinder, so perhaps the already-delineated clusters can be filtered to remove alternative transcripts, but I suspect it would be more rigorous to re-do the clustering using pre-filtered gene sets.

6) Figure 3D. I could not find mention of the sizes of the regions not shown for Ppyr1.3 LG1. The first one you can work out must span from 28400 to 30700 (so 2300), the second from after 30800 to before 67450 (so a bit less than 36650). Given the size, especially of the second one, it would be fairer to the reader to explicitly label these, and probably also to enumerate the number of genes in these regions.

7) Figure 3. The density of the figure makes it rather hard to figure out, but could it be that the orientation of PpyrMGST is inverted in 3B compared to 3D? Also, could there be a PACS missing in 3B for *A. lateralis*?

8) Figure 4B. I must preface these next few comments with the fact that I am familiar with PAML's codeml approaches but *not* with HyPhy's aBSREL approach. I tried to read up on aBSREL but could not easily find good documentation on how to interpret the results. The main reason I was confused is that I don't know how to interpret an omega value of 147.67 labelled on the branch to LlumLuc+PjanLuc [after writing that and then looking at the actual aBSREL tree I realised that this label is in fact referring to the blue star branch – so if I misread that then others might too]. Although this node is not the main result, such a high omega value seems strange (if indeed omega represents dN/dS here) as it implies estimates along that branch of ~150 times more non-synonymous changes than synonymous changes. I'm guessing that with few sites (1.53%) it can send the ratio sky-high. In which case, is it even meaningful to report this value on the figure? Indeed it seems aBSREL results on github and in the aBSREL paper they usually show nothing higher than omega=10.

9) While familiarising myself with aBSREL I noted in the 2015 paper that it states "over 80% of branches in typical gene phylogenies can be adequately modeled with a single ω ratio model". The aBSREL analysis of the Elateridae luciferases and PACSs identified 20 branches with two rate classes and 17 with one – so is this gene tree particularly atypical? Or is there something else happening here?

10) Figure 4B. A second question that immediately sprang to mind – how do these results compare with similar tests using PAML instead? For example, PMID:28094282 reports results from both approaches for neofunctionalisation of Caf1-55 after duplication. Confirmation of this kind, even just in the supplement, would help to reassure those, like me, who are still relatively new to aBSREL. I was unable to access the files on FigShare so I could not take a look at the data myself – since writing that I now have access. This however added to my confusion somewhat: Figure 4 legend states that there were 3 branches with significant (p<0.001) evidence of positive selection, but the aBSREL_stout_results.txt file reports only *one* (PangLucV) with p<0.001.

11) Another question that Figure 4 raises is why the molecular adaptation analysis was only performed on the Elateridae (plus PpyrLuc1 as outgroup)? Does it mean that the rest of the tree showed no signals of episodic positive selection, or that this was not examined? Or was there some technical reason why this was not performed, e.g. too many branches to test? Even if the tests fail to identify any significant signals of episodic positive selection, would it not be intriguing to examine amino acids in the alignment that are common to all/most luciferases but not amongst the PACSs/ACSs – i.e. have the two neofunctionalisation events converged on a common or slightly different molecular solution to luciferase activity? What do residues previously linked to the active site or cleft look like? Do site tests (rather than branch tests) pick up any of these potentially interesting amino acid positions?

12) Definition of subfunctionalisation. I am personally not overly put off by the use of subfunctionalisation to describe the differentiation of firefly Luc1 and Luc2. Nevertheless, in some circles (and perhaps more so when discussing enzymes in particular) subfunctionalisation would more strictly refer to an ancestral function being partitioned between the duplicates – i.e. the ancestral function can now only be performed through both extant copies working to together. In this case the ancestral (molecular) function is luciferase activity, and the (molecular) functions of Luc1 and Luc2 are still both luciferase activity. What distinguishes them is where and when they are expressed – i.e. divergence of function between the two copies is in terms of expression rather than molecular function. Clarifying this, or avoiding using subfunctionalisation at all, would therefore perhaps be advisable.

13) Long-read sequence assembly of the firefly *Pyrocoelia pectoralis* genome. Fu et al., 2017. This very recent data publication offers the opportunity to at least examine the Luc1 locus in a second Lampyridae.

The level of detail provided in the supplement is overwhelming for a reviewer (I did not read all 140 pages), but at the same time it is very much appreciated as it clearly documents detailed background information and methods. I wish more manuscripts were this well supported with such detailed supplementary materials. Subsection “Opsin analysis” still has MS Word comments.

Finally, the characterisation the holobiomes of the three species and identify putative symbionts, a parasitoid fly, as well as bacteria and viruses: this is encouraging to see, as (a) it suggests contaminant screening was comprehensive, and (b) 'contaminants' can actually be very interesting!

*Reviewer #2:*

In this manuscript, Fallon et al. explore the evolution of bioluminescence within the beetles, focusing on newly generated genomic and transcriptomic data for two distantly related species of firefly in comparison with a bioluminescent click beetle. This work is very comprehensive, incorporating a variety of sequencing methods to generate decent to high quality genome assemblies, supported by empirical data ranging from tissue-specific RNA-seq and bisulfite sequencing to mass-spec and holobiont analyses. The conclusion that bioluminescence was independently acquired in the firefly and click beetle lineages is convincingly supported by phylogenetic, syntenic, and previous (anatomical) results. A key asset of this work is the integrated approach to exploring this evolutionary novelty, including discussion of which biochemical components may have been ancestral and predated independent acquisition of the associated subtraits required for the full manifestation of beetle bioluminescence. I also appreciated the efforts taken in assessing synteny across fairly old lineages that have experienced a fair amount of gene shuffling. The extensive supplementary reports and files provide detailed documentation of the work. To strengthen this manuscript for publication, there are a few places where the analyses could be rounded out (particularly across the three species, to make the text less *P. pyralis*-centric), where the presentation of the material ought to be polished for clarity and to best reach a broader readership, and where a few (apparent) contradictions need to be reconciled.

Specific rounding-out analyses:

1) In support of the evolutionary scenario presented in Figure 3B and in light of the rather high repetitive content in *P. pyralis* (42.6%), do the authors detect any harbinger repetitive or transposable element sequences flanking either Luc1 or Luc2 in either *P. pyralis* or *A. lateralis*? Here and elsewhere, please make cross-species assessments more accessible. While total repetitive content for *P. pyralis* is reported in the main text, it appears that the only way to determine this for *A. lateralis* and *I. luminosus* is by manually summing up the final columns of Appendix 2—table 2 and Appendix 3—table 3 deep within the supplement.

2) As the repetitive content of *P. pyralis* may be inflated due to heterozygosity (suggested by the BUSCO duplicates), is it possible to add a filtering step or otherwise take levels of heterozygosity into account so as to make a more accurate estimation of repeat load?

3) Given the branch lengths in Figure 3C, is Luc2 more slowly evolving/ conserved than Luc1? Please perform the same molecular adaptation analysis on firefly Luc1 and Luc2 as presented in Figure 4B for the click beetle lineage Luc genes.

4) Given the care with which P450 genes were identified and curated in *P. pyralis*, please provide documentation that ensures the same level of care was used in assessing P450s in the other two species, which is particularly important (a) to support the claim of a *P. pyralis*-specific expansion of the CYP303 family and (b) in light of the low-quality assembly of *I. luminosus*. Please also ensure that the relevant supplementary sections are indeed cited in the main text section on P450s. Lastly, how were pseudogenes determined?

5) Regarding the enzymatic (not "enzymological", Introduction, last paragraph) basis of bioluminescence, highly efficient enzymes with high protein stability need not also be highly expressed at the transcript level. The analysis presented in Figure 6 fruitfully considers four selection criteria for candidate genes, but the other three criteria alone, without high expression, are already informative. Removing the high expression criterion would ~double the number of potential candidate genes. Do the authors find any important candidates among this set? Although two expected candidate genes are confirmed in the current analysis, take care to avoid the impression of cherry picking in data presentation. Two genes comprise a rather small litmus test. Were other expected candidates not found?

6) In terms of both phylogeny and synteny, I do not find the designated Clade C (Figure 3C-D) well motivated as a distinct clade.

Aspects to restructure or clarify:

1) It is counterproductive and unnecessary to oversell the text. Citing Darwin three times throughout the manuscript (Introduction and Discussion) to highlight the historical interest in and speculation on independent origins of bioluminescence in beetles belies the unsupported claim in the Abstract that this is a "contentious" hypothesis. Indeed, several recent studies are cited that have already lent support to this hypothesis, including a 2016 paper from the same first author. Similarly, the superlative claims in the first paragraph about the interest and importance of firefly bioluminescence are doubtful in comparison with GFP from jellyfish. A more consistent and milder tone would be helpful. With the data, the *P. pyralis* genome assembly is indeed high quality, but while the *A. lateralis* assembly is good, it is not as good and certainly not on par with that of *Tribolium castaneum* (subsection “Sequencing and assembly yield high-quality genomes”, second paragraph, based on scaffold number and NG50 statistics in Figure 2F). The orthogroup analysis in Figure 1E does not show a striking difference in number of shared OGs among the two fireflies compared to among all three species or pairwise between either firefly and the click beetle, although there is a high number of click beetle-specific OGs.

2) Although individual figure panels are generally quite nice, the organization of the figures is incredibly poor. As near as I can see, the main text refers to material in the sequence 1A, C, B, 2F, E, A, B, D, C, 3C, A, D, B, 4, 6 before 5, then 5B, A, C, D. Reorder appropriately, and consider whether 3A truly is germane to that section of the manuscript. Similarly, the care in depicting the PTS1 motif in Figure 3 is only finally explained much later in the main text (subsection “Metabolic adaptation of the firefly lantern”, second paragraph). I also found the figures very dense, visually, with insufficient white space between panels and with some elements too small even when viewed digitally at >2x print size. For example, the arrowhead to the click beetle organs in 1C is too small and by physically touching what it is pointing to obscures rather than highlights. It took several passes before I found the Ppyr LG1 label above the schematic in 3D, and I initially mistook the quotation marks in the Figure 6 table to mean omissions rather than "ditto".

3) I found the extent to which the reader is continually referred to the supplement, without *any* main text Materials and methods section, very frustrating. Judicious inclusion in the main text of key specifics would be helpful, including: specifying that the sex determination system is XO (subsection “Sequencing and assembly yield high-quality genomes”, third paragraph), that the "targeted molecular evolution analysis" uses a specific, citable pipeline (Mesquite, suppl. reference 213, for main text subsection “Independent origins of firefly and click beetle luciferase”, last paragraph), whether the entire protein (or gene locus?) is considered in the molecular adaptation analysis, which non-luminescent tissues were used (subsection “Metabolic adaptation of the firefly lantern”, first paragraph), which BUSCO version/ taxonomic group/ number of genes was used (Figure 2 legend), etc.

4) Although extensive and well structured, please polish the supplement. For example, remove lingering track changes/comments, cross-reference across sections as appropriate (e.g., for P450s per species and in comparative analyses), and ensure that information is clearly presented. For example, the main text is vague on the nature of the non-bioluminescent tissues used for the DE analysis, yet the list of libraries in Appendix 1—table 1 is based on opaque internal identifiers ("OAG"?) that make it difficult to ascertain either tissue type or number of biological replicates. Similarly, although I appreciate the detail presented on gene structure in Appendix 4—figure 4, *'s or other annotations to highlight identities would make the alignment easier to assess.

5) Throughout, consider a broad readership and avoid ambiguous jargon. I am still uncertain how "evolutionary events" fit into the Hi-C "long-range" analysis (subsection “Sequencing and assembly yield high-quality genomes”). The intended meanings of the term "promiscuous" (Discussion) are unclear. If a single enzyme can process multiple substrates (functionally promiscuous), how does this equate with a "need" for duplication and subfunctionalization for high specificity? That there are multiple synthesis possibilities to produce luciferin is not "promiscuous". In the legend for Figure 3, "Color gradients indicate the TPM values of whole body" is not nearly as accessible to a non-bioinformatician interested in beetle evolution as "Color gradients denote gene expression levels (TPM) from whole-body samples…". And similar instances (e.g., define BSN-TPM in the legend to Figure 6)…

6) Although the first Results section is appropriately concise as a foundation for the subsequent sections, the paragraph breaks and text flow within this wide-ranging section are difficult to follow and would benefit from smoothing and textual transitions.

7) Please make clear whether the genome assembly or OGS was used for specific analyses (e.g., subsection “The genomic context of firefly luciferase evolution”, first and second paragraphs). Was there really a de novo scan of the assembly itself?

8) Not all PACS/ ACS genes included in the phylogeny in 3C are shown in 3D. Please comment on the gene nomenclature and numbering conventions used for the taxa presented here, and whether genes not depicted in 3D are scattered in the genome.

9) The comment about potential lysosomes and the biological significance of an opsin gene being expressed within the light organ itself are unclear (“Metabolic adaptation of the firefly lantern”, last paragraph).

10) New genomes are continuously forthcoming. Provide a reference for the 62 referred to in the last paragraph of the subsection “Genomic insights into firefly chemical defense”.

11) It would be interesting and helpful for the authors to slightly expand the discussion on the possible link between bioluminescence and lucibufagin evolution. Is this a belt and braces situation of being doubly sure between CYP303 expansion and the whole body glow afforded by Luc2?

12) *T. castaneum* is a long established insect species for genetics and genomics and would be the more taxonomically appropriate outgroup for the four-species orthogroup analysis shown in main text Figure 2E.

Issues to reconcile:

1) Given the availability of fresh *A. lateralis* material for applications such as RNA-seq, I find it weak that the flow cytometry measurements for this species are based on only a single biological specimen.

2) If PACS genes gave rise to firefly luciferase genes (Figure 3B), why is the opposite evolutionary direction assumed for IlumLuc (subsection “Independent origins of firefly and click beetle luciferase”, first paragraph)?

3) What is the evidence that IlumLuc is highly expressed in the abdominal lanterns (subsection “Independent origins of firefly and click beetle luciferase”, first paragraph) if the RNA-seq sample is only based on the prothorax (legend for Figure 3C)?

I provided extensive, specific details above, but an overall concern is presentation, especially figures that are too dense and with poorly ordered and possibly irrelevant material.

*Reviewer #3:*

The manuscript, entitled "Firefly genomes illuminate parallel origins of bioluminescence in beetles," presents a comparative genomics analysis (with supporting data) of the evolution of bioluminescence in beetles. Focal firefly taxa in two lineages (Lampyridae subfamilies Lampyrinae and Luciolinae) are examined, as well as a bioluminescent click beetle (family Elateridae). The analysis is novel in providing strong genetic evidence supporting the independent evolution of bioluminescent light organs in Lampyrid and Elaterid beetles, as well as new insights into the origin of luciferase enzymes and the light organ itself. High quality draft genomes are provided, as well as supporting data (chromatin analysis, transcriptomic data, structural comparisons), to underpin the manuscript. Additionally, the authors investigate aspects of chemical defense in these beetles related to the lucibufagin steroid family, using mass spec of different tissues and life stages. The data suggests that only some lineages possess this defense chemistry, and the biosynthesis may be generated using a unique subfamily of cytochrome P450 genes. This P450 group (CYP303) is uniquely expanded in the Lampyrinae firefly subfamily. Finally, the authors examine evidence for symbionts, finding a Tenericutes mollicute might play an important role in firefly metabolism.

The manuscript is well written and carefully composed, with strong supporting evidence for the majority of analyses. It addresses a fascinating question, the molecular and evolutionary origin of bioluminescence, while also developing important resources for comparative genomics (that will likely benefit a broader research community). In addition, the supporting experiments involving mass spec, RNAseq, and chromatin analysis provide a richer foundation of knowledge than many genome papers achieve. My general view is that this is a significant publication that merits acceptance in *eLife*.

Some (hopefully constructive) criticisms are as follows.

Different genome sequencing methods were used to construct the data for this project, resulting in unequal assembly quality. This possibly impacts the analysis of parallel evolution in the luciferase homology gene family (based on protein blast), as incorrectly assembled or missing data might bias results in the more limited reference genome of the elaterid. An alternative approach to searching for these genes might be undertaken to help confirm these results, i.e. PCR or the use of ATRAM using the raw reads (https://bmcbioinformatics.biomedcentral.com/articles/10.1186/s12859-015-0515-2)

Some aspects of the figures seem uninformative (at least to the content of this paper). In Figure 1, the geographical distribution and sampling of *P. pyralis* is shown. This could be moved to the supplement, as it has minimal bearing on the paper as a whole. In Figure 2, much of the content seems better for supplement. I liked Figure 3 (very dense information though), and also wondered if a model of the evolution of the lantern from a peroxisome might want a cartoon figure.

Figure 4B. The large values of omega emerge as a result of a lack of synonymous sites (no synonymous sites lead omega to be infinity). Y. Ziheng suggests reporting likelihood ratio test statistics as being more informative.

*Reviewer #4:*

This study analyses the genomes of two fireflies and a click beetle to investigate the origins of bioluminescence. The authors offer convincing evidence for two independent origins thus showing that bioluminescence most likely evolved convergently in fireflies and click beetles. This manuscript is well written and the study provides many novel and interesting findings. I have a few specific comments.

Subsection “Independent origins of firefly and click beetle luciferase”, last paragraph and Figure 4B: Why is the analysis of molecular evolution only carried out on click beetle luciferase and not also on the fireflies?

Furthermore, is it possible to identify the specific adaptive mutations? Are homologous positions evolving adaptively in both suspected origins of luciferase or at least within the same domain/structure? Structuring of supplementary information and its referencing within the manual need to be improved.

Overall, I was very pleased with the quality of figures and the density of information they convey. However, several points concerning their arrangement, ordering, relevance, etc.

[Editors' note: further revisions were requested prior to acceptance, as described below.]

Thank you for resubmitting your work entitled "Firefly genomes illuminate parallel origins of bioluminescence in beetles" for further consideration at *eLife*. Your revised article has been favorably evaluated by Diethard Tautz (Senior Editor), a Reviewing Editor, and four reviewers.

The manuscript has been improved but there are some remaining issues that need to be addressed before acceptance, as outlined below:

Thank you for the detailed responses to the queries raised by the four reviewers and your efforts to address them in the main text and supplementary information. For the most part the reviewers are generally pleased with the updates, clarifications, etc. and feel that the manuscript is much improved. As the feedback is mostly positive, I have not requested a formal second round of reviews, but instead have consulted the reviewers to compile the following list of issues which we feel were not completely addressed through your revisions or that require some further clarifications.

1) Orthology analyses

The filtering of isoforms prior to orthology analyses is rather convoluted, the standard practice (as indicated on the OrthoFinder GitHub page) is to select the longest transcript (coding sequence) per gene (unless some other evidence supports selecting a canonical transcript). Nevertheless, the applied strategy has removed many alternative transcripts, and it is clearly indicated in the legend, and properly detailed (with scripts) in the supplement, and this issue mainly affects *Tribolium* and *Drosophila*, and *Drosophila* is no longer shown in Figure 2E (only the most observant reader would find Appendix 4—figure 1 with 15'152 genes for *Drosophila* when in fact it has only 13'931), so it is acceptable, albeit imprecise. Please ensure that any result that relies on or makes reference to *Drosophila* or *Tribolium* orthologues (e.g. gene trees) are checked to make sure they are not impacted by the erroneous inclusion of alternative transcripts.

2) BUSCO results

Lower geneset than genome BUSCO completeness: please add a qualifier P5L116, along the lines of – Higher BUSCO completeness of the assemblies compared to the genesets suggests that future curation efforts will lead to improved annotation completeness.

3) Molecular adaptation analyses

If we understand correctly, MEME was used only to test the branch leading to EAncLuc – so what about PmeLucV and PangLucV branches? If we understand correctly, PAML was used to test all three of these branches, but in the supplement we see results from only one background vs. foreground test, so this leaves us confused. Essentially – did the complementary tests confirm all three branches identified with aBSREL or not? Given the very high omega values for PmeLucV and PangLucV branches, and the low proportions of sites, are these perhaps not trustworthy anyway? We would be more inclined to dismiss these as unreliable rather than explain them by evoking sexual selection.

4) semi-redundant legends in Figure 4A

In the figure itself it is not immediately clear that the bottom left legend refers to inferred states of branches and nodes while the boxed legend next to the Elateridae refers to the observed states of extant species. Please make this difference clear.

5) Figure 3a) Please check the Ppyr scale and labels, going by the scale bar the distance between tick marks 28,000 and 28,350 should correspond to 50 Kbp – yes?

b) "whether genes not depicted in 3D are scattered in the genome"

We feel that the description in the legend could be more precise. Specifically "About ten PACS and ACS genes flank the Luc1 gene in both firefly genomes." E.g. something like the following (but please check details): Nine of the 12 *A. lateralis* PACS/ACS genes flank AlatLuc1 on scaffold 228, while four of the 11 *P. pyralis* PACS/ACS genes are neighbours of PpyrLuc1 on LG1 with a further six 2.4 Mbp and 39.1 Mbp downstream.

c) Reviewer.1 – "Given the size, especially of the second one, it would be fairer to the reader to explicitly label these, and probably also to enumerate the number of genes in these regions." We feel it is important to state how many genes are in these not-shown regions (labelled 2.4 Mbp and 36.7 Mbp) because if they are packed with other genes it hints at more shuffling while if just 'expanded' with inserted TEs then the synteny with Alat seems more confident.

---

## [Author Response]

Essential revisions:1) Orthology analysesConcerns were raised regarding the possibility that gene sets were not first filtered to select one protein per gene. If this is the case then these analyses will need to be performed on filtered gene sets and all downstream analyses that employ the orthology results will need to be updated, with conclusions revised accordingly.

Geneset analyses.For the *P. pyralis, A. lateralis*, and *I. luminosus* genesets, all peptide sequences presented were the “canonical” isoform of a given gene, as we only calculated a single (most well supported) transcript model per gene. We considered the global characterization of alternative splice variants to be outside the scope of the current manuscript, as we did not see evidence for alternative splice variants in our gene families of interest. Alternative splicing will be addressed in a separate manuscript detailing the ongoing curation strategy for the genesets and genome assemblies.

Orthology analyses.The *D. melanogaster* and *T. castaneum* NCBI peptide genesets were used for OrthoFinder protein clustering. Indeed, these include all alternative splice variants for given genes. We agree that these genesets should be prefiltered to the canonical isoforms.

Unfortunately, we were unable to find such a file of “canonical isoforms” for *D. melanogaster* and *T. castaneum* on Flybase.org or BeetleBase.org, although BeetleBase.org was not accessible due to a server configuration error at the time of writing.

We next turned to the Uniprot reference proteomes, which seemingly filter down to a single isoform. However, we detected substational gene-isoform redundancy in Uniprot reference proteomes (see BUSCO table below). This is due to the fact that genes are filtered down to single canonical isoform only for those proteins that have been manually reviewed on Uniprot (the “SwissProt” database). As a minority of genes/proteins in *Drosophila* (3,483 out of 21,957), and very few genes/proteins in *Tribolium* (2 out of 18,505), are manually reviewed on Uniprot, there is substantial remaining gene-isoform redundancy. Therefore, we sought to filter the Uniprot Reference Proteomes down to single isoforms.

Because we were unable to find an existing tool to filter Uniprot Reference Proteome down to “canonical” isoforms, we generated a custom script that filters Uniprot Reference Proteomes down to single isoforms by utilizing heuristics which select the isoform with the best reported “existence” score on Uniprot and the best BLASTP alignment scores to the peptide sets of other species. This custom script is available on Github (https://github.com/photocyte/filter_uniprot_to_best_isoform). The results of this filtering are shown below:

*Drosophila* Uniprot reference proteome (unfiltered)

(21,957 protein isoforms for 14,213 protein coding genes) C:99.8%[S:67.0%,D:32.8%],F:0.2%,M:0.0%,n:2442

*Drosophila* Uniprot reference proteome (custom script filtered):

(15,152 proteins for 14,213 protein coding genes)

C:99.6%[S:92.8%,D:6.8%],F:0.3%,M:0.1%,n:2442

*Tribolium* Uniprot reference proteome (unfiltered)

(18,505 proteins for 16,578 protein coding genes)

C:98.1%[S:89.4%,D:8.7%],F:1.5%,M:0.4%,n:2442

*Tribolium* Uniprot reference proteome (custom script filtered)

(16,991 proteins for 16,578 protein coding genes)

C:98.0%[S:95.8%,D:2.2%],F:1.6%,M:0.4%,n:2442

The remaining redundancy in the filtered datasets above is likely due to either (*i*) different splice variants that do not have significantly different BLASTP scores versus their alternative variants (and thus, should not affect OrthoFinder clustering results), or (*ii*) special cases that should not be reduced (e.g. the *Drosophila* gene(s) Rpb4, which have the same gene name and the same transcriptional start site but different gene IDs).

We now present OrthoFinder clustering results using the above filtered Uniprot reference proteomes for *D. melanogaster* and *T. castaneum*, and the latest version of OrthoFinder (v2.1.2 -> v2.2.6). Dependent analyses have now been redone and are listed below for the reviewers’ convenience:

- Figure 2E (updated to present genes instead of orthogroups)

- Figure 5

- Appendix 4—figure 4

- Appendix 4—figure 8

- Appendix 4—table 6

These figures show improved overlaps in orthogroups across species as compared to their previous versions, suggesting that the results of the orthogroup clustering heuristics were improved by filtering to single isoforms.

2) Quality assessmentsThe assessments are comprehensive, although a few details need to be checked (see below), but concluding that all three genomes and the annotations are 'high-quality' is misleading. The best assembly (P. pyralis) likely contains non-collapsed haplotypes while I. luminosus remains fragmented (~90K scaffolds).

We intended this statement to give context to the quality assessments of our assemblies in relation to other beetle genome assemblies and apologise that the reviewers felt misled. We have now revised the language in the main text (subsection “Sequencing and assembly of firefly and click-beetle genomes”, second paragraph), as well as the content of Figure 2F, to present quality statistics of our assemblies in comparison to the model beetle, *T. castaneum*, thus enabling readers to make their own assessment of the quality of our assemblies.

And the annotated gene sets appear to be less complete than the assemblies suggest they should be.

Yes, there is a discrepancy between the BUSCO scores generated from our assemblies (BUSCO: Ppyr – 97.2%, Alat – 97.4%, Ilum – 94.8%) and those of our annotated genesets (BUSCO: Ppyr – 94.2%, Alat- 90.0%, Ilum – 91.8%), suggesting the underperformance of the genesets generated by EvidenceModeler using the combination of multiple lines of evidence (*ab initio*, de novo transcriptome assembly and alignment, reference-based transcriptome assembly and alignment).

We were able to obtain “better” BUSCO values for genesets using Augustus *ab initio* gene prediction alone. However, upon manual examination of our target genes (luciferase, PACSs, ACSs), these predictions showed many errors, especially with closely related tandem genes which were made chimeric. This formation of chimeras was especially problematic for the tandemly arrayed gene families presented in this work.

Thus, we felt justified in our use of an empirically-optimized (optimization based on manual examination of target genes models) EvidenceModeler annotations, which heavily relied on our RNA-Seq data and only sparingly used *ab initio* gene prediction. However, despite an extensive empirical testing of the steps and parameters of our EVM based gene annotation pipelines, we were not able to match the BUSCO score from the *ab initio* annotation alone. In our experience, either annotation route produced flaws, but EVM did produce fewer flaws in the subset of genes that we were interested in. We plan to continue the curation of the genesets in a future manuscript.

Reassurances of manual curation supporting analyses of specific gene families (e.g. the ACTSs, PACSs, P450s and DNMTs) should therefore be clearly presented.

We did manually curate the members of specific gene families analyzed in the study. The description of our process has now been strengthened in Appendix 1, subsection “Official coding geneset annotation (PPYR_OGS1.1)” We now state this in the main text as well (subsection “Sequencing and assembly of firefly and click-beetle genomes”, second paragraph).

Unfortunately, we do not have a good record of the genes that were manually curated in the manuscript preparation process beyond the particular gene families described in the manuscript. Therefore, to aid the transparent tracking of future geneset issues and revisions, we have created a Github repository for the genesets of the three genomes, and have used that to track any geneset changes since the reviews were received. The *P. pyralis, A. lateralis*, and *I. luminosus* official genesets can be found at https://github.com/photocyte/PPYR_OGS, https://github.com/photocyte/AQULA_OGS, and https://github.com/photocyte/ILUMI_OGS respectively. These repositories are now linked to from www.fireflybase.org.

3) Molecular adaptation analysesSeveral points were raised regarding these analyses and the results (see below for details). (i) presenting results only for the Elateridae leads to the assumption that no positive results were found for the Lampyridae, but this is not discussed at all.

We did not run the aBSREL analysis for the entire dataset of cantharoid (click beetle + firefly + rhagophthalmid + phengodid) luciferase homologs. Instead, we tested whether the reconstructed elaterid luciferase ancestral branch in Figure 4A shows signs of positive selection, which would be sufficient to support the neofunctionalization of luciferase activity at that branch, thus supporting the claim of independence of the *AncLuc* and *EAncLuc* evolutionary origins.

A comprehensive molecular evolution analysis of luciferase homologs across bioluminescent beetles and their relatives is an interesting and logical follow-up to the findings in this study. However, we feel that such a study is premature due to limited data from key related taxa that could inform the origins of the ancestral luciferase (no genomic data for members of Rhagophthalmidae, Phengodidae, or Cantharidae).

Specifically, in the case of applying the same analysis to the cantharoid *AncLuc* branch as the reviewers suggest, the situation is complicated: the *AncLuc* neofunctionalization is more ancient (~>100 Mya) than the putative elaterid neofunctionalization, and the other “cantharoid” (phengodid and rhagophthalmid) sequences come into play as well. Unlike the elaterids, which have had a fair amount of work characterizing non luminous luciferase homologs (e.g. Oba, Y., Iida, K., Ojika, M., and Inouye, S. (2008): Orthologous gene of beetle luciferase in non-luminous click beetle, Agrypnus binodulus (Elateridae), encodes a fatty acyl-CoA synthetase. Gene *407*, 169–175.), the phengodids and rhagophthalmids have not been studied to the same degree. While luciferases have been cloned from phengodids and rhagophthalmids – no non-luciferase luciferase homologs are known from these taxa. In other words, there is a fair amount of missing sequence and topological uncertainty about the branch ancestral to AncLuc from the Figure 4A gene tree. Therefore, we posit that results from selection analyses at the base of AncLuc branch may be a bit misleading and would require sequencing of genomes of both phengodids and rhagophthalmids. We have added text which clarifies our stance to the Discussion (first paragraph), but overall consider the analysis as outside the scope of the current manuscript. These molecular and functional analyses are specialities of several of us within the collaboration (Oba, Lower, Stanger-Hall) and hold promise for future rigorous study. We do plan to revisit this in a separate manuscript.

(ii) reporting of some of the results leads to some confusion, e.g. the employed p-value cut-off and very high omega values.

We apologize for the confusion. The issue with the p-value cut-off as mentioned by reviewer 1 was due to an unfortunate typo. We have fixed that typo, and the text in the resubmitted version is now internally consistent with the analysis. For the very high-omega values, we agree with reviewer 3 that it is likely is due to a limited number of selected sites in those particular branches. Reviewer 3 suggested that we should present the likelihood ratio test statistic, and have done so by adding the LRT statistics to Figure 4B, and for the entire analysis in the supplementary files provided on FigShare (https://figshare.com/s/21a50b49b95b83f938c6)

Several reviewers touched on the significant but high-omega branches (shown as blue stars in Figure 3B), but we did not comment on these branches in the previous submission, as we had not yet fully solidified our thoughts on these.

We have now come to the conclusion that these may be signals of sexual selection on the dorsal vs. ventral color of the elaterid luciferases. Like fireflies, (certain) bioluminescent elaterids use their light as a courtship signal (*I. luminosus* appears to be one of these species), but in contrast to fireflies elaterids exclusively use continuous glows as courtship signals. There is an expectation that the dorsal and ventral light organs in elaterids perform different functions. Specifically, the dorsal light emissions are thought to function as aposematic signals and/or for illumination of the flight paths and/or landing sites, while the ventral light emissions are thought to function in mate attraction.. As well known from fireflies, luciferase emission color can be changed by single amino acid changes (Niwa, K., Ichino, Y., Kumata, S., Nakajima, Y., Hiraishi, Y., Kato, D.-I., Viviani, V.R., and Ohmiya, Y. (2010). Quantum yields and kinetics of the firefly bioluminescence reaction of beetle luciferases. Photochemistry And Photobiology *86*, 1046–1049.), and such changes, as a result of a selective sweep, have been documented in Jamaican click beetles, (Stolz et al., 2003., small number of sites changing drastically is consistent with our hypothesis of color evolution associated with sexual selection. While this is interesting in itself, it is somewhat outside the scope of the core question of the manuscript on ancient evolutionary origins. Therefore, to address the reviewers’ concerns we have added a sentence to the Figure 4B legend which suggests this. We reproduce the figure legend section below for the reviewers convenience:

“Branches with blue stars may represent the post-neofunctionalization selection of a few sites via sexual selection of emission colors.”

The results also leave at least two of the reviewers wondering about whether any specific sequence changes could be identified that might be linked to acquiring luciferase activity.

We agree, this is an interesting aspect to address. aBSREL analysis estimates the proportion of sites under a particular selective regime (ω category), but cannot test for specific sites under selection (see this Github issue for more information: https://github.com/veg/hyphy/issues/707). This can be thought of as akin to how PAML analysis can detect a proportion of sites under selection using branch by site models, but must then use the Empirical Bayes estimation to identify specific sites under selection.

To address this comment, we have now performed Mixed Effect Model of Evolution (MEME) analysis to identify specific sites under selection and present the results in Appendix 4, subsection “Testing for ancestral selection of elaterid ancestral luciferase (Figure 4B)” (referenced in the legend of Figure 4B in the main text).

For readers more familiar with PAML analysis, we also present the results of branch by sites tests in the aforementioned subsection. PAML results were consistent with abSREL and MEME – they identified a class of sites under selection on the branch leading to the ancestral elaterid luciferase (*EAncLuc*). However, PAML identified fewer sites than the more-sensitive MEME analysis and only 19 selected sites were identified by both analyses. This is not surprising given known false positive rates of these methods (Smith et al., 2015 and the sparse taxon sampling of our study. We stress that these analyses are merely a starting point and functional analysis is required to pinpoint specific mutations linked the neofunctionalization of luciferase activity, which we plan to address in a separate manuscript.

4) The 'contentious' hypothesisGiven that several recent studies are cited that have already lent support to the hypothesis on the independent origins of bioluminescence in beetles, including a 2016 paper from the same first author, emphasising the claimed 'contention' is unnecessary.

To address the reviewers’ concerns, we have removed the reference to “contentious” from the manuscript Abstract.

Regarding the reference to the 2016 paper, we thank the reviewer for highlighting this, indeed the sentence in the previously submitted version which claimed the findings were consistent with the previous hypothesis of independent evolution from (Fallon et al., 2016), was not written accurately. The 2016 LST paper, hypothesizes that LST is absent from Elateridae, but that doesn't reflect whether the bioluminescence overall is independent or not. We have rewritten the relevant section to more properly convey this logic:

“While an direct ortholog of LST is present in *A. lateralis*, it is absent from *I. luminosus*, suggesting that LST, and the presumed luciferin storage it mediates, is an exclusive ancestral firefly trait. This finding is consistent with previous hypotheses of the absence of LST in Elateridae (Fallon et al., 2016), and with the overall hypothesis of independent evolution of bioluminescence between the Lampyridae and Elateridae.“

5) Repeat annotationThe main text only discusses repeats in P. pyralis, but does not consider if the repeat content could have been inflated by the inclusion of haplotypes in the assembly.

We do believe there are redundant haplotypes in the *P. pyralis* assembly, as the main text states. We agree with the reviewers that, the reported repeat content could be inflated from a perfect assembly. However, as repetitive elements are amongst the sequence elements most poorly reconstructed in a genome assembly (e.g. contraction of tandem repeats), and are amongst the most genetically variable elements as well (expansion or contraction of STRs), the repeat count could have been underestimated as well. To address the reviewers’ concern, we have emphasized potential duplicates in the annotation.We reproduce that sentence below for the reviewers convenience:

“Remaining redundancy in the *P. pyralis* assembly and annotation, as indicated by duplicates of the BUSCOs and the assembly size (Figure 2F; Appendix 4—table 2) is likely due to the heterozygosity of the outbred input libraries (Appendix 1).”

A complex repeat in I. luminosus is discussed in the supplement, but how it relates to reconstructing the luciferase/ACSs/PACSs locus is ignored in the main text.

We believe that the reconstruction of the luciferase locus (i.e. the manual scaffold merge across the *I. luminosus* complex repeat), is well documented in the supplemental material (Appendix 3, subsection “Ilumi1.2: Manual long-read scaffolding”). To address the reviewers’ concerns, we have added a sentence to the figure legend of Figure 3D to emphasize this manual scaffold merge for the readers. We reproduce that sentence below for the reviewers convenience:

“The full Ilumi1.2_Scaffold13255 was produced by a manual evidence-supported merge of two scaffolds (Appendix 3, subsection “Ilumi1.2: Manual long-read scaffolding”)”

It remains unclear whether the repeat annotations offer any insights into the rearrangement histories of these loci in the three beetle genomes.

Yes, it is possible to consider the role of repetitive elements in the evolution of these loci. However, the Luc1/Luc2 duplication in fireflies occured >100 Mya, therefore we expect that, unless there is a selected role for an adjacent repetitive element, any neutrally evolving signal (e.g. non-functionally selected) repetitive elements would have been lost since the divergence of the Luciolinae, Lampyrinae, and Elateridae. To test this expectation/assumption and to address the reviewers’ concerns, we have evaluated if the repetitive elements +/- 100 Kbp of the Luc1, Luc2, and IlumLuc locus, and IlumLuc Scaffold 9654 in *P. pyralis, A. lateralis*, and *I. luminosus* respectively, show any sign of synteny/orthology between and within the three species, via BLASTN with an evalue threshold of 1e-5. We did not find any similarity of *I. luminosus* Luc adjacent repetitive elements to the repetitive elements of the two fireflies. Between the fireflies, we do not find Luc adjacent repetitive elements that are similar across the two species, but within the two firefly species, and within the *I. luminosus* assembly, we do find repetitive elements that are similar between the Luc1 and Luc2 loci, and the IlumLuc and Scaffold 9654 respectively. These repetitive elements may have been involved in the original or possible subsequent rearrangements involving these loci. There is a putative TE element upstream of the *P. pyralis* Luc1 locus, but the TE and its encoded ORF (obtained from close relatives in the genome, as the ORF of the particular element upstream of Luc1 is frame-shifted) do not have any sequence similarity to known repetitive elements or to any proteins in public databases. In short, while it would be interesting to investigate the role of particular repetitive elements in the evolution of this locus, there is no clear insight at this time. We plan to revisit this in a separate manuscript.

The full reviews are appended below but you need only consider the summary and essential points above.Reviewer #1:[…] 1) The fact that, for all three beetle species, the BUSCO completeness scores of the annotated gene sets are lower than for the assembly assessments suggests that despite the seemingly well-designed annotation strategies, they have failed to produce the best-quality automated annotations for these assemblies. As annotation pipelines employ a lot more evidence than BUSCO (which just uses profiles with Augustus), they generally produce gene set annotations that score better, or as good as, assessing the assemblies directly e.g. in Appendix 4—table 3 Dmela from 99.4% to 99.8%, Tcast from 98.4% to 99.0%. Having said that, the gene set completeness is still fairly good, and the main focus of the study is only on a limited set of genes, where manual curation was performed, so this should not substantially affect the main conclusions but should probably nonetheless be noted.

(Addressed in response to Essential revisions).

2) Labelling all three as 'high-quality genomes' is perhaps a bit of a stretch.

We have removed this label. (Addressed in response to Essential revisions).

3) The very high levels of duplicated BUSCOs in the gene sets of Dmela and a Tcast in Appendix 4—table 3 seem to suggest that alternative transcripts were not first filtered from the gene sets. To obtain reliable estimates of real gene duplications the annotations must be first filtered to select one protein representative per gene. The other species may be less affected by this as they probably do not have many alternative transcripts annotated (a problem also for orthology analysis, see below). This needs to be fixed to ensure that conclusions about levels of gene duplication are based on sound and like-for-like analyses.

The reviewer is correct, we have fixed this issue in the resubmitted version (see Essential revisions response for more detail).

4) Figure 2F. The BUSCO C and D percentages do not seem to match those presented in the supplement, 93.7% vs. 94.8% and 1.3% vs. 1.4%.

We appreciate the reviewer’s thorough review. Indeed, the Figure 2F values for Ilumi1.2 had not yet been updated from a previous version. We have fixed that issue.

5) The numbers presented as part of the orthology analyses are confusing. For example, Total numbers per species in Figure 2E do not match those reported in the text (numbers in the text are much higher). This might be explained if the figure only shows clustered genes (rather than all genes), but if so then this is not very well explained and hence the confusion. Much more worrying is the total gene count in Figure 2E for Drosophila melanogaster, 16991 genes – how can this possibly be when the current annotation at FlyBase has only 13931 protein-coding genes? Appendix 4—figure 8 gives a clue as to how this could be, as it shows 8 isoforms (alternative transcripts) of a single D. melanogaster gene. This would suggest that orthology clustering did not first select one protein per gene, a procedure that is more-or-less standard for most orthology delineation approaches. I am not familiar with the inner-workings of OrthoFinder, so perhaps the already-delineated clusters can be filtered to remove alternative transcripts, but I suspect it would be more rigorous to re-do the clustering using pre-filtered gene sets.

Indeed, alternative splice isoform filtered datasets were not used, and this has been addressed by rerunning the OrthoFinder analyses with properly filtered datasets for *D. melanogaster* and *T. castaneum* (Addressed in more detail in Essential revisions). As the reviewer notes, the orthogroup venn diagrams show overlaps of orthogroups (which contain 1 or more genes from a given species), not genes themselves, therefore the #s will not match exactly. We have updated the Figure 2E panel to show genes, instead of orthogroups. Appendix 4—figure 1, while it was updated for the new analysis, still shows orthogroups.

6) Figure 3D. I could not find mention of the sizes of the regions not shown for Ppyr1.3 LG1. The first one you can work out must span from 28400 to 30700 (so 2300), the second from after 30800 to before 67450 (so a bit less than 36650). Given the size, especially of the second one, it would be fairer to the reader to explicitly label these, and probably also to enumerate the number of genes in these regions.

The scale bar at the bottom right in Figure 3D indicates the scale for all three genome: an interval between marks indicates 25 kb. We apologize for the misleading of the Figure 3D but there are 3 regions of Ppyr1.3 LG1 in Figure 3D. We have revised this figure for more clarity.

7) Figure 3. The density of the figure makes it rather hard to figure out, but could it be that the orientation of PpyrMGST is inverted in 3B compared to 3D? Also, could there be a PACS missing in 3B for A. lateralis?

The reviewer is correct, the Figure 3B version of PpyrMGST was oriented the wrong way, and has now been updated. The Figure 3D orientation was correct. For simplicity, we have not shown all the genes in Figure 3B, and have added a sentence to the figure legend to report this.

8) Figure 4B. I must preface these next few comments with the fact that I am familiar with PAML's codeml approaches but not with HyPhy's aBSREL approach. I tried to read up on aBSREL but could not easily find good documentation on how to interpret the results. The main reason I was confused is that I don't know how to interpret an omega value of 147.67 labelled on the branch to LlumLuc+PjanLuc [after writing that and then looking at the actual aBSREL tree I realised that this label is in fact referring to the blue star branch – so if I misread that then others might too]. Although this node is not the main result, such a high omega value seems strange (if indeed omega represents dN/dS here) as it implies estimates along that branch of ~150 times more non-synonymous changes than synonymous changes. I'm guessing that with few sites (1.53%) it can send the ratio sky-high. In which case, is it even meaningful to report this value on the figure? Indeed it seems aBSREL results on github and in the aBSREL paper they usually show nothing higher than omega=10.

Yes, the few number of sites seems to be the major reasons for the omega values being so large. We now show the LRT values in addition to the omega values to help with reader interpretation. We believed that not showing the blue-star selected branches would misrepresent the results, and have now added a sentence to the main text which addresses the possible interpretation of these branches. A longer explanation of our interpretation of these blue star branches is present in the Essential revisions.

9) While familiarising myself with aBSREL I noted in the 2015 paper that it states "over 80% of branches in typical gene phylogenies can be adequately modeled with a single ω ratio model". The aBSREL analysis of the Elateridae luciferases and PACSs identified 20 branches with two rate classes and 17 with one – so is this gene tree particularly atypical? Or is there something else happening here?

We do believe there is a lot of selection happening in the luciferase gene tree (initial neofunctionalization – and then indirect/sexual selection towards certain colors). So this gene family may be unusual yes when compared to a typical gene phylogeny, of which the majority of genes are under purifying selection alone and do not intersect with sexual selection.

10) Figure 4B. A second question that immediately sprang to mind – how do these results compare with similar tests using PAML instead? For example, PMID:28094282 reports results from both approaches for neofunctionalisation of Caf1-55 after duplication. Confirmation of this kind, even just in the supplement, would help to reassure those, like me, who are still relatively new to aBSREL.

We have now run PAML analysis in addition to the aBSREL analysis and present the results in Appendix 4, subsection “Testing for ancestral selection of elaterid ancestral luciferase (Fig. 4B)”. PAML results support elevated rates of evolution on these branches.

I was unable to access the files on FigShare so I could not take a look at the data myself – since writing that I now have access. This however added to my confusion somewhat: Figure 4 legend states that there were 3 branches with significant (p<0.001) evidence of positive selection, but the aBSREL_stout_results.txt file reports only one (PangLucV) with p<0.001.

The reviewer is correct, this is a typo (a previous version of the analysis, with fewer genes in the tree, did show that the elaterid ancestral luciferase branch was significant at p<0.001). The correct cutoff with the most recent analysis is p<0.01. Although the Appendix text mentioned the correct p-value cutoff, indeed the main text was not correct and has now been updated.

11) Another question that Figure 4 raises is why the molecular adaptation analysis was only performed on the Elateridae (plus PpyrLuc1 as outgroup)? Does it mean that the rest of the tree showed no signals of episodic positive selection, or that this was not examined? Or was there some technical reason why this was not performed, e.g. too many branches to test?

We did not perform this analysis, due to the necessity of a greater taxon sampling in Rhagophthalmidae and Phengodidae to avoid misleading results (Addressed in response to Essential revisions).

Even if the tests fail to identify any significant signals of episodic positive selection, would it not be intriguing to examine amino acids in the alignment that are common to all/most luciferases but not amongst the PACSs/ACSs – i.e. have the two neofunctionalisation events converged on a common or slightly different molecular solution to luciferase activity? What do residues previously linked to the active site or cleft look like? Do site tests (rather than branch tests) pick up any of these potentially interesting amino acid positions?

We agree this is intriguing, and plan to address it in a separate manuscript, with more taxa and experimental analyses. A limited exploration of the selected sites of the EAncLuc is now present in the manuscript (also addressed in response to Essential revisions).

12) Definition of subfunctionalisation. I am personally not overly put off by the use of subfunctionalisation to describe the differentiation of firefly Luc1 and Luc2. Nevertheless, in some circles (and perhaps more so when discussing enzymes in particular) subfunctionalisation would more strictly refer to an ancestral function being partitioned between the duplicates – i.e. the ancestral function can now only be performed through both extant copies working to together. In this case the ancestral (molecular) function is luciferase activity, and the (molecular) functions of Luc1 and Luc2 are still both luciferase activity. What distinguishes them is where and when they are expressed – i.e. divergence of function between the two copies is in terms of expression rather than molecular function. Clarifying this, or avoiding using subfunctionalisation at all, would therefore perhaps be advisable.

We understand the reviewers concern. Describing changes of function in a biology is a tricky intersection of ontology and general understandings. In our case, we are following the definition of subfunctionalization given in the review by Innan, H., and Kondrashov, F. (2010). The evolution of gene duplications: classifying and distinguishing between models. Nature Reviews Genetics *11*, 97–108. In this review, a divergence of function in terms of expression patterns is considered subfunctionalization. But, the point on subpartitioning of ancestral roles is well taken. The ancestor to Luc1 and Luc2 may or may not have been involved in both a sexual signalling role, and a “glow” / potential aposematic role. We have revised the sentence to explicitly refer to this. We reproduce this phrase below for the reviewers convenience: “subfunctionalized in its transcript expression pattern to give rise to *Luc2”.*

13) Long-read sequence assembly of the firefly Pyrocoelia pectoralis genome. Fu et al., 2017. This very recent data publication offers the opportunity to at least examine the Luc1 locus in a second Lampyridae.

We have inspected this Luc1 locus in *Pyroceolia pectoralis* genome from Fu et al. As none of the redundant haplotypes have not been filtered out from this assembly, it was a bit tricky to interpret. But, the Luc1 loci looks quite similar to the *P. pyralis* Luc1 loci (as would be expected as both *Photinus pyralis* and *Pyroceolia pectoralis* are subfamily Lampyrinae fireflies). We have added a sentence plus citation to the main text which highlights the value of the *Pyroceolia pectoralis* genome assembly for future studies.

The level of detail provided in the supplement is overwhelming for a reviewer (I did not read all 140 pages), but at the same time it is very much appreciated as it clearly documents detailed background information and methods. I wish more manuscripts were this well supported with such detailed supplementary materials. Subsection “Opsin analysis” still has MS Word comments.

We thank the reviewer for their appreciation of the supplemental materials. Indeed, our hope is that the supplementary information would be an adequate description of the full process of such genome assembly/annotation and comparative genomics analyses. We continue to modify and proofread the supplementary, and do plan (if accepted) to work with the *eLife* editors to ensure the Supplementary Information are well-formatted for readers. Although we did doublecheck removing the MS Word comments before the initial submission, we apologize that some slipped through, and will ensure that further comments are not included in the resubmitted version.

Finally, the characterisation the holobiomes of the three species and identify putative symbionts, a parasitoid fly, as well as bacteria and viruses: this is encouraging to see, as (a) it suggests contaminant screening was comprehensive, and (b) 'contaminants' can actually be very interesting!

We thank the reviewer for their assessment. We agree, many of the contaminants were quite interesting, and especially in the case of the parasitic flies, unexpected!

Reviewer #2:In this manuscript, Fallon et al. explore the evolution of bioluminescence within the beetles, focusing on newly generated genomic and transcriptomic data for two distantly related species of firefly in comparison with a bioluminescent click beetle. This work is very comprehensive, incorporating a variety of sequencing methods to generate decent to high quality genome assemblies, supported by empirical data ranging from tissue-specific RNA-seq and bisulfite sequencing to mass-spec and holobiont analyses. The conclusion that bioluminescence was independently acquired in the firefly and click beetle lineages is convincingly supported by phylogenetic, syntenic, and previous (anatomical) results. A key asset of this work is the integrated approach to exploring this evolutionary novelty, including discussion of which biochemical components may have been ancestral and predated independent acquisition of the associated subtraits required for the full manifestation of beetle bioluminescence. I also appreciated the efforts taken in assessing synteny across fairly old lineages that have experienced a fair amount of gene shuffling. The extensive supplementary reports and files provide detailed documentation of the work. To strengthen this manuscript for publication, there are a few places where the analyses could be rounded out (particularly across the three species, to make the text less P. pyralis-centric), where the presentation of the material ought to be polished for clarity and to best reach a broader readership, and where a few (apparent) contradictions need to be reconciled.

We thank the reviewer for their positive assessment. Indeed, the synteny analyses across such anciently diverged lineages challenged the assumptions of the authors during preparation, but we believe the current presentation is accurate.

Specific rounding-out analyses:1) In support of the evolutionary scenario presented in Figure 3B and in light of the rather high repetitive content in P. pyralis (42.6%), do the authors detect any harbinger repetitive or transposable element sequences flanking either Luc1 or Luc2 in either P. pyralis or A. lateralis?

Although we do detect Harbinger repetitive elements in the firefly genomes, they are not found adjacent to the Luc1/Luc2 loci in either firefly. We further discuss the potential role of repetitive elements in the evolution of the Luc loci in the Essential revisions.

Here and elsewhere, please make cross-species assessments more accessible. While total repetitive content for P. pyralis is reported in the main text, it appears that the only way to determine this for A. lateralis and I. luminosus is by manually summing up the final columns of Appendix 2—table 2 and Appendix 3—table 3 deep within the supplement.

We have added the repeat content statistics of *A. lateralis* and *I. luminosus* to the main text.

2) As the repetitive content of P. pyralis may be inflated due to heterozygosity (suggested by the BUSCO duplicates), is it possible to add a filtering step or otherwise take levels of heterozygosity into account so as to make a more accurate estimation of repeat load?

(Addressed in response to Essential revisions).

3) Given the branch lengths in Figure 3C, is Luc2 more slowly evolving/ conserved than Luc1? Please perform the same molecular adaptation analysis on firefly Luc1 and Luc2 as presented in Figure 4B for the click beetle lineage Luc genes.

Yes, the interpretation is the tree is that Luc2 is more similar to AncLuc, than Luc1 is to AncLuc. This was described in the early molecular phylogenetics of Luc2, so we did not address it directly. Our response to the reviewers request to perform more in-depth molecular adaptation analysis of the firefly luciferases is addressed in the Essential revisions.

4) Given the care with which P450 genes were identified and curated in P. pyralis, please provide documentation that ensures the same level of care was used in assessing P450s in the other two species, which is particularly important (a) to support the claim of a P. pyralis-specific expansion of the CYP303 family and (b) in light of the low-quality assembly of I. luminosus.

The P450s of *P. pyralis* were manually named and analyzed. The P450s of *I. luminosus* and *A. lateralis* were not manually named, as is a labor intensive process which would give a marginal benefit to the interpretation. Even in the case of the manually named *P. pyralis* P450 genes, the ultimate gene-models are largely from the mentioned geneset annotations, which are comprehensive and similarly applied across species.

To address the reviewers concern of a CYP303 expansion not being captured in *A. lateralis* and *I. luminosus*, we have performed a TBLASTN analysis of CYP with evalue 1e-5 against these two genomes. This revealed that all tblastn hsps were already captured in the CYP gene models from the official genesets, supporting the specific expansion of the CYP303s in *P. pyralis*.

Please also ensure that the relevant supplementary sections are indeed cited in the main text section on P450s.

We have added a citation to the P450 supplementary section from the official geneset coding annotation supplementary section. We feel it isn’t necessary to cite use of the official geneset at every use in the main text.

Lastly, how were pseudogenes determined?

Pseudogenes were determined by the mutation of amino acids which are conserved in all functional P450s. This statement has been added to the supplementary P450 annotation section.

5) Regarding the enzymatic (not "enzymological", Introduction, last paragraph) basis of bioluminescence, highly efficient enzymes with high protein stability need not also be highly expressed at the transcript level. The analysis presented in Figure 6 fruitfully considers four selection criteria for candidate genes, but the other three criteria alone, without high expression, are already informative. Removing the high expression criterion would ~double the number of potential candidate genes. Do the authors find any important candidates among this set?

Thank you, we have changed the term “enzymological”. We agree that it is definitely possible that enzymes that are not highly expressed (HE), could be directly involved in bioluminescence. HE is, after all, an arbitrary cutoff. That being said, the firefly photophore tissue is highly specialized towards light production: e.g. luciferase is the 2nd most highly expressed transcript. Using the HE criteria reduces the number of genes down to a reasonable level suitable for direct presentation in the manuscript as a main text table, and also is a good “short-list” for functional follow up. But we agree that the other intersections in our analysis could be interesting. To address the reviewers concerns, we now provide excel files in the Figure 5 FigShare file which list additional intersections of the analysis. The “annot_DE-Es” excel file specifically addresses the reviewers request, and recovers 81 orthogroups overall. It is hard to gauge if a candidate is important, but with a first pass nothing jumps out.

Although two expected candidate genes are confirmed in the current analysis, take care to avoid the impression of cherry picking in data presentation. Two genes comprise a rather small litmus test. Were other expected candidates not found?

We apologize for potentially presenting the data in such a way that cherry-picking may be perceived. In truth, the enzymes involved in the firefly bioluminescent system are not well described, so we don’t have so many candidates to use as “positive controls”. To date. there are only 3 enzymes that have been reported to be linked with the firefly bioluminescent system: luciferase (de Wet et al., 1985), luciferin-regenerating-enzyme / LRE (Gomi et al., 2001), and LST (Fallon et al., 2016). In the author’s opinion, only luciferase and LST are unambiguously associated with the firefly luminescent system. The LRE orthologs are not highly nor differentially expressed in the *P. pyralis* or *A. lateralis* light organ, LRE orthologs are found in non-fireflies (data not shown), and recent manuscripts have had a difficult time reproducing the original enzymatic characterization of luciferin-regenerating-enzyme as a oxyluciferin catabolizing enzyme, instead assigning an unclear indirect role involving altered redox potentials (Hosseinkhani, S., Emamgholi Zadeh, E., Sahebazzamani, F., Ataei, F., and Hemmati, R. (2017). Luciferin-Regenerating Enzyme Crystal Structure Is Solved but its Function Is Still Unclear. Photochemistry and Photobiology *93*, 429–435.). We believe that the characterization of LRE as an oxyluciferin catabolizing enzyme with a key role in firefly bioluminescence is not justified, and personally don’t believe its claimed functional role in the firefly lantern. But, we believed that “spending the text” in the main text challenging the assumptions about LRE in the current literature within our manuscript, would be a distraction, and so chose not to do so. On the other hand, to discuss LRE, without challenging its role, in our opinion perpetuates its poorly supported (and in our mind – incorrect) functional role. In short, we believe that Luciferase and LST are the only known genes that could have matched our 4 criteria, and reassuringly they do. For other described non-enzyme genes, e.g. the fatty acid binding protein (Goh and Li 2011), and the octopamine binding secreted hemocyanin (Nathanson et al., 1989), these were not referred to in the main text as they are not enzymes, but we do include these in the Appendix text with appropriate citations.

6) In terms of both phylogeny and synteny, I do not find the designated Clade C (Figure 3C-D) well motivated as a distinct clade.

This comment is unclear to the authors. We believe that Clade C is well supported by the presented maximum likelihood gene tree in Figure 3C. For example, all the nodes within Clade C have a bootstrap value > 90%. Beyond that, Clade C does match the definition of a Clade: A monophyletic group descended from a single common ancestor. We agree, that it would be more convincing if the *P. pyralis* Clade C genes were colocalized with the Luc1 locus like they are in the *A. lateralis* loci, but synteny, in of itself, is not sufficient to argue that genes are within a clade, and conversely the lack of synteny does not disprove genes being within a clade, as gene rearrangement / reorganization may have occurred, and evidently did occur. If the reviewer is instead arguing that Clade C should be rooted at a more ancient node, such as the node that is ancestral to the current LCA of Clade C, that is fine, although it doesn’t change the results (In fact, we have changed Clade C to this definition in the resubmitted version). We assume that the concerns of the reviewer may be due to the proximity of Clade C to the elaterid luciferases and their closely related homologs. But, the purpose of Clade C is to delineate those genes that descended from the Clade C gene which was present at the Luc1 locus in the ancestral Lampyrinae-Luciolinae firefly (Figure 3B), not a Clade-C ancestor which was present in both Elateridae and Lampyridae. So, to redefine Clade C from a more ancient LCA, would lose the purpose we defined Clade C for: a clade that defines the genes that are descended from genes that were at one time present at the loci that is orthologous to the ancestral luciferase / Luc1 loci (and are not apart of the Clade A and Clade B lineages). We have produced a new Clade, Clade D, containing the elaterid luciferase homologs to further clarify this distinction.

Aspects to restructure or clarify:1) It is counterproductive and unnecessary to oversell the text. Citing Darwin three times throughout the manuscript (Introduction and Discussion) to highlight the historical interest in and speculation on independent origins of bioluminescence in beetles belies the unsupported claim in the Abstract that this is a "contentious" hypothesis.

While indeed, there is an aspect of narrative building and emphasis of general interest in our citation of Darwin, we also argue that this is a relevant and interesting historical background to the question, and as such, have not removed these historical references. In contrast, we have removed the claim of a “contentious” hypothesis (addressed in Essential revisions).

Indeed, several recent studies are cited that have already lent support to this hypothesis, including a 2016 paper from the same first author.

(Addressed in Essential revisions).

Similarly, the superlative claims in the first paragraph about the interest and importance of firefly bioluminescence are doubtful in comparison with GFP from jellyfish. A more consistent and milder tone would be helpful.

While we agree that GFP has been a revolutionary tool for biomedical science, and almost certainly is used more widely that luciferase, we have some responses to this specific claim: While GFP is indeed involved in the native bioluminescence system of *Aequorea victoria* through FRET with aequorin, the reconstituted use of GFP in biomedical research is most commonly a case of fluorescence, not an authentic bioluminescence. In the case of authentic bioluminescence, we believe that firefly luciferase is the most widely used variant of such, and that our claims are well supported.

With the data, the P. pyralis genome assembly is indeed high quality, but while the A. lateralis assembly is good, it is not as good and certainly not on par with that of Tribolium castaneum (subsection “Sequencing and assembly yield high-quality genomes”, second paragraph, based on scaffold number and NG50 statistics in Figure 2F). The orthogroup analysis in Figure 1E does not show a striking difference in number of shared OGs among the two fireflies compared to among all three species or pairwise between either firefly and the click beetle, although there is a high number of click beetle-specific OGs.

We have redone Figure 2E and reinterpreted the results in the main text. (Other aspects of Orthogroup reassignment addressed in Essential revisions).

2) Although individual figure panels are generally quite nice, the organization of the figures is incredibly poor. As near as I can see, the main text refers to material in the sequence 1A, C, B, 2F, E, A, B, D, C, 3C, A, D, B, 4, 6 before 5, then 5B, A, C, D. Reorder appropriately.

We have reordered the panels of Figure 1, rearranged Figure 5 and 6. For Figure 3, Figure 2, Figure 5, these figures are defined by horizontal panels, which constrains the organization of the panels.

Consider whether 3A truly is germane to that section of the manuscript.

We have reevaluated the role of 3A. We think this chemical scheme helps a naive reader comprehend the enzymatic function of firefly luciferase and its ancestors.

Similarly, the care in depicting the PTS1 motif in Figure 3 is only finally explained much later in the main text (subsection “Metabolic adaptation of the firefly lantern”, second paragraph).

The inclusion of PTS1 is now explained in the Figure 3 legend.

I also found the figures very dense, visually, with insufficient white space between panels and with some elements too small even when viewed digitally at >2x print size.

We are working to ensure that the figures will be provided in a high resolution / vector format that is suitable for digital scaling. We have attempted to reformat figures to ensure additional white space is present between panels.

For example, the arrowhead to the click beetle organs in 1C is too small and by physically touching what it is pointing to obscures rather than highlights.

We have increased the size of the arrow in Figure 1C, and have moved it so it is not obscuring the lantern.

It took several passes before I found the Ppyr LG1 label above the schematic in 3D.

We apologize for the poor clarity of this figure. It was a challenge to present all the syntenic loci in the same figure panel, without having a very large space. We do acknowledge that the figures are very dense, and require several passes to understand them fully. We hope that by providing several figures in an overall coherent format (e.g. shared presentation and matching of colors in Figure 3B, C, D), that ultimately the “forest” of the synergistic interpretation of the analyses is more easily grasped, rather than the “trees” of individual panels.

And I initially mistook the quotation marks in the Figure 6 table to mean omissions rather than "ditto".

We have added additional explanation to the figure legend to improve clarity.

3) I found the extent to which the reader is continually referred to the supplement, without any main text Materials and methods section, very frustrating.

We apologize for the reviewer’s frustration. This was an intentional choice in the intermediate writing and review of the manuscript, and we do hope to address the ultimate reorganization of methods with the *eLife* editors (addressed in Essential revisions). The reason behind this choice was: having a Materials and methods section which was a “watered down” version of the appendix led to two problems: (1) The question of which analysis deserved a place in the main text Materials and methods, (2) The problem of keeping the (shortened) main text methods in sync with the longer supplementary methods. Over the long term, a large degree of effort was spent trying to keep both the Appendix, Methods and Main text Methods up to date, and ultimately these two sections became desynced, which led us to focus entirely on the Appendix information for simplicity.

Judicious inclusion in the main text of key specifics would be helpful, including: specifying that the sex determination system is XO (subsection “Sequencing and assembly yield high-quality genomes”, third paragraph).

We have added this.

That the "targeted molecular evolution analysis" uses a specific, citable pipeline (Mesquite, suppl. reference 213, for main text subsection “Independent origins of firefly and click beetle luciferase”, last paragraph).

We have modified this sentence to be plural, as the targeted molecular evolution analyses refers to both the ancestral reconstruction, and the selection analysis. That being said, the following sentence now has the Mesquite reference.

Whether the entire protein (or gene locus?) is considered in the molecular adaptation analysis.

We have added a reference to “utilizing the coding nucleotide sequence”

Which non-luminescent tissues were used (subsection “Metabolic adaptation of the firefly lantern”, first paragraph).

This varies between the two species, so is better stated in the Materials and methods section for brevity.

Which BUSCO version/ taxonomic group/ number of genes was used (Figure 2 legend), etc.

This have added this information to the Figure 2 legend.

4) Although extensive and well structured, please polish the supplement. For example, remove lingering track changes/comments, cross-reference across sections as appropriate (e.g., for P450s per species and in comparative analyses), and ensure that information is clearly presented.

We have added additional references to and within the Appendix when appropriate.

For example, the main text is vague on the nature of the non-bioluminescent tissues used for the DE analysis, yet the list of libraries in Appendix 1—table 1 is based on opaque internal identifiers ("OAG"?) that make it difficult to ascertain either tissue type or number of biological replicates.

In the revised version, we have moved the Sleuth differential expression analysis to Appendix 4, subsection “Comparative RNA-Seq differential expression analysis (Figure 5)”. Appendix 4 is intended to represent “comparative genomic” or cross-species analyses, and the DE analysis fits into this category. Previously it was in Appendix 1 (*P. pyralis* analyses) which wasn’t a good fit. This section now explicitly lists the tissues which were compared between. 3 biological replicates were used per condition. OAG is “other accessory glands”, and is now present in the table legend. We are unable to change these identifiers as these are datasets from previously published manuscripts.

Similarly, although I appreciate the detail presented on gene structure in Appendix 4—figure 4, *'s or other annotations to highlight identities would make the alignment easier to assess.

The main thing we are trying to tell with this figure is the complete conservation of the intron-exon boundaries, so for simplicity we have not added such an identity row to Appendix 4—figure 4.

5) Throughout, consider a broad readership and avoid ambiguous jargon. I am still uncertain how "evolutionary events" fit into the Hi-C "long-range" analysis (subsection “Sequencing and assembly yield high-quality genomes”).

We agree, this phrasing is unclear. We have rephrased this to now say “long-range genetic structure”, rather than evolutionary events. For the evolutionary events, our specific question was the possibility of a long-range duplication event for Luc2 from AncLuc, as would be suggested by being found on separate linkage groups. Another question we had (but wasn’t written down as it was a negative result), was whether the known bioluminescence genes would be clustered. We have added a section which addresses this.

The intended meanings of the term "promiscuous" (Discussion) are unclear. If a single enzyme can process multiple substrates (functionally promiscuous), how does this equate with a "need" for duplication and subfunctionalization for high specificity? That there are multiple synthesis possibilities to produce luciferin is not "promiscuous".

In the first use of the term, we do mean substrate promiscuity. In the second use of the term, we have changed “promiscuous” to “sporadic low-level”, which better conveys our thinking.

In the legend for Figure 3, "Color gradients indicate the TPM values of whole body" is not nearly as accessible to a non-bioinformatician interested in beetle evolution as "Color gradients denote gene expression levels (TPM) from whole-body samples…". And similar instances (e.g., define BSN-TPM in the legend to Figure 6)…

We appreciate the reviewer’s suggestion and have made these changes.

6) Although the first Results section is appropriately concise as a foundation for the subsequent sections, the paragraph breaks and text flow within this wide-ranging section are difficult to follow and would benefit from smoothing and textual transitions.

We have added additional textual transitions to improve flow in the first Results section.

7) Please make clear whether the genome assembly or OGS was used for specific analyses (e.g., subsection “The genomic context of firefly luciferase evolution”, first and second paragraphs). Was there really a de novo scan of the assembly itself?

This result is predicted from the OGS, but is also the same result we found in many intermediate analyses / de novo transcriptomes. The OGS is in a sense a de novo / *ab initio* scan of the assembly, as our OGSs incorporates results from the *ab initio* gene finder Augustus. In general, if we do not state the source of the genes in an analysis, they are from the OGS.

8) Not all PACS/ ACS genes included in the phylogeny in 3C are shown in 3D.

Yes, the synteny analysis in Figure 3D only includes those genes that are co-localized with Luc1 (for the fireflies), and the scaffolds which contains the closely related homologs for *I. luminosus*.

Please comment on the gene nomenclature and numbering conventions used for the taxa presented here.

There are two gene numbering conventions at play in the manuscript

1) The systematic official gene set naming scheme (PPYR_00001-PA)

2) The “Human readable” naming scheme (e.g. PpyrLuc1, PpyrPACS1,2,3)

In both cases, they are numbered in a monotonically ascending order from left to right (using the scaffold orientation). In the case of Figure 3D, for some scaffolds the naming scheme goes from right to left, as what is actually in the figure is the reverse complement of the scaffold to match the organization of the *A. lateralis* orthologous loci.

And whether genes not depicted in 3D are scattered in the genome.

Yes, if these genes are not within the Figure 3D synteny diagram, they are outside those loci and likely found in various genomic locations.

9) The comment about potential lysosomes and the biological significance of an opsin gene being expressed within the light organ itself are unclear (“Metabolic adaptation of the firefly lantern”, last paragraph).

We believe these are intriguing findings from the study, which we mention in passing to highlight them for future researchers, but consider deeper characterization outside the scope of the current manuscript.

10) New genomes are continuously forthcoming. Provide a reference for the 62 referred to in the last paragraph of the subsection “Genomic insights into firefly chemical defense”.

We have added a reference for OrthoDB. The new value is 94/97 of the sequenced winged-insect genomes have a single CYP303 family gene within OrthoDB.

11) It would be interesting and helpful for the authors to slightly expand the discussion on the possible link between bioluminescence and lucibufagin evolution. Is this a belt and braces situation of being doubly sure between CYP303 expansion and the whole body glow afforded by Luc2?

This comment is unclear to the authors. The CYP303 expansion presumably came after the gene-duplication/subfunctionalization and fixation of the Luc1-Luc2 roles (already present and functionally equivalent to the extant taxa in the Luciolinae-Lampyrinae common ancestor). We think a deeper discussion on the link of aposematism to bioluminescence is better suited for its own manuscript.

12) T. castaneum is a long established insect species for genetics and genomics and would be the more taxonomically appropriate outgroup for the four-species orthogroup analysis shown in main text Figure 2E.

We have updated Figure 2E with *T. castaneum* instead. This figure has also been updated to show gene overlaps and numbers, rather than orthogroups.

Issues to reconcile:1) Given the availability of fresh A. lateralis material for applications such as RNA-seq, I find it weak that the flow cytometry measurements for this species are based on only a single biological specimen.

We understand the reviewer’s concerns. We are attempting to obtain more *A. lateralis* specimens for such a measurement, but there are certain practical concerns which may preclude this result from being completed in a timely fashion and therefore from included in the resubmitted version. In the meantime, we have added a kmer based genome-size measurement for *A. lateralis*, in Appendix 2—figure 2. The size as measured by flow cytometry is 940 Mbp, whereas the genome size inference via kmer analysis is 772 Mbp. We acknowledge that these values are quite different, but we highlight that in most genome papers in our experience, a genome-size measurement is usually provided by kmer analysis, or flow cytometry, but not both. As both methods have systematic biases which are not addressed, and the values are often presented as is without even proper addressing of the experimental variability, we argue that the discrepancy in genome size between the two values is due to the systematic error in such methods, and that chasing down the “exactly right” genome size measurement would not be informative to the main conclusions of the paper.

2) If PACS genes gave rise to firefly luciferase genes (Figure 3B), why is the opposite evolutionary direction assumed for IlumLuc (subsection “Independent origins of firefly and click beetle luciferase”, first paragraph)?

Upon rereading those lines, we agree the terms used were a bit unclear as to what we were trying to say. We say PACS genes descended from genes ancestral to IlumLuc (which were also PACSs genes – not stated). We have rephrased that section for greater clarity.

3) What is the evidence that IlumLuc is highly expressed in the abdominal lanterns (subsection “Independent origins of firefly and click beetle luciferase”, first paragraph) if the RNA-seq sample is only based on the prothorax (legend for Figure 3C)?

We have RNA-Seq data from both the prothorax, and abdomen, both of which show high levels of IlumLuc expression. We have updated Figure 3C with an additional row presenting the expression quantification from both the abdominal and prothorax lanterns.

I provided extensive, specific details above, but an overall concern is presentation, especially figures that are too dense and with poorly ordered and possibly irrelevant material.

We appreciate the reviewer’s concerns. We have reevaluated the main text figures to remove/reduce the figure density, and will continue to consider these types of manipulations after the review process (addressed in Essential revisions).

Reviewer #3:eLife[…] Some (hopefully constructive) criticisms are as follows.Different genome sequencing methods were used to construct the data for this project, resulting in unequal assembly quality. This possibly impacts the analysis of parallel evolution in the luciferase homology gene family (based on protein blast), as incorrectly assembled or missing data might bias results in the more limited reference genome of the elaterid. An alternative approach to searching for these genes might be undertaken to help confirm these results, i.e. PCR or the use of ATRAM using the raw reads (https://bmcbioinformatics.biomedcentral.com/articles/10.1186/s12859-015-0515-2)

We thank the reviewer for the comments. Indeed, there is always a concern with a non-finished genome that something may be missing or mis-annotated. But, we are relatively confident with the genome quality of the elaterid. For example, we captured direct orthologs of all the bioluminescent elaterid luciferase homologs cloned to date (e.g. PangPACS), as well as two previously unknown homologs (IlumPACS8 and IlumPACS9). IlumPACS8 and IlumPACS9 are particularly informative as they are descended from the ancestral gene to the elaterid ancestral luciferase. As this is a linked-reads based genome with DNA that was measured to have a >15 kbp mean input DNA size, we think we have scaffolded over most repetitive elements of 10 Kbp in size.

If the reviewer means degenerate PCR of cDNA to detect additional luciferase homologs, that approach was the previously applied approach for gene discovery in elaterids, and we would argue that our genome assembly mediated discovery of IlumPACS8/IlumPACS9 demonstrates we have approved on this approach. Additionally, several of the luciferase homologs are barely expressed by RNA-Seq, so a cDNA cloning based approach would be very difficult in this or similar cases.

If the reviewer means degenerate PCR of genomic DNA, we would argue that for those genes which may not be captured in our assembly, they likely have very long repetitive elements in introns (e.g. 15 Kbp+), or tandemly duplicated arrangements, which would prevent a PCR based approach at discovery.

For a targeted assembly, such as ATRAM, while we agree that targeted assemblies can be great for quick evaluations of a given gene family, we think our full genome assembly using the additional information provided by the linked-reads barcodes (which the Supernova assembler does use), improves on any assembly approach that only uses the short read information.

To address the reviewers concerns of mis-annotation, we have searched the Ilumi1.2 assembly using a tblastn of the *I. luminosus* luciferase peptide sequence (evalue cutoff = 1e-10). Such a tblastn search captures 49 scaffolds. We have manually inspected the tblastn HSPs for each of these 49 scaffolds, and their intersection with the existing gene models, and the *ab initio* and de novo transcriptome and reference guided transcriptome evidence, and used that to manually improve the existing gene models. Importantly this process picked up 3 gene models which were present in the *ab initio* gene prediction, but were erroneously absent in the ILUMI_OGS1.1. We have fixed these gene models and the fixes are available in the resubmitted version as ILUMI_OGS1.2. Relevant dependent analyses, such as expression quantification, InterProScan protein function annotation, Figure 3C, Figure 3D, have been redone.

Some aspects of the figures seem uninformative (at least to the content of this paper). In Figure 1, the geographical distribution and sampling of P. pyralis is shown. This could be moved to the supplement, as it has minimal bearing on the paper as a whole. In Figure 2, much of the content seems better for supplement.

We thank the reviewer for their comments. We believe the supplementary is already quite expansive, and that Figure 1 and Figure 2 provide relevant introductory information. We would argue that moving Figure 1 and Figure 2 content into the supplementary would likely make this information quite hard to find. But we will continue to evaluate such reorganization.

I liked Figure 3 (very dense information though), and also wondered if a model of the evolution of the lantern from a peroxisome might want a cartoon figure.

We thank the reviewer for their assessment. Indeed, it is a tough task to present such a density of data in a coherent figure. For the reviewer's comment on the evolution of the lantern from a peroxisome, this is unclear to us. Please allow us to clarify for the reviewer in case there is confusion: the lantern / light organ refers to the luminous tissue/organ (made up of specialized cells – photocytes, amongst several other cell types, e.g. neurons and tracheal cells), rather than the luminous organelle (perixosome) within the photocyte. The luminous organelle of the photocyte, to our knowledge, is a standard peroxisome, though it certainly filled with a lot of luciferase, and may be larger than a standard peroxisome, and are very abundant in the photocyte cells. We do think that considering the tissue level evolution of the bioluminescent system is a valuable goal, but perhaps outside the scope of the correct manuscript.

Figure 4B. The large values of omega emerge as a result of a lack of synonymous sites (no synonymous sites lead omega to be infinity). Y. Ziheng suggests reporting likelihood ratio test statistics as being more informative.

We have updated Figure 4B to include the likelihood ratio test statistics, in addition to the omega statistics.

Reviewer #4:This study analyses the genomes of two fireflies and a click beetle to investigate the origins of bioluminescence. The authors offer convincing evidence for two independent origins thus showing that bioluminescence most likely evolved convergently in fireflies and click beetles. This manuscript is well written and the study provides many novel and interesting findings. I have a few specific comments.Subsection “Independent origins of firefly and click beetle luciferase”, last paragraph and Figure 4B: Why is the analysis of molecular evolution only carried out on click beetle luciferase and not also on the fireflies?

Addressed in response to Essential revisions).

Furthermore, is it possible to identify the specific adaptive mutations?

(Addressed in response to Essential revisions).

Are homologous positions evolving adaptively in both suspected origins of luciferase or at least within the same domain/structure?

We have not performed the adaptive site analysis for the firefly homologs (Addressed in Essential revisions). As for whether sites are in the same domain, we presume they would be, beetles luciferases have the same domain structure amongst one another and with the ancestral acyl CoA synthetases, which is generally considered to be an N-terminal domain, and C-terminal domain, with the active site at the interface.

Structuring of supplementary information and its referencing within the manual need to be improved.Overall, I was very pleased with the quality of figures and the density of information they convey. However, several points concerning their arrangement, ordering, relevance, etc.

We thank the reviewer for their constructive comments, and have partially reorganized the main text figures to improve their arrangement, ordering, and relevance, and will continue to evaluate such reorganization

[Editors' note: further revisions were requested prior to acceptance, as described below.]

The manuscript has been improved but there are some remaining issues that need to be addressed before acceptance, as outlined below:Thank you for the detailed responses to the queries raised by the four reviewers and your efforts to address them in the main text and supplementary information. For the most part the reviewers are generally pleased with the updates, clarifications, etc. and feel that the manuscript is much improved. As the feedback is mostly positive, I have not requested a formal second round of reviews, but instead have consulted the reviewers to compile the following list of issues which we feel were not completely addressed through your revisions or that require some further clarifications.1) Orthology analysesThe filtering of isoforms prior to orthology analyses is rather convoluted, the standard practice (as indicated on the OrthoFinder GitHub page) is to select the longest transcript (coding sequence) per gene (unless some other evidence supports selecting a canonical transcript).

We feel that longest isoform selection, while admittedly a straightforward and easily applied heuristic, is prone to error. Here, we assume that a typical error mode for alternative splice variant structural annotation would be to produce an erroneously long transcript / peptide model, hence our more complicated approach, which places primary importance on isoforms with the best annotated evidence, seems reliable. The most reliable best practice might be to rely on the human-annotated canonical isoform for every gene, however to our knowledge no such FASTA file is provided by the *D. melanogaster* and *T. castaneum* genome databases.

Nevertheless, the applied strategy has removed many alternative transcripts, and it is clearly indicated in the legend, and properly detailed (with scripts) in the supplement, and this issue mainly affects Tribolium and Drosophila, and Drosophila is no longer shown in Figure 2E (only the most observant reader would find Appendix 4—figure 1 with 15'152 genes for Drosophila when in fact it has only 13'931), so it is acceptable, albeit imprecise.

Yes, the reviewers requested the change from *Drosophila* to *Tribolium* in Figure 2E in the last revision.

Please ensure that any result that relies on or makes reference to Drosophila or Tribolium orthologues (e.g. gene trees) are checked to make sure they are not impacted by the erroneous inclusion of alternative transcripts.

We have reviewed the following gene trees in the manuscript:

Figure 3C: No redundant isoforms

Figure 4A/B: No redundant isoforms

Appendix 1—figure 13: No redundant isoforms (doesn’t refer to *D. melanogaster* or *T. castaneum* sequences)

Appendix 4—figure 2: No redundant isoforms

Appendix 4—figure 6: No redundant isoforms

Appendix 4—figure 8: No redundant isoforms

Appendix 4—figure 9: Redundant isoforms, but doesn’t affect results and documented in metadata.

2) BUSCO resultsLower geneset than genome BUSCO completeness: please add a qualifier P5L116, along the lines of – Higher BUSCO completeness of the assemblies compared to the genesets suggests that future curation efforts will lead to improved annotation completeness.

We have added an equivalent sentence to the end of the second paragraph of the Results:

“The higher BUSCO completeness of the assemblies as compared to the genesets (Appendix 4—table 3), suggests that future manual curation efforts will lead to improved annotation completeness.”

3) Molecular adaptation analysesIf we understand correctly, MEME was used only to test the branch leading to EAncLuc – so what about PmeLucV and PangLucV branches? If we understand correctly, PAML was used to test all three of these branches, but in the supplement we see results from only one background vs. foreground test, so this leaves us confused.

The reviewers are correct — for the selected-sites analysis, MEME was used to test only the branch ancestral to EAncLuc, whereas for the PAML-BEB analysis, all three branches (EAncLuv, PmeLucV, Panluc) were assigned to the foreground. To make the PAML-BEB analysis analogous to the MEME analysis, we have updated this analysis and figures so the foreground branch is now only that of the branch ancestral to EAncLuc. Figure S4.3.4.3 now shows the union of selected sites detected by both PAML-BEB and MEME, rather than the intersection as was previously shown. We highlight that this is a preliminary analysis, and that functional experiments are required to definitively comment on any selected sites suggested by this analysis.

Essentially – did the complementary tests confirm all three branches identified with aBSREL or not?

The PAML and aBSREL analyses for branch selection were roughly concordant. PAML branch tests (not shown) supported the free-ratio model as the best fit, and showed elevated rates of directed evolution of the EAncLuc, PmeLucV, and PangLucV branches, with omega values ~0.6+, compared to a background elsewhere in the phylogeny of <0.1. However, the most strongly selected branch in the PAML branch analysis was that of the ancestral node to PmeLucD and PplagLucD. This branch was also found in the aBSREL analysis, but was below our chosen significance threshold. We argue that, because of the important function of the gene, we expect most of the luciferase to be conserved, but with elevated rates of evolution in specific domains or at specific active sites within the molecule. Thus, branch-by-site models are more appropriate, and PAML results from this analysis were concordant with the abSREL and MEME results (see above).

Given the very high omega values for PmeLucV and PangLucV branches, and the low proportions of sites, are these perhaps not trustworthy anyway? We would be more inclined to dismiss these as unreliable rather than explain them by evoking sexual selection.

We don’t think the PmeLucV and PangLucV branches are unreliable, as a low proportion of positively selected sites is the result we would expect for episodic selection of luciferase emission color. This is because mutation of relatively few key sites in the luciferase active site is known to strongly modulate the emission color. We also did not mention, both PangLucV and PmeLucV are in fact red-shifted luciferase variants (emission of PangLucV=566 nm, PmeLucV = 554 nm, compared to a presumed ancestral green color of ~530-550 nm).

We have updated the Figure 4B legend to better describe this thinking:

“As the selected branches with blue stars are red-shifted elaterid luciferases (Yuichi Oba, Kumazaki, and Inouye 2010; Stolz et al., 2003), they may represent the post-neofunctionalization selection of a few key sites via sexual selection of emission colors.”

4) semi-redundant legends in Figure 4AIn the figure itself it is not immediately clear that the bottom left legend refers to inferred states of branches and nodes while the boxed legend next to the Elateridae refers to the observed states of extant species. Please make this difference clear.

We have added additional figure legend text to emphasize this difference, which we reproduce here for the reviewers’ convenience:

“Luciferase activity (top right figure key; black: luciferase activity, white: no luciferase activity, shaded: undetermined) was annotated on extant firefly luciferase homologs via literature review or inference via direct orthology. The ancestral states of luciferase activity within the putative ancestral nodes were then reconstructed with an unordered parsimony framework and a maximum likelihood (ML) framework (bottom left figure key; Appendix 4, subsection “Ancestral state reconstruction of luciferase activity (Figure 4A)”).”

5) Figure 3a) Please check the Ppyr scale and labels, going by the scale bar the distance between tick marks 28,000 and 28,350 should correspond to 50 Kbp – yes?

Our apologies, the 28,000 value is a typo, and should instead be 28,300. We have redrawn the three *P. pyralis* loci in Figure 3D from scratch to preclude any other issues.

Through this redrawing process, we have recognized and included two non-PACS/ACS family genes (PPYR_01208; PPYR_01209) that were excluded from the previous figure as they did not have homologs in the other *A. lateralis* and *I. luminosus* scaffolds in Figure 3D. We have indicated these genes with a white background surrounded by a black border, and indicated this in the Figure 3 legend.

Recognizing the reviewer concerns about the gaps between the *P. pyralis* luciferase homolog loci being misleading or hard to see, we have removed extraneous non-genic scaffold space, and instead use that space to emphasize the length of the gaps in the figure.

b) "whether genes not depicted in 3D are scattered in the genome"We feel that the description in the legend could be more precise. Specifically "About ten PACS and ACS genes flank the Luc1 gene in both firefly genomes." E.g. something like the following (but please check details): Nine of the 12 A. lateralis PACS/ACS genes flank AlatLuc1 on scaffold 228, while four of the 11 P. pyralis PACS/ACS genes are neighbours of PpyrLuc1 on LG1 with a further six 2.4 Mbp and 39.1 Mbp downstream.

We agree, this sentence should be more descriptive. We have modified the sentence in-line with the reviewers’ suggestion, and present the revised sentence below for convenience:

“Nine of the 14 *A. lateralis* PACS/ACS genes closely flank AlatLuc1 on scaffold 228, while four of the 13 *P. pyralis* PACS/ACS genes are close neighbors of PpyrLuc1 on LG1, with a further seven genes 2.4 Mbp and 39.1 Mbp away on the same linkage-group.”

c) Reviewer 1 – "Given the size, especially of the second one, it would be fairer to the reader to explicitly label these, and probably also to enumerate the number of genes in these regions." We feel it is important to state how many genes are in these not-shown regions (labelled 2.4 Mbp and 36.7 Mbp) because if they are packed with other genes it hints at more shuffling while if just 'expanded' with inserted TEs then the synteny with Alat seems more confident.

The situation is the former case, there are many genes in these gapped regions:

*P. pyralis* 2.4 Mbp Gap1: 630 genes

*P. pyralis* 36.7 Mbp Gap2: 4511 genes

We have not added the number of genes into the figure itself for a practical consideration: If we were to add gene counts to the *P. pyralis* gapped loci, we would also need to add them for all the gapped edges, and there simply isn’t space in the figure for this additional text for all gapped areas. We do emphasize that this data is available to readers even without its explicit presentation: this information is available from the genome browser on www.fireflybase.org, and parsing of the official geneset GFF3 files (linked through www.fireflybase.org, currently hosted on Github).